# Beyond NTK with Vanilla Gradient Descent: A Mean-Field Analysis of Neural Networks with Polynomial Width, Samples, and Time

**Arvind Mahankali**[*]
Stanford University
amahanka@stanford.edu

**Jeff Z. HaoChen**[†]
Stanford University
jhaochen@stanford.edu

**Kefan Dong**
Stanford University
kefandong@stanford.edu

**Margalit Glasgow**
Stanford University
mglasgow@stanford.edu

**Tengyu Ma**
Stanford University
tengyuma@stanford.edu

## Abstract

Despite recent theoretical progress on the non-convex optimization of two-layer neural networks, it is still an open question whether gradient descent on neural networks without unnatural modifications can achieve better sample complexity than kernel methods. This paper provides a clean mean-field analysis of projected gradient flow on polynomial-width two-layer neural networks. Different from prior works, our analysis does not require unnatural modifications of the optimization algorithm. We prove that with sample size $n = O(d^{3.1})$ where $d$ is the dimension of the inputs, the network trained with projected gradient flow converges in $\text{poly}(d)$ time to a non-trivial error that is not achievable by kernel methods using $n \ll d^4$ samples, hence demonstrating a clear separation between unmodified gradient descent and NTK. As a corollary, we show that projected gradient descent with a positive learning rate and a polynomial number of iterations converges to low error with the same sample complexity.

## 1 Introduction

Training neural networks requires optimizing non-convex losses, which is often practically feasible but still not theoretically understood. The lack of understanding of non-convex optimization limits the design of new principled optimizers for training neural networks that use theoretical insights.

Early analysis on optimizing neural networks with linear or quadratic activations [29, 36, 51, 35, 65, 38, 48, 39, 60] relies on linear algebraic tools that do not extend to nonlinear and non-quadratic activations. The neural tangent kernel (NTK) approach analyzes nonconvex optimization under certain hyperparameter settings, e.g., when the initialization scale is large and the learning rate is small (see, e.g., Du et al. [30], Jacot et al. [43], Li and Liang [50], Arora et al. [10], Daniely et al. [26]). However, subsequent research shows that neural networks trained with practical hyperparameter settings typically outperform their corresponding NTK kernels [9]. Furthermore, the initialization and learning rate under the NTK regime does not yield optimal generalization guarantees [76, 22, 77, 34].

Many recent works study modified versions of stochastic gradient descent (SGD) and prove sample complexity and runtime guarantees beyond NTK [23, 2, 58, 52, 1, 7, 5, 79, 78, 25, 6, 67, 70, 20, 59]. These modified algorithms often contain multiple stages that optimize different blocks of parameters

---

[*]Equal Contribution

[†]Equal Contribution

37th Conference on Neural Information Processing Systems (NeurIPS 2023).

and/or use different learning rates or regularization strengths. For example, the work of Li et al. [52] uses a non-standard parameterization for two-layer neural networks and runs a two-stage algorithm with sharply changing gradient clipping strength; Damian et al. [23] use one step of gradient descent with a large learning rate and then optimize only the last layer of the neural net. Nichani et al. [59] construct non-standard regularizers based on the NTK feature covariance matrix, in order to prevent the movement of weights in certain directions which hinder generalization. Additionally, Abbe et al. [2] only train the hidden layer for $O(1)$ time in the first stage of their algorithm, and then train the second layer to convergence in the second stage. They do not study how the population loss decreases during the first stage. However, oftentimes vanilla (stochastic) gradient descent with a constant learning rate empirically converges to a minimum with good generalization error. Thus, the modifications are arguably artifacts tailored to the analysis, and to some extent, over-using the modification may obscure the true power of gradient descent.

Another technique to analyze optimization dynamics is the mean-field approach [21, 56, 57, 61, 28, 2, 44, 62, 76], which views the collection of weight vectors as a (discrete) distribution over $\mathbb{R}^d$ (where $d$ is the input dimension) and approximates its evolution by an infinite-width neural network, where the weight vector distribution is continuous and evolves according to a partial differential equation. However, these works do not provide an end-to-end polynomial runtime bound on the convergence to a global minimum in the concrete setting of two-layer neural networks (without modifying gradient descent). For example, the works of Chizat and Bach [21] and Mei et al. [56] do not provide a concrete bound on the number of iterations needed for convergence. Mei et al. [57] provide a coupling between the trajectories of finite and infinite-width networks with exponential growth of coupling error but do not apply it to a concrete setting to obtain a global convergence result with sample complexity guarantees. (See more discussion below and in Section 3.) Wei et al. [76] achieve a polynomial number of iterations but require exponential width and an artificially added noise. In other words, these works, without modifying gradient descent, cannot prove convergence to a global minimizer with polynomial width and iterations. We also discuss additional related works in Appendix A.

In this paper, we provide a mean-field analysis of projected gradient flow on two-layer neural networks with *polynomial width* and quartic activations. Under a simple data distribution, we demonstrate that the network converges to a non-trivial error in *polynomial time* with *polynomial samples*. Notably, our results show a sample complexity that is superior to the NTK approach. As a corollary, we show that projected gradient descent with polynomially small step size and polynomially many iterations can converge to a low error, with sample complexity that is superior to NTK. Our proof is similar to the standard bound on the error of Euler's method.

Concretely, the neural network is assumed to have unit-norm weight vectors and no bias, and the second-layer weights are all $\frac{1}{m}$ where $m$ is the width of the neural network. The data distribution is uniform over the sphere. The target function is of the form $y(x) = h(q_\star^\top x)$ where $h$ is an *unknown* quartic link function and $q_\star$ is an unknown unit vector. Our main result (Theorem 3.4) states that with $n = O(d^{3.1})$ samples, a polynomial-width neural network with random initialization converges in polynomial time to a non-trivial error, which is statistically not achievable by any kernel method with an inner product kernel using $n \ll d^4$ samples. To the best of our knowledge, our result is the first to demonstrate the advantage of *unmodified* gradient descent on neural networks over kernel methods.

The rank-one structure in the target function, also known as the single-index model, has been well-studied in the context of neural networks as a simplified case to demonstrate that neural networks can learn a latent feature direction $q_\star$ better than kernel methods [58, 2, 23, 16]. Many works on single-index models study (stochastic) gradient descent in the setting where only a single vector (or "neuron") in $\mathbb{R}^d$ is trained. This includes earlier works where the link function is monotonic, or the convergence is analyzed with quasi-convexity [46, 40, 55, 64], along with more recent work [69, 72, 11] for more general link functions. Since these works only train a single neuron, they have limited expressivity in comparison to a neural network, and can only achieve zero loss when the link function equals the activation. We stress that in our setting, the link function $h$ is unknown, not necessarily monotonic, and does not need to be equal to the activation function. We show that in this setting, the first layer weights will converge to a *distribution* of neurons that are correlated with but not exactly equal to $q_\star$, so that even without bias terms, their mixture can represent the link function $h$. Our analysis demonstrates that gradient flow, and gradient descent with a consistent inverse-polynomial learning rate, can *simultaneously* learn the feature $q_\star$ and the link function $h$, which is a key challenge that is side-stepped in previous works on neural-networks which use two-stage algorithms [58, 1, 2, 23, 14].

The main novelty of our population dynamics analysis is designing a potential function that shows that the iterate stays away from the saddle points.

Our sample complexity results leverage a coupling between the dynamics on the empirical loss for a finite-width neural network and the dynamics on the population loss for an infinite-width neural network. The main challenge stems from the fact that some exponential coupling error growth is inevitable over a certain period of time when the dynamics resemble a power method update. Heavily inspired by Li et al. [52], we address this challenge by using a direct and sharp comparison between the growth of the coupling error and the growth of the signal. In contrast, a simple exponential growth bound on the coupling error similar to the bound of Mei and Montanari [54] would result in a $d^{O(1)}$ sample complexity, which is not sufficient to outperform NTK.

## 2 Preliminaries and Notations

We use $O(\cdot), \lesssim, \gtrsim$ to hide only absolute constants. Formally, every occurrence of $O(x)$ in this paper can be simultaneously replaced by a function $f(x)$ where $|f(x)| \leq C|x|, \forall x \in \mathbb{R}$ for some universal constant $C > 0$ (each occurrence can have a different universal constant $C$ and $f$). We use $a \lesssim b$ as a shorthand for $a \leq O(b)$. Similarly, $\Omega(x)$ is a placeholder for some $g(x)$ where $|g(x)| \geq |x|/C, \forall x \in \mathbb{R}$ for some universal constant $C > 0$. We use $a \gtrsim b$ as a shorthand for $a \geq \Omega(b)$ and $a \asymp b$ as a shorthand to indicate that $a \gtrsim b$ and $a \lesssim b$ simultaneously hold.

**Legendre Polynomials.** We summarize the necessary facts about Legendre polynomials below, and present related background more comprehensively in Appendix C. Let $P_{k,d} : [-1, 1] \to \mathbb{R}$ be the degree-$k$ *un-normalized* Legendre polynomial [12], and $\overline{P}_{k,d}(t) = \sqrt{N_{k,d}} P_{k,d}(t)$ be the *normalized* Legendre polynomial, where $N_{k,d} \triangleq \binom{d+k-1}{d-1} - \binom{d+k-3}{d-1}$ is the normalizing factor. The polynomials $\overline{P}_{k,d}(t)$ form an orthonormal basis for the set of square-integrable functions over $[-1, 1]$ with respect to the measure $\mu_d(t) \triangleq (1 - t^2)^{\frac{d-3}{2}} \frac{\Gamma(d/2)}{\Gamma((d-1)/2)} \frac{1}{\sqrt{\pi}}$, i.e., the density of $u_1$ when $u = (u_1, \cdots, u_d)$ is uniformly drawn from sphere $\mathbb{S}^{d-1}$. Hence, for every function $h : [-1, 1] \to \mathbb{R}$ such that $\mathbb{E}_{t\sim\mu_d}[h(t)^2] < \infty$, we can define $\hat{h}_{k,d} \triangleq \mathbb{E}_{t\sim\mu_d}[h(t)\overline{P}_{k,d}(t)]$ and consequently, we have $h(t) = \sum_{k=0}^{\infty} \hat{h}_{k,d}\overline{P}_{k,d}(t)$.

## 3 Main Results

We will formally define the data distribution, neural networks, projected gradient flow, and assumptions on the problem-dependent quantities and then state our main theorems.

*Target function.* The ground-truth function $y(x) : \mathbb{R}^d \to \mathbb{R}$ that we aim to learn has the form $y(x) = h(q_\star^\top x)$, where $h : \mathbb{R} \to \mathbb{R}$ is an *unknown* one-dimensional *even* quartic polynomial (which is called a link function), and $q_\star$ is an *unknown* unit vector in $\mathbb{R}^d$. Note that $h(s)$ has the Legendre expansion $h(s) = \hat{h}_{0,d} + \hat{h}_{2,d}\overline{P}_{2,d}(s) + \hat{h}_{4,d}\overline{P}_{4,d}(s)$.

*Two-layer neural networks.* We consider a two-layer neural network where the first-layer weights are all unit vectors and the second-layer weights are fixed and all the same. Let $\sigma(\cdot)$ be the activation function, which can be different from $h(\cdot)$. Using the mean-field formulation, we describe the neural network using the distribution of first-layer weight vectors, denoted by $\rho$:

$$f_\rho(x) \triangleq \mathbb{E}_{u\sim\rho}[\sigma(u^\top x)]. \tag{3.1}$$

For example, when $\rho = \text{unif}(\{u_1, \ldots, u_m\})$, is a discrete, uniform distribution supported on $m$ vectors $\{u_1, \ldots, u_m\}$, then $f_\rho(x) = \frac{1}{m}\sum_{i=1}^{m} \sigma(u_i^\top x)$, i.e. $f_\rho$ corresponds to a finite-width neural network whose first-layer weights are $u_1, \ldots, u_m$. For a continuous distribution $\rho$, the function $f_\rho(\cdot)$ can be viewed as an infinite-width neural network where the weight vectors are distributed according to $\rho$ (and can be viewed as taking the limit as $m \to \infty$ of the finite-width neural network). We assume that the weight vectors have unit norms, i.e. the support of $\rho$ is contained in $\mathbb{S}^{d-1}$. The activation $\sigma : \mathbb{R} \to \mathbb{R}$ is assumed to be a fourth-degree polynomial with Legendre expansion $\sigma(s) = \sum_{k=0}^{4} \hat{\sigma}_{k,d}\overline{P}_{k,d}(s)$.

The simplified neural network defined in Eq. (3.1), even with infinite width (corresponding to a continuous distribution $\rho$), has a limited expressivity due to the lack of biases and trainable second-layer weights. We characterize the expressivity by the following lemma:

**Lemma 3.1** (Expressivity). *Let $\gamma_2 = \hat{h}_{2,d}/\hat{\sigma}_{2,d}$ and $\gamma_4 = \hat{h}_{4,d}/\hat{\sigma}_{4,d}$. Suppose for some $d$ and $q_\star$, there exists a network $\rho$ such that $f_\rho(x) = h(q_\star^\top x)$ on $\mathbb{S}^{d-1}$. Then we have $\hat{\sigma}_{0,d} = \hat{h}_{0,d}$, and $0 \leq \gamma_2^2 \leq \gamma_4 \leq \gamma_2 \leq 1$. Moreover, if this condition holds with strict inequalities, then for sufficiently large $d$, there exists a network $\rho$ such that $f_\rho(x) = h(q_\star^\top x)$ on $\mathbb{S}^{d-1}$. (A more explicit version is stated in Appendix G.1.)*

Informally, an almost sufficient and necessary condition to have $f_\rho(x) = h(q_\star^\top x)$ for some $\rho$ is that there exists a random variable $w$ supported on $[0, 1]$ such that $\mathbb{E}[w^2] \approx \gamma_2$ and $\mathbb{E}[w^4] \approx \gamma_4$, which is equivalent to $0 \leq \gamma_2^2 \leq \gamma_4 \leq \gamma_2 \leq 1$. In particular, assuming the existence of such a random variable $w$, the perfectly-fit network $\rho$ that fits the target function has the form

$$q_\star^\top u \stackrel{d}{=} w, \text{ and } u - q_\star q_\star^\top u \mid q_\star^\top u \text{ is uniformly distributed in the subspace orthogonal to } q_\star. \quad (3.2)$$

Motivated by this lemma, we will assume that $\gamma_2, \gamma_4$ are universal constants that satisfy $0 \leq \gamma_2^2 \leq \gamma_4 \leq \gamma_2 \leq 1$, and $d$ is chosen to be sufficiently large (depending on the choice of $\gamma_2$ and $\gamma_4$). In addition, we also assume that $\gamma_4 \leq O(\gamma_2^2)$, that is, the equality $\gamma_2^2 \leq \gamma_4$ is somewhat tight — this ensures that the distribution of $w = q_\star^\top u$ under the perfectly-fitted neural network is not too spread-out. We also assume that $\gamma_2$ is smaller than a sufficiently small universal constant. This ensures that the distribution of $q_\star^\top u$ under the perfectly-fitted network does not concentrate on 1, i.e. the distribution of $u$ is not merely a point mass around $q_\star$. In other words, this assumption restricts our setting to the most interesting case where the landscape has bad saddle points (and thus is fundamentally more challenging to analyze). We also assume for simplicity that $\hat{\sigma}_{0,d} = \hat{h}_{0,d}$ (because otherwise, the activation introduces a constant bias that prohibits perfect fitting), even though adding a trainable scalar to the neural network formulation can remove the assumption. If $\hat{\sigma}_{0,d} = \hat{h}_{0,d}$, then we can assume without loss of generality that $\hat{\sigma}_{0,d} = \hat{h}_{0,d} = 0$, since $\hat{\sigma}_{0,d}$ and $\hat{h}_{0,d}$ will cancel with each other in the population and empirical mean-squared losses (defined below). In summary, we make the following formal assumptions on the Legendre coefficients of the link function and the activation function.

**Assumption 3.2.** *Let $\gamma_2 = \hat{h}_{2,d}/\hat{\sigma}_{2,d}$ and $\gamma_4 = \hat{h}_{4,d}/\hat{\sigma}_{4,d}$. We first assume $\gamma_4 \geq 1.1 \cdot \gamma_2^2$. For any universal constant $c_1 > 1$, we assume that $\hat{\sigma}_{2,d}^2/c_1 \leq \hat{\sigma}_{4,d}^2 \leq c_1 \cdot \hat{\sigma}_{2,d}^2$, and $\gamma_4 \leq c_1 \gamma_2^2$. For a sufficiently small universal constant $c_2 > 0$ (which is chosen after $c_1$ is determined), we assume $0 \leq \gamma_2 \leq c_2$. We also assume that $d$ is larger than a sufficiently large constant $c_3$ (which is chosen after $c_1$ and $c_2$.) We also assume $\hat{h}_{0,d} = \hat{\sigma}_{0,d} = 0$, and $\hat{h}_{1,d} = \hat{h}_{3,d} = 0$.*

Our intention is to replicate the ReLU activation as well as possible with quartic polynomials; our assumption that $\hat{\sigma}_{2,d}^2 \asymp \hat{\sigma}_{4,d}^2$ is indeed satisfied by the quartic expansion of ReLU because $\widehat{\text{relu}}_{2,d} \asymp d^{-1/2}$ and $\widehat{\text{relu}}_{2,d} \asymp d^{-1/2}$ (see Proposition C.3). Following the convention defined in Section 2, we will simply write $\hat{\sigma}_{4,d}^2 \asymp \hat{\sigma}_{2,d}^2$, $\gamma_4 \geq 1.1 \cdot \gamma_2^2$, and $\gamma_4 \lesssim \gamma_2^2$.

Our assumptions rule out the case that $\gamma_4 = \gamma_2^2$, for the sake of simplicity. We believe that our analysis could be extended to this case with some modifications. Our analysis also rules out the case where $\gamma_2 = 0$ and $\gamma_4 \neq 0$, due to the restriction that $\gamma_4 \leq c_1 \gamma_2^2$. The case $\gamma_2 = 0$ and $\gamma_4 \neq 0$ would have a significantly different analysis, and potentially a different sample complexity, since our analysis in Section 4.2 and Section 5 makes use of the fact that the initial phase of the population and empirical dynamics behaves similarly to a power method, which follows from $\gamma_2$ being nonzero.

*Data distribution, losses, and projected gradient flow.* The population data distribution is assumed to be $\mathbb{S}^{d-1}$. We draw $n$ training examples $x_1, \ldots, x_n \stackrel{\text{i.i.d}}{\sim} \mathbb{S}^{d-1}$. Thus, the population and empirical mean-squared losses are:

$$L(\rho) = \frac{1}{2} \cdot \mathbb{E}_{x \sim \mathbb{S}^{d-1}} \left( f_\rho(x) - y(x) \right)^2, \quad \text{and} \quad \widehat{L}(\rho) = \frac{1}{2n} \sum_{i=1}^{n} \left( f_\rho(x_i) - y(x_i) \right)^2. \quad (3.3)$$

To ensure that the weight vectors remain on $\mathbb{S}^{d-1}$, we perform projected gradient flow on the empirical loss. We start by defining the gradient of the population loss $L$ with respect to a particle $u$ at $\rho$ and

the corresponding Riemannian gradient (which is simply the projection of the gradient to the tangent space of $\mathbb{S}^{d-1}$):

$$\nabla_u L(\rho) = \mathop{\mathbb{E}}_{x \sim \mathbb{S}^{d-1}} \left[ (f_\rho(x) - y(x))\sigma'(u^\top x)x \right] , \quad \text{and} \quad \text{grad}_u L(\rho) = (I - uu^\top)\nabla_u L(\rho) . \quad (3.4)$$

Here we interpret $L(\rho)$ as a function of a collection of particles (denoted by $\rho$) and $\nabla_u L(\rho)$ as the partial derivative with respect to a single particle $u$ evaluated at $\rho$. Similarly, the (Riemannian) gradient of the empirical loss $\widehat{L}$ with respect to the particle $u$ is defined as

$$\nabla_u \widehat{L}(\rho) = \frac{1}{n}\sum_{i=1}^{n}(f_\rho(x_i) - y(x_i))\sigma'(u^\top x_i)x_i , \quad \text{and} \quad \text{grad}_u \widehat{L}(\rho) = (I - uu^\top)\nabla_u \widehat{L}(\rho) . \quad (3.5)$$

**Population, Infinite-Width Dynamics.** Let the initial distribution $\rho_0$ of the infinite-width neural network be the uniform distribution over $\mathbb{S}^{d-1}$. We use $\chi$ to denote a particle sampled uniformly at random from the initial distribution $\rho_0$. A particle initialized at $\chi$ follows a deterministic trajectory afterwards — we use $u_t(\chi)$ to denote the location, at time $t$, of the particle that was initialized at $\chi$. Because $u_t(\cdot)$ is a deterministic function, we can use $\chi$ to index the particles at any time based on their initialization. The projected gradient flow on an infinite-width neural network and using population loss $L$ can be described as

$$\forall \chi \in \mathbb{S}^{d-1}, u_0(\chi) = \chi , \quad (3.6)$$

$$\frac{du_t(\chi)}{dt} = -\text{grad}_{u_t} L(\rho_t) , \quad (3.7)$$

$$\text{and } \rho_t = \text{distribution of } u_t(\chi) \text{ (where } \chi \sim \rho_0) . \quad (3.8)$$

**Empirical, Finite-Width Dynamics.** The training dynamics of a neural network with width $m$ can be described in this language by setting the initial distribution to be a discrete distribution uniformly supported on $m$ initial weight vectors. The update rule will maintain that at any time, the distribution of neurons is uniformly supported over $m$ items and thus still corresponds to a width-$m$ neural network. Let $\chi_1, \ldots, \chi_m \overset{\text{i.i.d}}{\sim} \mathbb{S}^{d-1}$ be the initial weight vectors of the width-$m$ neural network, and let $\hat\rho_0 = \text{unif}(\{\chi_1, \ldots, \chi_m\})$ be the uniform distribution over these initial neurons. We use $\chi \in \{\chi_1, \ldots, \chi_m\}$ to index neurons and denote a single initial neuron as $\hat u_0(\chi) = \chi$. Then, we can describe the projected gradient flow on the empirical loss $\widehat{L}$ with initialization $\{\chi_1, \ldots, \chi_m\}$ by:

$$\frac{d\hat u_t(\chi)}{dt} = -\text{grad}_{\hat u_t(\chi)} \widehat{L}(\hat\rho_t) , \quad (3.9)$$

$$\text{and } \hat\rho_t = \text{distribution of } \hat u_t(\chi) \text{ (where } \chi \sim \hat\rho_0) . \quad (3.10)$$

We first state our result on the population, infinite-width dynamics.

**Theorem 3.3** (Population, infinite-width dynamics). *Suppose Assumption 3.2 holds, and let $\epsilon \in (0, 1)$ be the target error. Let $\rho_t$ be the result of projected gradient flow on the population loss, initialized with the uniform distribution on $\mathbb{S}^{d-1}$, as defined in Eq. (3.7). Let $T_{*,\epsilon} = \inf\{t > 0 \mid L(\rho_t) \leq \frac{1}{2}(\hat\sigma_{2,d}^2 + \hat\sigma_{4,d}^2)\epsilon^2\}$ be the earliest time $t$ such that a loss of at most $\frac{1}{2}(\hat\sigma_{2,d}^2 + \hat\sigma_{4,d}^2)\epsilon^2$ is reached. Then, we have*

$$T_{*,\epsilon} \lesssim \frac{1}{\hat\sigma_{2,d}^2 \gamma_2} \log d + \frac{(\log\log d)^{20}}{\hat\sigma_{2,d}^2 \epsilon \gamma_2^8} \log\left(\frac{\gamma_2}{\epsilon}\right) . \quad (3.11)$$

The proof is given in Appendix D. The first term on the right-hand side of Eq. (3.11) corresponds to the burn-in time for the network to reach a region around where the Polyak-Lojasiewicz condition holds. We divide our analysis of this burn-in phase into two phases, Phase 1 and Phase 2, and we obtain tight control on the factor by which the signal component, $q_\star^\top u$, grows during Phase 1, while Phase 2 takes place for a comparatively short period of time. (This tight control is critical for our sample complexity bounds where we must show that the coupling error does not blow up too much — see more discussion below Lemma 5.4.) Phases 1 and 2 are mostly governed by the quadratic components in the activation and target functions, and the dynamics behave similarly to a power method update. After the burn-in phase, the dynamics operate for a short period of time (Phase 3) in a regime where the Polyak-Lojasiewicz condition holds. We explicitly prove the dynamics stay away from saddle points during this phase, as further discussed in Section 4.2.

We note that Theorem 3.3 provides a concrete polynomial runtime bound for projected gradient flow which is not achievable by prior mean-field analyses [21, 56, 2] using Wasserstein gradient flow techniques. For instance, while the population dynamics of Abbe et al. [2] only trains the hidden layer for $O(1)$ time, our projected gradient flow updates the hidden layer for $O(\log d)$ time. The main challenge in the proof is to deal with the saddle points that are not strict-saddle [32] in the loss landscape which cannot be escaped simply by adding noise in the parameter space [45, 49, 24].[3] Our analysis develops a fine-grained analysis of the dynamics that shows the iterates stay away from saddle points, which allows us to obtain the running time bound in Theorem 3.3 which can be translated to a polynomial-width guarantee in Theorem 3.4. In contrast, Wei et al. [76] escape the saddles by randomly replacing an exponentially small fraction of neurons, which makes the network require exponential width.

Next, we state the main theorem on projected gradient flow on empirical, finite-width dynamics.

**Theorem 3.4** (Empirical, finite-width dynamics). *Suppose Assumption 3.2 holds. Suppose $\epsilon = \frac{1}{\log \log d}$ is the target error and $T_{*,\epsilon}$ is the running time defined in Theorem 3.3. Suppose $n \geq d^\mu (\log d)^{\Omega(1)}$ for any constant $\mu > 3$, and the network width $m$ satisfies $m \gtrsim d^{2.5+\mu/2}(\log d)^{\Omega(1)}$, and $m \leq d^C$ for some sufficiently large universal constant $C$. Let $\hat{\rho}_t$ be the projected gradient flow on the empirical loss, initialized with $m$ uniformly sampled weights, defined in Eq. (3.10). Then, with probability at least $1 - \frac{1}{d^{\Omega(\log d)}}$ over the randomness of the data and the initialization of the finite-width network, we have that $L(\hat{\rho}_{T_{*,\epsilon}}) \lesssim (\hat{\sigma}_{2,d}^2 + \hat{\sigma}_{4,d}^2)\epsilon^2$.*

Plugging in $\mu = 3.1$, Theorem 3.4 suggests that, when the network width is at least $d^{4.05}$ (up to logarithmic factors), the empirical dynamics of gradient descent could achieve $\hat{\sigma}_{\max}^2 (1/\log \log d)^2$ population loss with $d^{3.1}$ samples (up to logarithmic factors).

In our analysis, we will establish a coupling between neurons in the empirical dynamics and neurons in the population dynamics. The main challenge is to bound the coupling error during Phase 1 where the population dynamics are similar to the power method. During this phase, we show that the coupling error (i.e., the distance between coupled neurons) remains small by showing that it can be upper bounded in terms of the growth of the signal $q_\star^T u$ in the population dynamics. Such a delicate relationship between the growth of the error and that of the signal is the main challenge to proving a sample complexity bound better than NTK. Even an additional constant factor in the growth rate of the coupling error would lead to an additional poly($d$) factor in the sample complexity. Prior work (e.g. Mei et al. [57], Abbe et al. [2]) also establishes a coupling between the population and empirical dynamics. However, these works obtain bounds on the coupling error which are exponential in the running time — specifically, their bounds have a factor of $e^{KT}$ where $K$ is a universal constant and $T$ is the time. In many settings, including ours, $\Omega(\log d)$ time is necessary to achieve a non-trivial error, (as we further discuss in Section 4.2) and thus this would lead to a $d^{O(1)}$ bound on the sample complexity, where $O(1)$ is a unspecified and likely loose constant, which cannot outperform NTK. Our work addresses this challenge by comparing the growth of the coupling error with the growth of the signal, which enables us to control the constant factor in the growth rate of the coupling error.

**Projected Gradient Descent with $1/\mathbf{poly}(d)$ Step Size.** As a corollary of Theorem 3.4, we show that projected gradient descent with a small learning rate can also achieve low population loss in poly($d$) iterations. We first define the dynamics of projected gradient descent as follows. As before, we let $\chi_1, \ldots, \chi_m \overset{\text{i.i.d}}{\sim} \mathbb{S}^{d-1}$ be the initial weight vectors, and we let $\tilde{\rho} = \text{unif}(\{\chi_1, \ldots, \chi_m\})$ denote the uniform distribution over these vectors. Thus, we can write a single neuron at initialization as $\tilde{u}_0(\chi) = \chi$ for $\chi \in \{\chi_1, \ldots, \chi_m\}$. The dynamics of projected gradient descent, on a finite-width neural network and using the empirical loss $\widehat{L}$, can be then described as

$$\tilde{u}_{t+1}(\chi) = \frac{\tilde{u}_t(\chi) - \eta \cdot \text{grad}_{\tilde{u}_t(\chi)} \widehat{L}(\tilde{\rho}_t)}{\|\tilde{u}_t(\chi) - \eta \cdot \text{grad}_{\tilde{u}_t(\chi)} \widehat{L}(\tilde{\rho}_t)\|_2}, \tag{3.12}$$

$$\text{and } \tilde{\rho}_t = \text{distribution of } \tilde{u}_t(\chi) \text{ (where } \chi \sim \tilde{\rho}_0). \tag{3.13}$$

where $\eta > 0$ is the learning rate.[4] We show that projected gradient descent with a $1/\text{poly}(d)$ learning rate can achieve a low population loss in poly($d$) iterations:

---

[3]There are a long list of prior works on the loss landscape of neural networks [55, 64, 72, 46, 16, 15, 11, 66, 71, 33, 80, 18].

[4]See Chapter 3, page 20 of Boumal [17], accessed on August 21, 2023.

**Theorem 3.5** (Projected Gradient Descent). *Suppose we are in the setting of Theorem 3.4. Let $\tilde{\rho}_t$ be the discrete-time projected gradient descent on the empirical loss, initialized with $m$ weight vectors sampled uniformly from $\mathbb{S}^{d-1}$, defined in Eq. (3.13). Finally, assume that $\eta \leq \frac{1}{(\hat{\sigma}_{2,d}^2 + \hat{\sigma}_{4,d}^2) d^B}$ for a sufficiently large universal constant $B$. Then, $L(\tilde{\rho}_{T_{*,\epsilon}/\eta}) \lesssim (\hat{\sigma}_{2,d}^2 + \hat{\sigma}_{4,d}^2)\epsilon^2$.*

Theorem 3.5 follows from Theorem 3.4 together with a standard inductive argument to bound the discretization error. The full proof is in Appendix F.

The following theorem states that in the setting of Theorem 3.4, kernel methods with any inner product kernel require $\Omega(d^4(\ln d)^{-6})$ samples to achieve a non-trivial population loss. (Note that the zero function has loss $\mathbb{E}_{x \sim \mathbb{S}^{d-1}}[y(x)^2] \geq (\hat{h}_{4,d})^2$.)

**Theorem 3.6** (Sample complexity lower bound for kernel methods). *Let $K$ be an inner product kernel. Suppose $d$ is larger than a universal constant and $n \lesssim d^4(\ln d)^{-6}$. Then, with probability at least $1/2$ over the randomness of $n$ i.i.d data points $\{x_i\}_{i=1}^n$ drawn from $\mathbb{S}^{d-1}$, any estimator of $y$ of the form $f(x) = \sum_{i=1}^n \beta_i K(x_i, x)$ of $\{x_i\}_{i=1}^n$ must have a large error: $\mathbb{E}_{x \sim \mathbb{S}^{d-1}}(y(x) - f(x))^2 \geq \frac{3}{4}(\hat{h}_{4,d})^2$.*

Theorem 3.4 and Theorem 3.6 together prove a clear sample complexity separation between gradient flow and NTK. When $\hat{h}_{4,d} \asymp \hat{\sigma}_{\max}$ (i.e., $\gamma_2, \gamma_4 \asymp 1$), gradient flow with finite-width neural networks can achieve $(\hat{h}_{4,d})^2 (\log \log d)^{-2}$ population error with $d^{3.1}$ samples, while kernel methods with any inner product kernel (including NTK) must have an error at least $(\hat{h}_{4,d})^2/2$ with $d^{3.9}$ samples.

On a high level, Abbe et al. [2, 3] prove similar lower bounds in a different setting where the target function is drawn randomly and the data points can be arbitrary (also see Kamath et al. [47], Hsu et al. [42], Hsu [41]). In comparison, our lower bound works for a fixed target function by exploiting the randomness of the data points. In fact, we can strengthen Theorem 3.6 by proving a $\Omega(\hat{h}_{k,d}^2)$ loss lower bound for any universal constant $k \geq 0$ (Theorem H.2). We defer the proof of Theorem 3.6 to Appendix H.

# 4 Analysis of Population Dynamics

## 4.1 Symmetry of the Population Dynamics

A key observation is that due to the symmetry in the data, the population dynamics $\rho_t$ has a symmetric distribution in the subspace orthogonal to the vector $q_\star$. As a result, the dynamics $\rho_t$ can be precisely characterized by the dynamics in the direction of $q_\star$.

Recall that $w = q_\star^\top u$ denotes the projection of a weight vector $u$ in the direction of $q_\star$. Let $z = (I - q_\star q_\star^\top)u$ be the remaining component. For notational convenience, without loss of generality, we can assume that $\boldsymbol{q_\star} = \boldsymbol{e_1}$ and write $u = (w, z)$, where $w \in [-1, 1]$ and $z \in \sqrt{1 - w^2} \cdot \mathbb{S}^{d-2}$. *We will use this convention throughout the rest of the paper.* We will use $w_t(\chi)$ to refer to the first coordinate of the particle $u_t(\chi)$, and $z_t(\chi)$ to refer to the last $(d-1)$ coordinates.

**Definition 4.1** (Rotational invariance and symmetry). *We say a neural network $\rho$ is rotationally invariant if for $u = (w, z) \sim \rho$, the distribution of $z \mid w$ is uniform over $\sqrt{1 - w^2}\mathbb{S}^{d-2}$ almost surely. We also say $\rho$ is symmetric w.r.t a variable $w$ if the density of $w$ is an even function.*

We note that any polynomial-width neural network (e.g., $\hat{\rho}$) is very far from rotationally invariant, and therefore the definition is specifically used for population dynamics. If $\rho$ is rotationally invariant and symmetric, then $L(\rho)$ has a simpler form that only depends on the marginal distribution of $w$.

**Lemma 4.2.** *Let $\rho$ be a rotationally invariant neural network. Then, for any target function $h$ and any activation $\sigma$,*

$$L(\rho) = \frac{1}{2} \sum_{k=0}^\infty \left( \hat{\sigma}_{k,d} \mathbb{E}_{u \sim \rho}[P_{k,d}(w)] - \hat{h}_{k,d} \right)^2. \tag{4.1}$$

*In addition, suppose $h$ and $\sigma$ satisfy Assumption 3.2 and $\rho$ is symmetric. Then*

$$L(\rho) = \frac{\hat{\sigma}_{2,d}^2}{2} \left( \mathbb{E}_{u \sim \rho}[P_{2,d}(w)] - \gamma_2 \right)^2 + \frac{\hat{\sigma}_{4,d}^2}{2} \left( \mathbb{E}_{u \sim \rho}[P_{4,d}(w)] - \gamma_4 \right)^2. \tag{4.2}$$

The proof of Lemma 4.2 is deferred to Appendix D.1. Eq. (4.1) says that if $\rho$ is rotationally invariant, then $L(\rho)$ only depends on the marginal distribution of $w$. Eq. (4.2) says that if the distribution of $w$ is additionally symmetric and $\sigma$ and $h$ are quartic, then the terms corresponding to odd $k$ and the higher order terms for $k > 4$ in Eq. (4.1) vanish. Note that $P_{2,d}(s) \approx s^2$ and $P_{4,d}(s) \approx s^4$ by Eq. (C.4). Thus, $L(\rho)$ essentially corresponds to matching the second and fourth moments of $w$ to some desired values $\gamma_2$ and $\gamma_4$. Inspired by this lemma, we define the following key quantities: for any time $t \geq 0$, we define $D_{2,t} = \mathbb{E}_{u \sim \rho_t}[P_{2,d}(w)] - \gamma_2$ and $D_{4,t} = \mathbb{E}_{u \sim \rho_t}[P_{4,d}(w)] - \gamma_4$, where $\rho_t$ is defined according to the population, infinite-width dynamics (Eq. (3.6), Eq. (3.7) and Eq. (3.8)).

We next show that the rotational invariance and symmetry properties of $\rho_t$ are indeed maintained:

**Lemma 4.3.** *Suppose we are in the setting of Theorem 3.3. At any time $t \in [0, \infty)$, $\rho_t$ is symmetric and rotationally invariant.*

Rotational invariance follows from the rotational invariance of the data. To show symmetry, we use Eq. (4.1), and the facts that $P_{k,d}$ is an odd polynomial for odd $k$ and $\rho_t$ is symmetric at initialization. Using Lemma 4.2 and Lemma 4.3, we obtain a simple formula for the dynamics of $w_t$:

**Lemma 4.4** (1-dimensional dynamics). *Suppose we are in the setting of Theorem 3.3. Then, for any $\chi \in \mathbb{S}^{d-1}$, writing $w_t := w_t(\chi)$, we have*

$$\frac{dw_t}{dt} = \underbrace{-(1 - w_t^2) \cdot (P_t(w_t) + Q_t(w_t))}_{\triangleq v(w_t)},\tag{4.3}$$

*where for any $w \in [-1, 1]$, we have $P_t(w) = 2\hat{\sigma}_{2,d}^2 D_{2,t} w + 4\hat{\sigma}_{4,d}^2 D_{4,t} w^3$, and $Q_t(w) = \lambda_d^{(1)} w + \lambda_d^{(3)} w^3$, where $|\lambda_d^{(1)}|, |\lambda_d^{(3)}| \lesssim \frac{\hat{\sigma}_{2,d}^2 |D_{2,t}| + \hat{\sigma}_{4,d}^2 |D_{4,t}|}{d}$. More specifically, $\lambda_d^{(1)} = 2\hat{\sigma}_{2,d}^2 D_{2,t} \cdot \frac{1}{d-1} - 2\hat{\sigma}_{4,d}^2 D_{4,t} \cdot \frac{6d+12}{d^2-1}$ and $\lambda_d^{(3)} = 4\hat{\sigma}_{4,d}^2 D_{4,t} \cdot \frac{6d+9}{d^2-1}$.*

Eq. (4.3) is a properly defined dynamics for $w$ because the update rule for $w_t$ only depends on $w_t$ and the quantities $D_{2,t}$ and $D_{4,t}$ — additionally, $D_{2,t}$ and $D_{4,t}$ only depend on the distribution of $w$. We henceforth refer to the $w_t(\chi)$ as particles — this is well-defined by Lemma 4.4.

## 4.2 Analysis of One-Dimensional Population Dynamics

We also use $\iota \in [-1, 1]$ to refer to the first coordinate of $\chi$, the initialization of a particle under the population dynamics. We note that $\iota = \langle \chi, e_1 \rangle$ and therefore, the distribution of $\iota$ is $\mu_d$. For any time $t \geq 0$, we use $w_t(\iota)$ to refer to $w_t(\chi)$ for any $\chi \in \mathbb{S}^{d-1}$. This notation is also well-defined by Lemma 4.4. We divide our proof of Theorem 3.3 into three phases, which are defined as follows. For ease of presentation, we will only consider particles $w_t(\iota)$ for $\iota > 0$ in the following discussion. Our argument also applies for $\iota < 0$ by the symmetry of $\rho_t$ (Lemma 4.3).

**Definition 4.5** (Phase 1). *Let $w_{\max} = \frac{1}{\log d}$ and $\iota_U = \frac{\log d}{\sqrt{d}}$. Let $T_1 > 0$ be the minimum time such that $w_{T_1}(\iota_U) = w_{\max} := \frac{1}{\log d}$. We refer to the time interval $[0, T_1]$ as Phase 1.*

Note that essentially all of the particles are less than $\iota_U = \frac{\log d}{\sqrt{d}}$ at initialization by tail bounds for $\mu_d$. During Phase 1, the term corresponding to $D_{2,t}$ in the velocity dominates, and all particles grow by a $\frac{\sqrt{d}}{(\log d)^2}$ factor. During Phase 1, the loss does not decrease much, but a large portion of the particles have grown by a large factor. During Phase 2, the particles and their velocity will become large.

**Definition 4.6** (Phase 2). *Let $T_2 > 0$ be the minimum time such that either $D_{2,T_2} = 0$ or $D_{4,T_2} = 0$. We refer to the time interval $[T_1, T_2]$ as Phase 2. Note that $T_2 > T_1$ by Lemma D.8.*

**Definition 4.7** (Phase 3). *Phase 3 is defined as the time interval $[T_2, \infty)$.*

We divide this phase into two cases, which are both more challenging to analyze than Phases 1 and 2: (i) $D_{2,T_2} = 0$ and $D_{4,T_2} < 0$, and (ii) $D_{2,T_2} < 0$ and $D_{4,T_2} = 0$. Here we discuss Case 1 — the analyses of Cases 1 and 2 are in Appendix D.5 and Appendix D.6 respectively. Suppose Case 1 holds, i.e. $D_{2,T_2} = 0$ and $D_{4,T_2} < 0$. Then, for all $t \geq T_2$, we will have $D_{2,t} \geq 0$ and $D_{4,t} \leq 0$ (Lemma D.19 and Lemma D.20), meaning that for $t \geq T_2$ a poly($\gamma_2$) fraction of particles will be far from 0 and 1 (Lemma D.17). However, this does not guarantee a large average velocity — unlike

in Phases 1 and 2, the velocity $v(w) = P_t(w) + Q_t(w)$ defined in Lemma 4.4 may have a positive root $r \in (0, 1)$, and the root $r$ can give rise to bad stationary points because for certain values of $r$, if $\rho$ degenerates to the singleton distribution on the root $r$, the velocity at $r$ would be exactly 0. Indeed, we can construct such an $r$, intuitively because $D_{2,t}$ and $D_{4,t}$ have different signs in $P_t(w)$ and $Q_t(w) \approx O(1/d)$ is a lower order term (Lemma D.35).

Thus, we must leverage some information about the trajectory to show that the average velocity is large when $D_{2,t}, D_{4,t}$ have different signs. To this end, we prove that $\mathbb{E}_{u \sim \rho_t}[(w - r)^2] \gtrsim \mathbb{E}_{w, w' \sim \rho_t}(w - w')^2$ is large by designing a novel potential function $\Phi(w) := \log(\frac{w}{\sqrt{1-w^2}})$. We show that $|\Phi(w) - \Phi(w')|$ is always increasing for any two particles $w, w'$ (Lemma D.13). Because $\Phi$ is Lipschitz on an interval bounded away from 0 and 1, a lower bound on $|\Phi(w) - \Phi(w')|$ also leads to a lower bound on $(w - w')^2$, and thus the particles away from 0 and 1 will have a large variance. Recall that when $D_{2,t} > 0$ and $D_{4,t} < 0$, a large portion of particles are away from 0 and 1, and hence, the average velocity is large. In Appendix B, we include simulations to illustrate the effects of Phases 1, 2 and 3.

# 5 Analysis of Empirical, Finite-Width Dynamics

**Coupling Between Empirical and Population Dynamics.** To analyze the difference between the empirical and population dynamics, we define an intermediate process $\bar{\rho}_t$, where the initial particles are from $\hat{\rho}_0$, but the dynamics of the particles then follow the population trajectories:

$$\bar{\rho}_0 = \hat{\rho}_0 = \text{unif}(\{\chi_1, \ldots, \chi_m\}),$$
$$\text{and } \bar{\rho}_t = \text{distribution of } u_t(\chi) \text{ (where } \chi \sim \bar{\rho}_0). \tag{5.1}$$

For $\chi \sim \text{unif}(\{\chi_1, \ldots, \chi_m\})$, let $\bar{u}_t(\chi) = u_t(\chi)$. Let $\Gamma_t$ be the joint distribution of $(\bar{u}_t(\chi), \hat{u}_t(\chi))$. Then, $\Gamma_t$ forms a natural coupling between $\bar{\rho}_t$ and $\hat{\rho}_t$. We will use $\bar{u}_t$ and $\hat{u}_t$ as shorthands for the random variables $\bar{u}_t(\chi), \hat{u}_t(\chi)$ respectively in the rest of this section. We define the average distance $\overline{\Delta}_t^2 := \mathbb{E}_{(\hat{u}_t, \bar{u}_t) \sim \Gamma_t}[\|\hat{u}_t - \bar{u}_t\|_2^2]$. Intuitively, $f_{\rho_t}(x)$ and $f_{\hat{\rho}_t}(x)$ are close when $\hat{u}_t$ and $\bar{u}_t$ are close, which is formalized by the following lemma:

**Lemma 5.1.** *In the setting of Theorem 3.4, let $T \leq \frac{d^{O(1)}}{\hat{\sigma}_{2,d}^2 + \hat{\sigma}_{4,d}^2}$. Then, with probability at least $1 - \exp(-d^2)$ over the initialization, we have for all $t \in [0, T]$ that $\mathbb{E}_{x \sim \mathbb{S}^{d-1}}[(f_{\rho_t}(x) - f_{\bar{\rho}_t}(x))^2] \lesssim \frac{(\hat{\sigma}_{2,d}^2 + \hat{\sigma}_{4,d}^2)d^2(\log d)^{O(1)}}{m}$ and $\mathbb{E}_{x \sim \mathbb{S}^{d-1}}[(f_{\hat{\rho}_t}(x) - f_{\bar{\rho}_t}(x))^2] \lesssim (\hat{\sigma}_{2,d}^2 + \hat{\sigma}_{4,d}^2)\overline{\Delta}_t^2$.*

The proof of Lemma 5.1 is deferred to Appendix E. As a simple corollary of Lemma 5.1, we can upper bound $\mathbb{E}_{x \sim \mathbb{S}^{d-1}}[(f_{\rho_t}(x) - f_{\hat{\rho}_t}(x))^2]$ by the triangle inequality. Thus, so long as $\overline{\Delta}_{T_{*,\epsilon}}$ is small, we can show that the empirical and population dynamics achieve similar test error at time $T_{*,\epsilon}$.

**Upper Bound for $\overline{\Delta}_{T_{*,\epsilon}}$.** The following lemma gives our upper bound on $\overline{\Delta}_{T_{*,\epsilon}}$:

**Lemma 5.2** (Final Bound on $\overline{\Delta}_{T_{*,\epsilon}}$)**.** *In the setting of Theorem 3.4, we have $\overline{\Delta}_{T_{*,\epsilon}} \leq d^{-\frac{\mu-1}{4}}$.*

We prove Lemma 5.2 by induction over the time $t \leq T_{*,\epsilon}$. Let us define the maximal distance $\Delta_{\max,t} := \max_{(\hat{u}_t, \bar{u}_t) \in \text{supp}(\Gamma_t)} \|\hat{u}_t - \bar{u}_t\|_2$ where the max is over the *finite* support of the coupling $\Gamma_t$. We will control $\overline{\Delta}_t$ and $\Delta_{\max,t}$ simultaneously using induction. The inductive hypothesis is:

**Assumption 5.3** (Inductive Hypothesis)**.** *For some universal constant $1 > \phi > 1/2$ and $\psi \in (0, \phi - \frac{1}{2})$, we say that the inductive hypothesis holds at time $T$ if, for all $0 \leq t \leq T$, we have $\overline{\Delta}_t \leq d^{-\phi}$ and $\Delta_{\max,t} \leq d^{-\psi}$.*

Showing that the inductive hypothesis holds for all $t$ requires studying the growth of the error $\|\hat{u}_t - \bar{u}_t\|$. We analyze it using the decomposition[5] $\frac{d}{dt}\|\hat{u}_t - \bar{u}_t\|_2^2 = A_t + B_t + C_t$, where $A_t := -2\langle \text{grad}_{\hat{u}} L(\rho_t) - \text{grad}_{\bar{u}} L(\rho_t), \hat{u}_t - \bar{u}_t \rangle$, $B_t := -2\langle \text{grad}_{\hat{u}} L(\hat{\rho}_t) - \text{grad}_{\hat{u}} L(\rho_t), \hat{u}_t - \bar{u}_t \rangle$, and $C_t := -2\langle \text{grad}_{\hat{u}} \widehat{L}(\hat{\rho}_t) - \text{grad}_{\hat{u}} L(\hat{\rho}_t), \hat{u}_t - \bar{u}_t \rangle$. Intuitively, $A_t$ captures the growth of the coupling error due to the population dynamics, $B_t$ the growth due to the discrepancy between the gradients from the

---

[5]Note that $\hat{u}_t - \bar{u}_t$ and $A_t, B_t$ and $C_t$ implicitly depend on the initialization $\chi$, and could be written as $A_t(\chi), B_t(\chi), C_t(\chi)$, and $\delta_t(\chi)$, but we omit this dependence for ease of presentation.

finite-width and infinite-width networks, and $C_t$ the growth due to the discrepancy between finite samples and infinite samples. The following lemma gives upper bounds on these terms:

**Lemma 5.4** (Bounds on $A_t, B_t, C_t$). *In the setting of Theorem 3.4, suppose the inductive hypothesis (Assumption 5.3) holds up to time $t$, for some $t \leq T_{*,\epsilon}$. Let $\delta_t := \hat{u}_t - \bar{u}_t$. Let $T_1$ be the runtime of Phase 1 (where Phase 1 is defined in Definition 4.5). Then, we have*

$$
A_t \leq \begin{cases} 4\hat{\sigma}_{2,d}^2 |D_{2,t}| \, \|\delta_t\|^2 + O\big(\frac{\hat{\sigma}_{2,d}^2}{\log d}|D_{2,t}| \, \|\delta_t\|^2\big) & \text{if } 0 \leq t \leq T_1 \\ O(1) \cdot (\hat{\sigma}_{2,d}^2 + \hat{\sigma}_{4,d}^2)\gamma_2 \, \|\delta_t\|^2 & \text{if } T_1 \leq t \leq T_{*,\epsilon} \end{cases}. \tag{5.2}
$$

*Additionally, with probability at least $1 - \frac{1}{d^{\Omega(\log d)}}$ over the initialization and dataset, we have*

$B_t \lesssim (\hat{\sigma}_{2,d}^2 + \hat{\sigma}_{4,d}^2) \cdot \left( \frac{d(\log d)^{O(1)}}{\sqrt{m}} + \overline{\Delta}_t \right) \|\delta_t\|_2, \; \mathbb{E}[B_t] \lesssim \frac{(\hat{\sigma}_{2,d}^2 + \hat{\sigma}_{4,d}^2)d(\log d)^{O(1)}}{\sqrt{m}} \overline{\Delta}_t + (\hat{\sigma}_{2,d}^2 + \hat{\sigma}_{4,d}^2)\overline{\Delta}_t^3$

*and* $C_t \lesssim (\hat{\sigma}_{2,d}^2 + \hat{\sigma}_{4,d}^2)(\log d)^{O(1)} \cdot \left( \sqrt{\frac{d}{n}}\|\delta_t\|_2 + \frac{d^{2-2\psi}}{n}\overline{\Delta}\|\delta_t\|_2^2 + \frac{d^{2.5-2\psi}}{n}\overline{\Delta}^2\|\delta_t\|_2 + \frac{d^{4-4\psi}}{n}\overline{\Delta}^2\|\delta_t\|_2^2 \right).$

The proof is in Appendix E. We now give an explanation for each of these bounds. For $A_t$, we establish two separate bounds, one which holds during Phase 1, and one which holds during Phases 2 and 3. Intuitively, the signal part in a neuron (i.e. $w^2$) grows at a rate $4\hat{\sigma}_{2,d}^2|D_{2,t}|$ during Phase 1 (which follows from Lemma 4.4) and in Eq. (5.2) we show that the growth rate of the coupling error due to the growth in the signal is also at most $4\hat{\sigma}_{2,d}^2|D_{2,t}|$. Intuitively, the growth rates match because the signal parts of the neurons follow similar dynamics to a power method during Phase 1. For example, in the worst case when $\delta_t$ is mostly in the direction of $e_1$, the factor by which $\delta_t$ grows is the same as that of $w$. More mathematically, this is due to the fact that the first-order term in Lemma 4.4 is the dominant term in $v(w) - v(\hat{w})$ for any two particles $w, \hat{w}$ (where $v(w)$ is the one-dimensional velocity defined in the previous section). To get a sample complexity better than NTK, the precise constant factor 4 in the growth rate $4\hat{\sigma}_{2,d}^2|D_{2,t}|w$ is important — a larger constant would lead to $\delta_{T_1}$ being larger by a poly($d$) factor, thus increasing the sample complexity by poly($d$).

The same bound for $A_t$ no longer applies during Phases 2 and 3. This is because the dynamics of $w$ are no longer similar to a power method, and the higher-order terms may make a larger contribution to $v(w) - v(\hat{w})$ than to the growth of $w$, or vice versa. Thus we use a looser bound $O(1) \cdot (\hat{\sigma}_{2,d}^2 + \hat{\sigma}_{4,d}^2)\gamma_2 \, \|\delta_t\|^2$ in Eq. (5.2). However, since the remaining running time after Phase 1, $T_{*,\epsilon} - T_1$, is very short (only poly($\log \log d$) — this corresponds to the second term of the running time in Theorem 3.3), the total growth of the coupling error is at most $\exp(\text{poly}(\log \log d))$, which is sub-polynomial and does not contribute to the sample complexity. Note that if the running time during Phases 2 and 3 is longer (e.g. $O(\log d)$) then the coupling error would grow by an additional $d^{O(1)}$ factor during Phases 2 and 3, which would make our sample complexity worse than NTK.

Our upper bounds on $B_t$ and $C_t$ capture the growth of coupling error due to the finite width and samples. We establish a stronger bound for $\mathbb{E}[B_t]$ than for $B_t$. In our proof, we use the bounds for $B_t$ and $\mathbb{E}[B_t]$ to prove the inductive hypothesis for $\overline{\Delta}_t$ and $\Delta_{\max,t}$ respectively. We prove the bound for $C_t$ by expanding the error due to finite samples into second-order and fourth-order polynomials and applying concentration inequalities for higher moments (Lemma I.7).

All of these bounds together control the growth of $\|\hat{u}_t - \bar{u}_t\|$ at time $t$. Using Lemma 5.4 we can show that for some properly chosen $\phi$ and $\psi$, the inductive hypothesis in Assumption 5.3 holds until $T_{*,\epsilon}$ (see Lemma E.3 for the statement), which naturally leads to the bound in Lemma 5.2. The full proof can be found in Appendix E.

## 6 Conclusion

In this paper, we prove a clear sample complexity separation between vanilla gradient flow and kernel methods with any inner product kernel, including NTK. Our work leads to several directions for future research. The first question is to generalize our results to the ReLU activation. Another question is whether gradient descent can achieve less than $d^3$ sample complexity in our setting. The work of Arous et al. [11] shows that if the population loss has information exponent 2, then a sample complexity of $d$ (up to logarithmic factors) can be achieved with a single-neuron student network — it is an open question if a similar sample complexity could be obtained in our setting. A final open question is whether two-layer neural networks can be shown to attain arbitrarily small generalization error $\epsilon$ — one limitation of our analysis is that we require $\epsilon \geq \text{poly}(1/\log \log d)$.

## Acknowledgments

The authors would like to thank Zhiyuan Li for helpful discussions. The authors would like to thank the support of NSF IIS 2045685, NSF Grant CCF-1844628 and a Sloan Research Fellowship.

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

# List of Appendices

# A  Additional Related Works

In addition to the related works on the NTK approach, mean-field analyses, and other works analyzing the training dynamics of neural networks that we discussed in Section 1, we also discuss the following additional related works.

There are several works which study mean-field Langevin dynamics [19, 68]. These works show "uniform-in-time propagation of chaos" estimates, i.e. their results imply bounds on the coupling error between finite-width and infinite-width trajectories which do not grow exponentially in the time $T$. However, it is likely non-trivial to apply these analyses to directly obtain good test error and sample complexity in the setting of two-layer neural networks, since mean-field Langevin dynamics may require many iterations to obtain good population loss. More concretely, Theorem 4 of Mei et al. [56] suggests that the inverse temperature $\lambda$ for mean-field Langevin dynamics has to be at least proportional to the dimension $d$ in order for Langevin dynamics to achieve low population loss. This would cause the log-Sobolev constant in Suzuki et al. [68] to be on the order of $e^{-d}$, leading to a running time of $e^{-d}$. In comparison, we show in Theorem 3.5 that projected gradient descent with $\text{poly}(d)$ iterations can achieve a low population loss. We also note that Chen et al. [19] do not study the impact of finite samples on the coupling error.

We also note that there are several works which reduce the population dynamics to a lower-dimensional dynamics [2, 37, 8], as we do in Section 4.1. The focus of our work is on the analysis of the population dynamics (Section 4.2) and coupling error between the empirical and population dynamics (Section 5), rather than the dimension-free dynamics.

Finally, we note that our setting goes beyond that of Abbe et al. [2]: our target function does not satisfy the merged-staircase property required by Abbe et al. [2], since the lowest-degree term in our target function has degree 2. The work of Abbe et al. [3] studies functions which have higher "leap complexity," meaning that terms of the target function can introduce more than one new coordinate at once, or introduce a new coordinate using a Hermite polynomial with degree greater than 1 (in the case where the inputs are Gaussian). However, Abbe et al. [3] use a non-standard algorithm which separates the coordinates based on whether they are large or small, and applying different projections to each subset of coordinates.

# B  Simulations for Population Dynamics

Here we include some simulations in a setting which is a simplified version of our 1-dimensional dynamics, to illustrate the different phases. Here, the goal is to find a weight vector $w \in \mathbb{R}^d$, where $d = 1000$, that minimizes the following loss:

$$L(w) = \left(\frac{1}{d} \sum_{i=1}^{d} w_i^2 - \gamma_2\right)^2 + \left(\frac{1}{d} \sum_{i=1}^{d} w_i^4 - \gamma_4\right)^2 \tag{B.1}$$

where $\gamma_2 = 0.8$ and $\gamma_4 = 0.7$. Intuitively, the $w_i$ play the role of the particles $w$ in the 1-dimensional dynamics of Section 4.2. We initialize $w$ as follows — each coordinate is initialized according to a standard Gaussian distribution with mean 0 and standard deviation 0.0001. We then train $w$ using gradient descent on $L(w)$ with learning rate 0.1, for $100,000$ iterations. We define $D_2 = \frac{1}{d} \sum_{i=1}^{d} w_i^2 - \gamma_2$ and $D_4 = \frac{1}{d} \sum_{i=1}^{d} w_i^4 - \gamma_4$.

The dynamics are illustrated by Figure 1, Figure 2, Figure 3 and Figure 4. At initialization, since the standard deviation is 0.0001, all of the $w_i$ are very small, as shown in Figure 1. Now, as shown in Figure 4, while both $D_2$ and $D_4$ grow rapidly after a certain point, $D_4$ reaches 0 first. At the point where $D_4$ reaches 0, the distribution of the coordinates of $w$ is shown in Figure 2. At this point, the coordinates $w_i$ have grown substantially, with a substantial fraction of coordinates around 0.5 or larger. This illustrates the intuition that all of the coordinates grow by a large factor during Phases 1 and 2. The weight histogram after many iterations have passed during Phase 3 is shown in Figure 3. In our infinite-width 1-dimensional population dynamics, when $D_4 > 0$ and $D_2 < 0$, the particles $w$ which are larger than the root $r$ of the velocity will move towards 0, while the particles $w$ which are smaller than the root $r$ of the velocity will move away from 0. This intuition is reflected in Figure 3, where not as many particles are concentrated around 0 compared to at the end of Phase 2, but the particles have not grown larger than 1 or $-1$ in absolute value.

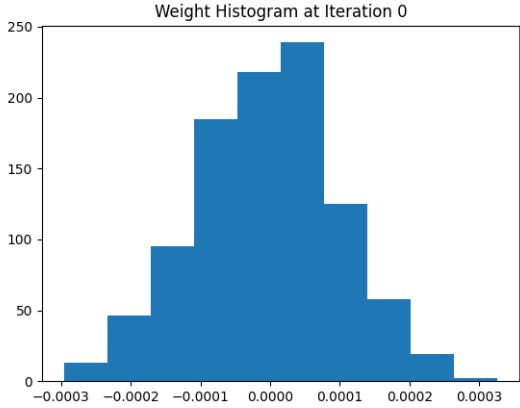

Figure 1: Distribution of coordinates of $w$ at initialization

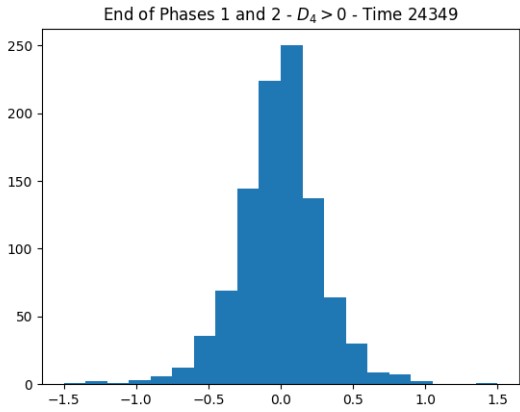

Figure 2: Distribution of coordinates of $w$ after Phases 1 and 2

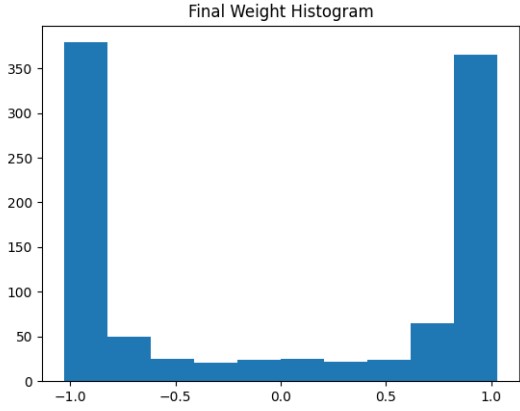

Figure 3: Distribution of coordinates of $w$ after time passes in Phase 3

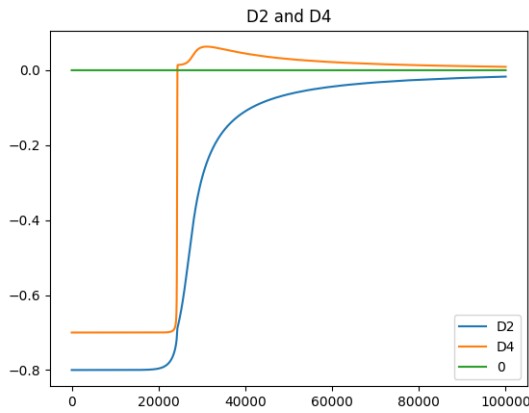

Figure 4: Evolution of $D_2$ and $D_4$

## C  Background on Spherical Harmonics

In the following, we summarize the background needed for this paper based on Atkinson and Han [12, Section 2].

**Spherical Harmonics.** Spherical harmonics are the eigenfunctions of any inner product kernels on the unit sphere $\mathbb{S}^{d-1}$. The eigenfunctions corresponding to the $k$-th eigenvalue are degree-$k$ polynomials, and form a Hilbert space $\mathbb{Y}_k^d$ with inner product $\langle f, g \rangle = \mathbb{E}_{x \sim \mathbb{S}^{d-1}}[f(x)g(x)]$. The dimension of $\mathbb{Y}_k^d$ is $N_{k,d} = \binom{d+k-1}{d-1} - \binom{d+k-3}{d-1}$. For any universal constant $k$, $N_{k,d} \asymp d^k$. Spherical harmonics with different degrees are orthogonal to each other, and their linear combinations can represent all square-integrable functions over $\mathbb{S}^{d-1}$.

The following theorem states that $\mathbb{Y}_k^d$ is the eigenspace corresponding to the $k$-th eigenvalue for any inner product kernel on the sphere.

**Theorem C.1** (Funk-Hecke formula [12]). *Let $h : [-1, 1] \to \mathbb{R}$ be any one-dimensional function with $\int_{-1}^{1} |h(t)|\mu_d(t)\mathrm{d}t < \infty$, and $\lambda_k = N_{k,d}^{-1/2} \mathbb{E}_{t \sim \mu_d}[h(t)\overline{P}_{k,d}(t)]$. Then for any function $Y_k \in \mathbb{Y}_{k,d}$,*

$$\forall x \in \mathbb{S}^{d-1}, \quad \mathbb{E}_{z \sim \mathbb{S}^{d-1}}[h(\langle x, z \rangle)Y_k(z)] = \lambda_k Y_k(x). \tag{C.1}$$

**Legendre Polynomials.** The un-normalized Legendre polynomial is defined recursively by

$$P_{0,d}(t) = 1, \quad P_{1,d}(t) = t, \tag{C.2}$$

$$P_{k,d}(t) = \frac{2k + d - 4}{k + d - 3}tP_{k-1,d}(t) - \frac{k-1}{k+d-3}P_{k-2,d}(t), \quad \forall k \geq 2. \tag{C.3}$$

In particular, we have

$$P_{2,d}(t) = \frac{d}{d-1}t^2 - \frac{1}{d-1}, \quad P_{4,d}(t) = \frac{(d+2)(d+4)}{d^2-1}t^4 - \frac{6d+12}{d^2-1}t^2 + \frac{3}{d^2-1}. \tag{C.4}$$

For any $u \in \mathbb{S}^{d-1}$, $P_{k,d}(\langle u, x \rangle)$ is a spherical harmonic of degree $k$ and, for every $k, k' \geq 0$ and $u_1, u_2 \in \mathbb{S}^{d-1}$, $\mathbb{E}_{x \sim \mathbb{S}^{d-1}}[\overline{P}_{k,d}(\langle u_1, x \rangle)\overline{P}_{k',d}(\langle u_2, x \rangle)] = \mathbf{1}[k = k'] P_{k,d}(\langle u_1, u_2 \rangle)$ (Lemma C.5).

**Projection Operator.** Let $\Pi_k$ be the projection operator to $\mathbb{Y}_k^d$. For any function $f : \mathbb{S}^{d-1} \to \mathbb{R}$, the projection operator $\Pi_k$ is given by

$$(\Pi_k f)(x) = \sqrt{N_{k,d}} \mathbb{E}_{\xi \sim \mathbb{S}^{d-1}}[\overline{P}_{k,d}(\langle x, \xi \rangle)f(\xi)]. \tag{C.5}$$

In addition, if $f$ is a function of the form $f(x) = h(\langle u, x \rangle)$ for some fixed vector $u \in \mathbb{S}^{d-1}$ and one-dimensional function $h : [-1, 1] \to \mathbb{R}$, the Funk-Hecke formula (Theorem C.1) implies

$$(\Pi_k f)(x) = \hat{h}_{k,d}\overline{P}_{k,d}(\langle u, x \rangle), \quad \forall x \in \mathbb{S}^{d-1}, \tag{C.6}$$

where $\hat{h}_{k,d} = \mathbb{E}_{t \sim \mu_d}[h(t)\overline{P}_{k,d}(t)]$ is the $k$-th coefficient in the Legendre decomposition of $h$.

Note that by the orthogonality of Legendre polynomials, we have

$$\forall x \in \mathbb{S}^{d-1}, \quad \mathop{\mathbb{E}}_{u \sim \mathbb{S}^{d-1}}[\overline{P}_{k,d}(\langle u, x \rangle)] = \mathop{\mathbb{E}}_{t \sim \mu_d}[\overline{P}_{k,d}(t)] = \mathop{\mathbb{E}}_{t \sim \mu_d}[\overline{P}_{k,d}(t)\overline{P}_{0,d}(t)] = \mathbf{1}\,[k=0]. \quad \text{(C.7)}$$

The following lemma computes the expectation of $\overline{P}_{k,d}(\langle u, x \rangle)$ with respect to a distribution on $u$ which is rotationally invariant on the last $(d-1)$ coordinates.

**Lemma C.2.** *Let $\rho$ be a distribution on $\mathbb{S}^{d-1}$ that is rotationally invariant as in Definition 4.1. Then, for any $x \in \mathbb{S}^{d-1}$ we have $\mathbb{E}_{u \sim \rho}[P_{k,d}(\langle u, x \rangle)] = \mathbb{E}_{u \sim \rho}[P_{k,d}(\langle e_1, u \rangle)]P_{k,d}(\langle e_1, x \rangle)$.*

*Proof.* We prove this lemma by invoking Eq. (2.167) of Atkinson and Han [12], which states that for every $s, t \in [-1, 1]$ and $k \geq 0, d \geq 3$,

$$\int_{-1}^{1} P_{k,d}(st + (1-s^2)^{1/2}(1-t^2)^{1/2}\xi)\mu_{d-1}(\xi)\mathrm{d}\xi = P_{k,d}(s)P_{k,d}(t). \quad \text{(C.8)}$$

In the following, for $x \in \mathbb{S}^{d-1}$, we write $x = (\langle e_1, x \rangle, \zeta)$, and for $u \sim \rho$ we write $u = (w, z)$. Note that when $\rho$ is symmetric and $u \sim \rho$, conditioned on $u_1$ we have $\langle z, \zeta \rangle \sim (1-u_1^2)^{1/2}(1-x_1^2)^{1/2}\mu_{d-1}$ for any fixed $x$. As a result, plugging in $t = w, s = \langle e_1, x \rangle$ and $\xi = (1-u_1^2)^{-1/2}(1-x_1^2)^{-1/2}\langle z, \zeta \rangle$ we get

$$\mathop{\mathbb{E}}_{u \sim \rho}[P_{k,d}(\langle u, x \rangle) \mid u_1] = P_{k,d}(w)P_{k,d}(\langle e_1, x \rangle). \quad \text{(C.9)}$$

Finally, taking expectation over $w$ we prove the desired result. $\qquad \square$

**Additional Useful Lemmas.** In the following, we present some useful lemmas about spherical harmonics and Legendre polynomials.

The following proposition computes the coefficient in the Legendre polynomial decomposition of ReLU activation.

**Proposition C.3** (Lemma C.2 of Dong and Ma [27]). *Let $\sigma(t) = \max\{t, 0\}$ be the ReLU activation. For every $d \geq 3$ and even $k$, we have*

$$\left| \mathop{\mathbb{E}}_{t \sim \mu_d}[\sigma(t)\overline{P}_{k,d}(t)] \right| \asymp d^{1/4}k^{-5/4}(d+k)^{-3/4}. \quad \text{(C.10)}$$

*As a corollary, when $k$ is an absolute constant we have*

$$\left| \mathop{\mathbb{E}}_{t \sim \mu_d}[\sigma(t)\overline{P}_{k,d}(t)] \right| \asymp d^{-1/2}. \quad \text{(C.11)}$$

**Theorem C.4** (Addition Theorem [12]). *Let $\{Y_{k,j} : 1 \leq j \leq N_{k,d}\}$ be an orthonormal basis of $\mathbb{Y}_{k,d}$, that is,*

$$\mathop{\mathbb{E}}_{x \sim \mathbb{S}^{d-1}}[Y_{k,j}(x)Y_{k,j'}(x)] = \mathbf{1}\,[j = j']. \quad \text{(C.12)}$$

*Then for all $x, z \in \mathbb{S}^{d-1}$*

$$\sum_{j=1}^{N_{k,d}} Y_{k,j}(x)Y_{k,j}(z) = N_{k,d}P_{k,d}(\langle x, z \rangle). \quad \text{(C.13)}$$

**Lemma C.5.** *For every $k, k' \geq 0, d \geq 3$ and $u, v \in \mathbb{S}^{d-1}$, we have*

$$\mathop{\mathbb{E}}_{\xi \sim \mathbb{S}^{d-1}}[\overline{P}_{k,d}(\langle u, \xi \rangle)\overline{P}_{k',d}(\langle v, \xi \rangle)] = \mathbf{1}\,[k = k']\,P_{k,d}(\langle u, v \rangle). \quad \text{(C.14)}$$

*Proof.* Let $\{Y_{k,j} : 1 \leq j \leq N_{k,d}\}$ be an orthonormal basis of $\mathbb{Y}_{k,d}$ with

$$\mathop{\mathbb{E}}_{\xi \sim \mathbb{S}^{d-1}}[Y_{k,j}(\xi)Y_{k,j'}(\xi)] = \mathbf{1}\,[j = j']. \quad \text{(C.15)}$$

Similarly define $\{Y_{k',j} : 1 \leq j \leq N_{k',d}\}$ for $\mathbb{Y}_{k',d}$. The addition theorem (Theorem C.4) implies that

$$P_{k,d}(\langle u, \xi \rangle) = \frac{1}{N_{k,d}} \sum_{j=1}^{N_{k,d}} Y_{k,j}(u) Y_{k,j}(\xi). \tag{C.16}$$

$$P_{k',d}(\langle u, \xi \rangle) = \frac{1}{N_{k',d}} \sum_{j=1}^{N_{k',d}} Y_{k',j}(u) Y_{k',j}(\xi). \tag{C.17}$$

Recall that for $k \neq k'$, $\mathbb{Y}_{k,d}$ and $\mathbb{Y}_{k',d}$ are orthononal subspaces. Consequently,

$$\mathop{\mathbb{E}}_{\xi \sim \mathbb{S}^{d-1}}[\overline{P}_{k,d}(\langle u, \xi \rangle) \overline{P}_{k',d}(\langle v, \xi \rangle)] = \frac{1}{\sqrt{N_{k,d} N_{k',d}}} \sum_{i=1}^{N_{k,d}} \sum_{j=1}^{N_{k',d}} \mathop{\mathbb{E}}_{\xi \sim \mathbb{S}^{d-1}}[Y_{k,i}(u) Y_{k,i}(\xi) Y_{k',j}(v) Y_{k',j}(\xi)] \tag{C.18}$$

$$= \frac{\mathbf{1}\,[k = k']}{N_{k,d}} \sum_{j=1}^{N_{k,d}} \mathop{\mathbb{E}}_{\xi \sim \mathbb{S}^{d-1}}[Y_{k,j}(u) Y_{k,j}(v)] \tag{C.19}$$

$$= \mathbf{1}\,[k = k']\, P_{k,d}(\langle u, v \rangle). \tag{C.20}$$

$\square$

**Lemma C.6.** *For any $k \geq 0, d \geq 3$, there exists a feature mapping $\phi : \mathbb{S}^{d-1} \to \mathbb{R}^{N_{k,d}}$ such that for every $u, v \in \mathbb{S}^{d-1}$,*

$$\langle \phi(u), \phi(v) \rangle = N_{k,d} P_{k,d}(\langle u, v \rangle), \tag{C.21}$$

$$\|\phi(u)\|_2^2 = N_{k,d}, \tag{C.22}$$

$$\mathop{\mathbb{E}}_{u \sim \mathbb{S}^{d-1}}[\phi(u) \phi(u)^\top] = I. \tag{C.23}$$

*Proof.* Let $Y_1, \cdots, Y_{N_{k,d}} : \mathbb{S}^{d-1} \to \mathbb{R}$ be an orthonormal basis for the degree-$k$ spherical harmonics $\mathbb{Y}_{k,d}$. We let $\phi(x) = (Y_l(x))_{l \in [N_{k,d}]}$ be the feature mapping, and in the following we verify Eqs. (C.21)-(C.23). By the addition theorem (Theorem C.4), for every $u, v \in \mathbb{S}^{d-1}$ we have

$$\langle \phi(u), \phi(v) \rangle = \sum_{j=1}^{N_{k,d}} Y_j(u) Y_j(v) = N_{k,d} P_{k,d}(\langle u, v \rangle), \tag{C.24}$$

which proves Eq. (C.21). Note that for every $u \in \mathbb{S}^{d-1}$, $P_{k,d}(\langle u, u \rangle) = P_{k,d}(1) = 1$. Hence Eq. (C.22) follows directly. To prove Eq. (C.23), note that $Y_1, \cdots, Y_{N_{k,d}}$ are orthonormal. Therefore for every $i, j \in [N_{k,d}]$

$$\mathop{\mathbb{E}}_{u \sim \mathbb{S}^{d-1}}[Y_i(u) Y_j(u)] = \mathbf{1}\,[i = j]. \tag{C.25}$$

Hence, Eq. (C.23) follows directly. $\square$

**Lemma C.7.** *Let $\mu_d$ be the distribution of $\langle u, v \rangle$ when $v$ is drawn uniformly at random from the unit sphere $\mathbb{S}^{d-1}$ and $u \in \mathbb{S}^{d-1}$ is a fixed vector. For fixed $n \geq 1$, let $x_1, \cdots, x_n \in \mathbb{S}^{d-1}$ be (not necessarily independent) random variables drawn from the distribution $\mu_d$. Then there exists a universal constant $c > 0$ such that for any integer $k \geq 1$,*

$$\forall t \geq 4^{1+k}(k \ln d)^k, \quad \Pr\left(\max_{i \leq n} d^k P_{k,d}(x_i)^2 \geq t\right) \leq 2n \exp\left(-\frac{t^{1/k}}{32}\right). \tag{C.26}$$

*In addition, we also have*

$$\mathbb{E}\left[\max_{i \leq n} d^k P_{k,d}(x_i)^2\right] \lesssim (32k \ln(dn))^k. \tag{C.27}$$

*Proof.* To prove the first part of this lemma, we first invoke Proposition C.8, which states that

$$\forall x \in \left[ \sqrt{\frac{k \ln d}{d}}, 1 \right], \quad |P_{k,d}(x)| \leq 2^{1+k} x^k. \tag{C.28}$$

When $x \sim \mu_d$, Proposition I.5 states that

$$\forall t > 0, \quad \Pr(|x| \geq t) \leq 2 \exp\left(-\frac{t^2 d}{2}\right). \tag{C.29}$$

Therefore for every $t \geq 4^{1+k}(k \ln d)^k$ we have

$$\Pr\left(P_{k,d}(x)^2 d^k \geq t\right) \leq \Pr\left(|x| \geq t^{1/2k} d^{-1/2} 2^{-1-1/k}\right) \tag{C.30}$$

$$\leq 2 \exp\left(-\frac{1}{2} t^{1/k} 4^{-1-1/k}\right) \tag{C.31}$$

$$\leq 2 \exp(-t^{1/k}/32). \tag{C.32}$$

By union bound we get

$$\forall t \geq 4^{1+k}(k \ln d)^k, \quad \Pr\left(\max_{i \leq n} d^k P_{k,d}(x_i)^2 \geq t\right) \leq 2n \exp\left(-\frac{t^{1/k}}{32}\right). \tag{C.33}$$

which proves the first part of this lemma.

As a corollary, for any fixed $\epsilon > 4^{1+k}(k \ln d)^k$ we have

$$\mathbb{E}\left[\max_{i \leq n} d^k P_{k,d}(x_i)^2\right] \leq \epsilon + \int_\epsilon^\infty \Pr\left(\max_{i \leq n} d^k P_{k,d}(x_i)^2 \geq t\right) dt \tag{C.34}$$

$$\leq \epsilon + n \int_\epsilon^\infty \Pr\left(d^k P_{k,d}(t)^2 \geq t\right) dt \tag{C.35}$$

$$\leq \epsilon + 2n \int_\epsilon^\infty \exp\left(-\frac{t^{1/k}}{32}\right) dt. \tag{C.36}$$

Now we upper bound the integral by changing variables. Letting $u = t^{1/k}/32$, we get

$$\int_\epsilon^\infty \exp\left(-\frac{t^{1/k}}{32}\right) dt \leq \int_{\frac{\epsilon^{1/k}}{32}}^\infty 32^k k u^{k-1} \exp(-u) \, du \tag{C.37}$$

$$\leq k^2 \exp\left(-\frac{\epsilon^{1/k}}{32}\right) 32^k \left(\frac{\epsilon^{1/k}}{32}\right)^{k-1} \tag{C.38}$$

$$\leq 32 k^2 \exp\left(-\frac{\epsilon^{1/k}}{32}\right) \epsilon \tag{C.39}$$

$$\tag{C.40}$$

where the last inequality comes from an upper bound for the incomplete gamma functions [31, Theorem 4.4.3]. Therefore we get

$$\mathbb{E}\left[\max_{i \leq n} d^k P_{k,d}(x_i)^2\right] \leq \epsilon + 64 n \epsilon k^2 \exp\left(-\frac{\epsilon^{1/k}}{32}\right). \tag{C.41}$$

Finally taking $\epsilon = (32k \ln(dn))^k$, we prove the desired result. □

**Proposition C.8.** *For any $k \geq 0, d \geq 2$, we have*

$$\forall t \in \left[ \sqrt{\frac{k \ln d}{d}}, 1 \right], \quad |P_{k,d}(t)| \leq 2^{1+k} t^k. \tag{C.42}$$

*Proof.* Let $\mu_d(t) \triangleq (1 - t^2)^{\frac{d-3}{2}} \frac{\Gamma(d/2)}{\Gamma((d-1)/2)} \frac{1}{\sqrt{\pi}}$ be the density of $u_1$ when $u = (u_1, \cdots, u_d)$ is drawn uniformly from $\mathbb{S}^{d-1}$. By Atkinson and Han [12, Theorem 2.24] we have

$$P_{k,d}(t) = \underset{s \sim \mu_{d-1}}{\mathbb{E}} \left[ (t + i(1 - t^2)^{1/2} s)^k \right]. \tag{C.43}$$

As a result,

$$|P_{k,d}(t)| \leq \underset{s \sim \mu_{d-1}}{\mathbb{E}} \left[ |t + i(1 - t^2)^{1/2} s|^k \right] \leq \underset{s \sim \mu_{d-1}}{\mathbb{E}} \left[ (t^2 + (1 - t^2)s^2)^{k/2} \right]. \tag{C.44}$$

By Proposition I.5 we have,

$$\Pr(\sqrt{d-1}|s| \leq 2\sqrt{k \ln d}) \leq 2\exp(-2k \ln d) \leq d^{-k}. \tag{C.45}$$

Consequently,

$$\underset{s \sim \mu_{d-1}}{\mathbb{E}} \left[ (t^2 + (1 - t^2)s^2)^{k/2} \right] \leq d^{-k} + \underset{s \sim \mu_{d-1}}{\mathbb{E}} \left[ \mathbf{1} \left[ s \leq 2\sqrt{k \ln d}/\sqrt{d-1} \right] (t^2 + s^2)^{k/2} \right] \tag{C.46}$$

$$\leq d^{-k} + \left( t^2 + \frac{2k \ln d}{d - 1} \right)^{k/2}. \tag{C.47}$$

Hence, when $t \geq \sqrt{\frac{k \ln d}{d}}$ we get

$$|P_{k,d}(t)| \leq d^{-k} + \left( t^2 + \frac{2k \ln d}{d - 1} \right)^{k/2} \leq 2^{1+k} t^k. \tag{C.48}$$

$\square$

# D   Proofs for Population Dynamics

## D.1   Proof of Lemma 4.2

In the following, we prove Lemma 4.2, which provides a way to simplify $L(\rho)$ when $\rho$ is rotationally invariant and symmetric.

*Proof of Lemma 4.2.* Recall that the population loss is

$$L(\rho) = \frac{1}{2} \underset{x \sim \mathbb{S}^{d-1}}{\mathbb{E}} [(f_\rho(x) - h(\langle x, e_1 \rangle))^2], \tag{D.1}$$

and the activation function $\sigma : \mathbb{R} \to \mathbb{R}$ and target function $h : \mathbb{R} \to \mathbb{R}$ has the following decomposition:

$$\sigma(t) = \sum_{k=0}^{\infty} \hat{\sigma}_{k,d} \overline{P}_{k,d}(t), \tag{D.2}$$

$$h(t) = \sum_{k=0}^{\infty} \hat{h}_{k,d} \overline{P}_{k,d}(t). \tag{D.3}$$

Therefore we have

$$2L(\rho) = \underset{x \sim \mathbb{S}^{d-1}}{\mathbb{E}} [(f_\rho(x) - h(\langle x, e_1 \rangle))^2] \tag{D.4}$$

$$= \underset{x \sim \mathbb{S}^{d-1}}{\mathbb{E}} \left( \underset{u \sim \rho}{\mathbb{E}} [\sigma(\langle u, x \rangle)] - h(\langle x, e_1 \rangle) \right)^2 \tag{D.5}$$

$$= \underset{x \sim \mathbb{S}^{d-1}}{\mathbb{E}} \left( \underset{u \sim \rho}{\mathbb{E}} \left[ \sum_{k=0}^{\infty} \hat{\sigma}_{k,d} \overline{P_{k,d}}(\langle u, x \rangle) \right] - \sum_{k=0}^{\infty} \hat{h}_{k,d} \overline{P}_{k,d}(\langle x, e_1 \rangle) \right)^2 \tag{D.6}$$

$$= \underset{x \sim \mathbb{S}^{d-1}}{\mathbb{E}} \left( \sum_{k=0}^{\infty} \left( \hat{\sigma}_{k,d} \underset{u \sim \rho}{\mathbb{E}} [\overline{P}_{k,d}(\langle u, x \rangle)] - \hat{h}_{k,d} \overline{P}_{k,d}(\langle x, e_1 \rangle) \right) \right)^2. \tag{D.7}$$

When the neural network $\rho$ is rotationally invariant, we can simplify the above equation by invoking Lemma C.2, which states that

$$\mathbb{E}_{u\sim\rho}[\overline{P}_{k,d}(\langle u, x\rangle)] = \mathbb{E}_{u\sim\rho}[P_{k,d}(\langle e_1, u\rangle)]\overline{P}_{k,d}(\langle e_1, x\rangle). \tag{D.8}$$

Continuing Eq. (D.4) above we have,

$$2L(\rho) = \mathbb{E}_{x\sim\mathbb{S}^{d-1}} \left( \sum_{k=0}^{\infty} \hat{\sigma}_{k,d} \mathbb{E}_{u\sim\rho}[P_{k,d}(\langle e_1, u\rangle)]\overline{P}_{k,d}(\langle e_1, x\rangle) - \hat{h}_{k,d}\overline{P}_{k,d}(\langle e_1, x\rangle) \right)^2 \tag{D.9}$$

$$= \mathbb{E}_{x\sim\mathbb{S}^{d-1}} \left( \sum_{k=0}^{\infty} (\hat{\sigma}_{k,d} \mathbb{E}_{u\sim\rho}[P_{k,d}(\langle e_1, u\rangle)] - \hat{h}_{k,d})\overline{P}_{k,d}(\langle e_1, x\rangle) \right)^2 \tag{D.10}$$

Now we expand the square by invoking Lemma C.5. In particular, Lemma C.5 states that

$$\mathbb{E}_{x\sim\mathbb{S}^{d-1}}[\overline{P}_{k,d}(\langle e_1, x\rangle)\overline{P}_{k',d}(\langle e_1, x\rangle)] = \mathbf{1}\left[k = k'\right] P_{k,d}(\langle e_1, e_1\rangle) = \mathbf{1}\left[k = k'\right]. \tag{D.11}$$

Consequently,

$$2L(\rho) = \sum_{k=0}^{\infty} \left( \hat{\sigma}_{k,d} \mathbb{E}_{u\sim\rho}[P_{k,d}(\langle e_1, u\rangle)] - \hat{h}_{k,d} \right)^2, \tag{D.12}$$

as desired.

To prove the second part of this lemma, in the following, we assume that (1) $\hat{h}_{k,d} = 0, \forall k \notin \{0, 2, 4\}$, (2) $\hat{h}_{0,d} = \hat{\sigma}_{0,d}$, and (3) that $\sigma$ is a degree-4 polynomial. Then we get

$$L(\rho) = \frac{\hat{\sigma}_{2,d}^2}{2} \left( \mathbb{E}_{u\sim\rho}[P_{2,d}(w)] - \gamma_2 \right)^2 + \frac{\hat{\sigma}_{4,d}^2}{2} \left( \mathbb{E}_{u\sim\rho}[P_{4,d}(w)] - \gamma_4 \right)^2 + \sum_{k\in\{1,3\}} \frac{\hat{\sigma}_{k,d}^2}{2} \left( \mathbb{E}_{u\sim\rho}[P_{k,d}(w)] \right)^2. \tag{D.13}$$

Since $\rho$ is symmetric and $P_{k,d}(w)$ is an odd function when $k$ is odd, we get

$$\forall k \in \{1, 3\}, \quad \mathbb{E}_{u\sim\rho}[P_{k,d}(w)] = 0. \tag{D.14}$$

It follows directly that

$$L(\rho) = \frac{\hat{\sigma}_{2,d}^2}{2} \left( \mathbb{E}_{u\sim\rho}[P_{2,d}(w)] - \gamma_2 \right)^2 + \frac{\hat{\sigma}_{4,d}^2}{2} \left( \mathbb{E}_{u\sim\rho}[P_{4,d}(w)] - \gamma_4 \right)^2 \tag{D.15}$$

$\square$

## D.2 Proof of Lemma 4.3 and Lemma 4.4

The goal of this subsection is to show Lemma 4.3, which states that $\rho_t$ remains symmetric and rotationally invariant under the infinite-width population dynamics, and to show Lemma 4.4 which describes the dynamics of the first coordinates of the particles in $\rho_t$.

The following lemma is a first step. Recall that when $\rho$ is rotationally invariant, then Lemma 4.2 allows us to rewrite $L(\rho) = \frac{1}{2} \mathbb{E}_{x\sim\mathbb{S}^{d-1}} \left( f_\rho(x) - y \right)^2$ as $\frac{1}{2} \sum_{k=0}^{\infty} \left( \hat{\sigma}_{k,d} \mathbb{E}_{u\sim\rho}[P_{k,d}(w)] - \hat{h}_{k,d} \right)^2$. The following lemma states that the gradient from the two formulas for $L(\rho)$ are the same when $\rho$ is rotationally invariant.

**Lemma D.1.** *Let $\rho$ be a distribution on $\mathbb{S}^{d-1}$ and let $\nu$ be a distribution on $[-1, 1]$. Define*

$$F(\nu) = \frac{1}{2} \sum_{k=0}^{\infty} \left( \hat{\sigma}_{k,d} \mathbb{E}_{w\sim\nu}[P_{k,d}(w)] - \hat{h}_{k,d} \right)^2. \tag{D.16}$$

*If $\rho$ is rotationally invariant as in Definition 4.1 and $\nu$ is the marginal distribution of $w$ for $u = (w, z) \sim \rho$, then for any particle $u = (w, z)$, we have*

$$(1 - w^2)\nabla_w F(\nu) = \mathrm{grad}_w L(\rho) \tag{D.17}$$

*where $\mathrm{grad}_w L(\rho)$ denotes the first coordinate of $\mathrm{grad}_u L(\rho)$.*

*Proof of Lemma D.1.* For any particle $u = (w, z)$, we write the first coordinate of $\nabla_u L(\rho)$ as $\nabla_w L(\rho)$, and the vector consisting of the last $(d-1)$ coordinates as $\nabla_z L(\rho)$. Observe that

$$\text{grad}_w L(\rho) = \langle e_1, \text{grad}_u L(\rho) \rangle \tag{D.18}$$

$$= \langle e_1, (I - uu^\top)\nabla_u L(\rho) \rangle \tag{D.19}$$

$$= \nabla_w L(\rho) - w \langle u, \nabla_u L(\rho) \rangle \tag{D.20}$$

$$= (1 - w^2)\nabla_w L(\rho) - w \langle z, \nabla_z L(\rho) \rangle \tag{D.21}$$

On the other hand, suppose $\rho$ satisfies the rotational invariance property, and that $\nu$ is the marginal distribution of the first coordinate under $\rho$. Then, we can write $\rho$ in terms of $\nu$ — for any $w \in [-1, 1]$, the distribution of $z$ conditioned on $w$ is $\sqrt{1-w^2}\mathbb{S}^{d-1}$. Thus, since $L(\rho) = F(\nu)$ by Lemma 4.2, we have by the chain rule that

$$\nabla_w F(\nu) = \mathbb{E}_{\substack{\xi \sim \mathbb{S}^{d-2} \\ u = we_1 + \sqrt{1-w^2}\xi}} \left[ \left\langle \nabla_u L(\rho), \frac{\partial u}{\partial w} \right\rangle \right] \tag{D.22}$$

$$= \nabla_w L(\rho) + \mathbb{E}_{\xi \sim \mathbb{S}^{d-2}} \left[ \left\langle \nabla_z L(\rho) \Big|_{z=\sqrt{1-w^2}\xi}, \frac{\partial z}{\partial w} \right\rangle \right] \tag{D.23}$$

$$= \nabla_w L(\rho) + \mathbb{E}_{\xi \sim \mathbb{S}^{d-2}} \left[ \left\langle \nabla_z L(\rho) \Big|_{z=\sqrt{1-w^2}\xi}, -\frac{w}{\sqrt{1-w^2}}\xi \right\rangle \right] \tag{D.24}$$

$$= \nabla_w L(\rho) + \mathbb{E}_{\xi \sim \mathbb{S}^{d-2}} \left[ \left\langle \nabla_z L(\rho) \Big|_{z=\sqrt{1-w^2}\xi}, -\frac{w}{1-w^2}z \right\rangle \right] \tag{D.25}$$

$$= \nabla_w L(\rho) - \frac{w}{1-w^2} \mathbb{E}_{\xi \sim \mathbb{S}^{d-2}}[\langle \nabla_z L(\rho), z \rangle] \tag{D.26}$$

Moreover, for any fixed $w \in [-1, 1]$, the quantity $\langle \nabla_z L(\rho), z \rangle$ does not depend on $z$, since by Lemma D.2, for any rotation matrix $A$ and $z' = Az$, we have $\langle \nabla_{z'} L(\rho), z' \rangle = \langle A \cdot \nabla_z L(\rho), Az \rangle = \langle \nabla_z L(\rho), z \rangle$. Thus, for any $w \in [-1, 1]$ and $z \in \sqrt{1-w^2}\mathbb{S}^{d-2}$, we can write

$$(1 - w^2)\nabla_w F(\nu) = (1 - w^2)\nabla_w L(\rho) - w \langle z, \nabla_z L(\rho) \rangle = \text{grad}_w L(\rho) \tag{D.27}$$

as desired. $\qquad\square$

In the proof of the above lemma, we used the following fact which we now prove: when $\rho$ is rotationally invariant, $\nabla_u L(\rho)$ is rotationally equivariant as a function of $u$, i.e. if $u$ is rotated by a rotation matrix $A$ which only modifies the last $(d-1)$ coordinates, then $\nabla_u L(\rho)$ is rotated by $A$ as well.

**Lemma D.2** (Gradient is Rotationally Equivariant). *Let $\rho$ be a rotationally invariant distribution as in Definition 4.1. Let $A \in \mathbb{R}^{d \times d}$ be a rotation matrix with $Ae_1 = e_1$, i.e. $A$ only modifies the last $(d-1)$ coordinates. Let $u \in \mathbb{S}^{d-1}$ and $u' = Au$. Then, $\nabla_{u'} L(\rho) = A \cdot \nabla_u L(\rho)$, and $\text{grad}_{u'} L(\rho) = A \cdot \text{grad}_u L(\rho)$.*

*Proof.* The proof is by a straightforward calculation:

$$\nabla_{u'} L(\rho) = \mathbb{E}_{x \sim \mathbb{S}^{d-1}}[(f_\rho(x) - y(x))\sigma'(\langle u', x \rangle)x] \tag{D.28}$$

$$= \mathbb{E}_{x \sim \mathbb{S}^{d-1}}[(f_\rho(x) - h(\langle e_1, x \rangle))\sigma'(\langle u', x \rangle)x] \tag{D.29}$$

$$= \mathbb{E}_{x \sim \mathbb{S}^{d-1}}[(f_\rho(Ax) - h(\langle e_1, Ax \rangle))\sigma'(\langle u', Ax \rangle)Ax]$$

(By rotational invariance of uniform distribution on $\mathbb{S}^{d-1}$)

$$= A \mathbb{E}_{x \sim \mathbb{S}^{d-1}}[(f_\rho(Ax) - h(\langle e_1, Ax \rangle))\sigma'(\langle u', Ax \rangle)x] \tag{D.30}$$

$$= A \mathbb{E}_{x \sim \mathbb{S}^{d-1}}[(f_\rho(Ax) - h(\langle e_1, Ax \rangle))\sigma'(\langle Au, Ax \rangle)x] \qquad \text{(By definition of } u')$$

$$= A \mathbb{E}_{x \sim \mathbb{S}^{d-1}}[(f_\rho(Ax) - h(\langle e_1, Ax \rangle))\sigma'(\langle u, x \rangle)x]$$

(B.c. $A$ is a rotation matrix so $A^\top A = I_{d \times d}$)

By the rotational invariance of $\rho$, we have $f_\rho(Ax) = f_\rho(x)$, meaning

$$\nabla_{u'} L(\rho) = A \mathop{\mathbb{E}}_{x \sim \mathbb{S}^{d-1}} [(f_\rho(x) - h(\langle e_1, Ax \rangle)) \sigma'(\langle u, x \rangle) x]$$

$$\text{(B.c. } f_\rho(Ax) = f_\rho(x) \text{ by rotational invariance of } \rho)$$

$$= A \mathop{\mathbb{E}}_{x \sim \mathbb{S}^{d-1}} [(f_\rho(x) - h(\langle e_1, x \rangle)) \sigma'(\langle u, x \rangle) x] \qquad \text{(B.c. } Ae_1 = e_1 \text{ and } A^\top A = I_{d \times d})$$

$$= A \cdot \nabla_u L(\rho) \tag{D.31}$$

This proves the first statement of the lemma. The second statement also follows from a similar calculation:

$$\text{grad}_{u'} L(\rho) = (I - u'(u')^\top) \cdot \nabla_{u'} L(\rho) \tag{D.32}$$

$$= (I - Auu^\top A^\top) \cdot A \cdot \nabla_u L(\rho) \tag{D.33}$$

$$= A \cdot \nabla_u L(\rho) - Auu^\top A^\top A \cdot \nabla_u L(\rho) \tag{D.34}$$

$$= A \cdot \nabla_u L(\rho) - Auu^\top \cdot \nabla_u L(\rho) \qquad \text{(B.c. } A^\top A = I_{d \times d}) \tag{D.35}$$

$$= A \cdot (I - uu^\top) \cdot \nabla_u L(\rho) \tag{D.35}$$

$$= A \cdot \text{grad}_u L(\rho) \tag{D.36}$$

as desired. $\qquad \square$

As a corollary, we obtain our first formula for the dynamics of the 1-dimensional particles:

**Lemma D.3** (One-Dimensional Velocity). *Suppose we are in the setting of Theorem 3.3 and that $\rho_t$ is rotationally invariant. Then, for any particle $u_t$ (where we omit the initialization $\chi$ for convenience), if $w_t$ is the first coordinate of $u_t$, then*

$$\frac{dw_t}{dt} = -(1 - w_t^2) \sum_{k=0}^{4} \left( \hat{\sigma}_{k,d} \mathop{\mathbb{E}}_{u \sim \rho_t} [P_{k,d}(w)] - \hat{h}_{k,d} \right) \cdot \hat{\sigma}_{k,d} P_{k,d}'(w) . \tag{D.37}$$

*Proof of Lemma D.3.* This follows directly from Lemma D.1, and because the activation $\sigma$ and the target $h$ only have nonzero coefficients $\hat{\sigma}_{k,d}$ and $\hat{h}_{k,d}$ for $k = 0, \dots, 4$. $\qquad \square$

We now complete the proof of Lemma 4.3, i.e. that $\rho_t$ remains rotationally invariant and symmetric if it is defined according to the population projected gradient flow as in Eq. (3.6) and Eq. (3.7). To prove that $\rho_t$ remains symmetric, we make use of the above formula for the 1-dimensional dynamics.

*Proof of Lemma 4.3.* First, we show the rotational invariance property. Assume inductively that at a time $t$, $\rho_t$ satisfies the desired rotational invariance property — it holds at initialization since $\rho_0$ is the uniform distribution on $\mathbb{S}^{d-1}$. Consider a particle $u_t$ and let $A \in \mathbb{R}^{d \times d}$ be a rotation matrix which only modifies the last $(d-1)$ coordinates. Let $u_t' = Au_t$ be another particle. Then, it follows from Lemma D.2 that

$$\frac{du_t'}{dt} = A \frac{du_t}{dt} \tag{D.38}$$

Thus, for a fixed $w \in [-1, 1]$, if $z \sim \sqrt{1 - w^2} \mathbb{S}^{d-2}$, then the distribution of $\frac{dz}{dt}$ is uniform on $c\mathbb{S}^{d-2}$ for a fixed scalar $c$ that depends on $w$. Therefore, $\rho$ will remain rotationally invariant.

Next, we prove that $\rho_t$ is symmetric. First observe that $\rho_t$ is symmetric at initialization, since it is the uniform distribution on $\mathbb{S}^{d-1}$. Now, suppose that $\rho_t$ is symmetric at time $t$. The polynomial $P_{k,d}$ is odd if $k$ is odd and even if $k$ is even — this can be seen from induction on Eq. (C.2) and Eq. (C.3). Thus, $\mathbb{E}_{u \sim \rho_t}[P_{k,d}(w)] = 0$ for odd $k$ by our assumption that $\rho_t$ is symmetric. Additionally, recall that we defined $h$ so that $\hat{\sigma}_{0,d} = \hat{h}_{0,d}$ and $\hat{h}_{k,d} = 0$ for odd $k$. Thus, by our definition of $D_{2,t}$ and $D_{4,t}$ in Section 4,

$$\frac{dw_t}{dt} = -(1 - w_t^2)(\hat{\sigma}_{2,d}^2 D_{2,t} P_{2,d}'(w_t) + \hat{\sigma}_{4,d}^2 D_{4,t} P_{4,d}'(w_t)) \tag{D.39}$$

by Lemma D.3. Since $P_{2,d}$ and $P_{4,d}$ are even polynomials, we know $P_{2,d}'$ and $P_{4,d}'$ are odd polynomials, and thus $\frac{dw_t}{dt}$ is an odd polynomial in $w_t$. In other words, if $w_t$ and $w_t'$ are the first

coordinates of two particles $u_t$ and $u'_t$ respectively, such that $w'_t = -w_t$, then $\frac{dw'_t}{dt} = -\frac{dw_t}{dt}$. Thus, the distribution of $\frac{dw_t}{dt}$ is symmetric, meaning that $\rho_t$ will remain symmetric. $\qquad\square$

In the rest of the proofs for the infinite-width population dynamics, we make use of the symmetry property. In particular, many of the lemmas are stated for $w > 0$ or $\iota > 0$ — however, the generalizations to $w < 0$ or $\iota < 0$ also hold. We use this symmetry implicitly throughout the proofs in this section.

In Lemma 4.4, we are simply rewriting our formula for the 1-dimensional dynamics in a more convenient form. The following proof shows the detailed calculation.

*Proof of Lemma 4.4.* By Lemma D.3 and the fact that only the second and fourth order terms are nonzero (which follows from Lemma 4.3, the fact that $P_{k,d}$ is an odd polynomial for odd $k$ (Eq. (C.2) and Eq. (C.3)) and because $\hat{h}_{1,d} = \hat{h}_{3,d} = 0$), the velocity of $w$ is equal to the following:

$$\mathrm{grad}_w L(\rho) = -(1 - w^2) \cdot \left( \hat{\sigma}_{2,d}^2 D_2(t) P_{2,d}{}'(w) + \hat{\sigma}_{4,d}^2 D_4(t) P_{4,d}{}'(w) \right) \qquad (D.40)$$

where we use $\mathrm{grad}_w L(\rho)$ to denote the first coordinate of $\mathrm{grad}_u L(\rho)$. For convenience, during the rest of the proof of this lemma, let $\eta_d{}^{(1)} = \frac{d}{d-1}$, $\eta_d{}^{(2)} = \frac{d^2 + 6d + 8}{d^2 - 1}$ and $\eta_d{}^{(3)} = \frac{6d+12}{d^2-1}$. Then, by Eq. (C.4),

$$\mathrm{grad}_w L(\rho) = -(1 - w^2) \cdot \left( \hat{\sigma}_{2,d}^2 D_2(t) P'_{2,d}(w) + \hat{\sigma}_{4,d}^2 D_4(t) P'_{4,d}(w) \right) \qquad (D.41)$$

$$= -(1 - w^2) \cdot \left( \hat{\sigma}_{2,d}^2 D_2(t) \cdot 2\eta_d{}^{(1)} w + \hat{\sigma}_{4,d}^2 D_4(t) \cdot (4\eta_d{}^{(2)} w^3 - 2\eta_d{}^{(3)} w) \right) \qquad (D.42)$$

$$= -(1 - w^2) \cdot \left( P_t(w) + 2\hat{\sigma}_{2,d}^2 D_2(t) \cdot (\eta_d{}^{(1)} - 1) w \right. \qquad (D.43)$$

$$\left. + 4\hat{\sigma}_{4,d}^2 D_4(t) \cdot (\eta_d{}^{(2)} - 1) w^3 - 2\hat{\sigma}_{4,d}^2 D_4(t) \eta_d{}^{(3)} w \right) \qquad (D.44)$$

$$= -(1 - w^2) \cdot \left( P_t(w) + Q_t(w) \right) \qquad (D.45)$$

Here we have chosen $\lambda_d{}^{(1)} = 2\hat{\sigma}_{2,d}^2(\eta_d{}^{(1)} - 1)D_2(t) - 2\hat{\sigma}_{4,d}^2 D_4(t)\eta_d{}^{(3)}$ and $\lambda_d{}^{(3)} = 4\hat{\sigma}_{4,d}^2 D_4(t)(\eta_d{}^{(2)} - 1)$. Observe that $|\lambda_d{}^{(1)}|, |\lambda_d{}^{(3)}| \leq O(\frac{\hat{\sigma}_{2,d}^2|D_{2,t}| + \hat{\sigma}_{4,d}^2|D_{4,t}|}{d})$, as desired. $\qquad\square$

### D.3 Analysis of Phase 1

Recall the formal definition of Phase 1 in Definition 4.5. Intuitively, during Phase 1, the particles are all growing at a similar rate — because the particles $w$ are all less than $w_{\max} = \frac{1}{\log d}$ in magnitude, the cubic term involving $w^3$ in the velocity of $w$ does not have a significant effect on the growth of $w$. We formalize this in the following lemma.

**Lemma D.4** (Uniform Growth Lemma). *Suppose we are in the setting of Theorem 3.3. Consider a time interval $[t_0, t_1] \in [0, T_2]$ in Phase 1 or 2. We are interested in quantifying the (relative) growth rate of the neurons indexed (or initialized) at $\iota_1$ and $\iota_2$ (where $0 < \iota_1 < \iota_2$), defined as*

$$F_1 \triangleq \frac{w_{t_1}(\iota_1)}{w_{t_0}(\iota_1)}$$

$$F_2 \triangleq \frac{w_{t_1}(\iota_2)}{w_{t_0}(\iota_2)}$$

*Let $\delta \in (0,1)$ such that $\frac{1}{\sqrt{d}} \lesssim \delta \leq \frac{1}{\log F_2}$ — here, we implicitly assume that $F_2 \leq e^d$. We also assume that $\delta \leq \frac{\gamma_2^2}{16}$. Further assume that $w_{t_1}(\iota_2) = \delta$. Finally assume that $\mathbb{P}(|\iota| \geq |\iota_2|) \leq \frac{\gamma_2^2}{16}$. Then,*

$$F_1 \asymp F_2 \qquad (D.46)$$

*and additionally,*

$$F_1 \asymp \exp\left( \int_{t_0}^{t_1} 2\hat{\sigma}_{2,d}^2 |D_{2,t}| dt \right) \qquad (D.47)$$

*Finally,*

$$t_1 - t_0 \lesssim \frac{1}{\hat{\sigma}_{2,d}^2 \gamma_2} \log F_2 \tag{D.48}$$

*We note that $F_1$ and $F_2$ are both at least $1$.*

Note that this lemma also holds for $\iota_2 < \iota_1 < 0$, and with $w_{t_1}(\iota_2) = -\delta$, by the symmetry of $\rho_t$.

*Proof of Lemma D.4.* In this proof, we define $\tau = \sqrt{\gamma_2}$ — then, by Assumption 3.2, we can write $\gamma_4 = \beta \cdot \tau^4$ where $1.1 \leq \beta$ and $\beta$ is less than some universal constant $c_1$. Suppose at time $t_1$, $w_{t_1}(\iota_2) = \delta$. One useful observation, which we use in the rest of this proof, is that for all $t \leq t_1$, $w_t(\iota_2) \leq \delta$, since for $w \gtrsim \frac{1}{\sqrt{d}}$, we have $\frac{dw}{dt} \geq 0$ by Lemma 4.4 (here we used the assumption that $\delta \gtrsim \frac{1}{\sqrt{d}}$). Thus,

$$\mathop{\mathbb{E}}_{w \sim \rho}[P_{2,d}(w)] = \mathbb{P}(|\iota| > |\iota_2|) \cdot \mathop{\mathbb{E}}_{w \sim \rho}[P_{2,d}(w) \mid |\iota| > |\iota_2|] + \mathbb{P}(|\iota| < |\iota_2|) \cdot \mathop{\mathbb{E}}_{w \sim \rho}[P_{2,d}(w) \mid |\iota| < |\iota_2|] \tag{D.49}$$

$$\leq \frac{\gamma_2^2}{16} + \mathop{\mathbb{E}}_{w \sim \rho}[P_{2,d}(w) \mid |\iota| < |\iota_2|]$$
(B.c. $|P_{2,d}(w)| \leq 1$ by Eq. (2.116) of Atkinson and Han [12] and $\mathbb{P}(|\iota| > |\iota_2|) \leq \delta$)

$$\leq \frac{\gamma_2^2}{16} + \delta^2 + O\left(\frac{1}{d}\right)$$
(By Eq. (C.4) and b.c. $|w_t(\iota)| \leq \delta$ if $|\iota| \leq |\iota_2|$ by Proposition D.37)

$$\leq \frac{\tau^4}{8} + O\left(\frac{1}{d}\right)$$
(B.c. $\delta \leq \frac{\gamma_2^2}{16}$ and $\gamma_2 = \tau^2$)

Thus, writing $\gamma_2 = \tau^2$, we have

$$|D_{2,t}| = |\mathop{\mathbb{E}}_{w \sim \rho}[P_{2,d}(w)] - \tau^2| \geq \left|\frac{\tau^4}{8} + O\left(\frac{1}{d}\right) - \tau^2\right| \geq \frac{\tau^2}{2} \tag{D.50}$$

where the last inequality is by Assumption 3.2 b.c. $d \geq c_3$ and $\gamma_4 \lesssim \gamma_2$. By a similar argument,

$$|D_{4,t}| = |\mathop{\mathbb{E}}_{w \sim \rho}[P_{4,d}(w)] - \beta\tau^4| \tag{D.51}$$

$$\leq \beta\tau^4 + \frac{\gamma_2^2}{16} + \delta^4 + O\left(\frac{1}{d}\right)$$
(Using $|P_{4,d}(w)| \leq 1$, $\mathbb{P}(|\iota| > |\iota_2|) \leq \frac{\gamma_2^2}{16}$, $\delta \leq \frac{\tau^4}{16}$ and Eq. (C.4))

$$\leq (\beta + 1/8)\tau^4$$
(B.c. $\delta \leq \frac{\gamma_2^2}{16} \leq \frac{\beta\tau^4}{16}$)

$$\leq |D_{2,t}| \tag{D.52}$$

Thus, for any time $t \in [t_0, t_1]$, for any $\iota \in [\iota_1, \iota_2]$, by Lemma 4.4 (and because $w_t(\iota) \leq \delta$ for any $\iota \in [\iota_1, \iota_2]$ by Proposition D.37), we have

$$v_t(w_t(\iota)) = -(1 - w_t(\iota)^2) \cdot \left(2\hat{\sigma}_{2,d}^2 D_{2,t} w_t(\iota) \cdot (1 \pm O(\delta)) + Q_t(w_t(\iota))\right)$$
(B.c. $\hat{\sigma}_{4,d}^2 \lesssim \hat{\sigma}_{2,d}^2$ by Assumption 3.2, $|D_{4,t}| \leq |D_{2,t}|$, and $w_t(\iota) \leq \delta$)

$$= 2\hat{\sigma}_{2,d}^2 |D_{2,t}| w_t(\iota) \cdot (1 \pm O(\delta)) - O\left(\frac{\hat{\sigma}_{2,d}^2 |D_{2,t}|}{d}\right) |w_t(\iota)|$$
(B.c. $|w_t(\iota)| \leq \delta$ and by bound on coefficients of $Q_t$ from Lemma 4.4)

$$= 2\hat{\sigma}_{2,d}^2 |D_{2,t}| w_t(\iota) \cdot (1 \pm O(\delta))$$
(B.c. $\delta \geq \frac{1}{d}$)

In summary,

$$v_t(w_t(\iota)) = 2\hat{\sigma}_{2,d}^2 |D_{2,t}| w_t(\iota) \cdot (1 \pm O(\delta)) \tag{D.53}$$

Thus,

$$\frac{v(w_t(\iota_2))}{w_t(\iota_2)} \leq \frac{1 + O(\delta)}{1 - O(\delta)} \frac{v(w_t(\iota_1))}{w_t(\iota_1)} \leq (1 + O(\delta)) \cdot \frac{v(w_t(\iota_1))}{w_t(\iota_1)} \tag{D.54}$$

and $w_t(\iota_2)$ grows by a factor

$$F_2 = \exp\left(\int_{t_0}^{t_1} \frac{v(w_t(\iota_2))}{w_t(\iota_2)} dt\right) \leq \exp\left((1 + O(\delta)) \int_{t_0}^{t_1} \frac{v(w_t(\iota_1))}{w_t(\iota_1)} dt\right) = F_1^{1+O(\delta)} \tag{D.55}$$

Rearranging gives

$$F_1 \geq F_2^{1-O(\delta)} = \frac{F_2}{F_2^{O(\delta)}} \tag{D.56}$$

Finally, observe that $F_2^{O(\delta)} \leq O(1)$, since by the assumption that $\delta \leq \frac{1}{\log F_2}$,

$$F_2^{\delta} \leq F_2^{\frac{1}{\log F_2}} \tag{D.57}$$

and for any $x > 0$, $x^{\frac{1}{\ln x}} = e$, meaning that $F_2^{\delta} \leq O(1)$. This implies that $F_1 \gtrsim F_2$. From the above calculations, we can also obtain

$$\exp\left(\int_{t_0}^{t_1} 2\hat{\sigma}_{2,d}^2 |D_{2,t}| dt\right) \gtrsim \exp\left(\int_{t_0}^{t_1} \frac{1}{1 + O(\delta)} \cdot \frac{v_t(w_t(\iota_2))}{w_t(\iota_2)} dt\right) \quad \text{(By Eq. (D.53))}$$

$$\gtrsim \exp\left((1 - O(\delta)) \cdot \int_{t_0}^{t_1} \frac{v_t(w_t(\iota_2))}{w_t(\iota_2)} dt\right) \tag{D.58}$$

$$\gtrsim F_2^{1-O(\delta)} \tag{D.59}$$

$$\gtrsim F_2 \quad \text{(By Eq. (D.57))}$$

as desired. Similarly, we can obtain

$$\exp\left(\int_{t_0}^{t_1} 2\hat{\sigma}_{2,d}^2 |D_{2,t}| dt\right) \lesssim \exp\left(\int_{t_0}^{t_1} (1 + O(\delta)) \cdot \frac{v_t(w_t(\iota_2))}{w_t(\iota_2)} dt\right) \tag{D.60}$$

$$\lesssim F_2^{1+O(\delta)} \tag{D.61}$$

$$\lesssim F_2 \quad \text{(B.c. } \delta \leq \frac{1}{\log F_2}\text{)}$$

Finally,

$$F_1 \lesssim \exp\left(\int_{t_0}^{t_1} (1 + O(\delta)) \cdot \frac{v_t(w_t(\iota_2))}{w_t(\iota_2)} dt\right) \tag{D.62}$$

$$\lesssim F_2^{1+O(\delta)} \tag{D.63}$$

$$\lesssim F_2 \quad \text{(B.c. } \delta \leq \frac{1}{\log F_2}\text{)}$$

as desired. Finally, we show the running time bound. Since $\delta \leq \frac{\gamma_2^2}{16}$ and $\gamma_2$ is less than a sufficiently small universal constant by Assumption 3.2, for any time $t$ during this interval, $v(w_t(\iota_2)) \gtrsim \hat{\sigma}_{2,d}^2 D_2(t) w_t(\iota_2) \gtrsim \hat{\sigma}_{2,d}^2 \tau^2 w_t(\iota_2)$ (where the second inequality is by Eq. (D.50)). Thus, by Gronwall's inequality (Fact I.13), the time elapsed when $w_t(\iota_2)$ grows by a factor of $F_2$ is at most $\frac{1}{\hat{\sigma}_{2,d}^2 \tau^2} \log F_2$. $\qquad\square$

### D.3.1 Summary of Phase 1

As a direct consequence of the above lemma, we can show Lemma D.5, which characterizes how fast the particles $w_t$ grow during Phase 1.

**Lemma D.5** (Phase 1). *Suppose we are in the setting of Theorem 3.3. At the end of phase 1, i.e. at time $T_1$, for all $\iota \in (0, \iota_U)$, we have $w_{T_1}(\iota) \asymp \frac{\sqrt{d}}{(\log d)^2} \iota$. Furthermore, $T_1 \lesssim \frac{1}{\hat{\sigma}_{2,d}^2 \gamma_2} \log d$. Additionally, we have $\exp\left(\int_0^{T_1} 2\hat{\sigma}_{2,d}^2 |D_{2,t}| dt\right) \asymp \frac{\sqrt{d}}{(\log d)^2}$.*

*Proof of Lemma D.5.* To show the lower bound on $w_{T_1}(\iota)$, we apply Lemma D.4, with $t_0 = 0$ and $t_1 = T_1$, with $\delta = \frac{1}{\log d}$, and with $\iota_1 = \iota$ and $\iota_2 = \iota_U$. Let us verify that the assumptions of Lemma D.4 hold:

- Observe that $\frac{w_{T_1}(\iota_{\mathrm{U}})}{\iota_{\mathrm{U}}} = \frac{1/\log d}{\log d/\sqrt{d}} = \frac{\sqrt{d}}{(\log d)^2}$, meaning that $\log F_2 \le \log d$ and $\frac{1}{\log F_2} \ge \delta$.

- By Assumption 3.2, $\delta = \frac{1}{\log d} \le \frac{\gamma_2^2}{16}$, since $d \ge c_3$.

- The assumption that $\mathbb{P}(|\iota| \ge \iota_{\mathrm{U}}) \le \frac{\gamma_2^2}{16}$ holds by a standard bound on the tail of $\mu_d$.

Using the notation of Lemma D.4,

$$F_2 = \frac{w_{T_1}(\iota_{\mathrm{U}})}{\iota_{\mathrm{U}}} = \frac{w_{\max}}{(\log d)/\sqrt{d}} = \frac{\sqrt{d}}{(\log d)^2} \tag{D.64}$$

where $w_{\max}$ is as defined in Definition 4.5. By Eq. (D.46), we therefore know that

$$\frac{\sqrt{d}}{(\log d)^2} \gtrsim \frac{w_{T_1}(\iota)}{\iota} \gtrsim \frac{\sqrt{d}}{(\log d)^2} \tag{D.65}$$

For the bound on the running time, observe that by Eq. (D.48), $T_1 \lesssim \frac{1}{\hat{\sigma}_{2,d}^2 \gamma_2} \log \frac{\sqrt{d}}{(\log d)^2}$. The final statement holds because $F_2 = \frac{\sqrt{d}}{(\log d)^2}$ and from the second statement of Lemma D.4. This completes the proof. $\square$

## D.4 Analysis of Phase 2

Recall the formal definition of Phase 2 from Definition 4.6. Our analysis in this section has two main goals: (1) to show that the running time of Phase 2 is small, and (2) to show that a majority of the particles become at least $\frac{1}{\log \log d}$ in magnitude, up to a constant factor. We will achieve goal (2) using Lemma D.4 — then, we show that goal (1) holds using goal (2). Additionally, goal (2) will be useful for the subsequent phases as well.

First, the following proposition defines a range of particles which will be relevant during Phase 2 and Phase 3. This range of particles is a strict subset of the interval $(0, \iota_{\mathrm{U}})$ which we considered during Phase 1 — we will show that this new range of particles grows uniformly during the beginning of Phase 2, until the largest of them becomes about $\frac{1}{\log \log d}$ in magnitude. Specifically, we define $\iota_{\mathrm{L}} \triangleq \frac{\kappa}{\sqrt{d}}$ and $\iota_{\mathrm{R}} \triangleq \frac{1}{\kappa\sqrt{d}}$ where $\kappa \asymp \frac{1}{\log \log d}$. The following proposition states that at initialization, a very large fraction of the $\iota$'s have magnitude between $\iota_{\mathrm{L}}$ and $\iota_{\mathrm{R}}$.

**Proposition D.6.** *Let $\kappa \in (0, 1)$, and suppose $\iota$ is drawn uniformly at random from $\mu_d$. Define $\iota_{\mathrm{L}} = \frac{\kappa}{\sqrt{d}}$ and $\iota_{\mathrm{R}} = \frac{1}{\kappa\sqrt{d}}$. Then, $\mathbb{P}(|\iota| \notin [\iota_{\mathrm{L}}, \iota_{\mathrm{R}}]) \lesssim \kappa$.*

*Proof of Proposition D.6.* First, we upper bound the probability that $|\iota| \ge \iota_{\mathrm{R}}$. By Markov's inequality,

$$\mathbb{P}(|\iota| \ge \iota_{\mathrm{R}}) \le \frac{\mathbb{E}[\iota^2]}{\iota_{\mathrm{R}}^2} = \frac{1/d}{1/(\kappa^2 d)} = \kappa^2 \tag{D.66}$$

Now, we upper bound the probability that $|\iota| \le \iota_{\mathrm{L}}$. By Eqs. (1.16) and (1.18) of Atkinson and Han [12], the probability density function of $\iota$ is $p(\iota) = \frac{\Gamma(d/2)}{\sqrt{\pi}\Gamma((d-1)/2)}(1 - \iota^2)^{\frac{d-3}{2}}$. Thus, we can simply upper bound the density by its maximum value:

$$\mathbb{P}(|\iota| \le \iota_{\mathrm{L}}) \lesssim \frac{\Gamma(d/2)}{\sqrt{\pi}\Gamma((d-1)/2)} \int_{-\iota_{\mathrm{L}}}^{\iota_{\mathrm{L}}} (1 - t^2)^{\frac{d-3}{2}} \, dt \tag{D.67}$$

$$\lesssim \frac{\Gamma(d/2)}{\sqrt{\pi}\Gamma((d-1)/2)} \cdot 2\iota_{\mathrm{L}} \tag{D.68}$$

$$\lesssim \sqrt{d} \cdot \iota_{\mathrm{L}} \qquad \text{(By Stirling's Formula)}$$

$$\lesssim \sqrt{d} \cdot \frac{\kappa}{\sqrt{d}} \tag{D.69}$$

$$\lesssim \kappa \tag{D.70}$$

as desired. $\square$

Next, as a corollary of Lemma D.4, we show that for a portion of Phase 2, the particles initialized at $\iota \in [\iota_{\mathrm{L}}, \iota_{\mathrm{R}}]$ grow by a similar factor. Specifically, this is true until the largest of these particles is $\xi = \frac{1}{2 \log \log d}$ in magnitude.

**Lemma D.7** (Corollary of Lemma D.4). *Suppose we are in the setting of Theorem 3.3. Let $T_{\mathrm{R}} > T_1$ be the first time such that $w_{T_{\mathrm{R}}}(\iota_{\mathrm{R}}) = \xi = \frac{1}{2 \log \log d}$. Then, for all $\iota$ with $|\iota| \in [\iota_{\mathrm{L}}, \iota_{\mathrm{R}}]$, we have $|w_{T_{\mathrm{R}}}(\iota)| \gtrsim \xi \kappa^2$, and additionally, $\frac{w_{T_{\mathrm{R}}}(\iota)}{w_{T_1}(\iota)} \asymp \xi \kappa (\log d)^2$. Furthermore, $T_{\mathrm{R}} - T_1 \lesssim \frac{1}{\hat{\sigma}_{2,d}^2 \gamma_2} \log \log d$.*

*Proof of Lemma D.7.* By Lemma D.5, $w_{T_1}(\iota_{\mathrm{R}}) \asymp \frac{\sqrt{d}}{(\log d)^2} \iota_{\mathrm{R}} = \frac{1}{\kappa (\log d)^2}$. Let $\iota_{\mathrm{L}} = \frac{\kappa}{\sqrt{d}}$ and $\iota_{\mathrm{R}} = \frac{1}{\kappa \sqrt{d}}$, where $\kappa = \frac{1}{\log \log d}$. Now, we apply Lemma D.4, with $t_0 = T_1$ and $t_1 = T_{\mathrm{R}}$, and with $\iota_1 = \iota$ for some $\iota \in [\iota_{\mathrm{L}}, \iota_{\mathrm{R}}]$ and $\iota_2 = \iota_{\mathrm{R}}$, and with $\delta = \frac{1}{2 \log \log d}$. Note that we assume without loss of generality that $\iota > 0$ — the statement for $\iota < 0$ follows by the symmetry of $\rho_t$. Let us verify that the assumptions hold:

- First, $\delta = \frac{1}{2 \log \log d} \gtrsim \frac{1}{\sqrt{d}}$. Addditionally, observe that by Lemma D.5, $\frac{w_{T_1}(\iota_{\mathrm{R}})}{\iota_{\mathrm{R}}} \asymp \frac{\sqrt{d}}{(\log d)^2}$, meaning that $w_{T_1}(\iota_{\mathrm{R}}) \asymp \frac{1}{\kappa (\log d)^2}$. Thus, using the notation of Lemma D.4, we have $F_2 \lesssim \frac{1/ \log \log d}{1/(\kappa (\log d)^2)} = \frac{\kappa (\log d)^2}{\log \log d}$, and $\log F_2 \leq 2 \log \log d - 2 \log \log \log d + C < \frac{1}{\delta}$, as desired. (Here $C$ is a universal constant, and the last inequality holds by Assumption 3.2 since $d \geq c_3$.)

- By Assumption 3.2, $\delta = \frac{1}{2 \log \log d} \lesssim \frac{\gamma_2^2}{16}$ since $d \geq c_3$.

- Additionally, the probability that $|\iota| \geq |\iota_{\mathrm{R}}|$ is at most $\kappa = \frac{1}{\log \log d}$ up to constant factors by Proposition D.6, and this is clearly at most $\frac{\gamma_2^2}{16}$ by Assumption 3.2.

Using Lemma D.4, we obtain

$$\frac{w_{T_{\mathrm{R}}}(\iota)}{w_{T_1}(\iota)} \asymp \frac{w_{T_{\mathrm{R}}}(\iota_{\mathrm{R}})}{w_{T_1}(\iota_{\mathrm{R}})} \asymp \frac{\xi}{w_{T_1}(\iota_{\mathrm{R}})} \asymp \frac{\xi}{\left( \frac{1}{\kappa (\log d)^2} \right)} \asymp \xi \kappa (\log d)^2 \tag{D.71}$$

Thus, using the conclusion of Lemma D.5 that $\frac{w_{T_1}(\iota)}{\iota} \gtrsim \frac{\sqrt{d}}{(\log d)^2}$, we obtain

$$w_{T_{\mathrm{R}}}(\iota) \gtrsim \frac{w_{T_{\mathrm{R}}}(\iota)}{w_{T_1}(\iota)} \cdot \frac{w_{T_1}(\iota)}{\iota} \cdot \iota \gtrsim \xi \kappa (\log d)^2 \cdot \frac{\sqrt{d}}{(\log d)^2} \cdot \frac{\kappa}{\sqrt{d}} \gtrsim \xi \kappa^2 \tag{D.72}$$

as desired. Thus, we obtain the first statement of this lemma. In addition, by Eq. (D.48), and since $w_{T_{\mathrm{R}}}(\iota_{\mathrm{R}}) = \xi$ and $w_{T_1}(\iota_{\mathrm{R}}) \gtrsim \frac{1}{\kappa (\log d)^2}$, the running time bound we obtain for this part of Phase 2 is $T_{\mathrm{R}} - T_1 \lesssim \frac{1}{\hat{\sigma}_{2,d}^2 \tau^2} \log \frac{\xi}{1/(\kappa (\log d)^2)} \lesssim \frac{1}{\hat{\sigma}_{2,d}^2 \tau^2} (\log \log d)$, as desired. $\square$

We now prove some auxiliary lemmas. First, we show an upper bound on $L(\rho_0)$, which is needed in order to later show an upper bound on the time required to have $L(\rho_t) \leq (\hat{\sigma}_{2,d}^2 + \hat{\sigma}_{4,d}^2)\epsilon^2$. In the process, we also show a reasonably tight bound on the magnitude of $D_{2,t}$ and $D_{4,t}$ during the first part of Phase 2.

**Lemma D.8.** *Suppose we are in the setting of Theorem 3.3 and Lemma D.7, and suppose $0 \leq t \leq T_{\mathrm{R}}$. Then, $-\frac{3\gamma_2}{2} \leq D_{2,t} \leq -\frac{\gamma_2}{2}$ and $-\frac{3\gamma_4}{2} \leq D_{4,t} \leq -\frac{\gamma_4}{2}$. As a corollary, for all $t \geq 0$, $L(\rho_t) \lesssim (\hat{\sigma}_{2,d}^2 + \hat{\sigma}_{4,d}^2)\gamma_2^2$.*

*Proof.* Since $t \leq T_{\mathrm{R}}$, $w_t(\iota_{\mathrm{R}}) \leq \xi = \frac{1}{2 \log \log d}$ (because for $w \gtrsim \frac{1}{\sqrt{d}}$, we have $\frac{dw}{dt} \geq 0$ by Lemma 4.4). By Proposition D.37, for all $\iota$ with $|\iota| \leq \iota_{\mathrm{R}}$, $|w_t(\iota)| \leq \xi$. Thus, by Proposition D.6,

$$\mathbb{E}_{w \sim \rho_t}[w^2] \leq \mathbb{P}_{\iota \sim \rho_0}(|\iota| \leq \iota_{\mathrm{R}}) \cdot \mathbb{E}[w^2 \mid |\iota| \leq \iota_{\mathrm{R}}] + \mathbb{P}_{\iota \sim \rho_0}(|\iota| \geq \iota_{\mathrm{R}}) \cdot \mathbb{E}[w^2 \mid |\iota| \geq \iota_{\mathrm{R}}] \tag{D.73}$$

$$\leq \xi^2 + O(\kappa) \tag{By Proposition D.6}$$

Thus, $D_{2,t} = O(\xi^2 + \kappa) - \gamma_2$, meaning that $-\frac{3\gamma_2}{2} \leq D_2 \leq -\frac{\gamma_2}{2}$, and similarly, $-\frac{3\gamma_4}{2} \leq D_4 \leq -\frac{\gamma_4}{2}$. The corollary follows from the fact that $\gamma_4 \lesssim \gamma_2^2$ by Assumption 3.2. $\square$

Next, we show that the particles are in fact increasing in magnitude during Phases 1 and 2. This makes use of our bound on the magnitudes of $D_{2,t}$ and $D_{4,t}$ up to time $T_R$.

**Lemma D.9.** *Suppose we are in the setting of Theorem 3.3, and suppose $[t_0, t_1]$ is a time interval with $0 \leq t_0 < t_1 \leq T_2$. Then, for all $\iota$ with $|\iota| \in [\iota_L, \iota_R]$, $|w_t(\iota)|$ is increasing on $[t_0, t_1]$.*

*Proof.* Recall that by Lemma 4.4,

$$v_t(w) = -(1 - w^2)(P_t(w) + Q_t(w)) \tag{D.74}$$

and for $w \geq 0$, it suffices to show that $P_t(w) + Q_t(w) \leq 0$. For $t \leq T_R$,

$$P_t(w) + Q_t(w) \leq 2\hat{\sigma}_{2,d}^2 D_{2,t} w + 4\hat{\sigma}_{4,d}^2 D_{4,t} w^3 + O\left(\frac{\hat{\sigma}_{2,d}^2 |D_{2,t}| + \hat{\sigma}_{4,d}^2 |D_{4,t}|}{d}\right)|w| \tag{D.75}$$

$$\leq -\hat{\sigma}_{2,d}^2 \gamma_2 w + O\left(\frac{\hat{\sigma}_{2,d}^2 |D_{2,t}| + \hat{\sigma}_{4,d}^2 |D_{4,t}|}{d}\right)|w|$$

(By Lemma D.8, b.c. $D_{4,t} \leq 0$)

$$\leq 0 \qquad \text{(B.c. } \hat{\sigma}_{4,d}^2 \lesssim \hat{\sigma}_{2,d}^2 \text{ and } \gamma_2 \gtrsim \frac{|D_{2,t}|}{d} \text{ and } \gamma_4 \gtrsim \frac{|D_{4,t}|}{d} \text{ (Assumption 3.2))}$$

For $t \geq T_R$, suppose $w \gtrsim \xi \kappa^2$ (which holds at time $T_R$ for $\iota \in [\iota_L, \iota_R]$ by Lemma D.7). Then,

$$P_t(w) + Q_t(w) \leq 2\hat{\sigma}_{2,d}^2 D_{2,t} w + 4\hat{\sigma}_{4,d}^2 D_{4,t} w^3 + O\left(\frac{\hat{\sigma}_{2,d}^2 |D_{2,t}| + \hat{\sigma}_{4,d}^2 |D_{4,t}|}{d}\right)|w|$$

(By Lemma 4.4)

$$\leq \hat{\sigma}_{2,d}^2 D_{2,t}\left(2w - \frac{Cw}{d}\right) + \hat{\sigma}_{4,d}^2 D_{4,t}\left(4w^3 - \frac{Cw}{d}\right) \tag{D.76}$$

for some universal constant $C > 0$. The final expression on the right-hand side is negative — to see this, note that $2w - \frac{w}{d} > 0$ if $w \gtrsim \xi \kappa^2$, and $4w^3 - \frac{w}{d} > 0$ if $w \gtrsim \xi \kappa^2$ since $d \geq c_3$ by Assumption 3.2. This completes the proof of the lemma when $\iota \geq 0$. For $\iota < 0$, the lemma holds by the symmetry of $\rho_t$. $\qquad\square$

### D.4.1 Summary of Phase 2

Finally, we show Lemma D.10 which concludes the analysis of Phase 2. For times $t \geq T_R$ within Phase 2, the particles with $|\iota| \in [\iota_L, \iota_R]$ satisfy $|w_t(\iota)| \gtrsim \xi \kappa^2$. Using this observation, and the fact that $D_{2,t}$ and $D_{4,t}$ have the same sign (meaning that the two terms of $P_t(w) = 2\hat{\sigma}_{2,d}^2 D_{2,t} w + 4\hat{\sigma}_{4,d}^2 D_{4,t} w^3$ do not cancel) we show that the velocity $v(w)$ is large on average. By Lemma D.39, this allows us to show that the loss decreases quickly at any time $t \geq T_R$ in Phase 2, meaning that either the loss goes below $(\hat{\sigma}_{2,d}^2 + \hat{\sigma}_{4,d}^2)\epsilon^2$ quickly, or we exit Phase 2 quickly.

**Lemma D.10** (Phase 2 Summary). *Suppose we are in the setting of Theorem 3.3. Let $T_2$ be the minimum time such that either $D_{2,T_2} = 0$ or $D_{4,T_2} = 0$, i.e. $T_2$ is the end of phase 2. Then, either $T_2 - T_1 \lesssim \frac{(\log\log d)^{18}}{(\hat{\sigma}_{2,d}^2 + \hat{\sigma}_{4,d}^2)} \log\left(\frac{\gamma_2}{\epsilon}\right)$ or $T_{*,\epsilon} - T_1 \lesssim \frac{(\log\log d)^{18}}{(\hat{\sigma}_{2,d}^2 + \hat{\sigma}_{4,d}^2)} \log\left(\frac{\gamma_2}{\epsilon}\right)$.*

*Proof of Lemma D.10.* First, let $T_R$ be as defined in Lemma D.7. By Lemma D.7, $T_R - T_1 \lesssim \frac{1}{\hat{\sigma}_{2,d}^2 \gamma_2} \log\log d$. Furthermore, at time $T_R$, for all $\iota$ such that $|\iota| \in [\iota_L, \iota_R]$, $|w_t(\iota)| \gtrsim \xi \kappa^2$.

Our proof strategy is now to show that $v(w)$ is large on average, and then apply Lemma D.39. At any time $t \in [T_R, T_2]$, since $D_{2,t} \leq 0$, Proposition D.38 implies that $\mathbb{E}_{w \sim \rho_t}[w^2] \leq \gamma_2 + O\left(\frac{1}{d}\right)$. Thus, by Markov's inequality, and because $d \geq c_3$ (where $c_3$ is defined in Assumption 3.2),

$$\mathbb{P}_{w \sim \rho_t}(|w| \geq 1/2) \leq 4 \cdot \left(\gamma_2 + \frac{1}{d}\right) \leq 5\tau^2 \tag{D.77}$$

For all $\iota$ with $|\iota| \in [\iota_L, \iota_R]$, $|w_t(\iota)|$ is increasing as a function of $t$ on $[0, T_2]$, by Lemma D.9. Thus, by Proposition D.6 and a union bound, $\mathbb{P}_{w \sim \rho_t}(\xi \kappa^2 \leq |w_t(\iota)| \leq 1/2) \geq 1 - O(\kappa) - 5\gamma_2 \geq \frac{1}{2}$ for all $t \geq T_R$ and all $\iota$ with $|\iota| \in [\iota_L, \iota_R]$. (Note that $1 - O(\kappa) - 5\gamma_2 \geq \frac{1}{2}$ holds because $\gamma_2$ is less than a

sufficiently small constant by Assumption 3.2.) Finally, for all $w$ satisfying $\xi\kappa^2 \le |w| \le \frac{1}{2}$, using Lemma 4.4 we can compute that

$$|v_t(w)| \ge (1 - w^2) \cdot (|P_t(w)| - |Q_t(w)|) \qquad \text{(By Lemma 4.4)}$$

$$\ge (1 - w^2) \cdot \left|2\hat{\sigma}_{2,d}^2|D_{2,t}||w| + 4\hat{\sigma}_{4,d}^2|D_{4,t}||w^3|\right| - O\left(\frac{\hat{\sigma}_{2,d}^2|D_{2,t}| + \hat{\sigma}_{4,d}^2|D_{4,t}|}{d}\right)|w| \qquad \text{(By Lemma 4.4)}$$

$$\ge \frac{3}{4} \cdot (2\hat{\sigma}_{2,d}^2|D_{2,t}|\xi\kappa^2 + 4\hat{\sigma}_{4,d}^2|D_{4,t}| \cdot \xi^3\kappa^6) - O\left(\frac{\hat{\sigma}_{2,d}^2|D_{2,t}| + \hat{\sigma}_{4,d}^2|D_{4,t}|}{d}\right)|w| \qquad \text{(B.c. } \xi\kappa^2 \le |w| \le \frac{1}{2})$$

$$\ge \frac{1}{2} \cdot (2\hat{\sigma}_{2,d}^2|D_{2,t}|\xi\kappa^2 + 4\hat{\sigma}_{4,d}^2|D_{4,t}| \cdot \xi^3\kappa^6) \qquad \text{(B.c. } d \text{ is sufficiently large (Assumption 3.2))}$$

Thus, by Lemma D.39,

$$\frac{dL(\rho_t)}{dt} \lesssim -(\hat{\sigma}_{2,d}^2|D_{2,t}|\xi\kappa^2 + \hat{\sigma}_{4,d}^2|D_{4,t}|\xi^3\kappa^6)^2 \tag{D.78}$$

$$\lesssim -\xi^6\kappa^{12}(\hat{\sigma}_{2,d}^2|D_{2,t}| + \hat{\sigma}_{4,d}^2|D_{4,t}|)^2 \tag{D.79}$$

$$\lesssim -\xi^6\kappa^{12}(\hat{\sigma}_{2,d}^2 + \hat{\sigma}_{4,d}^2)(\hat{\sigma}_{2,d}^2|D_{2,t}| + \hat{\sigma}_{4,d}^2|D_{4,t}|) \cdot (|D_{2,t}| + |D_{4,t}|) \tag{D.80}$$

$$\lesssim -\xi^6\kappa^{12}(\hat{\sigma}_{2,d}^2 + \hat{\sigma}_{4,d}^2)(\hat{\sigma}_{2,d}^2|D_{2,t}|^2 + \hat{\sigma}_{4,d}^2|D_{4,t}|^2) \tag{D.81}$$

$$\lesssim -\xi^6\kappa^{12}(\hat{\sigma}_{2,d}^2 + \hat{\sigma}_{4,d}^2)L \qquad \text{(By Lemma 4.2)} \tag{D.82}$$

Thus, since $L(\rho_{T_R}) \lesssim (\hat{\sigma}_{2,d}^2 + \hat{\sigma}_{4,d}^2)\gamma_2$ (by Lemma D.8), by Gronwall's inequality (Fact I.13), we know that for $t \in [T_R, T_2]$,

$$L(\rho_t) \le L(\rho_{T_R}) \cdot \exp\left(\int_{T_R}^t -\xi^6\kappa^{12}(\hat{\sigma}_{2,d}^2 + \hat{\sigma}_{4,d}^2)ds\right) \tag{D.82}$$

$$= L(\rho_{T_R})e^{-(t-T_R)\xi^6\kappa^{12}(\hat{\sigma}_{2,d}^2+\hat{\sigma}_{4,d}^2)} \tag{D.83}$$

Thus, if $L(\rho_t) \gtrsim (\hat{\sigma}_{2,d}^2 + \hat{\sigma}_{4,d}^2)\epsilon^2$, this implies that

$$e^{-(t-T_R)\xi^6\kappa^{12}(\hat{\sigma}_{2,d}^2+\hat{\sigma}_{4,d}^2)} \gtrsim \frac{\epsilon^2}{\gamma_2} \tag{D.84}$$

and rearranging gives $t - T_R \le \frac{1}{(\hat{\sigma}_{2,d}^2+\hat{\sigma}_{4,d}^2)\xi^6\kappa^{12}}\log(\frac{\gamma_2}{\epsilon^2})$. Thus, either $T_2 - T_R \le \frac{1}{(\hat{\sigma}_{2,d}^2+\hat{\sigma}_{4,d}^2)\xi^6\kappa^{12}}\log(\frac{\gamma_2}{\epsilon^2})$, or there exists a time $t \in [T_R, T_2]$ such that $L(\rho_t) \lesssim (\hat{\sigma}_{2,d}^2 + \hat{\sigma}_{4,d}^2)\epsilon^2$, and such that $t - T_R \le \frac{1}{(\hat{\sigma}_{2,d}^2+\hat{\sigma}_{4,d}^2)\xi^6\kappa^{12}}\log(\frac{\gamma_2}{\epsilon})$. This completes the proof of the lemma. $\square$

## D.5 Phase 3, Case 1

Phase 3, Case 1 is the case where $D_{2,T_2} = 0$ and $D_{4,T_2} < 0$, i.e. the case where $D_{2,t}$ reaches 0 first. As we later show in Lemma D.19 and Lemma D.20, if this occurs, then for all $t \ge T_2$, it will be the case that $D_{2,t} \ge 0$ and $D_{4,t} \le 0$. We now outline our analysis of this case — the goal of our analysis is to show that either $|D_{2,t}| \le \epsilon$ or $|D_{4,t}| \le \epsilon$ holds within $\frac{1}{\hat{\sigma}_{2,d}^2+\hat{\sigma}_{4,d}^2} \cdot \text{poly}\left(\frac{\log\log d}{\epsilon}\right)$ time after the start of Phase 3, Case 1.

**Proof Strategy for Phase 3, Case 1** Our overall proof strategy in this section is to show that $|v(w)|$ is large for a large portion of particles $w$, and then to apply Lemma D.39 to show that the loss decreases quickly. Recall that

$$v_t(w) = -(1 - w^2)(P_t(w) + Q_t(w)) \tag{D.85}$$

and intuitively, we can ignore $Q_t(w)$ due to the factor of $\frac{1}{d}$ in its coefficients. Thus, we can write

$$v_t(w) \approx -(1 - w^2)P_t(w) \tag{D.86}$$

$$\approx -(1 - w^2) \cdot w \cdot (2\hat{\sigma}_{2,d}^2 D_{2,t} + 4\hat{\sigma}_{4,d}^2 D_{4,t}w^2). \tag{D.87}$$

Since $D_{4,t} < 0$ and $D_{2,t} > 0$, we can further rewrite the above as

$$v_t(w) \approx -(1 - w^2) \cdot w \cdot 4\hat{\sigma}_{4,d}^2 |D_{4,t}|(w - r)(w + r) \tag{D.88}$$

where $r = \sqrt{-\frac{\hat{\sigma}_{2,d}^2 D_{2,t}}{2\hat{\sigma}_{4,d}^2 D_{4,t}}}$ is the positive root of the last factor $(2\hat{\sigma}_{2,d}^2 D_{2,t} + 4\hat{\sigma}_{4,d}^2 D_{4,t}w^2)$. Thus, to show that the velocity $v_T(w)$ is large, we want to show that there is a large fraction of particles $w \in [0, 1]$ simultaneously satisfying the following: (1) $w$ is far from 1, (2) $w$ is far from 0, and (3) $w$ is far from $r$. Indeed, in Lemma D.17, we show that out of the particles $w \in [0, 1]$, at least a $\frac{\gamma_2}{2}$ fraction of these particles is in $[\gamma_2^{3/4}, 1 - \gamma_2^{1/2}]$, enabling us to bound the distance of these particles $w$ from 0 and 1.

Thus, to show that $v_t(w)$ is large for these particles, it suffices to show that $|w - r|$ is large on average for the particles $w \in [\gamma_2^{3/4}, 1 - \gamma_2^{1/2}]$, i.e. the conditional expectation $\mathbb{E}_{w \sim \rho_t}[(w - r)^2 \mid w \in [\gamma_2^{3/4}, 1 - \gamma_2^{1/2}]]$ is large. In turn, this is at least the conditional variance of $w$ conditioned on $w$ being in $[\gamma_2^{3/4}, 1 - \gamma_2^{1/2}]$, so it suffices to show that $\mathrm{Var}(w \mid w \in [\gamma_2^{3/4}, 1 - \gamma_2^{1/2}])$ is large, and by Proposition D.40 it suffices to show that $\mathbb{E}[(w - w')^2 \mid w, w' \in [\gamma_2^{3/4}, 1 - \gamma_2^{1/2}]]$ is large.

To show that the quantity $\mathbb{E}[(w - w')^2 \mid w, w' \in [\gamma_2^{3/4}, 1 - \gamma_2^{1/2}]]$ is large, intuitively we would like to show that during Phase 3, Case 1, $w$ and $w'$ are getting farther apart. However, this is not true. If we write the velocity as $v_t(w) = (1 - w^2) \cdot w \cdot (4\hat{\sigma}_{4,d}^2 |D_{4,t}|w^2 - 2\hat{\sigma}_{2,d}^2 |D_{2,t}|)$, then the last factor is indeed increasing as a function of $w \in [0, 1]$, and thus "pushes" the different particles $w, w'$ further apart. This does not work as a formal argument, because as two particles $w, w'$ become closer to 0 or 1, their distance $|w - w'|$ will decrease again due to the factors $w$ and $(1 - w^2)$ in the velocity. However, we are only concerned with particles $w \in [\gamma_2^{3/4}, 1 - \gamma_2^{1/2}]$ — we must make precise the intuition that the behavior of the particles very close to 0 or 1 does not matter.

To make this proof strategy precise, we define the potential function $\Phi(w) = \log\left(\frac{w}{\sqrt{1-w^2}}\right)$ in Definition D.11 and show that $|\Phi(w) - \Phi(w')|$ is increasing in Phase 3, Case 1. As shown in Lemma D.12, we have $\frac{d}{dt}\Phi(w) \approx -2\hat{\sigma}_{2,d}^2 D_{2,t} - 4\hat{\sigma}_{4,d}^2 D_{4,t}w^2$ — this is simply because $\frac{d\Phi(w)}{dw} = \frac{1}{w(1-w^2)}$. In other words, $\frac{d}{dt}\Phi(w)$ is exactly $v(w)$, but without the factors $w$ and $(1 - w^2)$ — we have selected this potential function $\Phi(w)$ specifically to eliminate these factors. Using this observation, it is easy to show that $|\Phi(w) - \Phi(w')|$ is increasing and thus obtain a lower bound on $|\Phi(w) - \Phi(w')|$ for a large fraction of pairs $w, w'$. Finally, to convert this to a lower bound on $|w - w'|$, we use the fact that $\Phi$ is Lipschitz on the interval $[\gamma_2^{3/4}, 1 - \gamma_2^{1/2}]$ (Lemma D.14), which gives us $|w - w'| \geq \frac{1}{\gamma_2^{O(1)}}|\Phi(w) - \Phi(w')|$.

We now repeat the definition of the potential function:

**Definition D.11** (Potential Function $\Phi$). *We define $\Phi : [0, 1] \to (-\infty, \infty)$ by $\Phi(w) = \log\left(\frac{w}{\sqrt{1-w^2}}\right)$.*

We note that $\frac{d\Phi(w)}{dw} = \frac{1}{w(1-w^2)}$. Next, we note the following useful fact about the potential function (we note that this does not require any assumption that we are in Phase 3, Case 1, and just follows from algebraic manipulations):

**Lemma D.12.** *In the setting of Theorem 3.3,*

$$\frac{d}{dt}\Phi(w_t) = -2\hat{\sigma}_{2,d}^2 D_{2,t} - 4\hat{\sigma}_{4,d}^2 D_{4,t}w^2 - \lambda_d^{(1)} - \lambda_d^{(3)}w^2 \tag{D.89}$$

*where $\lambda_d^{(1)}$ and $\lambda_d^{(3)}$ are as in the statement of Lemma 4.4.*

*Proof of Lemma D.12.* For any $w_t$,

$$\frac{d}{dt}\Phi(w_t) = \frac{d\Phi}{dw} \cdot \frac{dw_t}{dt} = \frac{1}{w(1-w^2)} \cdot \frac{dw_t}{dt} \tag{D.90}$$

Thus, writing $\frac{dw_t}{dt} = -(1 - w^2) \cdot (P_t(w) + Q_t(w))$ where $P_t$ and $Q_t$ are defined in Lemma 4.4, we obtain

$$\frac{d}{dt}\Phi(w_t) = \frac{1}{w(1 - w^2)} \cdot \frac{dw_t}{dt} \tag{D.91}$$

$$= -\frac{1}{w} \cdot (P_t(w) + Q_t(w)) \tag{D.92}$$

$$= -\frac{P_t(w)}{w} - \frac{Q_t(w)}{w} \tag{D.93}$$

$$= -2\hat{\sigma}_{2,d}^2 D_{2,t} - 4\hat{\sigma}_{4,d}^2 D_{4,t}w^2 - \frac{Q_t(w)}{w} \tag{D.94}$$

$$= -2\hat{\sigma}_{2,d}^2 D_{2,t} - 4\hat{\sigma}_{4,d}^2 D_{4,t}w^2 - \lambda_d^{(1)} - \lambda_d^{(3)}w^2 \tag{D.95}$$

where $\lambda_d^{(1)}$ and $\lambda_d^{(3)}$ are as in the statement of Lemma 4.4. $\qquad\square$

Intuitively, when $D_2 > 0$ and $D_4 < 0$, the main part of the velocity is $P_t(w) = -2\hat{\sigma}_{2,d}^2 D_{2,t}w - 4\hat{\sigma}_{4,d}^2 D_{4,t}w^3$ which is an upward sloping cubic polynomial — thus, the particles $w$ move away from the positive root of this polynomial, and the distance between any pair $w, w'$ of particles is increasing provided that $w$ and $w'$ are not too close to $0$ or $1$. In the following few lemmas, we make this intuition formal using the potential function $\Phi$. We first show that for two particles $w, w'$, the distance between $\Phi(w)$ and $\Phi(w')$ is increasing as long as $D_{4,t} \leq 0$. Note that this condition holds not only during Phase 3, Case 1, but also during Phases 1 and 2.

**Lemma D.13** ($\Phi(w)$ and $\Phi(w')$ Moving Apart)**.** *In the setting of Theorem 3.3, suppose $D_{4,t} \leq 0$ for all $t$ in some time interval $[t_0, t_1]$, and let $\iota_1, \iota_2 > 0$. Then,*

$$\left| \Phi(w_t(\iota_1)) - \Phi(w_t(\iota_2)) \right| \tag{D.96}$$

*is increasing on the interval $[t_0, t_1]$.*

*Proof of Lemma D.13.* Suppose $\iota_1, \iota_2 \in [0, 1]$ such that $\iota_1 < \iota_2$ — by Proposition D.37, we will always have $w_t(\iota_1) \leq w_t(\iota_2)$. For convenience, in the rest of this proof, we will write $w_{1,t} = w_t(\iota_1)$ and $w_{2,t} = w_t(\iota_2)$. Since $\Phi$ is an increasing function of $w$, it will also always be the case that $\Phi(w_{2,t}) \geq \Phi(w_{1,t})$. Thus, to show that $|\Phi(w_{2,t}) - \Phi(w_{1,t})|$ is increasing, it suffices to show that $\Phi(w_{2,t}) - \Phi(w_{1,t})$ is increasing. By Lemma D.12,

$$\frac{d}{dt}\Big(\Phi(w_{2,t}) - \Phi(w_{1,t})\Big) = \Big( -2\hat{\sigma}_{2,d}^2 D_{2,t} - 4\hat{\sigma}_{4,d}^2 D_{4,t}w_{2,t}^2 - \lambda_d^{(1)} - \lambda_d^{(3)}w_{2,t}^2 \Big) \tag{D.97}$$

$$- \Big( -2\hat{\sigma}_{2,d}^2 D_{2,t} - 4\hat{\sigma}_{4,d}^2 D_{4,t}w_{1,t}^2 - \lambda_d^{(1)} - \lambda_d^{(3)}w_{1,t}^2 \Big) \tag{D.98}$$

$$= -4\hat{\sigma}_{4,d}^2 D_{4,t}(w_{2,t}^2 - w_{1,t}^2) - \lambda_d^{(3)}(w_{2,t}^2 - w_{1,t}^2) \tag{D.99}$$

$$= 4\hat{\sigma}_{4,d}^2 |D_{4,t}|(w_{2,t}^2 - w_{1,t}^2) - \lambda_d^{(3)}(w_{2,t}^2 - w_{1,t}^2) \tag{D.100}$$

where $\lambda_d^{(1)}$ and $\lambda_d^{(3)}$ are as in Lemma 4.4. Recall from the statement of Lemma 4.4 that $|\lambda_d^{(3)}| \lesssim \hat{\sigma}_{4,d}^2 |D_{4,t}| \cdot \frac{1}{d}$. Therefore,

$$\frac{d}{dt}\Big(\Phi(w_{2,t}) - \Phi(w_{1,t})\Big) = 4\hat{\sigma}_{4,d}^2 |D_{4,t}|(w_{2,t}^2 - w_{1,t}^2) - \lambda_d^{(3)}(w_{2,t}^2 - w_{1,t}^2) \tag{D.101}$$

$$\geq 3\hat{\sigma}_{4,d}^2 |D_{4,t}|(w_{2,t}^2 - w_{1,t}^2) \tag{D.102}$$

and the right hand side is nonnegative since $w_{2,t} \geq w_{1,t} \geq 0$. Thus, $\Phi(w_{2,t}) - \Phi(w_{1,t})$ is increasing, as desired. $\qquad\square$

Next, we prove a helpful lemma showing that the potential function is bi-Lipschitz. This is useful both when we show that $|\Phi(w) - \Phi(w')|$ is initially large (as it allows us to leverage an initial lower bound on $|w - w'|$) and when we show that $|w - w'|$ is large later during Phase 3, Case 1 (as it allows us to leverage the lower bound we obtain on $|\Phi(w) - \Phi(w')|$).

**Lemma D.14.** *For all $w, w' \in [0,1]$, $|\Phi(w) - \Phi(w')| \geq |w - w'|$. Additionally, for $\eta < \frac{1}{2}$ and $w, w' \in [\eta, 1 - \eta]$, $|\Phi(w) - \Phi(w')| \leq \frac{1}{\eta^2}|w - w'|$.*

*Proof of Lemma D.14.* First, for all $w \in [0,1]$, $\frac{d\Phi(w)}{dw} = \frac{1}{w(1-w^2)} > 1$, and the first statement of the lemma follows. For the second statement of the lemma, observe that for $w \in [\eta, 1 - \eta]$,

$$\left|\frac{d\Phi(w)}{dw}\right| \leq \frac{1}{w(1-w^2)} \leq \frac{1}{\eta(1-(1-\eta)^2)} = \frac{1}{\eta(1-(1-2\eta+\eta^2))} = \frac{1}{\eta(2\eta-\eta^2)} < \frac{1}{\eta^2} \tag{D.103}$$

where the last equality is because $\eta > \eta^2$ (so $2\eta - \eta^2 > \eta$). Thus, $\Phi$ is $\frac{1}{\eta^2}$-Lipschitz on the interval $[\eta, 1 - \eta]$. $\qquad\square$

We next need to show that the potential initially has a large value, at the beginning of Phase 3, Case 1. We show this in the next two lemmas, by showing that for any two particles $w, w'$, the distance between them grows by a large factor during Phase 2 (specifically, by time $T_R$) and then using this to obtain a lower bound on $|\Phi(w) - \Phi(w')|$ by the bi-Lipschitzness of $\Phi$.

**Lemma D.15.** *In the setting of Theorem 3.3, let $T_R$ be as defined in Lemma D.7. Then, for $\iota, \iota' \in [\iota_L, \iota_R]$,*

$$\frac{w_{T_R}(\iota') - w_{T_R}(\iota)}{\iota' - \iota} \gtrsim \xi\kappa\sqrt{d} \tag{D.104}$$

*Proof of Lemma D.15.* Let $\iota, \iota' \in [\iota_L, \iota_R]$, with $\iota < \iota'$. For convenience, we will write $w_t = w_t(\iota)$ and $w_t' = w_t(\iota')$. Our goal is to obtain a lower bound on the factor by which $w_t' - w_t$ grows by time $T_R$. First, define $P_t$ and $Q_t$ as in Lemma 4.4, and observe that

$$\frac{d}{dt}(w_t' - w_t) = -(1 - (w_t')^2)(P_t(w_t') + Q_t(w_t')) - (1 - w_t^2)(P_t(w_t) + Q_t(w_t)) \tag{D.105}$$

$$= -\Big(P_t(w_t') - P_t(w_t)\Big) + \Big((w_t')^2 P_t(w_t') - w_t^2 P_t(w_t)\Big) - \Big(Q_t(w_t') - Q_t(w_t)\Big) \tag{D.106}$$

$$+ \Big((w_t')^2 Q_t(w_t') - w_t^2 Q_t(w_t)\Big) \tag{D.107}$$

We first obtain a lower bound on the first term $-\Big(P_t(w_t') - P_t(w_t)\Big)$, and then show that the other terms are lower-order terms as long as $t \leq T_R$. For $t \in [0, T_R]$,

$$-(P_t(w_t') - P_t(w_t)) = -\Big(2\hat{\sigma}_{2,d}^2 D_{2,t} w_t' + 4\hat{\sigma}_{4,d}^2 D_{4,t}(w_t')^3\Big) + \Big(2\hat{\sigma}_{2,d}^2 D_{2,t} w_t + 4\hat{\sigma}_{4,d}^2 D_{4,t} w_t^3\Big) \tag{Lemma 4.4}$$

$$= -2\hat{\sigma}_{2,d}^2 D_{2,t}(w_t' - w_t) - 4\hat{\sigma}_{4,d}^2 D_{4,t}((w_t')^3 - w_t^3) \tag{D.108}$$

$$= 2\hat{\sigma}_{2,d}^2 |D_{2,t}|(w_t' - w_t) + 4\hat{\sigma}_{4,d}^2 |D_{4,t}|((w_t')^3 - w_t^3)$$
$$\text{(B.c. } t \leq T_R,\ D_{2,t}, D_{4,t} < 0 \text{ by Lemma D.8)}$$

$$= 2\hat{\sigma}_{2,d}^2 |D_{2,t}|(w_t' - w_t) + 4\hat{\sigma}_{4,d}^2 |D_{4,t}|((w_t')^2 + w_t w_t' + w_t^2)(w_t' - w_t) \tag{D.109}$$

$$\geq 2\hat{\sigma}_{2,d}^2 |D_{2,t}|(w_t' - w_t)$$
$$\text{(B.c. omitted term is nonnegative — } w_t' > w_t \text{ by Proposition D.37)}$$

Next, let us show that the remaining terms in Eq. (D.107) are lower-order terms which will not decrease the growth rate of $w_t' - w_t$ too much. First let us deal with the absolute value of the second

term in Eq. (D.107). For convenience, let $w_{R,t} = w_t(\iota_R)$. Then,

$$\left|(w_t')^2 P_t(w_t') - w_t^2 P_t(w_t)\right| \lesssim \left|(2\hat{\sigma}_{2,d}^2 D_{2,t}(w_t')^3 + 4\hat{\sigma}_{4,d}^2 D_{4,t}(w_t')^5)\right. \tag{D.110}$$

$$\left. - (2\hat{\sigma}_{4,d}^2 D_{2,t}(w_t)^3 + 4\hat{\sigma}_{4,d}^2 D_{4,t}(w_t)^5)\right| \tag{D.111}$$

$$\lesssim 2\hat{\sigma}_{2,d}^2 |D_{2,t}||(w_t')^3 - w_t^3| + 4\hat{\sigma}_{4,d}^2 |D_{4,t}||(w_t')^5 - w_t^5|$$
$$\text{(By Lemma 4.4)}$$

$$\lesssim 2\hat{\sigma}_{2,d}^2 |D_{2,t}||(w_t')^2 + w_t' w_t + w_t^2||w_t' - w_t| \tag{D.112}$$

$$+ 4\hat{\sigma}_{4,d}^2 |D_{4,t}||(w_t')^4 + (w_t')^3 w_t + (w_t')^2 w_t^2 + w_t' w_t^3 + w_t^4||w_t' - w_t| \tag{D.113}$$

$$\lesssim \hat{\sigma}_{2,d}^2 |D_{2,t}| w_{R,t}^2 |w_t' - w_t|$$
$$\text{(B.c. } |w_t|, |w_t'| \leq w_{R,t} \text{ by Proposition D.37)}$$

where the last inequality is also because $\hat{\sigma}_{4,d}^2 \lesssim \hat{\sigma}_{2,d}^2$, and $\gamma_4 \lesssim \gamma_2^2$ (Assumption 3.2) and $|D_2| \geq \frac{\gamma_2}{2}$ and $|D_4| \leq \frac{3\gamma_4}{2}$ for $t \leq T_R$ (Lemma D.8), meaning that the first term in Eq. (D.113) dominates up to universal constant factors. Next, let us deal with the absolute value of the third term in Eq. (D.107). By Lemma I.16,

$$|Q_t(w_t)' - Q_t(w_t)| \lesssim \frac{\hat{\sigma}_{2,d}^2 |D_{2,t}| + \hat{\sigma}_{4,d}^2 |D_{4,t}|}{d} \cdot |w_t' - w_t|$$
$$\text{(By definition of } \lambda_d^{(1)}, \lambda_d^{(3)}, \text{ Lemma 4.4)}$$

$$\lesssim \frac{\hat{\sigma}_{2,d}^2 |D_{2,t}|}{d} |w_t' - w_t|$$
$$\text{(B.c. } \hat{\sigma}_{4,d}^2 \lesssim \hat{\sigma}_{2,d}^2 \text{ and } |D_4| \lesssim \gamma_4 \lesssim \gamma_2^2 \lesssim |D_2| \text{ for } t \leq T_R \text{ (Lemma D.8))}$$

Using the same argument with Lemma I.16, we can bound the fourth term in Eq. (D.107):

$$|(w_t')^2 Q_t(w_t') - w_t^2 Q_t(w_t)| \lesssim \frac{\hat{\sigma}_{2,d}^2 |D_{2,t}|}{d} |w_t' - w_t| \tag{D.114}$$

Combining the bounds we obtained for all the terms of Eq. (D.107), we obtain

$$\frac{d}{dt}(w_t' - w_t) \geq 2\hat{\sigma}_{2,d}^2 |D_{2,t}|(w_t' - w_t) - O\left(\hat{\sigma}_{2,d}^2 |D_{2,t}|\left(w_{R,t}^2 + 1/d\right)(w_t' - w_t)\right) \tag{D.115}$$

$$= \left(2\hat{\sigma}_{2,d}^2 |D_{2,t}| - O\left(\hat{\sigma}_{2,d}^2 |D_{2,t}|(w_{R,t}^2 + 1/d)\right)\right)(w_t' - w_t) \tag{D.116}$$

and therefore,

$$\frac{\frac{d}{dt}(w_t' - w_t)}{w_t' - w_t} \geq 2\hat{\sigma}_{2,d}^2 |D_{2,t}| - O\left(\hat{\sigma}_{2,d}^2 |D_{2,t}|(w_{R,t}^2 + 1/d)\right) \tag{D.117}$$

First, let us consider how much $w_t' - w_t$ grows during Phase 1. At this point, we apply Lemma D.4 with $\iota_2 = \iota_U$ and $0 < \iota_1 < \iota_U$, with $\delta = \frac{1}{\log d}$. Let us first verify that the assumptions of Lemma D.4 are satisfied. By Lemma D.5, $F_2 \asymp \frac{\sqrt{d}}{(\log d)^2}$, using the notation of Lemma D.4. Thus, $\log F_2 \leq \log d$, meaning $\delta \leq \frac{1}{\log F_2}$. By Assumption 3.2, $\delta \leq \frac{\gamma_2^2}{16}$, and by Proposition I.5, $\mathbb{P}(|\iota| > \iota_U) \leq \frac{1}{\log d} \leq \frac{\gamma_2^2}{16}$ clearly holds. Thus, the assumptions of Lemma D.4 are satisfied. As a consequence of Lemma D.4 (specifically Eq. (D.47)) we obtain

$$\exp\left(\int_0^{T_1} 2\hat{\sigma}_{2,d}^2 |D_{2,t}| dt\right) \gtrsim F_2 \gtrsim \frac{\sqrt{d}}{(\log d)^2} \tag{D.118}$$

For $t \leq T_1$, we have $w_{R,t}^2 \leq \delta^2 \leq O(\frac{1}{\log d})$ by Definition 4.5 (and because $w_t(\iota_R) \leq w_t(\iota_U)$ by Proposition D.37). Thus,

$$\frac{w'_{T_1} - w_{T_1}}{w'_0 - w_0} \gtrsim \exp\left(\int_0^{T_1} 2\hat{\sigma}_{2,d}^2 |D_{2,t}| \cdot \left(1 - O\left(\frac{1}{\log d}\right)\right) dt\right) \qquad \text{(By Eq. (D.117))}$$

$$\gtrsim \left(\frac{\sqrt{d}}{(\log d)^2}\right)^{1 - O(1/\log d)} \qquad \text{(By Eq. (D.118))}$$

$$\gtrsim \frac{\sqrt{d}}{(\log d)^2} \qquad \text{(B.c. } d^{1/\log d} \leq O(1)\text{)}$$

Additionally, we apply Lemma D.4 with $\iota_2 = \iota_R$ (and an arbitrary $\iota_1 \in [\iota_L, \iota_R]$) with $\delta = \xi = \frac{1}{2\log\log d}$, and $t_0 = T_1$ and $t_1$ being the time $T_R$ such that $w_{T_R}(\iota_R) = \xi$. We verify that the conditions of Lemma D.4 hold. Using the notation of Lemma D.4, $F_2 = \frac{w_{T_R}(\iota_R)}{w_{T_1}(\iota_R)} = \xi\kappa(\log d)^2$ by Lemma D.7, and thus $\log F_2 \leq \frac{1}{\delta}$. Again by Assumption 3.2, $\delta \leq \frac{\gamma_2^2}{16}$, and by Proposition D.6, $\mathbb{P}(|\iota| > \iota_R) \leq O(\kappa) \leq \frac{\gamma_2^2}{16}$. Thus, we apply Lemma D.4 to obtain

$$\exp\left(\int_{T_1}^T 2\hat{\sigma}_{2,d}^2 |D_{2,t}| dt\right) \gtrsim F_2 \gtrsim \xi\kappa(\log d)^2 \qquad (D.119)$$

For $t \leq T_R$, $w_{R,t}^2 \leq \xi^2 \leq \delta^2$, meaning that

$$\frac{w'_{T_R} - w_{T_R}}{w'_{T_1} - w_{T_1}} \gtrsim \exp\left(\int_{T_1}^{T_R} 2\hat{\sigma}_{2,d}^2 |D_{2,t}| \cdot (1 - O(\delta^2)) dt\right) \qquad \text{(By Eq. (D.117))}$$

$$\gtrsim (\xi\kappa(\log d)^2)^{1 - O(\delta^2)} \qquad \text{(By Eq. (D.119))}$$

$$\gtrsim \xi\kappa(\log d)^2 \qquad \text{(B.c. } (\log d)^{O(\delta)} = (\log d)^{O(\frac{1}{\log\log d})} \leq O(1)\text{)}$$

In summary, since $w'_t - w_t$ grows by $\frac{\sqrt{d}}{(\log d)^2}$ (up to a constant factor) from time $0$ to time $T_1$, and by $\xi\kappa(\log d)^2$ (up to a constant factor) from time $T_1$ to time $T_R$, this means

$$\frac{w'_{T_R} - w_{T_R}}{w'_0 - w_0} \gtrsim \xi\kappa\sqrt{d} \qquad (D.120)$$

from time $0$ to time $T_R$, as desired. $\qquad\square$

We now use the previous lemma to obtain a lower bound on $|\Phi(w) - \Phi(w')|$ by using the bi-Lipschitzness of $\Phi$.

**Lemma D.16** (Initial Large Distance Between $w$ and $w'$). *Suppose we are in the setting of Theorem 3.3. Let $T_R$ be as in the statement of Lemma D.7. If $\iota, \iota'$ are sampled independently from $\rho_0$, then with probability at least $1 - O(\epsilon)$,*

$$|\Phi(w_{T_R}(\iota)) - \Phi(w_{T_R}(\iota'))| \gtrsim \xi\kappa^3 \qquad (D.121)$$

*Proof of Lemma D.16.* By Eqs. (1.16) and (1.18) of Atkinson and Han [12], the probability density function of $\iota$ is $p(\iota) = \frac{\Gamma(d/2)}{\sqrt{\pi}\Gamma((d-1)/2)}(1 - \iota^2)^{\frac{d-3}{2}}$. Thus, for $\iota \in [\iota_L, \iota_R]$, the conditional density function $p(\iota \mid \iota \in [\iota_L, \iota_R])$ can be upper bounded as

$$p(\iota \mid \iota \in [\iota_L, \iota_R]) \lesssim \frac{p(\iota)}{p(\iota \in [\iota_L, \iota_R])} \qquad (D.122)$$

$$\lesssim \frac{\Gamma(d/2)}{\sqrt{\pi}\Gamma((d-1)/2)} \cdot \frac{(1 - \iota^2)^{\frac{d-3}{2}}}{p(\iota \in [\iota_L, \iota_R])} \qquad (D.123)$$

$$\lesssim \frac{\Gamma(d/2)}{\sqrt{\pi}\Gamma((d-1)/2)} \cdot \frac{(1 - \iota^2)^{\frac{d-3}{2}}}{1 - O(\kappa)} \qquad \text{(By Proposition D.6)}$$

$$\lesssim \sqrt{d} \qquad \text{(By Stirling's Formula)}$$

In particular, for any interval $I$ of length at most $\frac{\kappa^2}{\sqrt{d}}$,

$$\mathbb{P}(\iota \in I \mid \iota \in [\iota_\mathrm{L}, \iota_\mathrm{R}]) \lesssim \sqrt{d} \cdot |I| \lesssim \kappa^2 \tag{D.124}$$

and from this it follows that

$$\mathbb{P}_{\iota,\iota' \sim \rho_0}\left(|\iota - \iota'| \leq \frac{\kappa^2}{\sqrt{d}} \mid \iota, \iota' \in [\iota_\mathrm{L}, \iota_\mathrm{R}]\right) \lesssim \kappa^2 \tag{D.125}$$

Next, suppose $\iota, \iota' \in [\iota_\mathrm{L}, \iota_\mathrm{R}]$ with $\iota' > \iota$, and $|\iota - \iota'| \geq \frac{\kappa^2}{\sqrt{d}}$. Then, by Lemma D.15, if $T_\mathrm{R}$ is the time $T$ mentioned in the statement of Lemma D.7, then

$$w_{T_\mathrm{R}}(\iota') - w_{T_\mathrm{R}}(\iota) \gtrsim \xi\kappa^3 \tag{D.126}$$

Suppose we sample $\iota, \iota'$ independently from $\rho_0$. By Proposition D.6, with probability at least $1 - O(\kappa)$, $\iota, \iota' \in [\iota_\mathrm{L}, \iota_\mathrm{R}]$. Thus, by Eq. (D.125), the probability that $\iota, \iota' \in [\iota_\mathrm{L}, \iota_\mathrm{R}]$ and $|\iota - \iota'| \geq \frac{\kappa^2}{\sqrt{d}}$ is at least $1 - O(\kappa)$. In summary, if we sample $\iota, \iota'$ independently from $\rho_0$, then with probability $1 - O(\kappa)$,

$$|\Phi(w_{T_\mathrm{R}}(\iota)) - \Phi(w_{T_\mathrm{R}}(\iota'))| \gtrsim \xi\kappa^3 \tag{D.127}$$

by Lemma D.14, as desired. $\qquad\square$

Next, we show that during Phase 3, Case 1, at any time, there is a significant fraction of particles whose distance from 0 or 1 can be bounded below.

**Lemma D.17.** *In the setting of Theorem 3.3, suppose $D_{4,t} \leq 0$ and $D_{2,t} \geq 0$. Then,*

$$\mathbb{P}_{w \sim \rho_t}(|w| \in [\gamma_2^{3/4}, 1 - \gamma_2^{1/2}]) \geq \frac{\gamma_2}{2} \tag{D.128}$$

*Proof of Lemma D.17.* For convenience, we define $\tau = \sqrt{\gamma_2}$ and $\beta \geq 1.1$ such that $\gamma_4 = \beta\tau^4$. (Here, $\beta$ and $\tau$ are well-defined by Assumption 3.2.) First, since $D_{2,t} \geq 0$ during Phase 3 Case 1, we know that

$$\underset{w_t \sim \rho_t}{\mathbb{E}}[P_{2,d}(w)] \geq \tau^2 \qquad\qquad \text{(By Definition of } D_{2,t})$$

and since $D_{4,t} \leq 0$ during Phase 3 Case 1, we know that

$$\underset{w_t \sim \rho_t}{\mathbb{E}}[P_{4,d}(w)] \leq \beta\tau^4 \qquad\qquad \text{(By Definition of } D_{4,t})$$

Rearranging using $P_{2,d}(t) = \frac{d}{d-1}t^2 - \frac{1}{d-1}$, we obtain

$$\frac{d}{d-1}\underset{w_t \sim \rho_t}{\mathbb{E}}[w^2] - \frac{1}{d-1} \geq \tau^2 \tag{D.129}$$

or

$$\underset{w_t \sim \rho_t}{\mathbb{E}}[w^2] \geq \frac{d-1}{d}\tau^2 + \frac{1}{d} \tag{D.130}$$

Similarly, rearranging using $P_{4,d}(t) = \frac{d^2+6d+8}{d^2-1}t^4 - \frac{6d+12}{d^2-1}t^2 + \frac{3}{d^2-1}$, we obtain

$$\frac{d^2+6d+8}{d^2-1}\underset{w_t \sim \rho_t}{\mathbb{E}}[w^4] - \frac{6d+12}{d^2-1}\underset{w_t \sim \rho_t}{\mathbb{E}}[w^2] + \frac{3}{d^2-1} \leq \beta\tau^4 \tag{D.131}$$

and rearranging gives

$$\underset{w_t \sim \rho_t}{\mathbb{E}}[w^4] \leq \beta\tau^4 + O\left(\frac{1}{d}\right) \tag{D.132}$$

because $w \leq 1$ and we can assume $\beta \tau^4 \leq 1$ due to Assumption 3.2. Suppose that with probability more than $1 - \frac{\tau^2}{2}$ under $\rho_t$, $w \notin [\tau^{3/2}, 1 - \tau]$. Then,

$$\mathop{\mathbb{E}}_{w \sim \rho_t} [w^2] = \mathbb{P}_{w \sim \rho_t}(w \geq 1 - \tau) \cdot \mathop{\mathbb{E}}_{w \sim \rho_t} [w^2 \mid w \geq 1 - \tau] \tag{D.133}$$

$$+ \mathbb{P}_{w \sim \rho_t}(w \in [\tau^{3/2}, 1 - \tau]) \mathop{\mathbb{E}}_{w \sim \rho_t} [w^2 \mid w \in [\tau^{3/2}, 1 - \tau]] \tag{D.134}$$

$$+ \mathbb{P}_{w \sim \rho_t}(w \leq \tau^{3/2}) \cdot \mathop{\mathbb{E}}_{w \sim \rho_t} [w^2 \mid w \leq \tau^{3/2}] \tag{D.135}$$

$$\leq \mathbb{P}_{w \sim \rho_t}(w \geq 1 - \tau) + \frac{\tau^2}{2} + \tau^3$$

(By assumption that $\mathbb{P}_{w \sim \rho_t}(w \in [\tau^{3/2}, 1 - \tau]) \leq \frac{\tau^2}{2}$)

$$\leq \mathbb{P}_{w \sim \rho_t}(w \geq 1 - \tau) + \frac{3\tau^2}{4}$$

(B.c. $\tau \leq 1/4$ (by Assumption 3.2, $\tau$ is sufficiently small))

which by Eq. (D.130) implies that

$$\mathbb{P}_{w \sim \rho_t}(w \geq 1 - \tau) \geq \mathop{\mathbb{E}}_{w \sim \rho_t} [w^2] - \frac{3\tau^2}{4} \geq \frac{d-1}{d}\tau^2 + \frac{1}{d} - \frac{3\tau^2}{4} = \frac{\tau^2}{4} - \frac{\tau^2}{d} + \frac{1}{d} \geq \frac{\tau^2}{4} \tag{D.136}$$

where the last inequality is because $\tau \leq 1$. On the other hand,

$$\mathop{\mathbb{E}}_{w \sim \rho_t} [w^4] = \mathbb{P}_{w \sim \rho_t}(w \geq 1 - \tau) \cdot \mathop{\mathbb{E}}_{w \sim \rho_t} [w^4 \mid w \geq 1 - \tau] \tag{D.137}$$

$$+ \mathbb{P}_{w \sim \rho_t}(w \in [\tau^{3/2}, 1 - \tau]) \mathop{\mathbb{E}}_{w \sim \rho_t} [w^4 \mid w \in [\tau^{3/2}, 1 - \tau]] \tag{D.138}$$

$$+ \mathbb{P}_{w \sim \rho_t}(w \leq \tau^{3/2}) \cdot \mathop{\mathbb{E}}_{w \sim \rho_t} [w^4 \mid w \leq \tau^{3/2}] \tag{D.139}$$

$$\geq \mathbb{P}_{w \sim \rho_t}(w \geq 1 - \tau) \cdot \mathop{\mathbb{E}}_{w \sim \rho_t} [w^4 \mid w \geq 1 - \tau] \tag{D.140}$$

$$\geq (1 - \tau)^4 \cdot \mathbb{P}_{w \sim \rho_t}(w \geq 1 - \tau) \tag{D.141}$$

$$\geq (1 - \tau)^4 \cdot \frac{\tau^2}{4} \qquad \text{(By Eq. (D.136))}$$

$$\geq \frac{\tau^2}{64} \qquad \text{(B.c. } \tau \leq 1/2 \text{ (see Assumption 3.2))}$$

By Eq. (D.132), this is a contradiction, since it implies that

$$\frac{\tau^2}{64} \leq \beta\tau^4 + O\left(\frac{1}{d}\right) \leq 2\beta\tau^4 \tag{D.142}$$

where the second inequality is because $\beta \geq 1.1$ and $d$ is sufficiently large (see Assumption 3.2). This implies that $128\beta\tau^2 \geq 1$, which contradicts Assumption 3.2 (since $\gamma_2$ is chosen to be sufficiently small). Thus, our original assumption that $\mathbb{P}_{w \sim \rho_t}(w \in [\tau^{3/2}, 1 - \tau]) \leq \frac{\tau^2}{2}$ is incorrect. $\qquad\square$

The purpose of the next three lemmas is to show that if Phase 3, Case 1 occurs, i.e. if $D_{2,T_2} = 0$ and $D_{4,T_2} < 0$, then for all $t \geq T_2$, we have $D_{2,t} \geq 0$ and $D_{4,t} \leq 0$. First, the following lemma states that if $D_{4,t}$ is much smaller than $D_{2,t}$ in absolute value, and $D_{4,t} < 0$ and $D_{2,t} > 0$, then the velocity of the particles $w \geq 0$ will be significantly negative.

**Lemma D.18.** *Suppose we are in the setting of Theorem 3.3, and suppose $D_{4,T_2} = 0$ and $t \geq T_2$ such that $D_{4,t} \leq 0$ and $D_{2,t} \geq 0$. If $|D_{4,t}| \leq \xi|D_{2,t}|$ where $\xi = \frac{1}{\log \log d}$, then for all $w \in [0, 1]$,*

$$v_t(w) = -(1 - w^2) \cdot (1 \pm O(\xi)) \cdot 2\hat{\sigma}_{2,d}^2 |D_{2,t}|w \tag{D.143}$$

*In particular, for $w \in [\gamma_2^{3/4}, 1 - \gamma_2^{1/2}]$,*

$$v_t(w) \lesssim -\gamma_2^{5/4}\hat{\sigma}_{2,d}^2 |D_{2,t}| \tag{D.144}$$

*Proof.* Suppose $|D_{4,t}| \leq \xi |D_{2,t}|$ where $\xi = \frac{1}{\log\log d}$. Let $P_t$ and $Q_t$ be as in Lemma 4.4. Then, for any $w \in [0, 1]$,

$$P_t(w) = (2\hat{\sigma}_{2,d}^2 D_{2,t} w + 4\hat{\sigma}_{4,d}^2 D_{4,t} w^3) \tag{D.145}$$

$$= (2\hat{\sigma}_{2,d}^2 |D_{2,t}| w - 4\hat{\sigma}_{4,d}^2 |D_{4,t}| w^3) \tag{D.146}$$

$$= \left( 2\hat{\sigma}_{2,d}^2 |D_{2,t}| w \pm O(\xi \hat{\sigma}_{4,d}^2 |D_{2,t}| w^3) \right) \tag{D.147}$$

$$= (1 \pm O(\xi)) \cdot 2\hat{\sigma}_{2,d}^2 |D_{2,t}| w \qquad \text{(By Assumption 3.2, } \hat{\sigma}_{4,d}^2 \lesssim \hat{\sigma}_{2,d}^2) \tag{D.148}$$

Thus, for $w \in [0, 1]$,

$$v_t(w) = -(1 - w^2)(P_t(w) + Q_t(w)) \qquad \text{(By Lemma 4.4)}$$

$$= -(1 - w^2)\left( |P_t(w)| \pm O\left( \frac{\hat{\sigma}_{2,d}^2 |D_{2,t}|}{d} w \right) \right)$$
$$\qquad \text{(B.c. } |D_{2,t}| \geq \xi |D_{4,t}| \text{ and } \hat{\sigma}_{4,d}^2 \lesssim \hat{\sigma}_{2,d}^2 \text{ by Assumption 3.2)}$$

$$= -(1 - w^2)\left( (1 \pm O(\xi)) \cdot 2\hat{\sigma}_{2,d}^2 |D_{2,t}| w \pm O\left( \frac{\hat{\sigma}_{2,d}^2 |D_{2,t}|}{d} w \right) \right) \qquad \text{(By Eq. (D.145))}$$

$$= -(1 - w^2) \cdot (1 \pm O(\xi)) \cdot 2\hat{\sigma}_{2,d}^2 |D_{2,t}| w \qquad \text{(B.c. } \xi \geq \frac{1}{d})$$

In particular, for $w \in [\gamma_2^{3/4}, 1 - \gamma_2^{1/2}]$,

$$v_t(w) \lesssim -(1 - (1 - \gamma_2^{1/2})^2) \cdot (1 - O(\xi)) \cdot 2\hat{\sigma}_{2,d}^2 |D_{2,t}| \gamma_2^{3/4} \tag{D.148}$$

$$\lesssim -(1 - (1 - \gamma_2^{1/2})^2) \cdot \hat{\sigma}_{2,d}^2 |D_{2,t}| \gamma_2^{3/4} \qquad \text{(B.c. } \xi = \frac{1}{\log\log d})$$

$$\lesssim -(2\gamma_2^{1/2} - \gamma_2) \cdot \hat{\sigma}_{2,d}^2 |D_{2,t}| \gamma_2^{3/4} \tag{D.149}$$

$$\lesssim -\gamma_2^{5/4} \hat{\sigma}_{2,d}^2 |D_{2,t}| \qquad \text{(B.c. } \gamma_2^{1/2} \geq \gamma_2 \text{ since } \gamma_2 \leq 1)$$

as desired. $\qquad \square$

The next lemma essentially implies that if at some point $D_{4,t} \leq 0$ and $D_{2,t} > 0$, then it will never be the case that $D_{4,t} > 0$ and $D_{2,t} > 0$.

**Lemma D.19.** *Suppose we are in the setting of Theorem 3.3, and suppose $D_{4,t} = 0$ and $D_{2,t} \geq 0$ for some $t \geq 0$. Then, $\frac{d}{ds} D_{4,s} \big|_{s=t} \leq 0$.*

*Proof.* We can calculate that

$$\frac{d}{dt} D_{4,t} = \frac{d}{dt} \mathbb{E}_{w \sim \rho_t} [P_{4,d}(w)] = \mathbb{E}_{w \sim \rho_t} [P_{4,d}'(w) \cdot v_t(w)] \tag{D.150}$$

Recall from Eq. (C.4) that

$$P_{4,d}(t) = \frac{d^2 + 6d + 8}{d^2 - 1} t^4 - \frac{6d + 12}{d^2 - 1} t^2 + \frac{3}{d^2 - 1} \tag{D.151}$$

meaning that

$$P_{4,d}'(t) = 4t^3 \pm O\left( \frac{1}{d} \right) \cdot (|t^3| + |t|) = 4t^3 \pm O\left( \frac{1}{d} \right) |t| \tag{D.152}$$

Observe that for $t \gtrsim \frac{1}{\sqrt{d}}$, $P_{4,d}{}'(t) > 0$ and for $t \lesssim \frac{1}{\sqrt{d}}$, it may be the case that $P_{4,d}{}'(t) < 0$. In particular,

$$\mathbb{E}_{w \sim \rho_t}[P_{4,d}{}'(w) \cdot v_t(w)] = \mathbb{P}(|w| \lesssim 1/\sqrt{d}) \cdot \mathbb{E}_{w \sim \rho_t}[P_{4,d}{}'(w) \cdot v_t(w) \mid |w| \lesssim 1/\sqrt{d}] \tag{D.153}$$

$$+ \mathbb{P}(|w| \gtrsim 1/\sqrt{d}) \, \mathbb{E}_{w \sim \rho_t}[P_{4,d}{}'(w) \cdot v_t(w) \mid |w| \gtrsim 1/\sqrt{d}]$$
$$\tag{D.154}$$

$$\leq O\Big(\frac{\hat{\sigma}_{2,d}^2 |D_{2,t}|}{d^2}\Big) + \mathbb{P}(|w| \gtrsim 1/\sqrt{d}) \, \mathbb{E}_{w \sim \rho_t}[P_{4,d}{}'(w) \cdot v_t(w) \mid |w| \gtrsim 1/\sqrt{d}]$$
$$\text{(B.c. } |w| \lesssim \tfrac{1}{\sqrt{d}} \text{ and by Lemma D.18)}$$

$$\leq O\Big(\frac{\hat{\sigma}_{2,d}^2 |D_{2,t}|}{d^2}\Big) + \mathbb{P}(|w| \in [\gamma_2^{3/4}, 1 - \gamma_2^{1/2}]) \tag{D.155}$$

$$\mathbb{E}_{w \sim \rho_t}[P_{4,d}{}'(w) \cdot v_t(w) \mid |w| \in [\gamma_2^{3/4}, 1 - \gamma_2^{1/2}]] \tag{D.156}$$

$$\leq O\Big(\frac{\hat{\sigma}_{2,d}^2 |D_{2,t}|}{d^2}\Big) - \mathbb{P}(|w| \in [\gamma_2^{3/4}, 1 - \gamma_2^{1/2}]) \gamma_2^{9/4} \cdot \gamma_2^{5/4} \hat{\sigma}_{2,d}^2 |D_{2,t}|$$
$$\text{(B.c. } P_{4,d}{}'(w) \gtrsim \gamma_2^{9/4}, \text{ and by Lemma D.18)}$$

$$\leq O\Big(\frac{\hat{\sigma}_{2,d}^2 |D_{2,t}|}{d^2}\Big) - \frac{\gamma_2}{2} \cdot \gamma_2^{14/4} \hat{\sigma}_{2,d}^2 |D_{2,t}| \qquad \text{(By Lemma D.17)}$$

$$< 0 \qquad \text{(By Assumption 3.2 since } d \text{ is sufficiently large)}$$

meaning that $\frac{d}{dt} D_{4,t} < 0$, as desired. $\qquad \square$

The next lemma essentially implies that if at some point $D_{4,t} \leq 0$ and $D_{2,t} > 0$, it will never be the case that $D_{4,t} \leq 0$ and $D_{2,t} < 0$.

**Lemma D.20.** *Suppose we are in the setting of Theorem 3.3, and suppose $D_{4,t} \leq 0$ and $D_{2,t} = 0$, for some $t \geq 0$. Then, $\frac{d}{ds} D_{2,s}\big|_{s=t} \geq 0$.*

*Proof.* Recall from Eq. (C.4) that $P_{2,d}(t) = \frac{d}{d-1} t^2 - \frac{1}{d-1}$ meaning that $P_{2,d}{}'(t) = \frac{2d}{d-1} t$. Thus,

$$\frac{d}{dt} D_{2,t} = \frac{d}{dt} \mathbb{E}_{w \sim \rho_t}[w^2] = \frac{d}{d-1} \mathbb{E}_{w \sim \rho_t}[2w \cdot v_t(w)] \tag{D.157}$$

If $D_{4,t} \leq 0$ and $D_{2,t} = 0$, then for any particle $w \in [0,1]$, we can write

$$v_t(w) = -(1 - w^2)(P_t(w) + Q_t(w)) \qquad \text{(By Lemma 4.4)}$$

$$= -(1 - w^2)\Big(4\hat{\sigma}_{4,d}^2 D_{4,t} w^3 \pm O\Big(\frac{\hat{\sigma}_{4,d}^2 |D_{4,t}|}{d}\Big)|w|\Big) \qquad \text{(By Lemma 4.4)}$$

$$= (1 - w^2)\Big(4\hat{\sigma}_{4,d}^2 |D_{4,t}| w^3 \pm O\Big(\frac{\hat{\sigma}_{4,d}^2 |D_{4,t}|}{d}\Big)|w|\Big) \tag{D.158}$$

where the last equality is because $D_{4,t} \leq 0$. Observe that $v_t(w) \geq 0$ for $w \gtrsim \frac{1}{\sqrt{d}}$, while it may be the case that $v_t(w) \leq 0$ for $w \lesssim \frac{1}{\sqrt{d}}$. Thus,

$$\mathbb{E}_{w \sim \rho_t}[2w \cdot v_t(w)] = \mathbb{P}(|w| \lesssim 1/\sqrt{d}) \cdot \mathbb{E}_{w \sim \rho_t}[2w \cdot v_t(w) \mid |w| \lesssim 1/\sqrt{d}] \tag{D.159}$$

$$+ \mathbb{P}(|w| \gtrsim 1/\sqrt{d}) \mathbb{E}_{w \sim \rho_t}[2w \cdot v_t(w) \mid |w| \lesssim 1/\sqrt{d}] \tag{D.160}$$

$$\geq -O\Big(\frac{\hat{\sigma}_{4,d}^2 |D_{4,t}|}{d^2}\Big) + \mathbb{P}(|w| \gtrsim 1/\sqrt{d}) \mathbb{E}_{w \sim \rho_t}[2w \cdot v_t(w) \mid |w| \gtrsim 1/\sqrt{d}]$$
$$\text{(By bounding Eq. (D.158) when } |w| \lesssim \tfrac{1}{\sqrt{d}})$$

$$\geq -O\Big(\frac{\hat{\sigma}_{4,d}^2 |D_{4,t}|}{d^2}\Big) + \mathbb{P}(|w| \in [\gamma_2^{3/4}, 1 - \gamma_2^{1/2}]) \tag{D.161}$$

$$\mathbb{E}_{w \sim \rho_t}[2w \cdot v_t(w) \mid |w| \in [\gamma_2^{3/4}, 1 - \gamma_2^{1/2}]] \tag{D.162}$$

$$\gtrsim -O\Big(\frac{\hat{\sigma}_{4,d}^2 |D_{4,t}|}{d^2}\Big) + \mathbb{P}(|w| \in [\gamma_2^{3/4}, 1 - \gamma_2^{1/2}]) \cdot \gamma_2^{9/4} \hat{\sigma}_{4,d}^2 |D_{4,t}|$$
$$\text{(By Eq. (D.158))}$$

$$\gtrsim -O\Big(\frac{\hat{\sigma}_{4,d}^2 |D_{4,t}|}{d^2}\Big) - \frac{\gamma_2}{2} \cdot \gamma_2^{9/4} \hat{\sigma}_{4,d}^2 |D_{4,t}| \qquad \text{(By Lemma D.17)}$$

$$\geq 0 \qquad \text{(By Assumption 3.2 since } d \text{ is sufficiently large)}$$

as desired. $\qquad\qquad\qquad\qquad\qquad\qquad\qquad\qquad\qquad\qquad\qquad\qquad\qquad\qquad\qquad\qquad\qquad\square$

Putting all of the previous lemmas together, we obtain the following invariant: a lower bound for $|\Phi(w) - \Phi(w')|$, for a large fraction of $w, w'$, which holds throughout Phase 3, Case 1. Note that in the proof of this invariant, we need to make use of the fact that once we enter the case where $D_{2,t} > 0$ and $D_{4,t} < 0$, we cannot leave this case. If this is not true, then we cannot use the fact that $|\Phi(w_t) - \Phi(w'_t)|$ is increasing as a function of $t$ (Lemma D.13) since if $D_{4,t} > 0$ at any point, then $|\Phi(w_t) - \Phi(w'_t)|$ could decrease, and we would not have control over this decrease.

**Lemma D.21** (Phase 3, Case 1 Invariant). *Suppose we are in the setting of Theorem 3.3. Let $T_{\mathrm{R}}$ be as defined in the statement of Lemma D.7, and assume that $D_{2,T_2} = 0$ and $D_{4,T_2} < 0$. Then, for all $t \geq T_{\mathrm{R}}$ (in particular, for all times $t$ during Phase 3 Case 1), if $\iota, \iota'$ are sampled independently from $\rho_0$,*

$$|\Phi(w_t(\iota)) - \Phi(w_t(\iota'))| \gtrsim \xi\kappa^3 \tag{D.163}$$

*with probability at least $1 - O(\kappa)$.*

*Proof of Lemma D.21.* By Lemma D.19 and Lemma D.20, if $D_{2,T_2} = 0$ and $D_{4,T_2} < 0$, then for all $t \geq T_2$, we will have $D_{2,t} \geq 0$ and $D_{4,t} \leq 0$. Thus, the lemma follows from Lemma D.13 and Lemma D.16, as well as the fact that $D_{4,t} \leq 0$ for $t \in [T_{\mathrm{R}}, T_2]$ (meaning that the potential difference is increasing in the time interval $[T_{\mathrm{R}}, T_2]$). $\qquad\qquad\qquad\qquad\square$

Finally, we show Lemma D.22 which gives a running time bound for Phase 3, Case 1.

**Lemma D.22** (Phase 3, Case 1 Summary). *In the setting of Theorem 3.3, suppose that $D_{2,T_2} = 0$ and $D_{4,T_2} < 0$ (i.e. Phase 3, Case 1 holds). Then, the total amount of time $t \geq T_2$ such that $L(\rho_t) \geq \frac{1}{2}(\hat{\sigma}_{2,d}^2 + \hat{\sigma}_{4,d}^2)\epsilon^2$ is at most $O(\frac{1}{\hat{\sigma}_{4,d}^2 \gamma_2^8 \xi^4 \kappa^6} \log(\frac{\gamma_2}{\epsilon}))$.*

*Proof of Lemma D.22.* Suppose $t \geq T_2$ and assume that $D_{2,T_2} = 0$ and $D_{4,t} < 0$ (i.e. Phase 3, Case 1 occurs). First, suppose $|D_{4,t}| \leq \xi|D_{2,t}|$ where $\xi = \frac{1}{\log \log d}$. Then, by Lemma D.18, if $|w| \in [\gamma_2^{3/4}, 1 - \gamma_2^{1/2}]$, we have

$$|v_t(w)| \gtrsim \gamma_2^{5/4} \hat{\sigma}_{2,d}^2 |D_{2,t}| \tag{D.164}$$

(note that the lemma is stated for $w \in [\gamma_2^{3/4}, 1 - \gamma_2^{1/4}]$, but holds for $|w| \in [\gamma_2^{3/4}, 1 - \gamma_2^{1/4}]$ since $v_t(w)$ is an odd function). Therefore,

$$\mathbb{E}_{w \sim \rho_t} |v_t(w)|^2 \gtrsim \mathbb{P}_{w \sim \rho_t}(|w| \in [\gamma_2^{3/4}, 1 - \gamma_2^{1/2}]) \cdot \mathbb{E}_{w \sim \rho_t}[|v_t(w)|^2 \mid |w| \in [\gamma_2^{3/4}, 1 - \gamma_2^{1/2}]] \quad \text{(D.165)}$$

$$\gtrsim \frac{\gamma_2}{2} \cdot \mathbb{E}_{w \sim \rho_t}[|v_t(w)|^2 \mid |w| \in [\gamma_2^{3/4}, 1 - \gamma_2^{1/2}]] \qquad \text{(By Lemma D.17)}$$

$$\gtrsim \frac{\gamma_2}{2} \cdot \gamma_2^{5/2} \hat{\sigma}_{2,d}^4 |D_{2,t}|^2 \qquad \text{(By Eq. (D.164))}$$

$$\gtrsim \gamma_2^{7/2} \hat{\sigma}_{2,d}^4 |D_{2,t}|^2 \qquad \text{(D.166)}$$

$$\gtrsim \gamma_2^{7/2} \hat{\sigma}_{2,d}^2 \cdot (\hat{\sigma}_{2,d}^2 |D_{2,t}|^2 + \hat{\sigma}_{4,d}^2 |D_{4,t}|^2)$$
$$\text{(B.c. } \hat{\sigma}_{4,d}^2 \lesssim \hat{\sigma}_{2,d}^2 \text{ (Assumption 3.2) and } |D_{4,t}| \leq \xi|D_{2,t}|)$$

$$\gtrsim \gamma_2^{7/2} \hat{\sigma}_{2,d}^2 L(\rho_t) \qquad \text{(D.167)}$$

Thus, by Lemma D.39, for any time $t$ during Phase 3, Case 1 such that $|D_{4,t}| \leq \xi|D_{2,t}|$, we have

$$\frac{dL(\rho_t)}{dt} \lesssim -\gamma_2^{7/2} \hat{\sigma}_{2,d}^2 L(\rho_t) \qquad \text{(D.168)}$$

Let $T_{*,\epsilon}$ be as in Theorem 3.3, and define

$$I_{|D_4| \leq \xi|D_2|} = \{t \leq T_{*,\epsilon} \text{ and } t \text{ in Phase 3, Case 1} \mid |D_{4,t}| \leq \xi|D_{2,t}|\} \qquad \text{(D.169)}$$

Then, by Eq. (D.168),

$$\frac{dL(\rho_t)}{dt} \lesssim -\gamma_2^{7/2} \hat{\sigma}_{2,d}^2 L(\rho_t) 1(t \in I_{|D_4| \leq \xi|D_2|}) \qquad \text{(D.170)}$$

Thus, by Gronwall's inequality (Fact I.13),

$$\frac{L(\rho_{T_{*,\epsilon}})}{L(\rho_0)} \leq \exp\left(-C \int_{T_2}^{T_{*,\epsilon}} \gamma_2^{7/2} \hat{\sigma}_{2,d}^2 1(t \in I_{|D_4| \leq \xi|D_2|}) dt\right) \qquad \text{(D.171)}$$

$$= \exp\left(-C\gamma_2^{7/2} \hat{\sigma}_{2,d}^2 \int_{I_{|D_4| \leq \xi|D_2|}} 1 dt\right) \qquad \text{(D.172)}$$

Since $L(\rho_0) \lesssim (\hat{\sigma}_{2,d}^2 + \hat{\sigma}_{4,d}^2)\gamma_2^2$ by Lemma D.8 and $L(\rho_{T_{*,\epsilon}}) \gtrsim (\hat{\sigma}_{2,d}^2 + \hat{\sigma}_{4,d}^2)\epsilon^2$, rearranging gives

$$\int_{I_{|D_4| \leq \xi|D_2|}} 1 dt \lesssim \frac{1}{\hat{\sigma}_{2,d}^2 \gamma_2^{7/2}} \log\left(\frac{\gamma_2^2}{\epsilon^2}\right) \qquad \text{(D.173)}$$

Now, suppose $t$ is such that $|D_{4,t}| \geq \xi|D_{2,t}|$. Let us lower bound the average velocity for $w \in [\gamma_2^{3/4}, 1 - \gamma_2^{1/4}]$ first (then, we can conclude the average velocity is large by Lemma D.17). By Lemma 4.4,

$$|v_t(w)| = (1 - w^2)|P_t(w) + Q_t(w)| \qquad \text{(D.174)}$$

and we can write

$$P_t(w) = 2\hat{\sigma}_{2,d}^2 D_{2,t} w + 4\hat{\sigma}_{4,d}^2 D_{4,t} w^3 = 4\hat{\sigma}_{4,d}^2 D_{4,t} w(w - r)(w + r) \qquad \text{(D.175)}$$

where $r = \sqrt{-\frac{\hat{\sigma}_{2,d}^2 D_{2,t}}{2\hat{\sigma}_{4,d}^2 D_{4,t}}}$ is the positive root of $P_t$. For $|w| \in [\gamma_2^{3/4}, 1 - \gamma_2^{1/2}]$, we have

$$1 - w^2 \geq 1 - (1 - \gamma_2^{1/2})^2 = 2\gamma_2^{1/2} - \gamma_2 \geq \gamma_2^{1/2}. \qquad \text{(D.176)}$$

Additionally, $|w| \geq \gamma_2^{3/4}$ and $|w+r| \geq \gamma_2^{3/4}$. Thus, for $w \in [\gamma_2^{3/4}, 1 - \gamma_2^{1/2}]$, we have

$$|v_t(w)|^2 \gtrsim (1-w^2)^2 \Big(P_t(w) + Q_t(w)\Big)^2 \tag{D.177}$$

$$\gtrsim \gamma_2 \Big(P_t(w) + Q_t(w)\Big)^2 \qquad \text{(B.c. } 1 - w^2 \geq \gamma_2^{1/2})$$

$$\gtrsim \gamma_2 \cdot \Big(P_t(w)^2 - 2Q_t(w)^2\Big) \qquad \text{(B.c. } (a+b)^2 \leq 2a^2 + 2b^2, \text{ so } a^2 - 2b^2 \leq 2(a+b)^2)$$

$$\gtrsim \gamma_2 \cdot \Big(P_t(w)^2 - O\Big(\frac{\hat{\sigma}_{2,d}^4 D_{2,t}^2 + \hat{\sigma}_{4,d}^4 D_{4,t}^2}{d^2}\Big)w^2\Big) \qquad \text{(By Lemma 4.4)}$$

$$\gtrsim \gamma_2 \cdot \Big(P_t(w)^2 - O\Big(\frac{(\hat{\sigma}_{2,d}^2 + \hat{\sigma}_{4,d}^2)L(\rho_t)}{d^2}\Big)w^2\Big) \qquad \text{(By Lemma 4.2)}$$

$$\gtrsim \gamma_2 \cdot \Big(\hat{\sigma}_{4,d}^4 D_{4,t}^2 w^2 (w+r)^2 (w-r)^2 - O\Big(\frac{(\hat{\sigma}_{2,d}^2 + \hat{\sigma}_{4,d}^2)L(\rho_t)}{d^2}\Big)w^2\Big)$$
$$\text{(By factorization of } P_t(w) \text{ above)}$$

$$\gtrsim \gamma_2 \cdot \Big(\hat{\sigma}_{4,d}^4 D_{4,t}^2 \gamma_2^3 (w-r)^2 - O\Big(\frac{(\hat{\sigma}_{2,d}^2 + \hat{\sigma}_{4,d}^2)L(\rho_t)}{d^2}\Big)w^2\Big) \tag{D.178}$$

where the last inequality is by the discussion above Eq. (D.177). Now, in order to take the expectation over $w \sim \rho_t$ (conditioned on $w \in [\gamma_2^{3/4}, 1 - \gamma_2^{1/2}]$), we first compute $\mathbb{E}_{w \sim \rho_t}[(w-r)^2 \mid w \in [\gamma_2^{3/4}, 1 - \gamma_2^{1/2}]]$:

$$\mathop{\mathbb{E}}_{w \sim \rho_t}[(w-r)^2 \mid w \sim [\gamma_2^{3/4}, 1 - \gamma_2^{1/2}]] \geq \mathop{\mathrm{Var}}_{w \sim \rho_t}(w \mid w \sim [\gamma_2^{3/4}, 1 - \gamma_2^{1/2}]) \tag{D.179}$$

$$\gtrsim \mathop{\mathbb{E}}_{w,w' \sim \rho_t}[(w-w')^2 \mid w, w' \in [\gamma_2^{3/4}, 1 - \gamma_2^{1/2}]] \tag{D.180}$$

where the last inequality is by Proposition D.40. By Lemma D.21,

$$\mathbb{P}_{w,w' \sim \rho_t}(|\Phi(w) - \Phi(w')| \gtrsim \xi\kappa^3) \geq 1 - O(\kappa) \tag{D.181}$$

By Lemma D.14, if $w, w' \in [\gamma_2^{3/4}, 1 - \gamma_2^{1/2}]$, then

$$|\Phi(w) - \Phi(w')| \leq \frac{1}{\gamma_2^{3/2}}|w - w'| \tag{D.182}$$

Thus, combining the previous two equations gives

$$\mathbb{P}_{w,w' \sim \rho_t}(|w - w'| < \gamma_2^{3/2}\xi\kappa^3 \mid w, w' \in [\gamma_2^{3/4}, 1 - \gamma_2^{1/2}]) \tag{D.183}$$

$$\lesssim \mathbb{P}_{w,w' \sim \rho_t}(|\Phi(w) - \Phi(w')| < \xi\kappa^3 \mid w, w' \in [\gamma_2^{3/4}, 1 - \gamma_2^{1/2}]) \tag{D.184}$$

$$\lesssim \frac{\mathbb{P}_{w,w' \sim \rho_t}(|\Phi(w) - \Phi(w')| < \xi\kappa^3 \text{ and } w, w' \in [\gamma_2^{3/4}, 1 - \gamma_2^{1/2}])}{\mathbb{P}_{w,w' \sim \rho_t}(w, w' \in [\gamma_2^{3/4}, 1 - \gamma_2^{1/2}])} \tag{D.185}$$

$$\lesssim \frac{\kappa}{\mathbb{P}_{w,w' \sim \rho_t}(w, w' \in [\gamma_2^{3/4}, 1 - \gamma_2^{1/2}])} \qquad \text{(By Lemma D.21)}$$

$$\lesssim \frac{\kappa}{\gamma_2} \qquad \text{(By Lemma D.17)}$$

and therefore, by Assumption 3.2, since $\kappa/\gamma_2 \lesssim \frac{1}{\gamma_2 \log\log d}$, this is at most $\frac{1}{2}$, since $d$ is sufficiently large. Thus,

$$\mathop{\mathbb{E}}_{w,w' \sim \rho_t}[(w-w')^2 \mid w, w' \in [\gamma_2^{3/4}, 1 - \gamma_2^{1/2}]] \tag{D.186}$$

$$\gtrsim \mathbb{P}_{w,w' \sim \rho_t}(|w - w'| \geq \gamma_2^{3/2}\xi\kappa^3 \mid w, w' \in [\gamma_2^{3/4}, 1 - \gamma_2^{1/2}]) \cdot \gamma_2^3\xi^2\kappa^6 \tag{D.187}$$

$$\gtrsim \gamma_2^3\xi^2\kappa^6 \tag{D.188}$$

where the last inequality is because $\mathbb{P}_{w,w'\sim\rho_t}(|w-w'| < \gamma_2^{3/2}\xi\kappa^3 \mid w, w' \in [\gamma_2^{3/4}, 1-\gamma_2^{1/2}]) \leq \frac{1}{2}$. By Eq. (D.178), we have

$$\mathbb{E}_{w\sim\rho_t}[|v_t(w)|^2 \mid w \in [\gamma_2^{3/4}, 1-\gamma_2^{1/2}]] \gtrsim \gamma_2 \cdot \left(\hat{\sigma}_{4,d}^4 D_{4,t}^2 \gamma_2^3 \mathbb{E}_{w\sim\rho_t}\left[(w-r)^2 \mid w \in [\gamma_2^{3/4}, 1-\gamma_2^{1/2}]\right]\right.$$
(D.189)

$$\left.- O\left(\frac{(\hat{\sigma}_{2,d}^2 + \hat{\sigma}_{4,d}^2)L(\rho_t)}{d^2}\right)\right)$$
(D.190)

$$\gtrsim \gamma_2 \cdot \left(\hat{\sigma}_{4,d}^4 D_{4,t}^2 \gamma_2^3 \cdot \gamma_2^3 \xi^2 \kappa^6 - O\left(\frac{(\hat{\sigma}_{2,d}^2 + \hat{\sigma}_{4,d}^2)L(\rho_t)}{d}\right)\right)$$
(By Eq. (D.180) and Eq. (D.188))

Since $\mathbb{P}_{w\sim\rho_t}(w \in [\gamma_2^{3/4}, 1-\gamma_2^{1/2}]) \geq \frac{\gamma_2}{4}$ by Lemma D.17 and the symmetry of $\rho_t$, this implies that

$$\mathbb{E}_{w\sim\rho_t}|v_t(w)|^2 \gtrsim \gamma_2^2 \cdot \left(\hat{\sigma}_{4,d}^4 D_{4,t}^2 \gamma_2^6 \xi^2 \kappa^6 - O\left(\frac{(\hat{\sigma}_{2,d}^2 + \hat{\sigma}_{4,d}^2)L(\rho_t)}{d}\right)\right)$$
(D.191)

$$\gtrsim \gamma_2^2 \cdot \left(\hat{\sigma}_{4,d}^2 \xi^2 L(\rho_t)\gamma_2^6 \xi^2 \kappa^6 - O\left(\frac{(\hat{\sigma}_{2,d}^2 + \hat{\sigma}_{4,d}^2)L(\rho_t)}{d}\right)\right)$$
(B.c. $\hat{\sigma}_{4,d}^2 \gtrsim \hat{\sigma}_{2,d}^2$ (Assumption 3.2) and $|D_{4,t}| \geq \xi|D_{2,t}|$ implies $\hat{\sigma}_{4,d}^2 D_{4,t}^2 \gtrsim \xi^2 L(\rho_t)$)

$$\gtrsim \gamma_2^2 \cdot \left(\hat{\sigma}_{4,d}^2 \gamma_2^6 \xi^4 \kappa^6 - O\left(\frac{\hat{\sigma}_{2,d}^2 + \hat{\sigma}_{4,d}^2}{d}\right)\right)L(\rho_t)$$
(D.192)

$$\gtrsim \hat{\sigma}_{4,d}^2 \gamma_2^8 \xi^4 \kappa^6 L(\rho_t)$$
(Because $d \geq c_3$ by Assumption 3.2)

Thus, by Lemma D.39, $\frac{dL(\rho_t)}{dt} \lesssim -\hat{\sigma}_{4,d}^2\gamma_2^8\xi^4\kappa^6 L(\rho_t)$. By a similar argument as for the subcase where $|D_{4,t}| \leq \xi|D_{2,t}|$, since $L(\rho_0) \lesssim \hat{\sigma}_{2,d}^2\gamma_2^2$ (by Lemma D.8), if we define

$$I_{|D_4|\geq\xi|D_2|} = \{t \leq T_{*,\epsilon} \text{ and } t \text{ in Phase 3, Case 1} \mid |D_{4,t}| \geq \xi|D_{2,t}|\}$$
(D.193)

then

$$\int_{I_{|D_4|\geq\xi|D_2|}} 1dt \lesssim \frac{1}{\hat{\sigma}_{4,d}^2\gamma_2^8\xi^4\kappa^6}\log\left(\frac{\gamma_2}{\epsilon}\right)$$
(D.194)

since $L(\rho_{T_{*,\epsilon}}) \gtrsim \hat{\sigma}_{2,d}^2\epsilon^2$. Combining Eq. (D.173) and Eq. (D.194), we find that the total time spent in Phase 3, Case 1 is at most $O(\frac{1}{\hat{\sigma}_{4,d}^2\gamma_2^8\xi^4\kappa^6}\log(\frac{\gamma_2}{\epsilon}))$, as desired. $\qquad\square$

## D.6 Phase 3, Case 2

Next, we analyze Phase 3, Case 2, where $D_{4,T_2} = 0$ and $D_{2,T_2} < 0$. In this case, as we will show, for all $t \geq T_2$, we will have $D_{4,t} \geq 0$ and $D_{2,t} \leq 0$ (Lemma D.28). We will first give an outline of our analysis in this case — our goal is again to show that either $|D_{2,t}| \leq \epsilon$ or $|D_{4,t}| \leq \epsilon$ within $\frac{1}{\hat{\sigma}_{2,d}^2 + \hat{\sigma}_{4,d}^2} \cdot \text{poly}(\frac{\log\log d}{\epsilon})$ time after the start of Phase 3, Case 2.

**Proof Strategy for Phase 3, Case 2** We again show that $|v(w)|$ is large on average and then use Lemma D.39 to show that the loss decreases quickly. However, our argument to show that $|v(w)|$ is large on average is significantly different from Phase 3, Case 1. We can approximately write

$$v_t(w) = -(1-w^2)(P_t(w) + Q_t(w))$$
(D.195)

$$\approx -(1-w^2)P_t(w)$$
(D.196)

$$\approx -(1-w^2)w(2\hat{\sigma}_{2,d}^2 D_{2,t} + 4\hat{\sigma}_{4,d}^2 D_{4,t}w^2)$$
(D.197)

$$\approx (1-w^2)w(2\hat{\sigma}_{2,d}^2|D_{2,t}| - 4\hat{\sigma}_{4,d}^2|D_{4,t}|w^2)$$
(B.c. $D_{2,t} \leq 0$ and $D_{4,t} \geq 0$)

$$\approx -(1-w^2)w(4\hat{\sigma}_{4,d}^2|D_{4,t}|w^2 - 2\hat{\sigma}_{2,d}^2|D_{2,t}|)$$
(D.198)

Letting $r = \sqrt{\frac{\hat{\sigma}_{2,d}^2|D_{2,t}|}{2\hat{\sigma}_{4,d}^2|D_{4,t}|}}$ be the positive root of $P_t(w)$, we obtain

$$v_t(w) \approx -4\hat{\sigma}_{4,d}^2|D_{4,t}|(w-r)(w+r)w(1-w^2)$$
(D.199)

This time, $v_t(w)$ has a negative slope at $r$, meaning that the particles are attracted towards $r$, and we cannot use the same argument as in Phase 3, Case 1.

Instead, the key point of our argument in this case is the following. Recall that $D_{4,t} \geq 0$ and $D_{2,t} \leq 0$ for all $t \geq T_2$ — these roughly imply that $\mathbb{E}_{w \sim \rho_t}[w^4] \geq \gamma_4$ and $\mathbb{E}_{w \sim \rho_t}[w^2] \leq \gamma_2$. By Assumption 3.2, we have $\gamma_4 \geq 1.1\gamma_2^2$ — in other words, $\mathbb{E}_{w \sim \rho_t}[w^4] \geq 1.1(\mathbb{E}_{w \sim \rho_t}[w^2])^2$. From this observation, we know that a significant fraction of the particles must be far from $r$ — otherwise, if all of the particles are concentrated at $r$, then we would have $\mathbb{E}_{w \sim \rho_t}[w^4] \approx (\mathbb{E}_{w \sim \rho_t}[w^2])^2$, which would give a contradiction.

One complication in showing that the velocity is large is that, a priori, nearly all of the particles could be close to $0$ or $1$ — their velocities could be small even if they are far from $r$. We make use of the potential function $\Phi$ defined in the analysis of Phase 3, Case 1 to avoid this issue. In this case, it turns out that for any two particles $w, w'$, their potential difference $|\Phi(w) - \Phi(w')|$ is decreasing (Lemma D.23). We can use this to show that a $1 - O(\gamma_4)$ fraction of particles $w$ with initializations in $[\iota_L, \iota_M]$ (where $\iota_M$ is defined later) cannot become too close to $0$ or $1$ in Lemma D.24 and Lemma D.25. Intuitively, this is because if $|\Phi(w) - \Phi(w')|$ is bounded by $\log B$ for two particles $w, w' > 0$, and $C$ is a sufficiently large constant, then $w \leq (\frac{1}{\log \log d})^C$ roughly implies that $w' \lesssim B(\frac{1}{\log \log d})^C$, simply by the definition $\Phi(w) = \log(\frac{w}{\sqrt{1-w^2}})$ together with some algebraic manipulations. Moreover, if all of the particles $w'$ in this group are at most $B(\frac{1}{\log \log d})^C$, then this implies that $D_2$ has a very large negative value, and this implies that all particles $w$ have a positive velocity (as argued in Lemma D.25), meaning that the particles $w$ in this group cannot become smaller than $B(\frac{1}{\log \log d})^C$. We can show by a similar argument that the particles having initializations in $[\iota_L, \iota_M]$ cannot become too close to $1$.

As long as the positive root $r$ of $P_t(w)$ is at least $\frac{1}{2}$, this group of particles will have a large velocity as argued in Lemma D.29, since a significant fraction of these particles must be at most $\frac{1}{3}$. We encounter a potential issue when the positive root $r$ is at most $\frac{1}{2}$: it is possible that the particles initialized in $[\iota_L, \iota_M]$ get stuck at $r$. We mentioned above that by the constraints $D_{2,t} < 0$ and $D_{4,t} > 0$, there must always be some mass away from $r$. However, this mass may only be a $\gamma_4^{5/4}$ fraction of the total mass, while the particles initialized at $[\iota_L, \iota_M]$ only have a $1 - O(\gamma_4)$ fraction of the total mass. Thus, once $r \leq \frac{1}{2}$, it is possible for the mass located far away from $r$ to be disjoint from the set of particles initialized at $[\iota_L, \iota_M]$. To resolve this, we argue in Lemma D.31 that the particles larger than $w_t(\iota_M)$ (which may be close to $1$) will move away from $1$ at an exponentially fast rate while $r \leq \frac{1}{2}$. Once these particles, which are larger than $w_t(\iota_M)$, are sufficiently far away from $1$, they will also contribute a significant amount to the velocity. Thus, we can argue as discussed above that there must be a significant fraction of particles away from $r$ due to the constraints $D_{2,t} \leq 0$ and $D_{4,t} \geq 0$ (Lemma D.32).

We now begin the formal proof. We first show that as long as $D_{4,t} \geq 0$, the potential difference between any two particles $w, w'$ is increasing. In particular, this holds during Phase 3, Case 2 (but note that this lemma is no longer applicable during Phase 1 or Phase 2).

**Lemma D.23** ($\Phi(w)$ and $\Phi(w')$ Move Closer in Phase 3 Case 2). *Suppose we are in the setting of Theorem 3.3, and suppose $D_{4,t} \geq 0$ for all $t$ in some time interval $[t_0, t_1]$. Then, for all $\iota, \iota'$,*

$$\left| \Phi(w_t(\iota)) - \Phi(w_t(\iota')) \right| \tag{D.200}$$

*is decreasing as a function of $t$ on the interval $[t_0, t_1]$.*

*Proof of Lemma D.23.* We mostly follow the proof of Lemma D.13. Suppose $\iota, \iota' \in [0, 1]$ such that $\iota \leq \iota'$. Then, $w_t(\iota') \geq w_t(\iota)$ for all times $t$ (by Proposition D.37), and since $\Phi$ is an increasing function, $\Phi(w_t(\iota')) \geq \Phi(w_t(\iota))$ for all times $t$. Thus, it suffices to show that $\Phi(w_t(\iota')) - \Phi(w_t(\iota))$ is decreasing. For convenience, we write $w_t' = w_t(\iota')$ and $w_t = w_t(\iota)$. We do a computation similar

to that in the proof of Lemma D.13:

$$\frac{d}{dt}(\Phi(w'_t) - \Phi(w_t)) = \left( - 2\hat{\sigma}^2_{2,d}D_{2,t} - 4\hat{\sigma}^2_{4,d}D_{4,t}(w'_t)^2 - \lambda_d^{(1)} - \lambda_d^{(3)}(w'_t)^2 \right) \tag{D.201}$$

$$- \left( - 2\hat{\sigma}^2_{2,d}D_{2,t} - 4\hat{\sigma}^2_{4,d}D_{4,t}w_t^2 - \lambda_d^{(1)} - \lambda_d^{(3)}w_t^2 \right)$$
$$\text{(By Lemma D.12)}$$

$$= -4\hat{\sigma}^2_{4,d}D_{4,t}((w'_t)^2 - w_t^2) - \lambda_d^{(3)}((w'_t)^2 - w_t^2) \tag{D.202}$$

According to the statement of Lemma 4.4, $\lambda_d^{(3)} = 4\hat{\sigma}^2_{4,d}D_{4,t} \cdot \frac{6d+9}{d^2-1}$, meaning that

$$\frac{d}{dt}(\Phi(w'_t) - \Phi(w_t)) = -4\hat{\sigma}^2_{4,d}D_{4,t}((w'_t)^2 - w_t^2) - 4\hat{\sigma}^2_{4,d}D_{4,t} \cdot \frac{6d+9}{d^2-1}((w'_t)^2 - w_t^2) \tag{D.203}$$

$$= -4\hat{\sigma}^2_{4,d}D_{4,t} \cdot \left(1 - \frac{6d+9}{d^2-1}\right) \cdot ((w'_t)^2 - w_t^2) \tag{D.204}$$

$$< 0 \qquad \text{(B.c. } D_{4,t} \geq 0 \text{ and } w'_t > w_t > 0, \text{ and } d \geq c_3 \text{ (Assumption 3.2))}$$

Thus, on any interval $[t_0, t_1]$ where $D_{4,t} \geq 0$, $\Phi(w'_t) - \Phi(w_t)$ is decreasing, as desired. $\qquad \square$

In the next lemma, we show an upper bound on $|\Phi(w) - \Phi(w')|$, at the beginning of Phase 3, Case 2, for particles $w, w'$ which are respectively initialized at $\iota, \iota' \in [\iota_L, \iota_M]$. Here $\iota_M$ is defined in the statement/proof of the following lemma.

**Lemma D.24.** *Suppose we are in the setting of Theorem 3.3. Assume that $D_{4,T_2} = 0$ and $D_{2,T_2} < 0$, i.e. we are in Phase 3, Case 2. Then, there exists $\iota_M \in [\iota_L, \iota_R]$ such that $\mathbb{P}_{\iota\sim\rho_0}(|\iota| > \iota_M) \lesssim \gamma_4$ and for all $\iota, \iota' \in [\iota_L, \iota_M]$,*

$$\left| \Phi(w_{T_2}(\iota)) - \Phi(w_{T_2}(\iota')) \right| \leq 2\log\frac{1}{\kappa} + \log\frac{1}{\xi} + C \tag{D.205}$$

*for some universal constant $C$.*

*Proof of Lemma D.24.* Since $D_{4,T_2} = 0$, we have $\mathbb{E}_{w\sim\rho_{T_2}}[P_{4,d}(w)] = \gamma_4$. Thus, by Eq. (C.4),

$$\frac{d^2+6d+8}{d^2-1} \mathop{\mathbb{E}}_{w\sim\rho_{T_2}}[w^4] - \frac{6d+12}{d^2-1} \mathop{\mathbb{E}}_{w\sim\rho_{T_2}}[w^2] + \frac{3}{d^2-1} = \gamma_4 \tag{D.206}$$

Since $|w_t(\iota)| \leq 1$ for all $t \geq 0$ and all $\iota \in [-1, 1]$, the left-hand side of the above equation is $\mathbb{E}_{w\sim\rho_t}[w^4] \pm O(\frac{1}{d})$, and rearranging gives

$$\mathop{\mathbb{E}}_{w\sim\rho_t}[w^4] \leq \gamma_4 + O\left(\frac{1}{d}\right) \tag{D.207}$$

By Markov's inequality,

$$\mathbb{P}_{w\sim\rho_{T_2}}(|w| \geq 1/2) \lesssim \gamma_4 + O\left(\frac{1}{d}\right) \lesssim \gamma_4 \tag{D.208}$$

where the second inequality is because $d \geq c_3$ by Assumption 3.2. Let $\iota_0$ be such that $w_{T_2}(\iota_0) = \frac{1}{2}$. Let $\iota_M = \min(\iota_0, \iota_R)$. Then, $\iota_M \in [\iota_L, \iota_R]$, and $\mathbb{P}_{\iota\sim\rho_0}(|\iota| > \iota_M) \lesssim \gamma_4$, because $\mathbb{P}_{\iota\sim\rho_0}(|\iota| > \iota_0) \lesssim \gamma_4$ and $\mathbb{P}_{\iota\sim\rho_0}(|\iota| > \iota_R) \lesssim O(\kappa)$ by Proposition D.6, meaning that $\mathbb{P}_{\iota\sim\rho_0}(|\iota| > \iota_M) \lesssim \max(\gamma_4, \kappa) \lesssim \gamma_4$ (where the last inequality is because $d \geq c_3$ by Assumption 3.2).

Now, let $\iota \in [\iota_L, \iota_M]$. Observe that on the interval $[0, T_2]$, for all $\iota$, $w_t(\iota)$ is strictly increasing as a function of $t$, for all $\iota$ (see Lemma D.9). Thus, if $T_R$ is as defined in Lemma D.7, then for all $\iota \in [\iota_L, \iota_R]$, $|w_{T_R}(\iota)| \gtrsim \xi\kappa^2$. In particular, for all $\iota \in [\iota_L, \iota_M]$, we can write $w_{T_2}(\iota) \geq C\xi\kappa^2$ for some universal constant $C$. Additionally, for $\iota \leq \iota_M$, we have $w_{T_2}(\iota) \leq \frac{1}{2}$ by Proposition D.37. Thus, since $\Phi(w)$ is increasing in $w$, for $\iota \in [\iota_L, \iota_M]$ we have

$$\Phi(w_{T_2}(\iota)) \geq \Phi(C\xi\kappa^2) = \log\left(\frac{C\xi\kappa^2}{\sqrt{1 - C^2\xi^2\kappa^4}}\right) \geq \log(C\xi\kappa^2) = \log C + \log\xi + 2\log\kappa \tag{D.209}$$

and

$$\Phi(w_{T_2}(\iota)) \le \Phi(1/2) = \log\left(\frac{1/2}{\sqrt{3/4}}\right) = \log(1/\sqrt{3}) = -\frac{1}{2}\log 3 \tag{D.210}$$

Thus, for any $\iota, \iota' \in [\iota_{\mathrm{L}}, \iota_{\mathrm{M}}]$,

$$\left|\Phi(w_{T_2}(\iota)) - \Phi(w_{T_2}(\iota'))\right| \le -\frac{1}{2}\log 3 - \left(\log C + \log \xi + 2\log \kappa\right) = 2\log\frac{1}{\kappa} + \log\frac{1}{\xi} + C' \tag{D.211}$$

where we have defined the constant $C' = -\frac{1}{2}\log 3 - \log C$. $\qquad\square$

In the next lemma, we show that for $\iota \in [\iota_{\mathrm{L}}, \iota_{\mathrm{M}}]$, during Phase 3, Case 2, $w_t(\iota)$ can be bounded away from 0 and 1, provided that the conclusion of Lemma D.24 holds at time $t$.

**Lemma D.25** (Lower and Upper Bounds on Particles Between $\iota_{\mathrm{L}}$ and $\iota_{\mathrm{M}}$). *Suppose that we are in the setting of Theorem 3.3. Let $s \ge T_2$, and suppose that for all $t \in [T_2, s]$, $D_{4,t} \ge 0$ and $D_{2,t} \le 0$. Further suppose that for all $t \in [T_2, s]$, the conclusion of Lemma D.24 holds, i.e.*

$$\left|\Phi(w_s(\iota)) - \Phi(w_s(\iota'))\right| \le 2\log\frac{1}{\kappa} + \log\frac{1}{\xi} + C \tag{D.212}$$

*for all $\iota \in [\iota_{\mathrm{L}}, \iota_{\mathrm{M}}]$, where $\iota_{\mathrm{M}}$ is such that $\mathbb{P}_{\iota \sim \rho_0}(|\iota| > \iota_{\mathrm{M}}) \lesssim \gamma_4$, and for some universal constant $C$. Then, for any $t \in [T_2, s]$, for all $\iota \ge \iota_{\mathrm{L}}$, $w_t(\iota) \ge \xi^2 \kappa^2$. Furthermore, for all $t \in [T_2, s]$, we have $w_t(\iota_{\mathrm{M}}) \le 1 - \kappa^4 \xi^3$.*

*Proof of Lemma D.25.* Consider some time $t \in [T_2, s]$. Suppose that for some $\iota \in [\iota_{\mathrm{L}}, \iota_{\mathrm{M}}]$, $w_t(\iota) \le \xi^2\kappa^2$. Also, suppose $\iota' \in [\iota_{\mathrm{L}}, \iota_{\mathrm{M}}]$ such that $\iota' > \iota$. Then,

$$\Phi(w_t(\iota')) \le \Phi(w_t(\iota)) + 2\log\frac{1}{\kappa} + \log\frac{1}{\xi} + C$$

$$\text{(By Eq. (D.212), and b.c. } \Phi \text{ is increasing, } \iota' > \iota \text{ and Proposition D.37)}$$

$$\le \log\left(\frac{\xi^2\kappa^2}{\sqrt{1-\xi^4\kappa^4}}\right) + 2\log\frac{1}{\kappa} + \log\frac{1}{\xi} + C \tag{D.213}$$

$$\le \log(2\xi^2\kappa^2) + 2\log\frac{1}{\kappa} + \log\frac{1}{\xi} + C \qquad \text{(B.c. } d \ge c_3, \text{ Assumption 3.2)}$$

$$= \log\xi + C' \qquad\qquad\qquad\qquad\qquad \text{(For some universal constant } C')$$

Thus, because $w \le \frac{w}{\sqrt{1-w^2}}$,

$$\log w_t(\iota') \le \log\left(\frac{w_t(\iota')}{\sqrt{1-(w_t(\iota'))^2}}\right) \le \log\xi + C' \tag{D.214}$$

Thus, $w_t(\iota') \lesssim \xi$ and for all $\iota \in [\iota_{\mathrm{L}}, \iota_{\mathrm{M}}]$, we have shown that $w_t(\iota) \lesssim \xi$. In addition, $\mathbb{P}_{\iota \sim \rho_0}(|\iota| > \iota_{\mathrm{M}}) \lesssim \gamma_4$. We can use this to calculate $D_{2,t}$ and $D_{4,t}$:

$$D_{2,t} = \mathop{\mathbb{E}}_{w\sim\rho_t}\left[\frac{d}{d-1}w^2 - \frac{1}{d-1}\right] - \gamma_2 \qquad\qquad \text{(By definition of } P_{2,d}, \text{ (Eq. (C.4)))}$$

$$\le \mathop{\mathbb{E}}_{w\sim\rho_t}[w^2] - \gamma_2 \qquad\qquad\qquad\qquad\qquad\qquad\qquad \text{(B.c. } w \le 1)$$

$$\le \mathbb{P}_{\iota\sim\rho_0}(|\iota| \le \iota_{\mathrm{M}})\,\mathbb{E}[w_t(\iota)^2 \mid |\iota| \le \iota_{\mathrm{M}}] + \mathbb{P}_{\iota\sim\rho_0}(|\iota| > \iota_{\mathrm{M}})\,\mathbb{E}[w_t(\iota)^2 \mid |\iota| \ge \iota_{\mathrm{M}}] - \gamma_2 \tag{D.215}$$

$$\le O(\xi^2) + \gamma_4 - \gamma_2 \qquad\qquad \text{(B.c. } w_t(\iota) \lesssim \xi \text{ for } \iota \in [\iota_{\mathrm{L}}, \iota_{\mathrm{M}}] \text{ and } \mathbb{P}(|\iota| > \iota_{\mathrm{M}}) \lesssim \gamma_4)$$

$$\lesssim -\gamma_2 \qquad \text{(B.c. } \gamma_4 \le c_1\gamma_2^2, \gamma_2 \le c_2 \text{ (}c_2 \text{ chosen after } c_1\text{) and } d \ge c_3, \text{ Assumption 3.2)}$$

and

$$D_{4,t} = \mathop{\mathbb{E}}_{w \sim \rho_t} \left[ \frac{d^2 + 6d + 8}{d^2 - 1} w^4 - \frac{6d + 12}{d^2 - 1} w^2 + \frac{3}{d^2 - 1} \right] - \gamma_4$$

(By definition of $P_{4,d}$, Eq. (C.4))

$$= \mathop{\mathbb{E}}_{w \sim \rho_t} [w^4] - \gamma_4 \pm O\left(\frac{1}{d}\right) \tag{D.216}$$

$$= \mathbb{P}_{\iota \sim \rho_0}(|\iota| \leq \iota_{\mathrm{M}}) \, \mathbb{E}[w_t(\iota)^4 \mid |\iota| \leq \iota_{\mathrm{M}}] + \mathbb{P}_{\iota \sim \rho_0}(|\iota| > \iota_{\mathrm{M}}) \, \mathbb{E}[w_t(\iota)^4 \mid |\iota| > \iota_{\mathrm{M}}] - \gamma_4 \pm O\left(\frac{1}{d}\right) \tag{D.217}$$

$$\leq O(\xi^4) + O(\gamma_4) - \gamma_4 \pm O\left(\frac{1}{d}\right) \qquad \text{(B.c. } w_t(\iota) \lesssim \xi \text{ for } \iota \in [\iota_{\mathrm{L}}, \iota_{\mathrm{M}}] \text{ and } \mathbb{P}(\iota > \iota_{\mathrm{M}}) \lesssim \gamma_4 \text{)}$$

$$\leq O(\xi^4 + \gamma_4) \tag{D.218}$$

$$\lesssim \gamma_4 \qquad \text{(B.c. } d \geq c_3, \text{ by Assumption 3.2)}$$

To complete the proof, we will show that if these bounds on $D_{2,t}$ and $D_{4,t}$ hold, then for any $w$, $v_t(w) \geq 0$. By Lemma 4.4, it suffices to show that $P_t(w) + Q_t(w) \leq 0$ for $w \geq 0$, where $P_t$ and $Q_t$ are as defined in Lemma 4.4. If $C_1$ and $C_2$ are two appropriately chosen positive universal constants, then by the above upper bounds on $D_{2,t}$ and $D_{4,t}$,

$$P_t(w) + Q_t(w) = 2\hat{\sigma}_{2,d}^2 D_{2,t} w + 4\hat{\sigma}_{4,d}^2 D_{4,t} w^3 + \lambda_d^{(1)} w + \lambda_d^{(3)} w^3 \tag{D.219}$$

$$\leq -C_1 \hat{\sigma}_{2,d}^2 \gamma_2 w + C_2 \hat{\sigma}_{4,d}^2 \cdot \gamma_4 w^3 + O\left(\frac{\hat{\sigma}_{2,d}^2 |D_{2,t}| + \hat{\sigma}_{4,d}^2 |D_{4,t}|}{d}\right) w \tag{D.220}$$

$$+ O\left(\frac{\hat{\sigma}_{4,d}^2 |D_{4,t}|}{d}\right) w^3$$

(By definition of $\lambda_d^{(1)}, \lambda_d^{(3)}$ in Lemma 4.4 and above bounds on $D_{2,t}, D_{4,t}$)

$$\leq -C_1' \hat{\sigma}_{2,d}^2 \gamma_2 w + O\left(\frac{\hat{\sigma}_{2,d}^2 |D_{2,t}| + \hat{\sigma}_{4,d}^2 |D_{4,t}|}{d}\right) |w|$$

(B.c. $\gamma_4 \lesssim \gamma_2^2 \leq \gamma_2$ and $\hat{\sigma}_{4,d}^2 \lesssim \hat{\sigma}_{2,d}^2$ by Assumption 3.2, with new constant $C_1'$)

$$\lesssim -\hat{\sigma}_{2,d}^2 \gamma_2 w \qquad \text{(B.c. } d \geq c_3, \text{ Assumption 3.2)}$$

In summary, if $w_t(\iota) = \xi^2 \kappa^2$ for some $\iota \in [\iota_{\mathrm{L}}, \iota_{\mathrm{M}}]$, then $v_t(w) > 0$ for all $w \in [0, 1]$. In addition, at time $T_2$, we have $w_t(\iota) \gtrsim \xi\kappa^2$ for all $\iota > \iota_{\mathrm{L}}$ by Lemma D.9 and Lemma D.7. Thus, for all $\iota \geq \iota_{\mathrm{L}}$, $w_t(\iota)$ will never go below $\xi^2 \kappa^2$ for $t \in [T_2, s]$, as desired.

To finish the proof, we will show the upper bound on $w_t(\iota_{\mathrm{M}})$. Suppose $1 - w_t(\iota_{\mathrm{M}}) \leq \kappa^4 \xi^3$. Then,

$$\Phi(w_t(\iota_{\mathrm{M}})) = \log \left( \frac{w_t(\iota_{\mathrm{M}})}{\sqrt{1 - w_t(\iota_{\mathrm{M}})^2}} \right) \tag{D.221}$$

$$\geq \log \frac{1}{2} - \frac{1}{2} \log(1 - w_t(\iota_{\mathrm{M}})^2) \qquad \text{(B.c. } w_t(\iota_{\mathrm{M}}) \geq \tfrac{1}{2} \text{)}$$

$$\geq \log \frac{1}{2} - \frac{1}{2} \log(1 - (1 - \kappa^4 \xi^3)^2) \tag{D.222}$$

$$= \log \frac{1}{2} - \frac{1}{2} \log(2\kappa^4 \xi^3 - \kappa^8 \xi^6) \tag{D.223}$$

$$\geq \log \frac{1}{2} - \frac{1}{2} \log(\kappa^4 \xi^3) \qquad \text{(B.c. } \kappa, \xi \leq 1 \text{ and } \kappa^4 \xi^3 - (\kappa^4 \xi^3)^2 \geq 0 \text{)}$$

$$= \log \frac{1}{2} + 2 \log \frac{1}{\kappa} + \frac{3}{2} \log \frac{1}{\xi} \tag{D.224}$$

Thus, for any $\iota \in [\iota_{\mathrm{L}}, \iota_{\mathrm{M}}]$,

$$\Phi(w_t(\iota)) \geq C + \frac{1}{2} \log \frac{1}{\xi} \tag{D.225}$$

for some universal constant $C$. In other words, letting $w_t := w_t(\iota)$ and rearranging using the definition of $\Phi$, we obtain

$$\frac{w_t}{\sqrt{1 - w_t^2}} \gtrsim \frac{1}{\xi^{1/2}} \tag{D.226}$$

or

$$\sqrt{1 - w_t^2} \lesssim \xi^{1/2} w_t \lesssim \xi^{1/2} \tag{D.227}$$

which finally gives $1 - w_t^2 \lesssim \xi$, or $w_t^2 \geq 1 - O(\xi)$. In other words, for all $\iota \geq \iota_{\mathrm{L}}$, we have $w_t(\iota) \geq 1 - O(\xi)$, which implies that

$$D_{2,t} = \mathbb{E}_{w \sim \rho_t}[P_{2,d}(w)] - \gamma_2 \tag{D.228}$$

$$\gtrsim \mathbb{E}_{w \sim \rho_t}[w^2] - \gamma_2 - O\Big(\frac{1}{d}\Big) \tag{D.229}$$

$$\geq \mathbb{P}_{\iota \sim \rho_0}(|\iota| \geq \iota_{\mathrm{L}}) \, \mathbb{E}_{w \sim \rho_t}[w^2 \mid |\iota| \geq \iota_{\mathrm{L}}] - \gamma_2 - O\Big(\frac{1}{d}\Big) \tag{D.230}$$

$$\geq (1 - O(\kappa)) \cdot (\mathbb{E}_{w \sim \rho_t}[w^2 \mid |\iota| \geq \iota_{\mathrm{L}}]) - \gamma_2 - O\Big(\frac{1}{d}\Big) \qquad \text{(By Proposition D.6)}$$

$$\geq (1 - O(\kappa))(1 - O(\xi)) - \gamma_2 - O\Big(\frac{1}{d}\Big) \qquad \text{(B.c. if } |\iota| > \iota_{\mathrm{L}}, \text{ then } w_t(\iota)^2 \geq 1 - O(\xi))$$

$$\geq 1 - O(\xi + \kappa) - \gamma_2 \qquad \text{(B.c. } 1/d \text{ is small by Assumption 3.2)}$$

$$> 0 \qquad \text{(By Assumption 3.2 b.c. } \gamma_2 \text{ small and } d \text{ large)}$$

and this contradicts the assumption that for all $t \in [T_2, s]$, $D_{4,t} \geq 0$ and $D_{2,t} \leq 0$. Thus, for all $t \in [T_2, s]$, $1 - w_t(\iota_{\mathrm{M}}) \geq \kappa^4 \xi^3$, i.e. $w_t(\iota_{\mathrm{M}}) \leq 1 - \kappa^4 \xi^3$. $\qquad\square$

The purpose of the next two lemmas is to show that if Phase 3, Case 2 occurs, then for all $t \geq T_2$, we will have $D_{2,t} \leq 0$ and $D_{4,t} \geq 0$, assuming the conclusion of Lemma D.25 holds. We will later put these next two lemmas with the above lemma, Lemma D.25, in order to inductively show that the hypothesis of Lemma D.25 holds for $t \geq T_2$ (note that Lemma D.25 assumes that the upper bound on the potential difference holds on an interval $[T_2, s]$, and more importantly assumes that $D_{2,t} \leq 0$ and $D_{4,t} \geq 0$ for all $t \in [T_2, s]$).

**Lemma D.26.** *Suppose we are in the setting of Theorem 3.3, and suppose $D_{4,t} = 0$ and $D_{2,t} < 0$ for some $t \geq 0$. Additionally, suppose that the conclusion of Lemma D.25 holds at time $t$, i.e. for all $\iota \geq \iota_{\mathrm{L}}$, $w_t(\iota) \geq \xi^2 \kappa^2$, and $w_t(\iota_{\mathrm{M}}) \leq 1 - \kappa^4 \xi^3$. Then, $\frac{d}{ds} D_{4,s}\big|_{s=t} > 0$.*

*Proof.* Recall from Eq. (C.4) that

$$P_{4,d}(t) = \frac{d^2 + 6d + 8}{d^2 - 1} t^4 - \frac{6d + 12}{d^2 - 1} t^2 + \frac{3}{d^2 - 1} \tag{D.231}$$

meaning that

$$P_{4,d}'(t) = 4t^3 \pm O\Big(\frac{1}{d}\Big)|t| \tag{D.232}$$

Next we compute the derivative of $D_{4,t}$:

$$\frac{d}{dt} D_{4,t} = \frac{d}{dt} \mathbb{E}_{w \sim \rho_t}[P_{4,d}(w)] = \mathbb{E}_{w \sim \rho_t}[P_{4,d}'(w) \cdot v_t(w)] \tag{D.233}$$

For positive particles $w \gtrsim \frac{1}{\sqrt{d}}$, we have $P_{4,d}'(w) \geq 0$, and for $w \lesssim \frac{1}{\sqrt{d}}$, it may be the case that $P_{4,d}'(w) \leq 0$. Furthermore, when $D_{4,t} = 0$ and $D_{2,t} \leq 0$, we have

$$v_t(w) = -(1 - w^2)\Big(2\hat{\sigma}_{2,d}^2 D_{2,t} w + O\Big(\frac{\hat{\sigma}_{2,d}^2 |D_{2,t}|}{d}\Big)|w|\Big)$$
$$\text{(By Lemma 4.4 and because } D_{4,t} = 0)$$

$$= (1 - w^2) \cdot \Big(2\hat{\sigma}_{2,d}^2 |D_{2,t}| w \pm O\Big(\frac{\hat{\sigma}_{2,d}^2 |D_{2,t}|}{d}\Big)|w|\Big) \qquad \text{(B.c. } D_{2,t} \leq 0)$$

$$= (1 - w^2) \cdot 2\hat{\sigma}_{2,d}^2 |D_{2,t}| w \cdot (1 \pm O(1/d)) \tag{D.234}$$

Finally, to lower bound $\frac{d}{dt}D_{4,t}$, we must show that there is a significant fraction of particles for which $w \gtrsim \frac{1}{\sqrt{d}}$ and $1 - w$ is large. Because we have $w_t(\iota) \geq \xi^2\kappa^2$ for all $\iota \geq \iota_{\mathrm{L}}$, and $\mathbb{P}(|\iota| < \iota_{\mathrm{L}}) \lesssim \kappa$ by Proposition D.6, using a union bound we therefore know that

$$\mathbb{P}_{w\sim\rho_t}(|w| \in [\xi^2\kappa^2, 1 - \kappa^4\xi^3]) \geq 1 - O(\gamma_4) - O(\kappa) \geq 1 - O(\gamma_4) \tag{D.235}$$

where the second inequality is by Assumption 3.2 since $d$ is sufficiently large. Thus,

$$\frac{d}{dt}D_{4,t} = \mathop{\mathbb{E}}_{w\sim\rho_t}[P_{4,d}{}'(w) \cdot v_t(w)] \tag{D.236}$$

$$= \mathbb{P}_{w\sim\rho_t}(|w| \lesssim 1/\sqrt{d}) \mathop{\mathbb{E}}_{w\sim\rho_t}[P_{4,d}{}'(w) \cdot v_t(w) \mid |w| \lesssim 1/\sqrt{d}] \tag{D.237}$$

$$+ \mathbb{P}_{w\sim\rho_t}(|w| \gtrsim 1/\sqrt{d}) \mathop{\mathbb{E}}_{w\sim\rho_t}[P_{4,d}{}'(w) \cdot v_t(w) \mid |w| \gtrsim 1/\sqrt{d}] \tag{D.238}$$

$$\geq -O\Big(\frac{1}{d^{3/2}} \cdot \frac{\hat{\sigma}_{2,d}^2|D_{2,t}|}{d^{1/2}}\Big) + \mathbb{P}_{w\sim\rho_t}(|w| \gtrsim 1/\sqrt{d}) \mathop{\mathbb{E}}_{w\sim\rho_t}[P_{4,d}{}'(w) \cdot v_t(w) \mid |w| \gtrsim 1/\sqrt{d}]$$
$$\text{(By bounding } |P_{4,d}{}'(w) \cdot v_t(w)| \text{ when } |w| \lesssim 1/\sqrt{d})$$

$$\geq -O\Big(\frac{\hat{\sigma}_{2,d}^2|D_{2,t}|}{d^2}\Big) + \mathbb{P}_{w\sim\rho_t}(|w| \in [\xi^2\kappa^2, 1 - \kappa^4\xi^3]) \tag{D.239}$$

$$\mathop{\mathbb{E}}_{w\sim\rho_t}[P_{4,d}{}'(w) \cdot v_t(w) \mid |w| \in [\xi^2\kappa^2, 1 - \kappa^4\xi^3]] \tag{D.240}$$

$$\geq -O\Big(\frac{\hat{\sigma}_{2,d}^2|D_{2,t}|}{d^2}\Big) + (1 - O(\gamma_4)) \mathop{\mathbb{E}}_{w\sim\rho_t}[P_{4,d}{}'(w) \cdot v_t(w) \mid |w| \in [\xi^2\kappa^2, 1 - \kappa^4\xi^3]]$$
$$\text{(By Eq. (D.235))}$$

$$\gtrsim -O\Big(\frac{\hat{\sigma}_{2,d}^2|D_{2,t}|}{d^2}\Big) + (\xi^2\kappa^2)^3 \cdot (1 - (1 - \kappa^4\xi^3)^2)\hat{\sigma}_{2,d}^2|D_{2,t}|(\xi^2\kappa^2) \tag{D.241}$$

where the last inequality is by Eq. (D.232) and applying the above bound on $v_t(w)$ from Eq. (D.234) for $|w| \in [\xi^2\kappa^2, 1 - \kappa^4\xi^3]$, and because $d \geq c_3$ by Assumption 3.2. Expanding the last line gives

$$\frac{d}{dt}D_{4,t} \gtrsim -O\Big(\frac{\hat{\sigma}_{2,d}^2|D_{2,t}|}{d^2}\Big) + \hat{\sigma}_{2,d}^2|D_{2,t}|\xi^6\kappa^6\kappa^4\xi^3\xi^2\kappa^2 \tag{D.242}$$

$$\gtrsim -O\Big(\frac{\hat{\sigma}_{2,d}^2|D_{2,t}|}{d^2}\Big) + \hat{\sigma}_{2,d}^2|D_{2,t}|\xi^{11}\kappa^{12} \tag{D.243}$$

and the right hand side is strictly positive since $d$ is sufficiently large by Assumption 3.2. $\qquad\square$

**Lemma D.27.** *Suppose we are in the setting of Theorem 3.3, and suppose $D_{4,t} > 0$ and $D_{2,t} = 0$ for some $t \geq 0$. Additionally, suppose that the conclusion of Lemma D.25 holds at time $t$, i.e. for all $\iota \geq \iota_{\mathrm{L}}$, $w_t(\iota) \geq \xi^2\kappa^2$, and $w_t(\iota_{\mathrm{M}}) \leq 1 - \kappa^4\xi^3$. Then, $\frac{d}{ds}D_{2,s}\big|_{s=t} < 0$.*

*Proof.* Recall from Eq. (C.4) that

$$P_{2,d}(t) = \frac{d}{d-1}t^2 - \frac{1}{d-1} \tag{D.244}$$

meaning that

$$\frac{d}{dt}D_{2,t} = \frac{d}{dt}\mathop{\mathbb{E}}_{w\sim\rho_t}[P_{2,d}(t)] = \frac{2d}{d-1}\mathop{\mathbb{E}}_{w\sim\rho_t}[w \cdot v_t(w)] \tag{D.245}$$

Suppose $D_{2,t} = 0$ and $D_{4,t} > 0$. Then, the velocity is

$$v_t(w) = -(1 - w^2)(P_t(w) + Q_t(w)) \qquad\qquad \text{(By Lemma 4.4)}$$

$$= -(1 - w^2)\Big(4\hat{\sigma}_{4,d}^2 D_{4,t}w^3 \pm O\Big(\frac{\hat{\sigma}_{4,d}^2|D_{4,t}|}{d}\Big)|w|\Big) \qquad \text{(By Lemma 4.4)}$$

$$= -(1 - w^2) \cdot 4\hat{\sigma}_{4,d}^2 D_{4,t}\Big(w^3 \pm \frac{|w|}{d}\Big) \tag{D.246}$$

Thus, for positive particles $w$, if $w \gtrsim \frac{1}{\sqrt{d}}$, $v_t(w) \leq 0$, and for $w \lesssim \frac{1}{\sqrt{d}}$, the velocity is potentially positive. Using this, we can upper bound $\frac{d}{dt} D_{2,t}$ by upper bounding $\mathbb{E}_{w \sim \rho_t}[w \cdot v_t(w)]$:

$$\mathop{\mathbb{E}}_{w \sim \rho_t}[w \cdot v_t(w)] = \mathbb{P}_{w \sim \rho_t}\Big(|w| \lesssim \frac{1}{\sqrt{d}}\Big) \mathop{\mathbb{E}}_{w \sim \rho_t}\Big[w \cdot v_t(w) \mid |w| \lesssim \frac{1}{\sqrt{d}}\Big] \tag{D.247}$$

$$+ \mathbb{P}_{w \sim \rho_t}\Big(|w| \gtrsim \frac{1}{\sqrt{d}}\Big) \mathop{\mathbb{E}}_{w \sim \rho_t}\Big[w \cdot v_t(w) \mid |w| \gtrsim \frac{1}{\sqrt{d}}\Big] \tag{D.248}$$

$$\leq O\Big(\frac{1}{d^{1/2}} \cdot \frac{\hat{\sigma}_{4,d}^2 |D_{4,t}|}{d^{3/2}}\Big) + \mathbb{P}_{w \sim \rho_t}\Big(|w| \gtrsim \frac{1}{\sqrt{d}}\Big) \mathop{\mathbb{E}}_{w \sim \rho_t}\Big[w \cdot v_t(w) \mid |w| \gtrsim \frac{1}{\sqrt{d}}\Big]$$
$$\text{(By bounding } w \cdot v_t(w) \text{ for } 0 \leq w \lesssim \tfrac{1}{\sqrt{d}})$$

$$\leq O\Big(\frac{\hat{\sigma}_{4,d}^2 |D_{4,t}|}{d^2}\Big) + \mathbb{P}_{w \sim \rho_t}\Big(|w| \in [\xi^2 \kappa^2, 1 - \kappa^4 \xi^3]\Big) \tag{D.249}$$
$$\mathop{\mathbb{E}}_{w \sim \rho_t}\Big[w \cdot v_t(w) \mid |w| \in [\xi^2 \kappa^2, 1 - \kappa^4 \xi^3]\Big]$$
$$\text{(B.c. for } w \gtrsim \tfrac{1}{\sqrt{d}}, v_t(w) < 0.)$$

$$\leq O\Big(\frac{\hat{\sigma}_{4,d}^2 |D_{4,t}|}{d^2}\Big) + (1 - O(\kappa) - O(\gamma_4)) \mathop{\mathbb{E}}_{w \sim \rho_t}\Big[w \cdot v_t(w) \mid |w| \in [\xi^2 \kappa^2, 1 - \kappa^4 \xi^3]\Big] \tag{D.250}$$

where the last inequality is because $\mathbb{P}(|\iota| > \iota_{\mathrm{M}}) \leq O(\gamma_4)$ (by the definition of $\iota_{\mathrm{M}}$, see statement of Lemma D.25) and $\mathbb{P}(|\iota| < \iota_{\mathrm{L}}) \leq O(\kappa)$ by Proposition D.6. Simplifying further, we obtain,

$$\mathop{\mathbb{E}}_{w \sim \rho_t}[w \cdot v_t(w)] \leq O\Big(\frac{\hat{\sigma}_{4,d}^2 |D_{4,t}|}{d^2}\Big) - (1 - O(\kappa) - O(\gamma_4))(\xi^2 \kappa^2) \cdot (1 - (1 - \kappa^4 \xi^3)^2)\hat{\sigma}_{4,d}^2 D_{4,t}(\xi^2 \kappa^2)^3$$
$$\text{(By Eq. (D.246) and b.c. } \tfrac{1}{d} \text{ is small (Assumption 3.2))}$$

$$\lesssim O\Big(\frac{\hat{\sigma}_{4,d}^2 |D_{4,t}|}{d}\Big) - \hat{\sigma}_{4,d}^2 D_{4,t} \cdot \xi^2 \kappa^2 \cdot \kappa^4 \xi^3 \xi^6 \kappa^6$$
$$\text{(By Assumption 3.2, } 1 - O(\kappa) - O(\gamma_4) \gtrsim 1)$$

$$\lesssim O\Big(\frac{\hat{\sigma}_{4,d}^2 |D_{4,t}|}{d}\Big) - \hat{\sigma}_{4,d}^2 D_{4,t} \xi^{11} \kappa^{12} \tag{D.251}$$

and the last expression is strictly negative, since $d \geq c_3$ by Assumption 3.2 and $D_{4,t} \geq 0$. This proves the lemma. $\qquad\square$

Now, we combine Lemma D.25, Lemma D.26 and Lemma D.27 to inductively show that for all $t \geq T_2$, we have $D_{2,t} \leq 0$, $D_{4,t} \geq 0$ and most importantly, $w_t(\iota)$ can be bounded away from 0 and 1 for $\iota \in [\iota_{\mathrm{L}}, \iota_{\mathrm{M}}]$.

**Lemma D.28** (Phase 3 Case 2 Invariants)**.** *Suppose we are in the setting of Theorem 3.3. Assume $D_{2,T_2} < 0$ and $D_{4,T_2} = 0$, i.e. Phase 3 Case 2 holds. Then, for all $t \geq T_2$, the following will hold:*

1. *$D_{2,t} \leq 0$ and $D_{4,t} \geq 0$*

2. *Let $\iota_{\mathrm{M}}$ be defined as in Lemma D.25. Then, $w_t(\iota) \geq \xi^2 \kappa^2$ for all $\iota \in [\iota_{\mathrm{L}}, \iota_{\mathrm{M}}]$ and $w_t(\iota_{\mathrm{M}}) \leq 1 - \kappa^4 \xi^3$.*

*Proof of Lemma D.28.* For the purpose of this proof, define

$$I = \{s \geq T_2 \mid D_{2,s} \leq 0 \text{ and } D_{4,s} \geq 0\} \tag{D.252}$$

and let $t_0 = \sup I$, meaning that for $t \in [T_2, t_0]$, $D_{2,t} \leq 0$ and $D_{4,t} \geq 0$. Assume for the sake of contradiction that $t_0 < \infty$. Then, by the continuity of $D_{2,t}$ and $D_{4,t}$ as functions of $t$, $D_{2,t} \leq 0$ and $D_{4,t} \geq 0$. Since $t_0 = \sup I$, either $D_{2,t_0} = 0$ or $D_{4,t_0} = 0$ (and one of these has to be nonzero since otherwise $t_0 = \infty$, which is a contradiction). By Lemma D.24 and Lemma D.23, for all $t \in [T_2, t_0]$, for all $\iota, \iota' \in [\iota_{\mathrm{L}}, \iota_{\mathrm{M}}]$,

$$\Big|\Phi(w_t(\iota)) - \Phi(w_t(\iota'))\Big| \leq 2 \log \frac{1}{\kappa} + \log \frac{1}{\xi} + C \tag{D.253}$$

where $C$ is a universal constant. Thus, applying Lemma D.25 with $s = t_0$, we find that for all $t \in [T_2, t_0]$, for all $\iota \geq \iota_L$, $w_t(\iota) \geq \xi^2 \kappa^2$, and furthermore, for all $t \in [T_2, t_0]$, $w_t(\iota_M) \leq 1 - \kappa^4 \xi^3$.

To finish the proof, there are two cases to address: either $D_{2,t_0} < 0$ and $D_{4,t_0} = 0$, or $D_{2,t_0} = 0$ and $D_{4,t_0} > 0$. In the first case, by Lemma D.26, $\frac{d}{dt} D_{4,t}\big|_{t=t_0} > 0$, meaning that for a sufficiently small open interval of time $J$ containing $t_0$, $D_{4,t}$ is increasing on this interval. This gives us a contradiction, since it implies that for some $t_0' > t_0$ (with $t_0' - t_0$ sufficiently small) $D_{4,t}$ is nonnegative on $[t_0, t_0']$, and since $D_{2,t_0} < 0$, by continuity, $D_{2,t} < 0$ for $t \in [t_0, t_0']$ as long as $t_0' - t_0 > 0$ is sufficiently small. Thus, $\sup I > t_0$, which is a contradiction. In the second case, we can apply the same argument using Lemma D.27 in place of Lemma D.26.

In summary, if $D_{4,T_2} = 0$ and $D_{2,T_2} < 0$, then for all $t \geq T_2$, $D_{2,t} \leq 0$ and $D_{4,t} \geq 0$. As a consequence of Lemma D.23, Lemma D.24, and Lemma D.25, for all $\iota \in [\iota_L, \iota_M]$, $w_t(\iota) \geq \xi^2 \kappa^2$, and furthermore, $w_t(\iota_M) \leq 1 - \kappa^4 \xi^3$, for all $t \geq T_2$. $\qquad\square$

We now proceed to bound the running time of Phase 3, Case 2, by dividing it into a few subcases, and showing that the running time that each of these subcases contribute is within our desired bound. The first lemma below bounds the running time spent in the subcase where the positive root $r$ of $P_t$ is larger than $\frac{1}{2}$ (and where $D_{4,t}$ is large relative to $D_{2,t}$).

**Lemma D.29.** *Suppose we are in the setting of Theorem 3.3, and assume that $D_{4,T_2} = 0$ and $D_{2,T_2} < 0$, i.e. we are in Phase 3, Case 2. Then, at most $\frac{1}{\hat{\sigma}_{2,d}^2 \xi^4 \kappa^6} \log\left(\frac{\gamma_2}{\epsilon}\right)$ time $t \geq T_2$ elapses such that the following hold:*

1. *If $r$ is the positive root of $P_t$ (as defined in Lemma 4.4) then $r \geq \frac{1}{2}$.*

2. *$|D_{4,t}| \geq \xi |D_{2,t}|$*

3. *$L(\rho_t) \geq (\hat{\sigma}_{2,d}^2 + \hat{\sigma}_{4,d}^2)\epsilon^2$*

*Proof of Lemma D.29.* Let $t \geq T_2$ such that $|D_{4,t}| \geq \xi |D_{2,t}|$. We can write

$$P_t(w) = 2\hat{\sigma}_{2,d}^2 D_{2,t} w + 4\hat{\sigma}_{4,d}^2 D_{4,t} w^3 = 4\hat{\sigma}_{4,d}^2 D_{4,t} w(w - r)(w + r) \qquad (D.254)$$

where $r$ is the positive root of $P_t(w)$. We are now going to lower bound the expected velocity while $r \geq \frac{1}{2}$ by lower bounding $P_t(w)$ for a large fraction of $w$.

For all $\iota \geq \iota_L$, and $t \geq T_2$, we have $w_t(\iota) \geq \xi^2 \kappa^2$, by Lemma D.28. Thus, $\mathbb{P}_{w \sim \rho_t}(w \geq \xi^2 \kappa^2) \leq 1 - O(\kappa)$ by Proposition D.6. Additionally, because $D_{2,t} \leq 0$, we have $\mathbb{E}_{w \sim \rho_t}[w^2] \leq \gamma_2 + O(1/d)$ by Proposition D.38, meaning that by Markov's inequality, $\mathbb{P}_{w \sim \rho_t}(|w| \geq \frac{1}{3}) \lesssim \gamma_2 + 1/d \leq \gamma_2$, where the last inequality is because $d \geq c_3$ by Assumption 3.2. By a union bound,

$$\mathbb{P}_{w \sim \rho_t}(|w| \in [\xi^2 \kappa^2, 1/3]) \geq 1 - O(\gamma_2 + \kappa) \qquad (D.255)$$

For particles $w$ such that $|w| \in [\xi^2 \kappa^2, 1/3]$, we can lower bound the velocity if $r \geq \frac{1}{2}$:

$$
\begin{aligned}
|v_t(w)| &\gtrsim (1 - w^2)|P_t(w) + Q_t(w)| && \text{(By Lemma 4.4)} \\
&\gtrsim |P_t(w) + Q_t(w)| && \text{(B.c. } w \leq 1/3\text{)} \\
&\gtrsim |P_t(w)| - |Q_t(w)| && \text{(D.256)} \\
&\gtrsim |2\hat{\sigma}_{2,d}^2 D_{2,t} w + 4\hat{\sigma}_{4,d}^2 D_{4,t} w^3| - |\lambda_d^{(1)} w + \lambda_d^{(3)} w^3| && \text{(By Lemma 4.4)} \\
&\gtrsim |2\hat{\sigma}_{2,d}^2 D_{2,t} w + 4\hat{\sigma}_{4,d}^2 D_{4,t} w^3| - O\left(\frac{\hat{\sigma}_{2,d}^2 |D_{2,t}| + \hat{\sigma}_{4,d}^2 |D_{4,t}|}{d}\right)|w| && \text{(By Lemma 4.4)} \\
&\gtrsim \hat{\sigma}_{4,d}^2 D_{4,t} |w||w - r||w + r| - O\left(\frac{\hat{\sigma}_{2,d}^2 |D_{2,t}| + \hat{\sigma}_{4,d}^2 |D_{4,t}|}{d}\right) && \text{(By definition of } r\text{)} \\
&\gtrsim \hat{\sigma}_{4,d}^2 D_{4,t} \cdot \xi^2 \kappa^2 \cdot \frac{1}{6} \cdot \frac{1}{2} - O\left(\frac{\hat{\sigma}_{2,d}^2 |D_{2,t}| + \hat{\sigma}_{4,d}^2 |D_{4,t}|}{d}\right) && \\
& && \text{(B.c. } w \in [\xi^2 \kappa^2, 1/3] \text{ and } r \geq 1/2\text{)} \\
&\gtrsim \hat{\sigma}_{4,d}^2 D_{4,t} \xi^2 \kappa^2 && \text{(D.257)}
\end{aligned}
$$

where the last inequality is because $|D_{4,t}| \geq \xi|D_{2,t}|$ and $d$ is sufficiently large by Assumption 3.2. Thus,

$$\mathbb{E}_{w\sim\rho_t} |v_t(w)|^2 \gtrsim \mathbb{P}_{w\sim\rho_t}(|w| \in [\xi^2\kappa^2, 1/3]) \cdot \mathbb{E}_{w\sim\rho_t}[|v_t(w)|^2 \mid |w| \in [\xi^2\kappa^2, 1/3]] \tag{D.258}$$

$$\gtrsim (1 - O(\gamma_2 + \kappa)) \cdot \hat{\sigma}_{4,d}^4 D_{4,t}^2 \xi^4 \kappa^4 \qquad \text{(By Eq. (D.255))}$$

$$\gtrsim \hat{\sigma}_{4,d}^4 D_{4,t}^2 \xi^4 \kappa^4 \qquad \text{(By Assumption 3.2 since } \gamma_2, d \text{ sufficiently small, large resp.)}$$

and

$$\frac{dL(\rho_t)}{dt} \lesssim -\hat{\sigma}_{4,d}^4 D_{4,t}^2 \xi^4 \kappa^4 \qquad \text{(By Lemma D.39)}$$

$$\lesssim -\hat{\sigma}_{4,d}^4 (D_{4,t}^2 + \xi^2 D_{2,t}^2)\xi^4\kappa^4 \qquad \text{(B.c. } |D_{4,t}| \geq \xi|D_{2,t}|)$$

$$\lesssim -\hat{\sigma}_{4,d}^4 (D_{4,t}^2 + D_{2,t}^2)\xi^4\kappa^6 \tag{D.259}$$

$$\lesssim -\hat{\sigma}_{4,d}^2 \xi^4\kappa^6 L(\rho_t) \qquad \text{(B.c. } \hat{\sigma}_{2,d}^2 \asymp \hat{\sigma}_{4,d}^2 \text{ by Assumption 3.2)}$$

Since $L(\rho_0) \asymp \hat{\sigma}_{4,d}^2\gamma_2^2$ by Lemma D.8 and Assumption 3.2, the amount of time spent in the case where $r(t) \geq \frac{1}{2}$, $|D_{4,t}| \geq \xi|D_{4,t}|$ and $L(\rho_t) \geq (\hat{\sigma}_{2,d}^2 + \hat{\sigma}_{4,d}^2)\epsilon^2$ after $t \geq T_2$ is at most $\frac{1}{\hat{\sigma}_{2,d}^2\xi^4\kappa^6} \log\left(\frac{\gamma_2}{\epsilon}\right)$ by Gronwall's inequality (Fact I.13). $\square$

Next, we deal with the subcase where $D_{4,t}$ is small relative to $D_{2,t}$.

**Lemma D.30.** *Suppose we are in the setting of Theorem 3.3, and $D_{4,T_2} = 0$ and $D_{2,T_2} < 0$, i.e. we are in Phase 3, Case 2. Then, at most $\frac{1}{\hat{\sigma}_{2,d}^2\xi^4\kappa^4} \log\left(\frac{\gamma_2}{\epsilon}\right)$ time $t \geq T_2$ elapses such that the following hold:*

1. $|D_{4,t}| \leq \xi|D_{2,t}|$

2. $L(\rho_t) \geq (\hat{\sigma}_{2,d}^2 + \hat{\sigma}_{4,d}^2)\epsilon^2$

*Proof of Lemma D.30.* Suppose $|D_{4,t}| \leq \xi|D_{2,t}|$ at some time $t \geq T_2$. Then, we can lower bound the velocity for a large fraction of $w$'s, as follows. First, we provide a bound on $P_t(w)$ (defined in Lemma 4.4):

$$P_t(w) = 2\hat{\sigma}_{2,d}^2 D_{2,t}w + 4\hat{\sigma}_{4,d}^2 D_{4,t}w^3 \qquad \text{(By Lemma 4.4)}$$

$$= -2\hat{\sigma}_{2,d}^2 |D_{2,t}|w + 4\hat{\sigma}_{4,d}^2 |D_{4,t}|w^3 \qquad \text{(B.c. } D_{2,t} < 0 \text{ and } D_{4,t} > 0 \text{ by Lemma D.28)}$$

$$= -2\hat{\sigma}_{2,d}^2 |D_{2,t}|w + O(\xi\hat{\sigma}_{4,d}^2 |D_{2,t}|w^3) \qquad \text{(B.c. } |D_{4,t}| \leq \xi|D_{2,t}|)$$

$$= -2\hat{\sigma}_{2,d}^2 |D_{2,t}|w + O(\xi\hat{\sigma}_{2,d}^2 |D_{2,t}|w^3) \qquad \text{(B.c. } \hat{\sigma}_{2,d}^2 \lesssim \hat{\sigma}_{4,d}^2 \text{ by Assumption 3.2)}$$

$$= -2\hat{\sigma}_{2,d}^2 |D_{2,t}|w \cdot (1 - O(\xi)) \tag{D.260}$$

Next, we provide a bound on $Q_t(w)$:

$$|Q_t(w)| = |\lambda_d^{(1)}w + \lambda_d^{(3)}w| \qquad \text{(By Lemma 4.4)}$$

$$= O\left(\frac{\hat{\sigma}_{2,d}^2 |D_{2,t}| + \hat{\sigma}_{4,d}^2 |D_{4,t}|}{d}\right)|w| \qquad \text{(By Lemma 4.4)}$$

$$= O\left(\frac{\hat{\sigma}_{2,d}^2 |D_{2,t}|}{d}\right)|w| \qquad \text{(B.c. } |D_{4,t}| \leq \xi|D_{2,t}| \text{ and } \hat{\sigma}_{4,d}^2 \lesssim \hat{\sigma}_{2,d}^2)$$

$$= O(\xi\hat{\sigma}_{2,d}^2 |D_{2,t}|)|w| \qquad \text{(B.c. } 1/d \leq \frac{1}{\log\log d} = \xi \text{ by Assumption 3.2)}$$

Thus,

$$|P_t(w) + Q_t(w)| \geq 2\hat{\sigma}_{2,d}^2 |D_{2,t}|w \cdot (1 - O(\xi)) \tag{D.261}$$

By Lemma D.28, we have $D_{2,t} \leq 0$, meaning that by Proposition D.38,

$$\mathbb{E}_{w\sim\rho_t}[w^2] \leq \gamma_2 + O(1/d) \tag{D.262}$$

Thus, by Markov's inequality and Assumption 3.2,

$$\mathbb{P}_{w\sim\rho_t}(|w| \geq 1/2) \lesssim \gamma_2 \tag{D.263}$$

Furthermore, by Lemma D.28, for all $\iota \geq \iota_{\mathrm{L}}$, $w_t(\iota) \geq \xi^2\kappa^2$ for all times $t \geq T_2$, meaning that $w_t \geq \xi^2\kappa^2$ with probability at least $1 - O(\kappa)$ under $\rho_t$, by Proposition D.6. Thus, by a union bound,

$$\mathbb{P}_{w\sim\rho_t}(|w| \in [\xi^2\kappa^2, 1/2]) \geq 1 - O(\gamma_2) - O(\kappa) \geq \frac{1}{2} \tag{D.264}$$

where the last inequality is by Assumption 3.2 since $\gamma_2$ and $d$ are sufficiently small and large respectively. Thus the average velocity is at least

$$\mathbb{E}_{w\sim\rho_t} |v_t(w)|^2 \gtrsim \mathbb{P}_{w\sim\rho_t}(\xi^2\kappa^2 \leq |w| \leq 1/2) \cdot \mathbb{E}_{w\sim\rho_t}[|v_t(w)|^2 \mid \xi^2\kappa^2 \leq |w| \leq 1/2] \tag{D.265}$$

$$\gtrsim \frac{1}{2} \cdot \mathbb{E}_{w\sim\rho_t}[|v_t(w)|^2 \mid \xi^2\kappa^2 \leq |w| \leq 1/2] \tag{D.266}$$

$$\gtrsim \mathbb{E}_{w\sim\rho_t}[|v_t(w)|^2 \mid \xi^2\kappa^2 \leq |w| \leq 1/2] \tag{D.267}$$

$$\gtrsim \mathbb{E}_{w\sim\rho_t}[(1-w^2)^2|P_t(w) + Q_t(w)|^2 \mid \xi^2\kappa^2 \leq |w| \leq 1/2] \tag{D.268}$$

$$\gtrsim \mathbb{E}_{w\sim\rho_t}[|P_t(w) + Q_t(w)|^2 \mid \xi^2\kappa^2 \leq |w| \leq 1/2] \tag{B.c. $|w| \leq 1/2$}$$

$$\gtrsim \mathbb{E}_{w\sim\rho_t}[\hat{\sigma}_{2,d}^4|D_{2,t}|^2 w^2 \mid \xi^2\kappa^2 \leq |w| \leq 1/2] \tag{By Eq. (D.261)}$$

$$\gtrsim \hat{\sigma}_{2,d}^4|D_{2,t}|^2\xi^4\kappa^4 \tag{D.269}$$

Thus, at any time $t$ where $|D_{4,t}| \leq \xi|D_{2,t}|$, we have

$$\frac{dL(\rho_t)}{dt} \lesssim -\hat{\sigma}_{2,d}^4|D_{2,t}|^2\xi^4\kappa^4 \tag{D.270}$$

$$\lesssim -\hat{\sigma}_{2,d}^2(\hat{\sigma}_{2,d}^2|D_{2,t}|^2 + \hat{\sigma}_{4,d}^2|D_{4,t}|^2)\xi^4\kappa^4$$
$$\text{(B.c. $|D_{4,t}| \leq \xi|D_{2,t}|$ and $\hat{\sigma}_{4,d}^2 \lesssim \hat{\sigma}_{2,d}^2$ by Assumption 3.2)}$$

$$\lesssim -\hat{\sigma}_{2,d}^2\xi^4\kappa^4 L(\rho_t) \tag{D.271}$$

Since the initial loss is $L(\rho_0) \lesssim \hat{\sigma}_{2,d}^2\gamma_2^2$ by Lemma D.8, by an argument similar to that used in Lemma D.29, using Gronwall's inequality (Fact I.13), we find that

$$\int_{T_2}^{\infty} \mathbb{1}(|D_{4,t}| \leq \xi|D_{2,t}|)\mathbb{1}(L(\rho_t) \geq (\hat{\sigma}_{2,d}^2 + \hat{\sigma}_{4,d}^2)\epsilon^2)dt \lesssim \frac{1}{\hat{\sigma}_{2,d}^2\xi^4\kappa^4} \log\left(\frac{\gamma_2}{\epsilon}\right) \tag{D.272}$$

i.e. the time spent in the subcase in the statement of this lemma is at most $\frac{1}{\hat{\sigma}_{2,d}^2\xi^4\kappa^4} \log\left(\frac{\gamma_2}{\epsilon}\right)$, as desired. $\qquad\square$

We have already considered the case where $|D_{4,t}| \geq \xi|D_{2,t}|$ and $r \geq \frac{1}{2}$. Next, we must deal with the case where $|D_{4,t}| \geq \xi|D_{2,t}|$, but where $r \leq \frac{1}{2}$. We further split this case into two subcases. In the next lemma, we deal with the subcase where $w_t(\iota_{\mathrm{R}}) \geq 1 - \xi$, meaning a non-negligible portion of the particles are close to 1.

**Lemma D.31.** *Suppose we are in the setting of Theorem 3.3, and $D_{4,T_2} = 0$ and $D_{2,T_2} < 0$, i.e. we are in Phase 3, Case 2. Then, the amount of time $t \geq T_2$ that elapses while the following conditions simultaneously hold:*

1. *$|D_{4,t}| \geq \xi|D_{2,t}|$*

2. *$r \leq \frac{1}{2}$ where $r$ is the positive root of $P_t$*

3. *$w_t(\iota_{\mathrm{R}}) \geq 1 - \xi$*

4. *$L(\rho_t) \geq (\hat{\sigma}_{2,d}^2 + \hat{\sigma}_{4,d}^2)\epsilon^2$*

*is at most $\frac{1}{\hat{\sigma}_{4,d}^2\epsilon\xi^7\kappa^{12}} \log\left(\frac{\gamma_2}{\epsilon}\right)$.*

*Proof of Lemma D.31.* If $w(\iota_R) \geq 1 - \xi$ at any time, then the subsequent times when it could possibly be moving to towards 1 are:

1. During phase 2 — note that before phase 2, $w_t(\iota_R) \leq \frac{1}{\log d}$, so it is not possible for $w_t(\iota_R)$ to be larger than $1 - \xi$ since, by Lemma D.9, $w_t(\iota_R)$ is increasing during Phase 1.

2. The times $t \geq T_2$ when $|D_{4,t}| \leq \xi|D_{2,t}|$

3. The times $t \geq T_2$ when $|D_{4,t}| \geq \xi|D_{2,t}|$ and $r \geq \frac{1}{2}$, where $r$ is the positive root of $P_t(w)$

The only case not mentioned above consists of the times $t \geq T_2$ when $|D_{4,t}| \geq \xi|D_{2,t}|$ and the positive root $r$ of $P_t(w)$ is less than $\frac{1}{2}$. In this case, $w_t(\iota_R)$ will be moving away from 1. For this reason, we also do not need to consider the case discussed in Lemma D.32.

Thus, we will use upper bounds on the velocity and time during the first 3 cases listed above, and use a lower bound on the velocity in the last. First note that the velocity can always be upper bounded as follows:

$$v_t(w) \lesssim (1-w)(1+w) \cdot |P_t(w) + Q_t(w)| \qquad \text{(By Lemma 4.4)}$$

$$\lesssim (1-w)(1+w) \cdot \left| 2\hat{\sigma}_{2,d}^2|D_{2,t}|w + 4\hat{\sigma}_{4,d}^2|D_{4,t}|w^3 + O\left(\frac{\hat{\sigma}_{2,d}^2|D_{2,t}| + \hat{\sigma}_{4,d}^2|D_{4,t}|}{d}\right)|w| \right| \qquad \text{(By Lemma 4.4)}$$

$$\lesssim (1-w) \cdot (\hat{\sigma}_{2,d}^2|D_{2,t}| + \hat{\sigma}_{4,d}^2|D_{4,t}|) \qquad \text{(B.c. } |w| \leq 1)$$

Thus,

$$\frac{d}{dt}(1-w) \gtrsim -(\hat{\sigma}_{2,d}^2|D_{2,t}| + \hat{\sigma}_{4,d}^2|D_{4,t}|)(1-w) \gtrsim -(\hat{\sigma}_{2,d}^2 + \hat{\sigma}_{4,d}^2)(1-w) \qquad \text{(D.273)}$$

**During Phase 2.** By Lemma D.10, the running time during Phase 2 is $T_2 - T_1 \lesssim \frac{1}{(\hat{\sigma}_{2,d}^2 + \hat{\sigma}_{4,d}^2)\xi^6\kappa^{12}} \log\left(\frac{\gamma_2}{\epsilon}\right)$. At the beginning of Phase 2, $w_t(\iota_R) \leq \frac{1}{\log d}$, by the definition of Phase 1. Thus, by Gronwall's inequality (Fact I.13), at time $T_2$,

$$1 - w_{T_2}(\iota_R) \geq \left(1 - \frac{1}{\log d}\right) \cdot \exp\left(-(\hat{\sigma}_{2,d}^2 + \hat{\sigma}_{4,d}^2) \cdot \frac{1}{(\hat{\sigma}_{2,d}^2 + \hat{\sigma}_{4,d}^2)\xi^6\kappa^{12}} \log\left(\frac{\gamma_2}{\epsilon}\right)\right)$$
$$\text{(By Fact I.13, Eq. (D.273) and bound on } T_2 - T_1)$$

$$\geq \left(1 - \frac{1}{\log d}\right) \cdot \exp\left(-\frac{1}{\xi^6\kappa^{12}} \log\left(\frac{\gamma_2}{\epsilon}\right)\right) \qquad \text{(D.274)}$$

**Times $t \geq T_2$ when $|D_{4,t}| \leq \xi|D_{2,t}|$ and $L(\rho_t) \geq (\hat{\sigma}_{2,d}^2 + \hat{\sigma}_{4,d}^2)\epsilon^2$.** By Lemma D.30, this case contributes at most $\frac{1}{\hat{\sigma}_{2,d}^2\xi^4\kappa^4} \log\left(\frac{\gamma_2}{\epsilon}\right)$ time to the overall running time. Thus, by Gronwall's inequality (Fact I.13), the additional factor that this case contributes to $1 - w_t(\iota_R)$ is larger than or equal to

$$\exp\left(-(\hat{\sigma}_{2,d}^2 + \hat{\sigma}_{4,d}^2) \cdot \frac{1}{\hat{\sigma}_{2,d}^2\xi^4\kappa^4} \log\left(\frac{\gamma_2}{\epsilon}\right)\right) \geq \exp\left(-\frac{C}{\xi^4\kappa^4} \log\left(\frac{\gamma_2}{\epsilon}\right)\right) \qquad \text{(D.275)}$$

for some universal constant $C$.

**Times $t \geq T_2$ when $|D_{4,t}| \geq \xi|D_{2,t}|$, $r(t) \geq \frac{1}{2}$ and $L(\rho_t) \geq (\hat{\sigma}_{2,d}^2 + \hat{\sigma}_{4,d}^2)\epsilon^2$.** By Lemma D.29, this case contributes at most $\frac{1}{\hat{\sigma}_{2,d}^2\xi^4\kappa^6} \log\left(\frac{\gamma_2}{\epsilon}\right)$ to the overall running time. Thus, by Gronwall's inequality (Fact I.13), the additional factor that this case contributes to $1 - w_t(\iota_R)$ is larger than or equal to

$$\exp\left(-(\hat{\sigma}_{2,d}^2 + \hat{\sigma}_{4,d}^2) \cdot \frac{1}{\hat{\sigma}_{2,d}^2\xi^4\kappa^6} \log\left(\frac{\gamma_2}{\epsilon}\right)\right) \geq \exp\left(-\frac{C}{\xi^4\kappa^6} \log\left(\frac{\gamma_2}{\epsilon}\right)\right) \qquad \text{(D.276)}$$

for some universal constant $C$.

**Times $t \geq T_2$ when $|D_{4,t}| \geq \xi|D_{2,t}|$, $r(t) \leq \frac{1}{2}$ and $w_{\mathrm{R}} \geq 1 - \xi$.** In this case, we obtain a lower bound on $\frac{d}{dt}(1 - w_{\mathrm{R}})$, where we have written $w_{\mathrm{R}} = w_t(\iota_{\mathrm{R}})$ for convenience. First let us obtain a lower bound on $P_t(w_{\mathrm{R}})$:

$$P_t(w_{\mathrm{R}}) \gtrsim 2\hat{\sigma}_{2,d}^2 D_{2,t} w_{\mathrm{R}} + 4\hat{\sigma}_{4,d}^2 D_{4,t} w_{\mathrm{R}}^3 \tag{D.277}$$

$$\gtrsim 4\hat{\sigma}_{4,d}^2 D_{4,t} w_{\mathrm{R}}(w_{\mathrm{R}} - r)(w_{\mathrm{R}} + r) \qquad \text{(B.c. } r \text{ is the positive root of } P_t)$$

$$\gtrsim \hat{\sigma}_{4,d}^2 D_{4,t}(1 - \xi - r) \qquad \text{(B.c. } D_{4,t} \geq 0 \text{ and } w_{\mathrm{R}} \geq 1 - \xi)$$

$$\gtrsim \hat{\sigma}_{4,d}^2 D_{4,t} \qquad \text{(B.c. } r \leq \frac{1}{2} \text{ and } \xi \leq \frac{1}{\log\log d})$$

Next let us obtain an upper bound on $Q_t(w_{\mathrm{R}})$:

$$|Q_t(w_{\mathrm{R}})| \lesssim |\lambda_d^{(1)}| + |\lambda_d^{(3)}| \qquad \text{(By Lemma 4.4)}$$

$$\lesssim \frac{\hat{\sigma}_{2,d}^2 |D_{2,t}| + \hat{\sigma}_{4,d}^2 |D_{4,t}|}{d} \qquad \text{(By Lemma 4.4)}$$

$$\lesssim \frac{\hat{\sigma}_{4,d}^2 |D_{4,t}|}{\xi d} \qquad \text{(B.c. } |D_{2,t}| \leq |D_{4,t}|/\xi \text{ and } \hat{\sigma}_{2,d}^2 \lesssim \hat{\sigma}_{4,d}^2 \text{ by Assumption 3.2)}$$

Thus, because $\frac{1}{\xi d} \asymp \frac{\log\log d}{d}$,

$$P_t(w_{\mathrm{R}}) + Q_t(w_{\mathrm{R}}) \gtrsim \hat{\sigma}_{4,d}^2 D_{4,t} \tag{D.278}$$

and when $w_{\mathrm{R}} \geq 1 - \xi$,

$$\frac{d}{dt}(1 - w_{\mathrm{R}}) \gtrsim -v_t(w_{\mathrm{R}}) \tag{D.279}$$

$$\gtrsim (1 - w_{\mathrm{R}})(1 + w_{\mathrm{R}})(P_t(w_{\mathrm{R}}) + Q_t(w_{\mathrm{R}})) \qquad \text{(By Lemma 4.4)}$$

$$\gtrsim (1 - w_{\mathrm{R}}) \cdot \hat{\sigma}_{4,d}^2 D_{4,t} \qquad \text{(By Eq. (D.278) and b.c. } w_{\mathrm{R}} \geq 1 - \xi)$$

Furthermore, since $|D_{4,t}| \geq \xi|D_{2,t}|$, this means that $|D_{4,t}| \gtrsim \xi\epsilon$, since otherwise, $|D_{2,t}| \lesssim \epsilon$ and $|D_{4,t}| \lesssim \xi\epsilon$, and the loss would be less than $(\hat{\sigma}_{2,d}^2 + \hat{\sigma}_{4,d}^2)\epsilon^2$. Thus, if time $T_{\mathrm{esc}}$ is spent in this case, then by Gronwall's inequality (Fact I.13), this case contributes a factor of at least

$$\exp(T_{\mathrm{esc}} \cdot \hat{\sigma}_{4,d}^2 D_{4,t}) \geq \exp(T_{\mathrm{esc}} \cdot \hat{\sigma}_{4,d}^2 \xi\epsilon) \tag{D.280}$$

to $1 - w_{\mathrm{R}}$.

**Summary of 4 Cases.** In summary, by Eq. (D.274), Eq. (D.275), Eq. (D.276) and Eq. (D.280),

$$1 - w_{\mathrm{R}} \geq \left(1 - \frac{1}{\log d}\right) \cdot \exp\left(-\frac{1}{\xi^6 \kappa^{12}} \log\left(\frac{\gamma_2}{\epsilon}\right)\right) \exp\left(-\frac{C}{\xi^4 \kappa^4} \log\left(\frac{\gamma_2}{\epsilon}\right)\right) \tag{D.281}$$

$$\exp\left(-\frac{C}{\xi^4 \kappa^6} \log\left(\frac{\gamma_2}{\epsilon}\right)\right) \exp(T_{\mathrm{esc}} \cdot \hat{\sigma}_{4,d}^2 \xi\epsilon) \tag{D.282}$$

where $T_{\mathrm{esc}}$ is the amount of time spent so far in the case where $|D_{4,t}| \geq \xi|D_{2,t}|$, $r(t) \geq \frac{1}{2}$ and $w_{\mathrm{R}} \geq 1 - \xi$. This is in fact a lower bound on $1 - w_{\mathrm{R}}$ when $L(\rho_t) \geq (\hat{\sigma}_{2,d}^2 + \hat{\sigma}_{4,d}^2)\epsilon^2$, since at all other times when $L(\rho_t) \geq (\hat{\sigma}_{2,d}^2 + \hat{\sigma}_{4,d}^2)\epsilon^2$, we have that $1 - w_{\mathrm{R}}$ is increasing (i.e. $w_{\mathrm{R}}$ is moving away from 1). In particular, if

$$T_{\mathrm{esc}} \gtrsim \frac{1}{\hat{\sigma}_{4,d}^2 \xi\epsilon}\left(\frac{1}{\xi^6 \kappa^{12}} \log\left(\frac{\gamma_2}{\epsilon}\right) + \frac{C}{\xi^4 \kappa^4} \log\left(\frac{\gamma_2}{\epsilon}\right) + \frac{C}{\xi^4 \kappa^6} \log\left(\frac{\gamma_2}{\epsilon}\right)\right) \gtrsim \frac{1}{\hat{\sigma}_{4,d}^2 \epsilon \xi^7 \kappa^{12}} \log\left(\frac{\gamma_2}{\epsilon}\right) \tag{D.283}$$

then $1 - w_{\mathrm{R}}$ will be at least $\xi$. $\qquad\square$

Finally, the following lemma deals with the last case, where the positive root $r$ of $P_t$ is at most $\frac{1}{2}$, and additionally, $w_t(\iota_{\mathrm{R}}) \leq 1 - \xi$.

**Lemma D.32.** *Suppose we are in the setting of Theorem 3.3. Assume $D_{4,T_2} = 0$ and $D_{2,T_2} < 0$, i.e. Phase 3 Case 2 holds. Then, the amount of time $t \geq T_2$ that elapses while the following conditions simultaneously hold:*

1. $|D_{4,t}| \geq \xi|D_{2,t}|$

2. $r \leq \frac{1}{2}$

3. $w_t(\iota_{\mathbf{R}}) \leq 1 - \xi$

4. $L(\rho_t) \geq (\hat{\sigma}_{2,d}^2 + \hat{\sigma}_{4,d}^2)\epsilon^2$

*is at most* $\frac{1}{\hat{\sigma}_{4,d}^2 \gamma_2^{11/2} \xi^8 \kappa^4} \log\left(\frac{\gamma_2}{\epsilon}\right)$.

*Proof of Lemma D.32.* For convenience and ease of presentation, in this proof we define $\tau := \sqrt{\gamma_2}$, and $\beta \geq 1.1$ such that $\gamma_4 = \beta\tau^4$. (Note that by Assumption 3.2, $\beta$ is at most a universal constant.) We first show that there is always a large fraction of particles which are far away from the positive root $r$ of $P_t(w)$. We consider two cases: (i) $r \leq \tau^{3/2}$ and (ii) $r \geq \tau^{3/2}$.

**Case 1: $r \leq \tau^{3/2}$.** Assume for the sake of contradiction that

$$\mathbb{P}_{w\sim\rho_t}(|w| \geq r + \tau^{3/2}) \leq \tau^4 \tag{D.284}$$

Then,

$$\mathbb{E}_{w\sim\rho_t}[w^4] \leq \mathbb{P}_{w\sim\rho_t}(|w| \geq r + \tau^{3/2}) + \mathbb{P}_{w\sim\rho_t}(|w| \leq r + \tau^{3/2}) \mathbb{E}_{w\sim\rho_t}[w^4 \mid |w| \leq r + \tau^{3/2}] \tag{D.285}$$

$$= \tau^4 + (r + \tau^{3/2})^4 \tag{D.286}$$

$$\leq \tau^4 + (2\tau^{3/2})^4 \qquad \text{(By assumption that } r \leq \tau^{3/2}) \tag{D.287}$$

$$= \tau^4 + 16\tau^6 \tag{D.287}$$

Thus,

$$D_{4,t} = \mathbb{E}_{w\sim\rho_t}[P_{4,d}(w)] - \beta\tau^4 \tag{D.288}$$

$$\leq \mathbb{E}_{w\sim\rho_t}[w^4] + O\left(\frac{1}{d}\right) - \beta\tau^4 \qquad \text{(By Eq. (C.4))}$$

$$\leq \tau^4 + 16\tau^6 + O\left(\frac{1}{d}\right) - \beta\tau^4 \tag{D.289}$$

$$= 16\tau^6 + O\left(\frac{1}{d}\right) - (\beta - 1)\tau^4 \tag{D.290}$$

$$\leq 17\tau^6 - (\beta - 1)\tau^4 \qquad \text{(B.c. } d \text{ is sufficiently large by Assumption 3.2)}$$

$$< 0 \qquad \text{(B.c. } (\beta - 1) > 17\tau^2 \text{ by Assumption 3.2)}$$

which contradicts the assumption that we are in Phase 3, Case 2 since $D_{4,t} > 0$ by Lemma D.28. Thus, if $r \leq \tau^{3/2}$, then

$$\mathbb{P}_{w\sim\rho_t}(|w| \geq r + \tau^{3/2}) \geq \tau^4 \tag{D.291}$$

**Case 2: $r \geq \tau^{3/2}$.** Suppose that

$$\mathbb{P}_{w\sim\rho_t}(r - \tau^{3/2} \leq |w| \leq r + \tau^{3/2}) \geq 1 - \tau^5 \tag{D.292}$$

Then,

$$\mathbb{E}_{w\sim\rho_t}[w^2] \geq \mathbb{P}_{w\sim\rho_t}(r - \tau^{3/2} \leq |w| \leq r + \tau^{3/2}) \mathbb{E}_{w\sim\rho_t}[w^2 \mid |w| \in [r - \tau^{3/2}, r + \tau^{3/2}]] \tag{D.293}$$

$$\geq (1 - \tau^5) \cdot (r - \tau^{3/2})^2 \tag{D.294}$$

Since $D_{2,t} < 0$ by Lemma D.28, this implies that $\mathbb{E}_{w\sim\rho_t}[w^2] \leq \tau^2 + O(1/d)$ by Proposition D.38. Thus,

$$(1 - \tau^5)(r - \tau^{3/2})^2 \leq \tau^2 + O(1/d) \tag{D.295}$$

and taking square roots gives

$$(1 - \tau^5)^{1/2}(r - \tau^{3/2}) \leq \tau + O(1/\sqrt{d}) \tag{D.296}$$

Thus,

$$r \leq \tau^{3/2} + \frac{1}{(1-\tau^5)^{1/2}} \cdot \left(\tau + O\left(\frac{1}{\sqrt{d}}\right)\right) \tag{D.297}$$

On the other hand,

$$\mathbb{E}_{w \sim \rho_t}[w^4] \leq \mathbb{P}_{w \sim \rho_t}(|w| \in [r - \tau^{3/2}, r + \tau^{3/2}]) \mathbb{E}_{w \sim \rho_t}[w^4 \mid |w| \in [r - \tau^{3/2}, r + \tau^{3/2}]] \tag{D.298}$$

$$+ \mathbb{P}_{w \sim \rho_t}(|w| \notin [r - \tau^{3/2}, r + \tau^{3/2}]) \tag{D.299}$$

$$\leq (r + \tau^{3/2})^4 + \tau^5 \qquad \text{(By the assumption in Eq. (D.292))}$$

and since $D_{4,t} \geq 0$ by Lemma D.28, $\mathbb{E}_{w \sim \rho_t}[w^4] \geq \beta\tau^4 - O(1/d)$ by Eq. (C.4) and because $w \sim \rho_t$ are bounded by 1 in absolute value. Thus, by the above equation,

$$(r + \tau^{3/2})^4 + \tau^5 \geq \beta\tau^4 - O\left(\frac{1}{d}\right) \tag{D.300}$$

Rearranging gives

$$(r + \tau^{3/2})^4 + \tau^5 + O\left(\frac{1}{d}\right) \geq \beta\tau^4 \tag{D.301}$$

and taking fourth roots gives

$$r + \tau^{3/2} + \tau^{5/4} + O\left(\frac{1}{d^{1/4}}\right) \geq \beta^{1/4}\tau \tag{D.302}$$

(by the inequality $\sqrt{x+y} \leq \sqrt{x} + \sqrt{y}$). Therefore,

$$r \geq \beta^{1/4}\tau - \tau^{3/2} - \tau^{5/4} - O\left(\frac{1}{d^{1/4}}\right) \tag{D.303}$$

Combining Eq. (D.297) and Eq. (D.303) gives

$$\beta^{1/4}\tau - \tau^{3/2} - \tau^{5/4} - O\left(\frac{1}{d^{1/4}}\right) \leq \tau^{3/2} + \frac{1}{(1-\tau^5)^{1/2}} \cdot \left(\tau + O\left(\frac{1}{\sqrt{d}}\right)\right) \tag{D.304}$$

Rearranging this equation gives

$$\left(\beta^{1/4} - \frac{1}{\sqrt{1-\tau^5}}\right)\tau \leq 2\tau^{3/2} + \tau^{5/4} + O\left(\frac{1}{d^{1/4}}\right) + \frac{1}{\sqrt{1-\tau^5}} \cdot O\left(\frac{1}{\sqrt{d}}\right) \tag{D.305}$$

$$\leq 3\tau^{5/4} + O\left(\frac{1}{d^{1/4}}\right) + \frac{1}{\sqrt{1-\tau^5}} \cdot O\left(\frac{1}{\sqrt{d}}\right)$$
$$\text{(B.c. } \tau \leq 1, \text{ we have } \tau^{3/2} \leq \tau^{5/4})$$

$$\leq 4\tau^{5/4} + \frac{1}{\sqrt{1-\tau^5}} \cdot O\left(\frac{1}{\sqrt{d}}\right)$$
$$\text{(B.c. } d \text{ is sufficiently large by Assumption 3.2)}$$

$$\leq 4\tau^{5/4} + O\left(\frac{1}{\sqrt{d}}\right) \qquad \text{(B.c. } \tau \leq \frac{1}{2} \text{ by Assumption 3.2)}$$

$$\leq 5\tau^{5/4} \qquad \text{(B.c. } d \text{ is sufficiently large by Assumption 3.2)}$$

Further rearranging gives

$$\beta^{1/4} - \frac{1}{\sqrt{1-\tau^5}} \leq 5\tau^{1/4} \tag{D.306}$$

This contradicts Assumption 3.2 — since $\beta \geq 1.1$, the left-hand side is larger than an absolute constant, while the right-hand side can be made sufficiently small (since $\gamma_2 \leq c_2$ and $c_2$ is chosen to be sufficiently small). Thus, in the case that $r \geq \tau^{3/2}$, we obtain

$$\mathbb{P}_{w \sim \rho_t}(|w| \notin [r - \tau^{3/2}, r + \tau^{3/2}]) \geq \tau^5 \tag{D.307}$$

In summary, whether $r \geq \tau^{3/2}$ or $r \leq \tau^{3/2}$, we can conclude that

$$\mathbb{P}_{w \sim \rho_t}(|w| \notin [r - \tau^{3/2}, r + \tau^{3/2}]) \geq \tau^5 \tag{D.308}$$

Further assume that $w_t(\iota_{\mathrm{R}}) \leq 1 - \xi$, and note that by Lemma D.28, for all $\iota > \iota_{\mathrm{L}}$, we have $w_t(\iota) \geq \xi^2 \kappa^2$. Thus, by Proposition D.6, with probability $1 - O(\kappa)$ under $\rho_t$, we have $\xi^2 \kappa^2 \leq |w| \leq 1 - \xi$. By a union bound,

$$\mathbb{P}_{w \sim \rho_t}(||w| - r| \geq \tau^{3/2} \text{ and } \xi^2 \kappa^2 \leq |w| \leq 1 - \xi) \geq \tau^5 - O(\kappa) \geq \frac{\tau^5}{2} \tag{D.309}$$

Suppose $w$ satisfies $||w| - r| \geq \tau^{3/2}$ and $\xi^2 \kappa^2 \leq |w| \leq 1 - \xi$, i.e. the conditions mentioned in Eq. (D.309), and write

$$P_t(w) = 2\hat{\sigma}_{2,d}^2 D_{2,t} w + 4\hat{\sigma}_{4,d}^2 D_{4,t} w^3 = 4\hat{\sigma}_{4,d}^2 D_{4,t} w(w - r)(w + r) \tag{D.310}$$

Then,

$$
\begin{aligned}
|v_t(w)| &\gtrsim |1 - w^2| \cdot |P_t(w) + Q_t(w)| & \text{(By Lemma 4.4)} \\
&\gtrsim |1 + w| \cdot |1 - w| \cdot |P_t(w) + Q_t(w)| & \text{(D.311)} \\
&\gtrsim \xi |P_t(w) + Q_t(w)| & \text{(B.c. } |w| \leq 1 - \xi) \\
&\gtrsim \xi \cdot \left| 4\hat{\sigma}_{4,d}^2 D_{4,t} w(w - r)(w + r) - O\left(\frac{\hat{\sigma}_{2,d}^2 |D_{2,t}| + \hat{\sigma}_{4,d}^2 |D_{4,t}|}{d}\right) |w| \right| \\
&& \text{(By Lemma 4.4)} \\
&\gtrsim \xi \left( 4\hat{\sigma}_{4,d}^2 |D_{4,t}| \xi^2 \kappa^2 \cdot \tau^{3/2} \cdot \tau^{3/2} - O\left(\frac{\hat{\sigma}_{2,d}^2 |D_{2,t}| + \hat{\sigma}_{4,d}^2 |D_{4,t}|}{d}\right) \right) \\
&& \text{(By triangle inequality and b.c. } |w| \geq \xi^2 \kappa^2 \text{ and } ||w| - r| \geq \tau^{3/2}) \\
&\gtrsim \xi \left( 4\hat{\sigma}_{4,d}^2 |D_{4,t}| \tau^3 \xi^2 \kappa^2 - O\left(\frac{\hat{\sigma}_{4,d}^2 |D_{4,t}|}{\xi d}\right) \right) \\
&& \text{(B.c. } |D_{2,t}| \leq |D_{4,t}|/\xi \text{ and } \hat{\sigma}_{2,d}^2 \lesssim \hat{\sigma}_{4,d}^2 \text{ by Assumption 3.2)} \\
&\gtrsim \xi \cdot 4\hat{\sigma}_{4,d}^2 |D_{4,t}| \tau^3 \xi^2 \kappa^2 \\
&& \text{(B.c. } 1/(\xi d) = \frac{\log \log d}{d} \text{ and } d \text{ is sufficiently large (Assumption 3.2))} \\
&\gtrsim \hat{\sigma}_{4,d}^2 \tau^3 \xi^3 \kappa^2 |D_{4,t}| & \text{(D.312)}
\end{aligned}
$$

and the average velocity is at least

$$
\begin{aligned}
\mathbb{E}_{w \sim \rho_t} |v_t(w)|^2 &\gtrsim \mathbb{P}_{w \sim \rho_t}(||w| - r| \geq \tau^{3/2} \text{ and } \xi^2 \kappa^2 \leq |w| \leq 1 - \xi) \cdot \hat{\sigma}_{4,d}^4 \tau^6 \xi^6 \kappa^4 |D_{4,t}|^2 \\
&& \text{(By Eq. (D.312))} \\
&\gtrsim \tau^5 \cdot \hat{\sigma}_{4,d}^4 \tau^6 \xi^6 \kappa^4 |D_{4,t}|^2 & \text{(By Eq. (D.309))} \\
&\gtrsim \hat{\sigma}_{4,d}^4 \tau^{11} \xi^6 \kappa^4 |D_{4,t}|^2 & \text{(D.313)} \\
&\gtrsim \hat{\sigma}_{4,d}^2 \tau^{11} \xi^6 \kappa^4 (\xi^2 L(\rho_t)) \\
&& \text{(B.c. } |D_{4,t}| \geq \xi |D_{2,t}| \text{ and } \hat{\sigma}_{4,d}^2 \gtrsim \hat{\sigma}_{2,d}^2 \text{ by Assumption 3.2)} \\
&\gtrsim \hat{\sigma}_{4,d}^2 \tau^{11} \xi^8 \kappa^4 L(\rho_t) & \text{(D.314)}
\end{aligned}
$$

Thus, by Lemma D.39,

$$\frac{dL(\rho_t)}{dt} \lesssim -\hat{\sigma}_{4,d}^2 \tau^{11} \xi^8 \kappa^4 L(\rho_t) \tag{D.315}$$

Since the initial loss is $L(\rho_0) \lesssim (\hat{\sigma}_{2,d}^2 + \hat{\sigma}_{4,d}^2)\gamma_2^2$ by Lemma D.8, we can calculate using Gronwall's inequality (Fact I.13) that the time spent when the conditions in the statement of this lemma simultaneously hold is at most $\frac{1}{\hat{\sigma}_{4,d}^2 \tau^{11} \xi^8 \kappa^4} \log\left(\frac{\gamma_2}{\epsilon}\right)$, as desired. $\qquad \square$

To conclude, we show Lemma D.33 which gives a bound on the running time during Phase 3, Case 2.

**Lemma D.33** (Phase 3, Case 2 Summary). *Suppose we are in the setting of Theorem 3.3. Assume $D_{4,T_2} = 0$ and $D_{2,T_2} < 0$, i.e. Phase 3 Case 2 holds. Then, at most $O\left(\frac{1}{\hat{\sigma}_{2,d}^2 \epsilon \gamma_2^{11/2} \xi^8 \kappa^{12}} \log\left(\frac{\gamma_2}{\epsilon}\right)\right)$ time $t \geq T_2$ elapses while $L(\rho_t) \gtrsim (\hat{\sigma}_{2,d}^2 + \hat{\sigma}_{4,d}^2)\epsilon^2$.*

*Proof of Lemma D.33.* By combining Lemma D.29, Lemma D.30, Lemma D.31, and Lemma D.32, we find that the overall running time during Phase 3 Case 2 is at most

$$O\left(\frac{1}{\hat{\sigma}_{2,d}^2\xi^4\kappa^6}\log\left(\frac{\gamma_2}{\epsilon}\right) + \frac{1}{\hat{\sigma}_{2,d}^2\xi^4\kappa^4}\log\left(\frac{\gamma_2}{\epsilon}\right) + \frac{1}{\hat{\sigma}_{4,d}^2\epsilon\xi^7\kappa^{12}}\log\left(\frac{\gamma_2}{\epsilon}\right) + \frac{1}{\hat{\sigma}_{4,d}^2\gamma_2^{11/2}\xi^8\kappa^4}\log\left(\frac{\gamma_2}{\epsilon}\right)\right)$$
(D.316)

and this can be upper bounded by $O\left(\frac{1}{\hat{\sigma}_{2,d}^2\epsilon\gamma_2^{11/2}\xi^8\kappa^{12}}\log\left(\frac{\gamma_2}{\epsilon}\right)\right)$, where we have used the fact that $\hat{\sigma}_{2,d}^2 \asymp \hat{\sigma}_{4,d}^2$ (see Assumption 3.2). $\qquad\square$

## D.7 Summary of Phase 3

The following lemma provides a bound on the running time after $T_2$ that is required for $L(\rho_t) \lesssim (\hat{\sigma}_{2,d}^2 + \hat{\sigma}_{4,d}^2)\epsilon^2$.

**Lemma D.34** (Phase 3)**.** *Suppose we are in the setting of Theorem 3.3. Then, $T_{*,\epsilon} - T_2 \lesssim \frac{(\log\log d)^{20}}{\hat{\sigma}_{2,d}^2\epsilon\gamma_2^8}\log\left(\frac{\gamma_2}{\epsilon}\right)$.*

*Proof of Lemma D.34.* By combining Lemma D.22 and Lemma D.33, we find that the overall running time during Phase 3 is at most

$$\max\left(O\left(\frac{1}{\hat{\sigma}_{4,d}^2\gamma_2^8\xi^4\kappa^6}\log\left(\frac{\gamma_2}{\epsilon}\right)\right), O\left(\frac{1}{\hat{\sigma}_{2,d}^2\epsilon\gamma_2^{11/2}\xi^8\kappa^{12}}\log\left(\frac{\gamma_2}{\epsilon}\right)\right)\right)$$
(D.317)

and this is at most $O\left(\frac{1}{\hat{\sigma}_{2,d}^2\epsilon\gamma_2^8\xi^8\kappa^{12}}\log\left(\frac{\gamma_2}{\epsilon}\right)\right)$, where we have used the fact that $\hat{\sigma}_{2,d}^2 \asymp \hat{\sigma}_{4,d}^2$ (see Assumption 3.2) $\qquad\square$

## D.8 Proof of Main Theorem for Population Case

Finally, we prove Theorem 3.3:

*Proof of Theorem 3.3.* We combine Lemma D.5, Lemma D.10 and Lemma D.34 to find that the total runtime is at most

$$T_{*,\epsilon} \lesssim \frac{1}{\hat{\sigma}_{2,d}^2\gamma_2}\log d + \frac{(\log\log d)^{18}}{(\hat{\sigma}_{2,d}^2 + \hat{\sigma}_{4,d}^2)}\log\left(\frac{\gamma_2}{\epsilon}\right) + \frac{(\log\log d)^{20}}{\hat{\sigma}_{2,d}^2\epsilon\gamma_2^8}\log\left(\frac{\gamma_2}{\epsilon}\right)$$
(D.318)

and we can further simplify the above bound to obtain:

$$T_{*,\epsilon} \lesssim \frac{1}{\hat{\sigma}_{2,d}^2\gamma_2}\log d + \frac{(\log\log d)^{20}}{\hat{\sigma}_{2,d}^2\epsilon\gamma_2^8}\log\left(\frac{\gamma_2}{\epsilon}\right)$$
(D.319)

where we have used the fact that $\hat{\sigma}_{2,d}^2 \asymp \hat{\sigma}_{4,d}^2$ by Assumption 3.2. $\qquad\square$

## D.9 Additional Lemmas

**Lemma D.35.** *Suppose Assumption 3.2 holds. There exists some $r$, with $r \gtrsim \frac{1}{\log\log d}$ and $r \leq 1$, such that if $\rho$ is the singleton distribution under which $r$ has probability $1$, then the velocity at $r$ is $0$.*

*Proof.* Let $\rho$ be such that $r$ has probability $1$ under $\rho$, for some $r \in (0, 1)$. Applying the definitions of $D_2$ and $D_4$ to $\rho$, we have $D_2 = P_{2,d}(r) - \gamma_2$ and $D_4 = P_{4,d}(r) - \gamma_4$. By Lemma D.3, the velocity of $r$ is

$$v(r) = -(1 - r^2)(\hat{\sigma}_{2,d}^2 D_2 P_{2,d}{}'(r) + \hat{\sigma}_{4,d}^2 D_4 P_{4,d}{}'(r))$$
(D.320)

The second factor is equal to

$$\hat{\sigma}_{2,d}^2(P_{2,d}(r) - \gamma_2)P_{2,d}{}'(r) + \hat{\sigma}_{4,d}^2(P_{4,d}(r) - \gamma_4)P_{4,d}{}'(r)$$
(D.321)

and we denote this polynomial of $r$ by $F(r)$. Note that $P_{2,d}(1) = P_{4,d}(1) = 1$ (see Equation (2.20) of Atkinson and Han [12]). Thus, $F(1) > 0$ by Assumption 3.2, since $P_{2,d}'(1) > 0$ and $P_{4,d}'(1) > 0$ by Eq. (C.4). On the other hand, suppose $r \asymp \frac{1}{\log \log d}$. Then,

$$|P_{2,d}(r)| = \left| \frac{d}{d-1} r^2 - \frac{1}{d-1} \right| \qquad \text{(By Eq. (C.4))}$$

$$\lesssim \frac{1}{(\log \log d)^2} \qquad \qquad \text{(D.322)}$$

Similarly, $|P_{4,d}(r)| \lesssim \frac{1}{(\log \log d)^4}$. Thus, $P_{2,d}(r) - \gamma_2 < 0$ and $P_{4,d}(r) - \gamma_4 < 0$ since $d$ can be chosen to be sufficiently large according to Assumption 3.2. However, by Eq. (C.4), one can show that $P_{2,d}'(r) \gtrsim \frac{1}{\log \log d}$ and $P_{4,d}'(r) \gtrsim \frac{1}{(\log \log d)^3}$ as long as $d$ is sufficiently large. Thus, $F(r) < 0$ for $r \asymp \frac{1}{\log \log d}$. In summary, $F$ has a root between $\frac{C}{\log \log d}$ and 1 for some universal constant $C$, and for this value of $r$, the velocity of $r$ is 0. $\qquad \square$

**Lemma D.36.** *In the setting of Theorem 3.3, $T_{*,\epsilon} - T_1 \lesssim \frac{(\log \log d)^{20}}{\hat{\sigma}_{2,d}^2 \epsilon \gamma_2^8} \log \left( \frac{\gamma_2}{\epsilon} \right)$. Thus, $(\hat{\sigma}_{2,d}^2 + \hat{\sigma}_{4,d}^2) \gamma_2 (T_{*,\epsilon} - T_1) \lesssim \frac{(\log \log d)^{20}}{\epsilon \gamma_2^7} \log \left( \frac{\gamma_2}{\epsilon} \right)$.*

*Proof.* This follows from Lemma D.10 and Lemma D.34. $\qquad \square$

## D.10 Helper Lemmas

In this subsection, we give some convenient facts that we use in the following subsections.

**Proposition D.37.** *If $0 \leq \iota_1 < \iota_2$, then for all $t \geq 0$, $w_t(\iota_1) < w_t(\iota_2)$.*

*Proof of Proposition D.37.* Assume this is not the case, and let $t$ be the infimum of all times $s$ such that $w_s(\iota_1) \geq w_s(\iota_2)$. Then, $w_t(\iota_1) = w_t(\iota_2)$, meaning that for all $t' \geq t$, $w_{t'}(\iota_1) = w_{t'}(\iota_2)$. For convenience, let $w_0 = w_t(\iota_1) = w_t(\iota_2)$. Note that $t > 0$ since $\iota_1 \neq \iota_2$. However, this implies that for $t' \in (0, t)$, $w_{t'}(\iota_1) < w_{t'}(\iota_2)$, meaning that for any $\delta > 0$, $w_s(\iota_1)$ and $w_s(\iota_2)$ are non-identical solutions to the initial value problem on the time interval $(t - \delta, t + \delta)$ given by the conditions $f(t) = w_0$ and $\frac{df}{dt} = v_t(f(t))$. This contradicts the Picard-Lindelof theorem, and thus there cannot exist a time $t$ such that $w_t(\iota_1) \geq w_t(\iota_2)$. $\qquad \square$

**Proposition D.38.** *Suppose we are in the setting of Theorem 3.3, and let $t \geq 0$ such that $D_{2,t} < 0$. Then, $\mathbb{E}_{w \sim \rho_t}[w^2] \leq \gamma_2 + O\left( \frac{1}{d} \right)$.*

*Proof of Proposition D.37.* Suppose $D_{2,t} \leq 0$. Then, $\mathbb{E}_{w \sim \rho_t}[P_{2,d}(w)] \leq \gamma_2$, which implies that

$$\frac{d}{d-1} \mathbb{E}_{w \sim \rho_t}[w^2] - \frac{1}{d-1} \leq \gamma_2 \qquad \qquad \text{(D.323)}$$

and rearranging gives

$$\mathbb{E}_{w \sim \rho_t}[w^2] \leq \frac{d-1}{d} \gamma_2 + \frac{1}{d} \leq \gamma_2 + O\left( \frac{1}{d} \right) \qquad \qquad \text{(D.324)}$$

as desired. $\qquad \square$

**Lemma D.39** (Decrease in Loss and Average Velocity). *Suppose we are in the setting of Theorem 3.3. Then, $\frac{dL}{dt} = -\mathbb{E}_{u \sim \rho_t}[\|\text{grad}_u L(\rho_t)\|_2^2]$.*

We note that a similar result was shown in Lemma E.11 of Wei et al. [75], but for gradient flow rather than projected gradient flow.

*Proof of Lemma D.39.* First, we can calculate

$$\frac{d}{dt} L(\rho_t) = \frac{1}{2} \frac{d}{dt} \mathbb{E}_{x \sim \mathbb{S}^{d-1}}[(f_{\rho_t}(x) - y(x))^2] = \mathbb{E}_{x \sim \mathbb{S}^{d-1}} \left[ (f_{\rho_t}(x) - y(x)) \frac{d}{dt} f_{\rho_t}(x) \right]. \qquad \text{(D.325)}$$

Furthermore, we can expand $\frac{d}{dt} f_{\rho_t}(x)$ as

$$\frac{d}{dt} \underset{u \sim \rho_t}{\mathbb{E}} [\sigma(u \cdot x)] = - \underset{u \sim \rho_t}{\mathbb{E}} \left[ \sigma'(u \cdot x) x \cdot \mathrm{grad}_u L(\rho_t) \right]. \tag{D.326}$$

Therefore,

$$\frac{d}{dt} L(\rho_t) = \underset{x \sim \mathbb{S}^{d-1}}{\mathbb{E}} \left[ (f_{\rho_t}(x) - y(x)) \frac{d}{dt} f_{\rho_t}(x) \right] \tag{D.327}$$

$$= - \underset{x \sim \mathbb{S}^{d-1}}{\mathbb{E}} \underset{u \sim \rho_t}{\mathbb{E}} \left[ (f_{\rho_t}(x) - y(x)) \sigma'(u \cdot x) x \cdot \mathrm{grad}_u L(\rho_t) \right] \tag{D.328}$$

$$= - \underset{u \sim \rho_t}{\mathbb{E}} \left[ \mathrm{grad}_u L(\rho_t) \cdot \underset{x \sim \mathbb{S}^{d-1}}{\mathbb{E}} [(f_{\rho_t}(x) - y(x)) \sigma'(u \cdot x) x] \right] \tag{D.329}$$

$$= - \underset{u \sim \rho_t}{\mathbb{E}} \left[ \mathrm{grad}_u L(\rho_t) \cdot \nabla_u L(\rho_t) \right] \tag{D.330}$$

$$= - \underset{u \sim \rho_t}{\mathbb{E}} \left[ \| \mathrm{grad}_u L(\rho_t) \|_2^2 \right].$$

$$\text{(B.c. } \mathrm{grad}_u = (I - uu^\top) \nabla_u L(\rho_t) \text{ and } (I - uu^\top)^2 = (I - uu^\top))$$

This completes the proof. $\square$

The above lemma implies that $L(\rho_t)$ is always decreasing. Note that frequently, when we apply the above lemma, we will only consider the first coordinate of $\mathrm{grad}_u L(\rho_t)$.

**Proposition D.40.** *For any distribution $\rho$ on $\mathbb{R}$,*

$$\underset{w, w' \sim \rho}{\mathbb{E}} [|w - w'|^2] = 2 \underset{w \sim \rho}{\mathrm{Var}}[w] \tag{D.331}$$

*Proof of Proposition D.40.* This follows from a short calculation:

$$\underset{w, w' \sim \rho}{\mathbb{E}} [|w - w'|^2] = \underset{w, w' \sim \rho}{\mathbb{E}} (w^2 - 2ww' + (w')^2) = 2 \underset{w \sim \rho}{\mathbb{E}} [w^2] - 2(\underset{w \sim \rho}{\mathbb{E}}[w])^2 = 2 \underset{w \sim \rho}{\mathrm{Var}}[w] \tag{D.332}$$

as desired. $\square$

# E Proofs for Finite-Samples and Finite-Width Dynamics

In this section we give the proof for results related to the empirical dynamics, where the number of samples and network width are both polynomial in $d$. The end goal is to prove Theorem 3.4, the proof of which is given below.

*Proof of Theorem 3.4.* We have

$$L(\hat{\rho}_{T_{*,\epsilon}}) = \underset{x \sim \mathbb{S}^{d-1}}{\mathbb{E}} [(f_{\hat{\rho}_{T_{*,\epsilon}}}(x) - y(x))^2] \tag{E.1}$$

$$\leq 3 \underset{x \sim \mathbb{S}^{d-1}}{\mathbb{E}} [(f_{\hat{\rho}_{T_{*,\epsilon}}}(x) - f_{\bar{\rho}_{T_{*,\epsilon}}}(x))^2] + 3 \underset{x \sim \mathbb{S}^{d-1}}{\mathbb{E}} [(f_{\bar{\rho}_{T_{*,\epsilon}}}(x) - f_{\rho_{T_{*,\epsilon}}}(x))^2] \tag{E.2}$$

$$+ 3 \underset{x \sim \mathbb{S}^{d-1}}{\mathbb{E}} [(f_{\rho_{T_{*,\epsilon}}}(x) - y(x))^2] \tag{E.3}$$

$$\lesssim (\hat{\sigma}_{2,d}^2 + \hat{\sigma}_{4,d}^2) \frac{1}{d^{\frac{\mu-1}{2}}} + \frac{(\hat{\sigma}_{2,d}^2 + \hat{\sigma}_{4,d}^2) d^2 (\log d)^{O(1)}}{m} + (\hat{\sigma}_{2,d}^2 + \hat{\sigma}_{4,d}^2) \epsilon^2. \tag{E.4}$$

where the first inequality is by Cauchy-Schwarz inequality, and the second inequality is by Lemma 5.1, Lemma 5.2 and Theorem 3.3. Since $\epsilon = \frac{1}{\log \log d}$, the last term dominates because the conditions of Theorem 3.4 imply that $m \gtrsim d^3$, and because $\frac{\mu-1}{2} > 1$, meaning that the first term is at most $\frac{\hat{\sigma}_{2,d}^2 + \hat{\sigma}_{4,d}^2}{d}$. $\square$

To complete the proof, we give proofs for results in Section 5. We first give a proof for Lemma 5.1:

*Proof of Lemma 5.1.* For the first statement, for any fixed $t \leq T$, by Lemma E.13, we have

$$\underset{x \sim \mathbb{S}^{d-1}}{\mathbb{E}}[(f_{\rho_t}(x) - f_{\bar{\rho}_t}(x))^2] \lesssim \frac{(\hat{\sigma}_{2,d}^2 + \hat{\sigma}_{4,d}^2)(d^2 + \log \frac{1}{\delta})(\log d)^C}{m} + \frac{\hat{\sigma}_{2,d}^2 + \hat{\sigma}_{4,d}^2}{d^{\Omega(\log d)}} \quad \text{(E.5)}$$

with probability $1 - \delta$ — we will choose $\delta$ shortly. In addition, for any $t, t' \in [0, T]$, by Lemma E.14, we have that $\|u_t(\chi) - u_{t'}(\chi)\|_2 \lesssim (\hat{\sigma}_{2,d}^2 + \hat{\sigma}_{4,d}^2)d^4|t - t'|$, which by Lemma E.12 implies that for any $x \in \mathbb{S}^{d-1}$,

$$|f_{\rho_t}(x) - f_{\rho_{t'}}(x)| \leq \underset{\chi \sim \rho_0}{\mathbb{E}} |\sigma(u_t(\chi) \cdot x) - \sigma(u_{t'}(\chi) \cdot x)| \quad \text{(E.6)}$$

$$\lesssim (|\hat{\sigma}_{2,d}|\sqrt{N_{2,d}} + |\hat{\sigma}_{4,d}|\sqrt{N_{4,d}}) \cdot (\hat{\sigma}_{2,d}^2 + \hat{\sigma}_{4,d}^2)d^4|t - t'| \quad \text{(E.7)}$$

In particular, if $|t - t'| \leq \frac{1}{(\hat{\sigma}_{2,d}^2 + \hat{\sigma}_{4,d}^2)d^{\Omega(\log d)}}$, then for all $x \in \mathbb{S}^{d-1}$ we have

$$|f_{\rho_t}(x) - f_{\rho_{t'}}(x)| \lesssim \frac{|\hat{\sigma}_{2,d}| + |\hat{\sigma}_{4,d}|}{d^{\Omega(\log d)}} \quad \text{(E.8)}$$

meaning that

$$\underset{x \sim \mathbb{S}^{d-1}}{\mathbb{E}}[(f_{\rho_t}(x) - f_{\rho_{t'}}(x))^2] \lesssim \frac{\hat{\sigma}_{2,d}^2 + \hat{\sigma}_{4,d}^2}{d^{\Omega(\log d)}} \quad \text{(E.9)}$$

By the same argument, we have that

$$\underset{x \sim \mathbb{S}^{d-1}}{\mathbb{E}}[(f_{\bar{\rho}_t}(x) - f_{\bar{\rho}_{t'}}(x))^2] \lesssim \frac{\hat{\sigma}_{2,d}^2 + \hat{\sigma}_{4,d}^2}{d^{\Omega(\log d)}} \quad \text{(E.10)}$$

Now, we perform a union bound over all times $t$ such that $t \leq T$ and $t$ is an integer multiple of $\frac{1}{(\hat{\sigma}_{2,d}^2 + \hat{\sigma}_{4,d}^2)d^{H \log d}}$ for some sufficiently large absolute constant $H > 0$. Since $T \leq \frac{d^C}{\hat{\sigma}_{2,d}^2 + \hat{\sigma}_{4,d}^2}$, the number of such times is at most $d^{C + H \log d}$. Thus, with probability at least $1 - \delta \cdot d^{C + H \log d}$, we have

$$\underset{x \sim \mathbb{S}^{d-1}}{\mathbb{E}}[(f_{\rho_t}(x) - f_{\bar{\rho}_t}(x))^2] \lesssim \frac{(\hat{\sigma}_{2,d}^2 + \hat{\sigma}_{4,d}^2)(d^2 + \log \frac{1}{\delta})(\log d)^{O(1)}}{m} + \frac{\hat{\sigma}_{2,d}^2 + \hat{\sigma}_{4,d}^2}{d^{\Omega(\log d)}} \quad \text{(E.11)}$$

for all $t \in [0, T]$. In particular, in order to have exponentially small failure probability, we set $\delta = \frac{1}{e^{d^2} d^{C + H \log d}}$, and by our assumption that $m \leq d^{O(\log d)}$, we obtain

$$\underset{x \sim \mathbb{S}^{d-1}}{\mathbb{E}}[(f_{\rho_t}(x) - f_{\bar{\rho}_t}(x))^2] \lesssim \frac{(\hat{\sigma}_{2,d}^2 + \hat{\sigma}_{4,d}^2)d^2(\log d)^{O(1)}}{m} \quad \text{(E.12)}$$

for all $t \leq T$, with probability at least $1 - e^{-d^2}$, since $\log \frac{1}{\delta} = d^2 + O((\log d)^2)$. This completes the proof of the first statement of the lemma. Additionally, by Lemma E.16, we have

$$\underset{x \sim \mathbb{S}^{d-1}}{\mathbb{E}}[(f_{\hat{\rho}_t}(x) - f_{\bar{\rho}_t}(x))^2] \lesssim (\hat{\sigma}_{2,d}^2 + \hat{\sigma}_{4,d}^2) \underset{(\bar{u},\hat{u}) \sim \Gamma}{\mathbb{E}}[\|\hat{u}_t - \bar{u}_t\|_2^2] \quad \text{(E.13)}$$

which proves the second statement of the lemma. $\qquad\square$

The rest of the section is dedicated to prove Lemma 5.2. To begin with, we first introduce the following lemma which formalizes the decomposition of $\frac{d}{dt}\|\hat{u}_t - \bar{u}_t\|_2^2$ discussed in Section 5.

**Lemma E.1** (Decomposition of $\frac{d}{dt}\|\hat{u}_t - \bar{u}_t\|_2^2$)**.** *Suppose we are in the setting of Theorem 3.4. Then, $\frac{d}{dt}\|\hat{u}_t - \bar{u}_t\|_2^2 = A_t + B_t + C_t$, where $A_t = -2\langle \text{grad}_{\hat{u}}L(\rho_t) - \text{grad}_{\bar{u}}L(\rho_t), \hat{u}_t - \bar{u}_t\rangle$, $B_t = -2\langle \text{grad}_{\hat{u}}L(\hat{\rho}_t) - \text{grad}_{\hat{u}}L(\rho_t), \hat{u}_t - \bar{u}_t\rangle$, and $C_t = -2\langle \text{grad}_{\hat{u}}\widehat{L}(\hat{\rho}_t) - \text{grad}_{\hat{u}}L(\hat{\rho}_t), \hat{u}_t - \bar{u}_t\rangle$.*

*Proof of Lemma E.1.* By the chain rule,

$$\frac{d}{dt}\|\hat{u}_t - \bar{u}_t\|_2^2 = 2\left\langle \frac{d}{dt}\hat{u}_t - \frac{d}{dt}\bar{u}_t, \hat{u}_t - \bar{u}_t\right\rangle = -2\langle \text{grad}_{\hat{\rho}}\widehat{L}(\hat{\rho}_t) - \text{grad}_{\bar{u}}L(\rho_t), \hat{u}_t - \bar{u}_t\rangle \quad \text{(E.14)}$$

We can expand the difference in gradients as

$$\operatorname{grad}_{\hat{\rho}}\widehat{L}(\hat{\rho}_t) - \operatorname{grad}_{\bar{u}}L(\rho_t) = \left(\operatorname{grad}_{\hat{u}}\widehat{L}(\hat{\rho}_t) - \operatorname{grad}_{\hat{u}}L(\hat{\rho}_t)\right) + \left(\operatorname{grad}L_{\hat{u}}(\hat{\rho}_t) - \operatorname{grad}L_{\hat{u}}(\rho_t)\right) \quad \text{(E.15)}$$

$$+ \left(\operatorname{grad}_{\hat{u}}L(\rho_t) - \operatorname{grad}_{\bar{u}}L(\rho_t)\right) \quad \text{(E.16)}$$

Thus,

$$\frac{d}{dt}\|\hat{u}_t - \bar{u}_t\|_2^2 = -2\langle\operatorname{grad}_{\hat{u}}\widehat{L}(\hat{\rho}_t) - \operatorname{grad}_{\hat{u}}L(\hat{\rho}_t), \hat{u}_t - \bar{u}_t\rangle - 2\langle\operatorname{grad}L_{\hat{u}}(\hat{\rho}_t) - \operatorname{grad}L_{\hat{u}}(\rho_t), \hat{u}_t - \bar{u}_t\rangle$$

$$\text{(E.17)}$$

$$- 2\langle\operatorname{grad}_{\hat{u}}L(\rho_t) - \operatorname{grad}_{\bar{u}}L(\rho_t), \hat{u}_t - \bar{u}_t\rangle \quad \text{(E.18)}$$

$$= C_t + B_t + A_t \quad \text{(E.19)}$$

as desired. $\qquad\square$

Next, we give a proof of Lemma 5.4, which gives an upper bound for each of $A_t$, $B_t$ and $C_t$.

*Proof of Lemma 5.4.* The proof directly follows from Lemma E.8, Lemma E.7, Lemma E.9, Lemma E.10 and Lemma E.11. $\qquad\square$

Combining the upper bounds in Lemma 5.4 and the decomposition in Lemma E.1, we have the following Lemma E.2 which upper bounds the growth of $\|\hat{u}_t - \bar{u}_t\|$. Recall that $T_{*,\epsilon}$ is the total runtime for the algorithm (see Theorem 3.3), and $T_1$ is the time such that $w_{T_1}(\iota_U) = \frac{1}{\log d}$ (see Definition 4.5).

**Lemma E.2** (Growth Rate of $\|\delta_t\|_2^2$ With Dependency on $n$). *In the setting of Theorem 3.4, suppose that the inductive hypothesis in Assumption 5.3 holds for some $\phi$ and $\psi$ up until $t$ for some $t \leq T_{*,\epsilon}$. Further assume that $\psi < \frac{1}{2}$ and $\phi > \frac{1}{2}$. Then, when $t \leq T_1$ we have*

$$\frac{d}{dt}\|\hat{u}_t - \bar{u}_t\|_2^2 \leq \left(4\hat{\sigma}_{2,d}^2|D_{2,t}|\left(1 + O\left(\frac{1}{\log d}\right)\right)\right)\|\hat{u}_t - \bar{u}_t\|_2^2 \quad \text{(E.20)}$$

$$+ (\hat{\sigma}_{2,d}^2 + \hat{\sigma}_{4,d}^2) \cdot \frac{d(\log d)^{O(1)}}{\sqrt{m}}\|\hat{u}_t - \bar{u}_t\|_2 \quad \text{(E.21)}$$

$$+ (\hat{\sigma}_{2,d}^2 + \hat{\sigma}_{4,d}^2)(\log d)^{O(1)}\left(\frac{1}{d^\phi} + \sqrt{\frac{d}{n}} + \frac{d^{2.5-2\psi-2\phi}}{n}\right)\|\hat{u}_t - \bar{u}_t\|_2 \quad \text{(E.22)}$$

$$+ (\hat{\sigma}_{2,d}^2 + \hat{\sigma}_{4,d}^2)(\log d)^{O(1)}\left(\frac{d^{2-2\psi-\phi}}{n} + \frac{d^{4-4\psi-2\phi}}{n}\right)\|\hat{u}_t - \bar{u}_t\|_2^2 \quad \text{(E.23)}$$

*and*

$$\frac{d}{dt}\overline{\Delta}_t^2 \leq \left(4\hat{\sigma}_{2,d}^2|D_{2,t}|\left(1 + O\left(\frac{1}{\log d}\right)\right)\right)\overline{\Delta}_t^2 \quad \text{(E.24)}$$

$$+ (\hat{\sigma}_{2,d}^2 + \hat{\sigma}_{4,d}^2) \cdot \frac{d(\log d)^{O(1)}}{\sqrt{m}}\overline{\Delta}_t \quad \text{(E.25)}$$

$$+ (\hat{\sigma}_{2,d}^2 + \hat{\sigma}_{4,d}^2)(\log d)^{O(1)}\left(\sqrt{\frac{d}{n}} + \frac{d^{2.5-2\psi-2\phi}}{n}\right)\overline{\Delta}_t \quad \text{(E.26)}$$

$$+ (\hat{\sigma}_{2,d}^2 + \hat{\sigma}_{4,d}^2)(\log d)^{O(1)}\left(\frac{1}{d^\phi} + \frac{d^{2-2\psi-\phi}}{n} + \frac{d^{4-4\psi-2\phi}}{n}\right)\overline{\Delta}_t^2. \quad \text{(E.27)}$$

*When $t \in [T_1, T_{*,\epsilon}]$, we have*

$$\frac{d}{dt}\|\hat{u}_t - \bar{u}_t\|_2^2 \lesssim (\hat{\sigma}_{2,d}^2 + \hat{\sigma}_{4,d}^2)\gamma_2\|\hat{u}_t - \bar{u}_t\|_2^2 \quad \text{(E.28)}$$

$$+ (\hat{\sigma}_{2,d}^2 + \hat{\sigma}_{4,d}^2) \cdot \frac{d(\log d)^{O(1)}}{\sqrt{m}}\|\hat{u}_t - \bar{u}_t\|_2 \quad \text{(E.29)}$$

$$+ (\hat{\sigma}_{2,d}^2 + \hat{\sigma}_{4,d}^2)(\log d)^{O(1)}\left(\frac{1}{d^\phi} + \sqrt{\frac{d}{n}} + \frac{d^{2.5-2\psi-2\phi}}{n}\right)\|\hat{u}_t - \bar{u}_t\|_2 \quad \text{(E.30)}$$

$$+ (\hat{\sigma}_{2,d}^2 + \hat{\sigma}_{4,d}^2)(\log d)^{O(1)}\left(\frac{d^{2-2\psi-\phi}}{n} + \frac{d^{4-4\psi-2\phi}}{n}\right)\|\hat{u}_t - \bar{u}_t\|_2^2 \quad \text{(E.31)}$$

*and*

$$\frac{d}{dt}\overline{\Delta}_t^2 \leq (\hat{\sigma}_{2,d}^2 + \hat{\sigma}_{4,d}^2)\gamma_2\overline{\Delta}_t^2 \tag{E.32}$$

$$+ (\hat{\sigma}_{2,d}^2 + \hat{\sigma}_{4,d}^2) \cdot \frac{d(\log d)^{O(1)}}{\sqrt{m}}\overline{\Delta}_t \tag{E.33}$$

$$+ (\hat{\sigma}_{2,d}^2 + \hat{\sigma}_{4,d}^2)(\log d)^{O(1)}\Big(\sqrt{\frac{d}{n}} + \frac{d^{2.5-2\psi-2\phi}}{n}\Big)\overline{\Delta}_t \tag{E.34}$$

$$+ (\hat{\sigma}_{2,d}^2 + \hat{\sigma}_{4,d}^2)(\log d)^{O(1)}\Big(\frac{1}{d^\phi} + \frac{d^{2-2\psi-\phi}}{n} + \frac{d^{4-4\psi-2\phi}}{n}\Big)\overline{\Delta}_t^2 \,. \tag{E.35}$$

*Proof of Lemma E.2.* This result follows directly from Lemma 5.4 and the inductive hypothesis. By the inductive hypothesis, we have $\overline{\Delta}_t \leq d^{-\phi}$. We apply this to the bound for $C_t$ in Lemma 5.4 to obtain the desired bounds above. Note that in our bounds for $\frac{d}{dt}\overline{\Delta}_t^2$, it is important that we use the upper bound for $\mathbb{E}[B_t]$ in Lemma 5.4 instead of the upper bound for $B_t$. $\qquad\square$

Using this bound on the growth rate, we can show that the inductive hypothesis is maintained.

**Lemma E.3** (IH is Maintained). *Suppose we are in the setting of Theorem 3.4. Suppose $\psi > 0$ and $1 > \phi > \frac{1}{2} + \psi$. Let $n = d^\mu(\log d)^C$ where $\mu \geq \max\{2 + 2\phi, 4 - 2\phi - 4\psi\}$ and $C$ is a sufficiently large universal constant. Suppose the inductive hypothesis in Assumption 5.3 holds for all $t \leq T$ where $T < T_{*,\epsilon}$. Then, when $m \gtrsim d^{3+2\phi}(\log d)^{\Omega(1)}$ and $m \leq d^C$ for a sufficiently large universal constant $C$, we have that the inductive hypothesis holds for all $t \leq T + c_{\mathrm{IH}}$ where $c_{\mathrm{IH}} > 0$ is a constant that only depends on $d$, $\phi$, $\psi$, $\hat{\sigma}_{2,d}$, $\hat{\sigma}_{4,d}$.*

*Proof.* Suppose the inductive hypothesis holds for all $t \leq T$ where $T \leq T_1$. Then, by Lemma E.2 we have

$$\frac{d}{dt}\|\hat{u}_t - \bar{u}_t\|_2^2 \leq \Big(4\hat{\sigma}_{2,d}^2|D_{2,t}|\Big(1 + O\Big(\frac{1}{\log d}\Big)\Big)\Big)\|\hat{u}_t - \bar{u}_t\|_2^2 \tag{E.36}$$

$$+ \frac{(\hat{\sigma}_{2,d}^2 + \hat{\sigma}_{4,d}^2)d(\log d)^{O(1)}}{\sqrt{m}}\|\hat{u}_t - \bar{u}_t\|_2 \tag{E.37}$$

$$+ O\Big(\frac{(\hat{\sigma}_{2,d}^2 + \hat{\sigma}_{4,d}^2)(\log d)^{O(1)}}{d^\phi}\|\hat{u}_t - \bar{u}_t\|_2\Big) \tag{E.38}$$

$$\leq \Big(4\hat{\sigma}_{2,d}^2|D_{2,t}|\Big(1 + O\Big(\frac{1}{\log d}\Big)\Big)\Big)\|\hat{u}_t - \bar{u}_t\|_2^2 \tag{E.39}$$

$$+ O\Big(\frac{(\hat{\sigma}_{2,d}^2 + \hat{\sigma}_{4,d}^2)(\log d)^{O(1)}}{d^\phi}\|\hat{u}_t - \bar{u}_t\|_2\Big) \,. \tag{E.40}$$

Here we have used the fact that $n = d^\mu(\log d)^{O(1)}$ where $\mu \geq \max(4 - 4\psi - 2\phi, 2 + 2\phi)$ as well as the fact that $\phi > \frac{1}{2}$, as follows:

- In the third term in the bound for $\frac{d}{dt}\|\hat{u}_t - \bar{u}_t\|_2^2$ from Lemma E.2, only the $\frac{1}{d^\phi}$ term is important — this term dominates the $\sqrt{\frac{d}{n}}$ and $\frac{d^{2.5-2\psi-2\phi}}{n}$ terms, using the fact that $\mu > 2 + 2\phi$.

- In the fourth term in the bound for $\frac{d}{dt}\|\hat{u}_t - \bar{u}_t\|_2^2$, the $\frac{d^{2-2\psi-\phi}}{n}\|\hat{u}_t - \bar{u}_t\|_2^2$ term is dominated by the $\frac{1}{d^\phi}\|\hat{u}_t - \bar{u}_t\|_2$ term — this is because $\mu \geq 2 + 2\phi$.

- Since $\mu \geq 4 - 4\psi - 2\phi$, if we increase $n$ by a $(\log d)^{O(1)}$ factor, then the $\frac{d^{4-4\psi-2\phi}}{n}\|\hat{u}_t - \bar{u}_t\|_2^2$ term can be absorbed into the first term, as it is dominated by the $4\hat{\sigma}_{2,d}^2|D_{2,t}|O\Big(\frac{1}{\log d}\Big)\|\hat{u}_t - \bar{u}_t\|_2^2$ term. (Recall that by Lemma D.8, $|D_{2,t}| \gtrsim \gamma_2$ during Phase 1).

In the second inequality above, we have used the fact that $m \gtrsim d^{3+2\phi}(\log d)^{\Omega(1)}$ — this implies that the term involving $m$ is dominated by the $\frac{(\log d)^{O(1)}}{d^\phi}\|\hat{u}_t - \bar{u}_t\|_2$ term.

Additionally, by Lemma E.2, we have

$$\frac{d}{dt}\overline{\Delta}_t^2 \leq \Big(4\hat{\sigma}_{2,d}^2|D_{2,t}|\Big(1 + O\Big(\frac{1}{\log d}\Big)\Big)\Big)\overline{\Delta}_t^2 + \frac{(\hat{\sigma}_{2,d}^2 + \hat{\sigma}_{4,d}^2)\cdot d(\log d)^{O(1)}}{\sqrt{m}}\overline{\Delta}_t \tag{E.41}$$

$$+ O\Big(\frac{(\hat{\sigma}_{2,d}^2 + \hat{\sigma}_{4,d}^2)}{d^{0.5+\phi}(\log d)^{O(1)}}\overline{\Delta}_t\Big) \tag{E.42}$$

$$\leq \Big(4\hat{\sigma}_{2,d}^2|D_{2,t}|\Big(1 + O\Big(\frac{1}{\log d}\Big)\Big)\Big)\overline{\Delta}_t^2 + O\Big(\frac{(\hat{\sigma}_{2,d}^2 + \hat{\sigma}_{4,d}^2)}{d^{0.5+\phi}(\log d)^{O(1)}}\overline{\Delta}_t\Big). \tag{E.43}$$

Here, we apply the bound for $\frac{d}{dt}\overline{\Delta}_t^2$ from Lemma E.2. In the third term in this bound, the $\sqrt{\frac{d}{n}}$ term dominates the $\frac{d^{2.5-2\psi-2\phi}}{n}$ term, using the fact that $\phi > \frac{1}{2}$ and $\mu \geq 2 + 2\phi > 3$, meaning $\frac{d^{2.5-2\psi-2\phi}}{n} \leq \frac{d^{1.5}}{n} \leq \frac{1}{\sqrt{n}}$. In the fourth term, we deal with the $\frac{d^{4-4\psi-2\phi}}{n}$ term similarly to our calculations for the bound for $\frac{d}{dt}\|\hat{u}_t - \bar{u}_t\|_2^2$. We can also deal with the $\frac{d^{2-2\psi-\phi}}{n}$ term similarly, since by our definition of $\mu$, this will be at most $\frac{1}{d^{3\phi}} \leq \frac{1}{(\log d)^{O(1)}}$, and will thus be absorbed into the first term involving $|D_{2,t}|$. Using $\mu \geq 2 + 2\phi$, we then obtain $\sqrt{\frac{d}{n}} \leq \frac{1}{d^{0.5+\phi}}$ as the coefficient of the $\overline{\Delta}_t$ term (i.e. the first-order term involving $\overline{\Delta}_t$). Note that here we need to have $m \gtrsim d^{3+2\phi}(\log d)^{\Omega(1)}$ in order to absorb the term involving $m$ into the $\frac{1}{d^{0.5+\phi}}\overline{\Delta}_t$ term.

Now using Lemma I.12 we have

$$\overline{\Delta}_t^2 \lesssim \frac{1}{(\log d)^{O(1)}}\Big(\frac{(\hat{\sigma}_{2,d}^2 + \hat{\sigma}_{4,d}^2)O(\frac{1}{d^{0.5+\phi}})}{4\hat{\sigma}_{2,d}^2 \min_t |D_{2,t}|}\Big)^2 \exp\Big(\int_0^T 4\hat{\sigma}_{2,d}^2|D_{2,t}|\Big(1 + O\Big(\frac{1}{\log d}\Big)\Big)dt\Big) \tag{E.44}$$

and

$$\Delta_{\max,T}^2 \lesssim (\log d)^{O(1)}\Big(\frac{(\hat{\sigma}_{2,d}^2 + \hat{\sigma}_{4,d}^2)O(\frac{1}{d^\phi})}{4\hat{\sigma}_{2,d}^2 \min_t |D_{2,t}|}\Big)^2 \exp\Big(\int_t^T 4\hat{\sigma}_{2,d}^2|D_{2,t}|\Big(1 + O\Big(\frac{1}{\log d}\Big)\Big)dt\Big) \tag{E.45}$$

By Lemma D.5 we have

$$\exp\Big(\int_{t=0}^T 4\hat{\sigma}_{2,d}^2|D_{2,t}|dt\Big) \lesssim d \tag{E.46}$$

and by Lemma D.8 we know that $|D_{2,t}| \gtrsim \gamma_2$ during Phase 1. Thus,

$$\Delta_{\max,T}^2 \leq O\Big(\frac{(\log d)^{O(1)}}{\gamma_2^2 d^{2\phi-1}}\Big). \tag{E.47}$$

Since $2\phi - 1 > 2\psi$ and $d$ is chosen to be sufficiently large after $\gamma_2$ is chosen (Assumption 3.2), we have that $\Delta_{\max,T} < \sqrt{\frac{1}{d^{\psi+\phi-1/2}}} = \frac{1}{d^{\frac{1}{2}\phi+\frac{1}{2}\psi-\frac{1}{4}}} < \frac{1}{d^\psi}$. Note that $\Delta_{\max,t}$ is continuous in $t$ and we have

$$\frac{d}{dt}\Delta_{\max,t} \leq \max_\chi \frac{d}{dt}\|\hat{u}_t - \bar{u}_t\|_2 \tag{E.48}$$

$$= \max_\chi \frac{1}{\|\hat{u}_t - \bar{u}_t\|_2}\frac{d}{dt}\|\hat{u}_t - \bar{u}_t\|_2^2 \tag{E.49}$$

$$\leq \max_\chi \frac{1}{\|\hat{u}_t - \bar{u}_t\|_2}|\langle \hat{u}_t - \bar{u}_t, \text{grad}_{\hat{u}_t}\widehat{L}(\hat{u}_t) - \text{grad}_{\bar{u}_t}L(\bar{u}_t)\rangle| \tag{E.50}$$

$$\leq \max_\chi \|\text{grad}_{\hat{u}_t}\widehat{L}(\hat{u}_t) - \text{grad}_{\bar{u}_t}L(\bar{u}_t)\|_2 \qquad \text{(By Cauchy-Schwarz Inequality)}$$

$$\leq (\hat{\sigma}_{2,d}^2 + \hat{\sigma}_{4,d}^2)d^4. \qquad \text{(By Lemma E.14)}$$

Thus, for $\mu_1 = \left( \frac{1}{d^\psi} - \frac{1}{d^{\frac{1}{2}\phi + \frac{1}{2}\psi - \frac{1}{4}}} \right) \cdot \frac{1}{(\hat{\sigma}_{2,d}^2 + \hat{\sigma}_{4,d}^2)d^4}$ we still have the inductive hypothesis for $\Delta_{\max}$ until time $T + \mu_1$, that is, $\Delta_{\max, T+\mu_1} \leq \frac{1}{d^\psi}$. Similarly, our calculation above using Lemma I.12 gives us

$$\overline{\Delta}_T^2 \leq O\Big( \frac{1}{\gamma_2^2 d^{2\phi}(\log d)^{O(1)}} \Big) \tag{E.51}$$

Thus, we have $\overline{\Delta}_T < \frac{1}{d^\phi(\log d)^{O(1)}} < \frac{1}{d^\phi}$ (where in the middle expression, we choose a smaller power of $(\log d)^{O(1)}$ so that the inequality holds). Observe that $\overline{\Delta}_t$ is continuous in $t$, and

$$\frac{d}{dt}\overline{\Delta}_t = \frac{1}{\overline{\Delta}_t} \frac{d}{dt}\overline{\Delta}_t^2 \tag{E.52}$$

$$= \frac{1}{\overline{\Delta}_t} \mathbb{E}_\chi[\langle \hat{u}_t - \bar{u}_t, \hat{u}_t' - \bar{u}_t' \rangle] \tag{E.53}$$

$$\leq \frac{1}{\overline{\Delta}_t} \mathbb{E}_\chi[\|\hat{u}_t - \bar{u}_t\|_2 \|\hat{u}_t' - \bar{u}_t'\|_2] \qquad \text{(By Cauchy-Schwarz Inequality)}$$

$$\leq \frac{(\hat{\sigma}_{2,d}^2 + \hat{\sigma}_{4,d}^2)d^4}{\overline{\Delta}_t} \mathbb{E}_\chi[\|\hat{u}_t - \bar{u}_t\|_2] \qquad \text{(By Lemma E.14)}$$

$$\leq (\hat{\sigma}_{2,d}^2 + \hat{\sigma}_{4,d}^2)d^4 . \tag{E.54}$$

Thus, for $\mu_2 = \left( \frac{1}{d^\phi} - \frac{1}{d^\phi(\log d)^{O(1)}} \right) \cdot \frac{1}{(\hat{\sigma}_{2,d}^2 + \hat{\sigma}_{4,d}^2)d^4}$ we still have the inductive hypothesis for $\overline{\Delta}$ until time $T + \mu_2$, i.e. $\overline{\Delta}_{T+\mu_2} \leq \frac{1}{d^\phi}$. As a result, assuming the inductive hypothesis holds up to time $T$, it holds up to time $T + \min(\mu_1, \mu_2)$.

Our above proof strategy applies to all times $T \leq T_1$. For the induction when $T \in [T_1, T_{*,\epsilon}]$ we follow a similar argument but instead use the bound on $A_t$ obtained from Lemma E.8. Since we know from Lemma D.36 and the assumption that $\epsilon = O(\frac{1}{\log \log d})$ that

$$\int_{T_1}^{T_{*,\epsilon}} (\hat{\sigma}_{2,d}^2 + \hat{\sigma}_{4,d}^2)\gamma_2 dt \leq (T_{*,\epsilon} - T_1)(\hat{\sigma}_{2,d}^2 + \hat{\sigma}_{4,d}^2)\gamma_2 \leq \frac{\text{poly}(\log \log d)}{\gamma_2^{O(1)}}, \tag{E.55}$$

we know that this additional phase will at most add another $\exp(\text{poly}(\log \log d))$ factor to the growth of $\Delta_{\max,t}$ which is smaller than any polynomial in $d$. Thus, Eq. (E.47) and Eq. (E.51) still hold for this period, which finishes the proof. $\qquad \square$

Finally, we prove Lemma 5.2:

*Proof of Lemma 5.2.* The proof directly follows Lemma E.3 with $\phi = \frac{\mu-1}{4}$ and $\psi = \frac{\mu-3}{8}$. $\qquad \square$

## E.1 Upper bound on $A_t$

The goal of this section is to obtain an upper bound on

$$A_t(\chi) := -2\langle \text{grad}_{\hat{u}}L(\rho_t) - \text{grad}_{\bar{u}}L(\rho_t), \hat{u}_t(\chi) - \bar{u}_t(\chi) \rangle . \tag{E.56}$$

When there is no risk of confusion, we simply use $A_t$, $\hat{u}_t$, and $\bar{u}_t$, omitting the dependence on $\chi$.

We first obtain a strong upper bound on this inner product during Phase 1 of the population projected gradient flow. To obtain an upper bound on this inner product, we separately consider the contributions from the first coordinates of the vectors, and the remaining $(d-1)$ coordinates — in other words, the gradients with respect to $w$ and the gradients with respect to $z$. For convenience, let $\text{grad}_w L(\rho)$ denote the gradient with respect to $w$ and $\text{grad}_z L(\rho)$ denote the gradient with respect to $z$. The following lemma is about the contribution of the first coordinates of the particles to $A_t$, and is useful during Phase 1.

**Lemma E.4** (Contribution of $w$'s). *Suppose we are in the setting of Theorem 3.4, and let $\rho_t$ be the projected gradient flow on population loss, initialized with the uniform distribution on $\mathbb{S}^{d-1}$, defined*

*in Eq. (3.7). Suppose $D_{2,t} < 0$ and $D_{4,t} < 0$, where $D_{2,t}$ and $D_{4,t}$ are as defined in Section 4.1. Then, the contribution of the first coordinates to $A_t$ can be bounded by*

$$-2\Big(\mathrm{grad}_{\hat{w}}L(\rho_t) - \mathrm{grad}_{\bar{w}}L(\rho_t)\Big) \cdot (\hat{w}_t - \bar{w}_t) \tag{E.57}$$

$$\leq 4\hat{\sigma}_{2,d}^2|D_{2,t}|(\hat{w}_t - \bar{w}_t)^2 + O(\hat{\sigma}_{4,d}^2|D_{4,t}|w_{\max}(\hat{w}_t - \bar{w}_t)^2) \tag{E.58}$$

$$+ O\Big(\hat{\sigma}_{4,d}^2|D_{4,t}|\Delta_{\max}(\hat{w}_t - \bar{w}_t)^2\Big) \tag{E.59}$$

$$+ O\Big(\frac{\hat{\sigma}_{2,d}^2|D_{2,t}| + \hat{\sigma}_{4,d}^2|D_{4,t}|}{d} \cdot (\hat{w}_t - \bar{w}_t)^2\Big). \tag{E.60}$$

*where $w_{\max}$ is as defined in Definition 4.5.*

The constant $4$ at the beginning of the first term is particularly important since it determines the growth rate of $\|\delta_t\|_2^2$, and in turn affects the number of samples we will need. At least during Phase 1, the contribution of the first coordinates will be the dominant term in $A_t$, as we will see in the next few lemmas.

*Proof of Lemma E.4.* The contribution of the first coordinate to the dot product defining $A_t$ is

$$-2\Big(\mathrm{grad}_{\hat{w}}L(\rho_t) - \mathrm{grad}_{\bar{w}}L(\rho_t)\Big) \cdot (\hat{w}_t - \bar{w}_t) \tag{E.61}$$

$$= 2(v(\hat{w}_t) - v(\bar{w}_t)) \cdot (\hat{w}_t - \bar{w}_t) \tag{E.62}$$

where $v(w)$ is as defined in Lemma 4.4. We can further simplify $v(\hat{w}_t) - v(\bar{w}_t)$ using Lemma 4.4. Let $P_t$ and $Q_t$ be defined as in Lemma 4.4. Then,

$$v(\hat{w}_t) - v(\bar{w}_t) = -(1 - \hat{w}_t^2)(P_t(\hat{w}_t) + Q_t(\hat{w}_t)) + (1 - \bar{w}_t^2)(P_t(\bar{w}_t) + Q_t(\bar{w}_t)). \tag{E.63}$$

Let us upper bound the term corresponding to $Q_t$ first. Since each coefficient of $Q_t$ is bounded above by $O(\frac{\hat{\sigma}_{2,d}^2|D_{2,t}| + \hat{\sigma}_{4,d}^2|D_{4,t}|}{d})$ (by the conclusion of Lemma 4.4) and $Q_t$ only has linear and cubic terms, we have

$$\Big| - (1 - \hat{w}_t^2)Q_t(\hat{w}_t) + (1 - \bar{w}_t^2)Q_t(\bar{w}_t)\Big| \cdot |\hat{w}_t - \bar{w}_t| \lesssim \frac{\hat{\sigma}_{2,d}^2|D_{2,t}| + \hat{\sigma}_{4,d}^2|D_{4,t}|}{d}|\hat{w}_t - \bar{w}_t|^2 \tag{E.64}$$

by Lemma I.16. The terms corresponding to $P_t(\bar{w}_t)$ in $(v(\hat{w}_t) - v(\bar{w}_t))(\hat{w}_t - \bar{w}_t)$ can be bounded as follows:

$$(-(1 - \hat{w}_t^2)P_t(\hat{w}_t) + (1 - \bar{w}_t^2)P_t(\bar{w}_t)) \cdot (\hat{w}_t - \bar{w}_t) \tag{E.65}$$

$$= 2\hat{\sigma}_{2,d}^2|D_{2,t}|(\hat{w}_t - \bar{w}_t)^2 + 4\hat{\sigma}_{4,d}^2|D_{4,t}|(\hat{w}_t - \bar{w}_t)^2(\hat{w}_t^2 + \hat{w}_t\bar{w}_t + \bar{w}_t^2) \tag{E.66}$$

$$- 2\hat{\sigma}_{2,d}^2|D_{2,t}|(\hat{w}_t - \bar{w}_t)^2(\hat{w}_t^2 + \hat{w}_t\bar{w}_t + \bar{w}_t^2) - 4\hat{\sigma}_{4,d}^2|D_{4,t}|(\hat{w}_t - \bar{w}_t)^2(\hat{w}_t^2 + \hat{w}_t\bar{w}_t + \bar{w}_t^2) \tag{E.67}$$

$$\leq 2\hat{\sigma}_{2,d}^2|D_{2,t}|(\hat{w}_t - \bar{w}_t)^2 + 4\hat{\sigma}_{4,d}^2|D_{4,t}|(\hat{w}_t - \bar{w}_t)^2(\hat{w}_t^2 + \hat{w}_t\bar{w}_t + \bar{w}_t^2)$$
$$\text{(Since last two terms are nonnegative)}$$

$$\leq 2\hat{\sigma}_{2,d}^2|D_{2,t}|(\hat{w}_t - \bar{w}_t)^2 + O(\hat{\sigma}_{4,d}^2|D_{4,t}||\hat{w}_t + \bar{w}_t|(\hat{w}_t - \bar{w}_t)^2) \qquad \text{(Since } \bar{w}_t, \hat{w}_t \leq 1)$$

Thus, the overall bound on the contribution of the first coordinates to $A_t$ is

$$2(v(\hat{w}_t) - v(\bar{w}_t)) \cdot (\hat{w}_t - \bar{w}_t) \leq 2 \cdot (-(1 - \hat{w}_t^2)P_t(\hat{w}_t) + (1 - \bar{w}_t^2)P_t(\bar{w}_t)) \cdot (\hat{w}_t - \bar{w}_t) \tag{E.68}$$

$$+ \Big| - (1 - \hat{w}_t^2)Q_t(\hat{w}_t) + (1 - \bar{w}_t^2)Q_t(\bar{w}_t)\Big| \cdot |\hat{w}_t - \bar{w}_t| \tag{E.69}$$

$$\leq 4\hat{\sigma}_{2,d}^2|D_{2,t}|(\hat{w}_t - \bar{w}_t)^2 + O(\hat{\sigma}_{4,d}^2|D_{4,t}||\hat{w}_t + \bar{w}_t|(\hat{w}_t - \bar{w}_t)^2) \tag{E.70}$$

$$+ O\Big(\frac{\hat{\sigma}_{2,d}^2|D_{2,t}| + \hat{\sigma}_{4,d}^2|D_{4,t}|}{d}(\hat{w}_t - \bar{w}_t)^2\Big) \tag{E.71}$$

We can simplify this bound further in terms of $w_{\max}$ and $\Delta_{\max,t}$. First observe that

$$|\hat{w}_t - \bar{w}_t| \leq \|\delta_t\|_2 \leq \Delta_{\max,t} \tag{E.72}$$

Thus, by the triangle inequality, we have $|\bar{w}_t| \leq w_{\max}$ and $|\hat{w}_t| \leq w_{\max} + \Delta_{\max,t}$ meaning

$$2(v(\hat{w}_t) - v(\bar{w}_t)) \cdot (\hat{w}_t - \bar{w}_t) \leq 4\hat{\sigma}_{2,d}^2 |D_{2,t}|(\hat{w}_t - \bar{w}_t)^2 + O(\hat{\sigma}_{4,d}^2 |D_{4,t}||\hat{w}_t + \bar{w}_t|(\hat{w}_t - \bar{w}_t)^2) \tag{E.73}$$

$$+ O\Big(\frac{\hat{\sigma}_{2,d}^2 |D_{2,t}| + \hat{\sigma}_{4,d}^2 |D_{4,t}|}{d}(\hat{w}_t - \bar{w}_t)^2\Big) \tag{E.74}$$

$$\leq 4\hat{\sigma}_{2,d}^2 |D_{2,t}|(\hat{w}_t - \bar{w}_t)^2 + O(\hat{\sigma}_{4,d}^2 |D_{4,t}| w_{\max}(\hat{w}_t - \bar{w}_t)^2) \tag{E.75}$$

$$+ O\Big(\hat{\sigma}_{4,d}^2 |D_{4,t}|\Delta_{\max,t}(\hat{w}_t - \bar{w}_t)^2\Big) \tag{E.76}$$

$$+ O\Big(\frac{\hat{\sigma}_{2,d}^2 |D_{2,t}| + \hat{\sigma}_{4,d}^2 |D_{4,t}|}{d}(\hat{w}_t - \bar{w}_t)^2\Big) \tag{E.77}$$

$$\tag{E.78}$$

as desired. $\qquad\square$

Next, we study the contribution of the last $(d-1)$ coordinates of the particles to $A_t(\chi)$ during Phase 1. We first introduce the following formula for the projected gradient of the population loss with respect to the last $(d-1)$ coordinates of a particle.

**Lemma E.5.** *Let $\rho$ be a distribution which is rotationally invariant, as in Definition 4.1. Let $\mathrm{grad}_u L(\rho)$ be defined as in Eq. (3.4). In addition, let $\mathrm{grad}_z L(\rho)$ denote the last $(d-1)$ coordinates of $\mathrm{grad}_u L(\rho)$, for any particle $u = (w, z)$. Then, for any particle $u = (w, z)$, we have $\mathrm{grad}_z L(\rho) = -\Big(\frac{w}{1-w^2} \cdot \mathrm{grad}_w L(\rho)\Big)z$.*

*Proof of Lemma E.5.* Notice that $w^2 + z^2 = 1$ for any $u = (w, z) \in \mathbb{S}^{d-1}$, and this equality is maintained by projected gradient descent on the population loss, as in Eq. (3.7). Also, by Lemma I.15, we have $\mathrm{grad}_z L(\rho) = cz$ for some scalar $c$, since $\rho$ is rotationally invariant. Thus,

$$0 = \frac{d}{dt}(w^2 + z^2) \tag{E.79}$$

$$= -2w\mathrm{grad}_w L(\rho) - 2\langle z, \mathrm{grad}_z L(\rho)\rangle \tag{E.80}$$

$$= -2w\mathrm{grad}_w L(\rho) - 2c\|z\|^2 \tag{E.81}$$

$$= -2w\mathrm{grad}_w L(\rho) - 2c(1 - w^2). \tag{E.82}$$

Rearranging gives us

$$c = -\frac{w}{1 - w^2}\mathrm{grad}_w L(\rho). \tag{E.83}$$

$\qquad\square$

By directly applying this formula for the last $(d-1)$ coordinates of the projected gradient, we have the following upper bound on the contribution of the last $(d-1)$ coordinates to $A_t$ during Phase 1.

**Lemma E.6** (Contribution of $z$'s During Phase 1.). *Suppose we are in the setting of Theorem 3.4. Let $t \leq T_1$ (where $T_1$ is as defined in Definition 4.5). Let $\rho_t$ be defined according to the population projected gradient flow as in Eq. (3.7). Let $\mathrm{grad}_u L(\rho)$ be as in Eq. (3.4), and let $\mathrm{grad}_z L(\rho)$ denote the last $(d-1)$ coordinates of $\mathrm{grad}_u L(\rho)$ for any particle $u = (w, z)$. Then, we have the following upper bound:*

$$-2\big\langle \mathrm{grad}_{\hat{z}} L(\rho_t) - \mathrm{grad}_{\bar{z}} L(\rho_t), \hat{z}_t - \bar{z}_t\big\rangle \tag{E.84}$$

$$\lesssim \hat{\sigma}_{2,d}^2 |D_{2,t}|(|\bar{w}_t| + |\hat{w}_t|)\|\bar{u}_t - \hat{u}_t\|^2 + \hat{\sigma}_{4,d}^2 |D_{4,t}|(|\bar{w}_t^3| + |\hat{w}_t^3|)\|\bar{u}_t - \hat{u}_t\|^2 \tag{E.85}$$

$$+ \frac{\hat{\sigma}_{2,d}^2 |D_{2,t}| + \hat{\sigma}_{4,d}^2 |D_{4,t}|}{d}\|\bar{u}_t - \hat{u}_t\|_2^2. \tag{E.86}$$

*Proof of Lemma E.6.* Using the formula for projected gradient in Lemma E.5, we have that

$$- \langle \mathrm{grad}_{\hat{z}} L(\rho_t) - \mathrm{grad}_{\bar{z}} L(\rho_t), \hat{z}_t - \bar{z}_t \rangle \tag{E.87}$$

$$= \Big\langle \frac{\hat{w}_t}{1 - \hat{w}_t^2} \cdot \mathrm{grad}_{\hat{w}} L(\rho_t) \cdot \hat{z}_t - \frac{\bar{w}_t}{1 - \bar{w}_t^2} \cdot \mathrm{grad}_{\bar{w}} L(\rho_t) \cdot \bar{z}_t, \hat{z}_t - \bar{z}_t \Big\rangle \tag{E.88}$$

$$= \Big( \frac{\hat{w}_t \mathrm{grad}_{\hat{w}} L(\rho_t)}{1 - \hat{w}_t^2} - \frac{\bar{w}_t \mathrm{grad}_{\bar{w}} L(\rho_t)}{1 - \bar{w}_t^2} \Big) \hat{z}^\top (\hat{z}_t - \bar{z}_t) + \frac{\bar{w}_t \mathrm{grad}_{\bar{w}} L(\rho_t)}{1 - \bar{w}_t^2} \| \hat{z}_t - \bar{z}_t \|_2^2 \tag{E.89}$$

The second term is always negative for $t \leq T_1$, because Lemma D.9 implies that $v(\bar{w}_t)$ has the same sign as $\bar{w}_t$, meaning that $\mathrm{grad}_{\bar{w}} L(\rho_t)$ has the opposite sign. We bound the first term as

$$\Big( \frac{\hat{w}_t \mathrm{grad}_{\hat{w}} L(\rho_t)}{1 - \hat{w}_t^2} - \frac{\bar{w}_t \mathrm{grad}_{\bar{w}} L(\rho_t)}{1 - \bar{w}_t^2} \Big) \hat{z}_t^\top (\bar{z}_t - \hat{z}_t) \tag{E.90}$$

$$\leq \Big| \frac{\hat{w}_t \mathrm{grad}_{\hat{w}} L(\rho_t)}{1 - \hat{w}_t^2} - \frac{\bar{w}_t \mathrm{grad}_{\bar{w}} L(\rho_t)}{1 - \bar{w}_t^2} \Big| \cdot \| \bar{z}_t - \hat{z}_t \| \qquad \text{(B.c. } |\hat{z}_t\|_2 \leq \|\hat{u}_t\|_2 = 1 )$$

$$= |\hat{w}_t (P_t(\hat{w}_t) + Q_t(\hat{w}_t)) - \bar{w}_t (P_t(\bar{w}_t) + Q_t(\bar{w}_t))| \cdot \| \bar{z}_t - \hat{z}_t \|_2$$
$$\text{(By Lemma 4.4)}$$

$$\leq 2\hat{\sigma}_{2,d}^2 |D_{2,t}| |\hat{w}_t^2 - \bar{w}_t^2| \| \bar{z}_t - \hat{z}_t \|_2 + 4\hat{\sigma}_{4,d}^2 |D_{4,t}| |\hat{w}_t^4 - \bar{w}_t^4| \| \bar{z}_t - \hat{z}_t \|_2 \tag{E.91}$$
$$+ |Q_t(\bar{w}_t) - Q_t(\hat{w}_t)| \cdot \| \bar{z}_t - \hat{z}_t \|_2 \qquad \text{(By Lemma 4.4 and Def. of } P_t)$$

$$\leq 2\hat{\sigma}_{2,d}^2 |D_{2,t}| |\hat{w}_t^2 - \bar{w}_t^2| \| \bar{z}_t - \hat{z}_t \|_2 + 4\hat{\sigma}_{4,d}^2 |D_{4,t}| |\hat{w}_t^4 - \bar{w}_t^4| \| \bar{z}_t - \hat{z}_t \|_2 \tag{E.92}$$

$$+ O\Big( \frac{\hat{\sigma}_{2,d}^2 |D_{2,t}| + \hat{\sigma}_{4,d}^2 |D_{4,t}|}{d} \Big) |\bar{w}_t - \hat{w}_t| \| \bar{z}_t - \hat{z}_t \|_2$$
$$\text{(By Lemma I.16 and Def. of } Q_t \text{ from Lemma 4.4)}$$

$$\lesssim \hat{\sigma}_{2,d}^2 |D_{2,t}| (|\bar{w}_t| + |\hat{w}_t|) \| \hat{u}_t - \bar{u}_t \|_2^2 + \hat{\sigma}_{4,d}^2 |D_{4,t}| (|\bar{w}_t|^3 + |\hat{w}_t|^3) \| \hat{u}_t - \bar{u}_t \|_2^2 \tag{E.93}$$

$$+ \frac{\hat{\sigma}_{2,d}^2 |D_{2,t}| + \hat{\sigma}_{4,d}^2 |D_{4,t}|}{d} \| \hat{u}_t - \bar{u}_t \|_2^2$$
$$\text{(B.c. } |\bar{w}_t - \hat{w}_t| \leq \|\bar{u}_t - \hat{u}_t\|_2 \text{ and } |\bar{z}_t - \hat{z}_t\|_2 \leq \|\bar{u}_t - \hat{u}_t\|_2 )$$

as desired. $\qquad \square$

We can combine Lemma E.4 and Lemma E.6 and get the following bound on $A_t$ during Phase 1.

**Lemma E.7.** *Suppose we are in the setting of Theorem 3.4. Let $t \leq T_1$ (where $T_1$ is as defined in Definition 4.5). Then,*

$$A_t \leq 4\hat{\sigma}_{2,d}^2 |D_{2,t}| (\hat{w}_t - \bar{w}_t)^2 + O((\hat{\sigma}_{2,d}^2 |D_{2,t}| + \hat{\sigma}_{4,d}^2 |D_{4,t}|) w_{\max}) \| \hat{u}_t - \bar{u}_t \|_2^2 \tag{E.94}$$

$$+ O\Big( (\hat{\sigma}_{2,d}^2 |D_{2,t}| + \hat{\sigma}_{4,d}^2 |D_{4,t}|) \Delta_{\max} \Big) \| \hat{u}_t - \bar{u}_t \|_2^2 \tag{E.95}$$

$$+ O\Big( \frac{\hat{\sigma}_{2,d}^2 |D_{2,t}| + \hat{\sigma}_{4,d}^2 |D_{4,t}|}{d} \Big) \| \hat{u}_t - \bar{u}_t \|_2^2 . \tag{E.96}$$

*Proof of Lemma E.7.* This directly follows from Lemma E.4 and Lemma E.6. $\qquad \square$

Next, we give a potentially looser bound for $A_t$ which holds even after Phase 1. The running time after Phase 1 is at most $(\log \log d)^{O(1)}$ (aside from the dependency on $\hat{\sigma}_{2,d}, \hat{\sigma}_{4,d}$, etc.), meaning that even the following growth rate is enough to ensure that the coupling error grows by only a sub-polynomial factor after Phase 1.

**Lemma E.8.** *Suppose we are in the setting of Theorem 3.4. Then, for all $t \geq 0$, we have*

$$|A_t| \lesssim (\hat{\sigma}_{2,d}^2 + \hat{\sigma}_{4,d}^2) \gamma_2 \| \bar{u}_t - \hat{u}_t \|^2 . \tag{E.97}$$

*Proof of Lemma E.8.* We first consider the part of $A_t$ that originates from the non-projected gradients, which is defined as

$$A_t' := -2 \langle \nabla_{\bar{u}} L(\rho_t) - \nabla_{\hat{u}} L(\rho_t), \bar{u}_t - \hat{u}_t \rangle \tag{E.98}$$

$$= -2 \mathbb{E}_{x \sim \mathbb{S}^{d-1}} \big[ (f_\rho(x) - y(x))(\sigma'(\bar{u}_t^\top x) - \sigma'(\hat{u}_t^\top x))(\bar{u}_t^\top x - \hat{u}_t^\top x) \big]. \tag{E.99}$$

By the Cauchy-Schwarz inequality, we have

$$|A_t'| \lesssim \mathbb{E}_{x \sim \mathbb{S}^{d-1}}[(f_\rho(x) - y(x))^2]^{\frac{1}{2}} \cdot \mathbb{E}_{x \sim \mathbb{S}^{d-1}}[(\sigma'(\bar{u}_t^\top x) - \sigma'(\hat{u}_t^\top x))^2 (\bar{u}_t^\top x - \hat{u}_t^\top x)^2]^{\frac{1}{2}}. \quad \text{(E.100)}$$

We bound the first factor in the equation above by the loss at initialization, which is

$$\frac{1}{2}\mathbb{E}_{x \sim \mathbb{S}^{d-1}}[(f_{\rho_0}(x) - y)^2] \lesssim (\hat{\sigma}_{2,d}^2 + \hat{\sigma}_{4,d}^2)\gamma_2^2. \quad \text{(E.101)}$$

by Lemma D.8. Thus, plugging Eq. (E.101) and Eq. (E.210) (from the statement of Lemma E.15) into Eq. (E.100) gives us

$$|A_t'| \lesssim (\hat{\sigma}_{4,d}^2 + \hat{\sigma}_{2,d}^2)\gamma_2 \|\delta_t\|^2. \quad \text{(E.102)}$$

Next, we bound the difference between $A_t$ and $A_t'$. We have

$$A_t - A_t' = -\langle \text{grad}_{\hat{u}}L(\rho_t) - \text{grad}_{\bar{u}}L(\rho_t), \hat{u}_t - \bar{u}_t \rangle + \langle \nabla_{\hat{u}}L(\rho_t) - \nabla_{\bar{u}}L(\rho_t), \hat{u}_t - \bar{u}_t \rangle \quad \text{(E.103)}$$
$$= \langle \nabla_{\hat{u}}L(\rho_t) - \text{grad}_{\hat{u}}L(\rho_t), \hat{u}_t - \bar{u}_t \rangle - \langle \nabla_{\bar{u}}L(\rho_t) - \text{grad}_{\bar{u}}L(\rho_t), \hat{u}_t - \bar{u}_t \rangle \quad \text{(E.104)}$$

For the first term of Eq. (E.104), we have

$$\langle \nabla_{\hat{u}}L(\rho_t) - \text{grad}_{\hat{u}}L(\rho_t), \hat{u}_t - \bar{u}_t \rangle = \langle \hat{u}_t \hat{u}_t^\top \nabla_{\hat{u}}L(\rho_t), \hat{u}_t - \bar{u}_t \rangle \quad \text{(E.105)}$$
$$= \langle \nabla_{\hat{u}}L(\rho_t), (\hat{u}_t \hat{u}_t^\top)\hat{u}_t - (\hat{u}_t \hat{u}_t^\top)\bar{u}_t \rangle \quad \text{(E.106)}$$
$$= \langle \nabla_{\hat{u}}L(\rho_t), \hat{u}_t \rangle \cdot (1 - \langle \hat{u}_t, \bar{u}_t \rangle) \quad \text{(E.107)}$$
$$= \langle \nabla_{\hat{u}}L(\rho_t), \hat{u}_t \rangle \cdot \frac{\|\hat{u}_t - \bar{u}_t\|_2^2}{2} \quad \text{(B.c. } \|\hat{u}_t\|_2 = \|\bar{u}_t\|_2 = 1)$$
$$= \mathbb{E}_{x \sim \mathbb{S}^{d-1}}[(f_{\rho_t}(x) - y(x))\sigma'(\hat{u}_t^\top x)\hat{u}_t^\top x] \cdot \frac{\|\hat{u}_t - \bar{u}_t\|_2^2}{2} \quad \text{(E.108)}$$
$$= \mathbb{E}_{x \sim \mathbb{S}^{d-1}}[(f_{\rho_t}(x) - y(x))\sigma'(\hat{u}_t^\top x)\hat{u}_t^\top x] \cdot \frac{\|\delta_t\|_2^2}{2} \quad \text{(E.109)}$$

and by an argument similar to the one used to bound $A_t'$ (here using Eq. (E.211) from Lemma E.15 instead of Eq. (E.210)), we can thus show that

$$|\langle \nabla_{\hat{u}}L(\rho_t) - \text{grad}_{\hat{u}}L(\rho_t), \bar{u}_t - \hat{u}_t \rangle| \lesssim (\hat{\sigma}_{2,d}^2 + \hat{\sigma}_{4,d}^2)\gamma_2 \|\delta_t\|^2. \quad \text{(E.110)}$$

Similarly, for the second term we have

$$|\langle \nabla_{\bar{u}}L(\rho_t) - \text{grad}_{\bar{u}}L(\rho_t), \bar{u}_t - \hat{u}_t \rangle| \lesssim (\hat{\sigma}_{2,d}^2 + \hat{\sigma}_{4,d}^2)\gamma_2 \|\delta_t\|^2. \quad \text{(E.111)}$$

Combining Eq. (E.110), Eq. (E.111) and Eq. (E.102) finishes the proof. $\square$

## E.2   Upper Bound on $B_t$

The goal of this section is to obtain an upper bound on

$$B_t(\chi) := -2\langle \text{grad}_{\hat{u}}L(\hat{\rho}_t) - \text{grad}_{\hat{u}}L(\rho_t), \hat{u}_t(\chi) - \bar{u}_t(\chi) \rangle. \quad \text{(E.112)}$$

When there is no risk of confusion, we simply use $B_t$, $\hat{u}_t$, and $\bar{u}_t$, omitting the $\chi$ dependency. In the next lemma, we first obtain a bound on $B_t(\chi)$, and after that, we obtain a bound on $\mathbb{E}_\chi[B_t(\chi)]$. The proof of the following lemma is mostly a straightforward calculation, and follows mostly from Lemma 5.1 and the definition of $\overline{\Delta}$.

**Lemma E.9.** *Suppose we are in the setting of Theorem 3.4. Suppose $T \lesssim \frac{d^C}{\hat{\sigma}_{2,d}^2 + \hat{\sigma}_{4,d}^2}$ for any absolute constant $C$. Assume the width $m$ of $\bar{\rho}, \hat{\rho}$ is at most $d^{O(\log d)}$. Then, with probability at least $1 - e^{-d^2}$, we have*

$$\mathbb{E}_{x \sim \mathbb{S}^{d-1}}[(f_{\rho_t}(x) - f_{\hat{\rho}_t}(x))^2] \lesssim \frac{(\hat{\sigma}_{2,d}^2 + \hat{\sigma}_{4,d}^2)d^2(\log d)^{O(1)}}{m} \quad \text{(E.113)}$$

*for all times $t \in [0, T]$. Additionally, with probability at least $1 - e^{-d^2}$, for all $t \in [0, T]$ and $\chi \in \hat{\rho}_0$, we have*

$$B_t(\chi) \lesssim (\hat{\sigma}_{2,d}^2 + \hat{\sigma}_{4,d}^2) \cdot \left(\frac{d(\log d)^{O(1)}}{\sqrt{m}} + \overline{\Delta}_t\right)\|\delta_t\|_2. \quad \text{(E.114)}$$

*Proof of Lemma E.9.* We first decompose $B_t$ into two terms:

$$B_t = -2\langle \operatorname{grad}_{\hat{u}} L(\hat{\rho}_t) - \operatorname{grad}_{\hat{u}} L(\rho_t), \hat{u}_t - \bar{u}_t \rangle \tag{E.115}$$

$$= -2\langle (I - \hat{u}_t \hat{u}_t^\top) \nabla_{\hat{u}} L(\hat{\rho}_t) - (I - \hat{u}_t \hat{u}_t^\top) \nabla_{\hat{u}} L(\rho_t), \hat{u}_t - \bar{u}_t \rangle \tag{E.116}$$

$$= -2\langle \nabla_{\hat{u}} L(\hat{\rho}_t) - \nabla_{\hat{u}} L(\rho_t), (I - \hat{u}_t \hat{u}_t^\top)(\hat{u}_t - \bar{u}_t) \rangle \tag{E.117}$$

$$= -2\langle \nabla_{\hat{u}} L(\hat{\rho}_t) - \nabla_{\hat{u}} L(\rho_t), \hat{u}_t - \bar{u}_t \rangle + 2\langle \nabla_{\hat{u}} L(\hat{\rho}_t) - \nabla_{\hat{u}} L(\rho_t), \hat{u}_t \hat{u}_t^\top (\hat{u}_t - \bar{u}_t) \rangle \tag{E.118}$$

$$= -2\langle \nabla_{\hat{u}} L(\hat{\rho}_t) - \nabla_{\hat{u}} L(\rho_t), \hat{u}_t - \bar{u}_t \rangle + 2\langle \nabla_{\hat{u}} L(\hat{\rho}_t) - \nabla_{\hat{u}} L(\rho_t), \hat{u}_t \rangle \cdot (1 - \langle \hat{u}_t, \bar{u}_t \rangle) \tag{E.119}$$

$$= -2\langle \nabla_{\hat{u}} L(\hat{\rho}_t) - \nabla_{\hat{u}} L(\rho_t), \hat{u}_t - \bar{u}_t \rangle + 2\langle \nabla_{\hat{u}} L(\hat{\rho}_t) - \nabla_{\hat{u}} L(\rho_t), \hat{u}_t \rangle \cdot \frac{\|\hat{u}_t - \bar{u}_t\|_2^2}{2} \tag{E.120}$$

where the last equality is because $\frac{\|\hat{u}_t - \bar{u}_t\|_2^2}{2} = 1 - \langle \hat{u}_t, \bar{u}_t \rangle$ since $\|\hat{u}_t\|_2 = \|\bar{u}_t\|_2 = 1$. For convenience, define $\delta_t := \hat{u}_t - \bar{u}_t$. Then, we have

$$B_t = -2\langle \nabla_{\hat{u}} L(\hat{\rho}_t) - \nabla_{\hat{u}} L(\rho_t), \delta_t \rangle + \langle \nabla_{\hat{u}} L(\hat{\rho}_t) - \nabla_{\hat{u}} L(\rho_t), \hat{u}_t \rangle \cdot \|\delta_t\|_2^2 \tag{E.121}$$

We first focus on the first term in Eq. (E.121), which is the main contributor to $B_t$.

**Analysis of First Term in Eq. (E.121).** For convenience, we refer to the first term in Eq. (E.121) as $B_t'$. Observe that we can expand $B_t'$ as

$$B_t' = -2\langle \nabla_{\hat{u}} L(\hat{\rho}_t) - \nabla_{\hat{u}} L(\rho_t), \delta_t \rangle \tag{E.122}$$

$$= -2\mathop{\mathbb{E}}_{x \sim \mathbb{S}^{d-1}}[(f_{\hat{\rho}}(x) - f_\rho(x))\sigma'(\langle \hat{u}_t, x \rangle)\langle \delta_t, x \rangle] \qquad \text{(By Eq. (3.4))}$$

$$= -2\mathop{\mathbb{E}}_{x \sim \mathbb{S}^{d-1}}[(f_{\bar{\rho}}(x) - f_\rho(x))\sigma'(\langle \hat{u}_t, x \rangle)\langle \delta_t, x \rangle] - 2\mathop{\mathbb{E}}_{x \sim \mathbb{S}^{d-1}}[(f_{\hat{\rho}}(x) - f_{\bar{\rho}}(x))\sigma'(\langle \hat{u}_t, x \rangle)\langle \delta_t, x \rangle] \tag{E.123}$$

Let us bound the first term of Eq. (E.123). By Lemma 5.1, with probability $e^{-d^2}$, for all $t \le T$ we have

$$\mathop{\mathbb{E}}_{x \sim \mathbb{S}^{d-1}}[(f_{\rho_t}(x) - f_{\bar{\rho}_t}(x))^2] \lesssim \frac{(\hat{\sigma}_{2,d}^2 + \hat{\sigma}_{4,d}^2)d^2(\log d)^{O(1)}}{m}. \tag{E.124}$$

Thus, we can bound the first term of Eq. (E.123) by

$$\left| \mathop{\mathbb{E}}_{x \sim \mathbb{S}^{d-1}}[(f_{\bar{\rho}}(x) - f_\rho(x))\sigma'(\langle \hat{u}_t, x \rangle)\langle \delta_t, x \rangle] \right| \tag{E.125}$$

$$\le \mathop{\mathbb{E}}_{x \sim \mathbb{S}^{d-1}}[(f_{\bar{\rho}}(x) - f_\rho(x))^2]^{1/2} \mathop{\mathbb{E}}_{x \sim \mathbb{S}^{d-1}}[\sigma'(\langle \hat{u}_t, x \rangle)^2 \langle \delta_t, x \rangle^2]^{1/2}$$
$$\text{(By Cauchy-Schwarz inequality)}$$

$$\lesssim \mathop{\mathbb{E}}_{x \sim \mathbb{S}^{d-1}}[(f_{\bar{\rho}}(x) - f_\rho(x))^2]^{1/2}(\hat{\sigma}_{2,d}^2 + \hat{\sigma}_{4,d}^2)^{1/2}\|\delta_t\|_2$$
$$\text{(By Lemma E.15, Eq. (E.211))}$$

$$\lesssim \frac{\sqrt{\hat{\sigma}_{2,d}^2 + \hat{\sigma}_{4,d}^2} \cdot d(\log d)^{O(1)}}{\sqrt{m}} \cdot (\hat{\sigma}_{2,d}^2 + \hat{\sigma}_{4,d}^2)^{1/2}\|\delta_t\|_2 \qquad \text{(By Eq. (E.124))}$$

$$\lesssim \frac{(\hat{\sigma}_{2,d}^2 + \hat{\sigma}_{4,d}^2) \cdot d(\log d)^{O(1)}}{\sqrt{m}}\|\delta_t\|_2. \tag{E.126}$$

Next, we bound the second term in Eq. (E.123). By applying the Cauchy-Schwarz inequality, we have

$$\left| \mathop{\mathbb{E}}_{x \sim \mathbb{S}^{d-1}}[(f_{\hat{\rho}}(x) - f_{\bar{\rho}}(x))\sigma'(\langle \hat{u}, x \rangle)\langle \delta_t, x \rangle] \right| \tag{E.127}$$

$$\le \mathop{\mathbb{E}}_{x \sim \mathbb{S}^{d-1}}[(f_{\hat{\rho}}(x) - f_{\bar{\rho}}(x))^2]^{1/2} \mathop{\mathbb{E}}_{x \sim \mathbb{S}^{d-1}}[\sigma'(\langle \hat{u}, x \rangle)^2 \langle \delta_t, x \rangle^2]^{1/2} \tag{E.128}$$

$$\lesssim (\hat{\sigma}_{2,d}^2 + \hat{\sigma}_{4,d}^2)^{1/2}\overline{\Delta}_t \cdot \mathop{\mathbb{E}}_{x \sim \mathbb{S}^{d-1}}[\sigma'(\langle \hat{u}_t, x \rangle)^2 \langle \delta_t, x \rangle^2]^{1/2} \qquad \text{(By Lemma 5.1)}$$

$$\lesssim (\hat{\sigma}_{2,d}^2 + \hat{\sigma}_{4,d}^2)^{1/2}\overline{\Delta}_t \cdot (\hat{\sigma}_{2,d}^2 + \hat{\sigma}_{4,d}^2)^{1/2}\|\delta_t\|_2 \qquad \text{(By Lemma E.15 (Eq. (E.211)))}$$

$$\lesssim (\hat{\sigma}_{2,d}^2 + \hat{\sigma}_{4,d}^2)\overline{\Delta}_t\|\delta_t\|_2 \tag{E.129}$$

Combining Eq. (E.126) and Eq. (E.129) with Eq. (E.123), we find that

$$|B_t'| \lesssim (\hat{\sigma}_{2,d}^2 + \hat{\sigma}_{4,d}^2) \cdot \Big(\frac{d(\log d)^{O(1)}}{\sqrt{m}} + \overline{\Delta}_t\Big)\|\delta_t\|_2\,. \tag{E.130}$$

**Analysis of Second Term in Eq. (E.121).** For convenience, we write this term as $B_t''$, i.e. we define $B_t'' := \langle \nabla_{\hat{u}} L(\hat{\rho}_t) - \nabla_{\hat{u}} L(\rho_t), \hat{u}_t\rangle \cdot \|\delta_t\|_2^2$. It suffices to show that the first dot product in the definition of $B_t''$ is at most $(\hat{\sigma}_{2,d}^2 + \hat{\sigma}_{4,d}^2)$ up to constant factors. To start, we have

$$|\langle \nabla_{\hat{u}} L(\hat{\rho}_t) - \nabla_{\hat{u}} L(\rho_t), \hat{u}_t\rangle| \lesssim \Big| \operatorname*{\mathbb{E}}_{x \sim \mathbb{S}^{d-1}}[(f_{\hat{\rho}}(x) - f_\rho(x))\sigma'(\langle \hat{u}_t, x\rangle)\langle \hat{u}_t, x\rangle]\Big| \tag{E.131}$$

$$\lesssim \Big| \operatorname*{\mathbb{E}}_{x \sim \mathbb{S}^{d-1}}[(f_\rho(x) - f_{\bar{\rho}}(x))\sigma'(\langle \hat{u}_t, x\rangle)\langle \hat{u}_t, x\rangle]\Big| \tag{E.132}$$

$$+ \Big| \operatorname*{\mathbb{E}}_{x \sim \mathbb{S}^{d-1}}[(f_{\bar{\rho}}(x) - f_{\hat{\rho}}(x))\sigma'(\langle \hat{u}_t, x\rangle)\langle \hat{u}_t, x\rangle]\Big| \tag{E.133}$$

Both terms in Eq. (E.133) can be bounded as before. For the first term we use Eq. (E.124):

$$\Big| \operatorname*{\mathbb{E}}_{x \sim \mathbb{S}^{d-1}}[(f_\rho(x) - f_{\bar{\rho}}(x))\sigma'(\langle \hat{u}_t, x\rangle)\langle \hat{u}_t, x\rangle]\Big| \tag{E.134}$$

$$\leq \operatorname*{\mathbb{E}}_{x \sim \mathbb{S}^{d-1}}[(f_\rho(x) - f_{\bar{\rho}}(x))^2]^{1/2} \operatorname*{\mathbb{E}}_{x \sim \mathbb{S}^{d-1}}[\sigma'(\langle \hat{u}_t, x\rangle)^2 \langle \hat{u}_t, x\rangle^2]^{1/2}$$

$$\text{(By Cauchy-Schwarz Inequality)}$$

$$\lesssim (\hat{\sigma}_{2,d}^2 + \hat{\sigma}_{4,d}^2) \cdot \frac{d(\log d)^{O(1)}}{\sqrt{m}}\,. \quad \text{(By Eq. (E.124) and Lemma E.15, Eq. (E.211))}$$

For the second term, we use Lemma E.16:

$$\Big| \operatorname*{\mathbb{E}}_{x \sim \mathbb{S}^{d-1}}[(f_{\bar{\rho}}(x) - f_{\hat{\rho}}(x))\sigma'(\langle \hat{u}_t, x\rangle)\langle \hat{u}_t, x\rangle]\Big| \tag{E.135}$$

$$\lesssim \operatorname*{\mathbb{E}}_{x \sim \mathbb{S}^{d-1}}[(f_{\bar{\rho}}(x) - f_{\hat{\rho}}(x))^2]^{1/2} \operatorname*{\mathbb{E}}_{x \sim \mathbb{S}^{d-1}}[\sigma'(\langle \hat{u}_t, x\rangle)^2 \langle \hat{u}_t, x\rangle^2]^{1/2}$$

$$\text{(By Cauchy-Schwarz Inequality)}$$

$$\lesssim (\hat{\sigma}_{2,d}^2 + \hat{\sigma}_{4,d}^2)\overline{\Delta}_t \qquad \text{(By Lemma E.16 and Lemma E.15, Eq. (E.211))}$$

Thus, we can conclude that

$$|B_t''| \lesssim (\hat{\sigma}_{2,d}^2 + \hat{\sigma}_{4,d}^2) \cdot \Big(\frac{d(\log d)^{O(1)}}{\sqrt{m}} + \overline{\Delta}_t\Big) \cdot \|\delta_t\|_2^2 \tag{E.136}$$

**Overall Bound on $B_t$.** Combining our bounds on $B_t'$ and $B_t''$, and using the fact that $\|\delta_t\|_2 \leq 2$ to absorb the bound for $B_t''$ into the bound for $B_t'$, we find that

$$|B_t| \lesssim (\hat{\sigma}_{2,d}^2 + \hat{\sigma}_{4,d}^2) \cdot \Big(\frac{d(\log d)^{O(1)}}{\sqrt{m}} + \overline{\Delta}_t\Big)\|\delta_t\|_2 \tag{E.137}$$

with probability $1 - e^{-d^2}$ (recall that Eq. (E.124) held with probability $1 - e^{-d^2}$). This completes the proof. $\qquad \square$

Next, we show a tighter bound when we take the expectation of $B_t$ over the neurons, i.e. over $\chi$. Taking the expectation allows us to obtain a tighter bound because of the term

$$-2 \operatorname*{\mathbb{E}}_{x \sim \mathbb{S}^{d-1}}[(f_{\hat{\rho}}(x) - f_{\bar{\rho}}(x))\sigma'(\langle \hat{u}_t, x\rangle)\langle \delta_t, x\rangle]\,. \tag{E.138}$$

For a single $\hat{u}_t$, the best bound we can obtain is by using the Cauchy-Schwarz inequality. However, the key point in the following lemma is that when we take the expectation over $(\hat{u}_t, \bar{u}_t) \sim \Gamma_t$, this term becomes

$$-2 \operatorname*{\mathbb{E}}_{x \sim \mathbb{S}^{d-1}}[(f_{\hat{\rho}}(x) - f_{\bar{\rho}}(x))^2] \leq 0 \tag{E.139}$$

together with a third-order term which is at most $(\hat{\sigma}_{2,d}^2 + \hat{\sigma}_{4,d}^2)\overline{\Delta}_t^3$, which is a significantly better bound than $(\hat{\sigma}_{2,d}^2 + \hat{\sigma}_{4,d}^2)\overline{\Delta}_t\|\delta_t\|_2$, which we obtained in the previous lemma. This is because to simplify Eq. (E.138), we can use

$$\sigma(\langle\bar{u}_t, x\rangle) = \sigma(\langle\hat{u}_t, x\rangle) + \sigma'(\langle\hat{u}_t, x\rangle)\langle\bar{u}_t - \hat{u}_t, x\rangle + O(\sigma''(\langle\hat{u}_t, x\rangle)\langle\delta_t, x\rangle^2) \quad \text{(E.140)}$$

meaning that we can essentially substitute $\sigma'(\langle\hat{u}_t, x\rangle)\langle\delta_t, x\rangle \approx \sigma(\langle\hat{u}_t, x\rangle) - \sigma(\langle\bar{u}_t, x\rangle)$ in Eq. (E.138). Note that we need this better bound on the growth rate of $\overline{\Delta}_t$ since our inductive hypothesis imposes a stricter condition on $\overline{\Delta}_t$.

**Lemma E.10.** *Suppose we are in the setting of Lemma E.9, and in particular, suppose $T$ satisfies the same assumptions as in the statement of Lemma E.9. Then, for all $t \leq T$, we have*

$$\mathbb{E}_{\chi\sim\hat{\rho}_0}[B_t(\chi)] \lesssim \frac{(\hat{\sigma}_{2,d}^2 + \hat{\sigma}_{4,d}^2)d(\log d)^{O(1)}}{\sqrt{m}}\overline{\Delta}_t + (\hat{\sigma}_{2,d}^2 + \hat{\sigma}_{4,d}^2)\overline{\Delta}_t^3. \quad \text{(E.141)}$$

*Proof of Lemma E.10.* We again make use of Eq. (E.121), and we recall from the proof of the previous lemma that $B_t = B_t' + B_t''$, where

$$B_t' = -2\mathbb{E}_{x\sim\mathbb{S}^{d-1}}[(f_{\bar{\rho}}(x) - f_\rho(x))\sigma'(\langle\hat{u}_t, x\rangle)\langle\delta_t, x\rangle] - 2\mathbb{E}_{x\sim\mathbb{S}^{d-1}}[(f_{\hat{\rho}}(x) - f_{\bar{\rho}}(x))\sigma'(\langle\hat{u}_t, x\rangle)\langle\delta_t, x\rangle] \quad \text{(E.142)}$$

and

$$B_t'' = \langle\nabla_{\hat{u}}L(\hat{\rho}_t) - \nabla_{\hat{u}}L(\rho_t), \hat{u}_t\rangle \cdot \|\delta_t\|_2^2. \quad \text{(E.143)}$$

The main difference between Lemma E.9 is that we now bound the second term of $B_t'$ in expectation. By Taylor's theorem, we can write

$$\sigma(\langle\bar{u}_t, x\rangle) = \sigma(\langle\hat{u}_t, x\rangle) + \sigma'(\langle\hat{u}_t, x\rangle)(\langle\bar{u}_t, x\rangle - \langle\hat{u}_t, x\rangle) + \frac{\sigma''(\lambda)}{2}(\langle\bar{u}_t, x\rangle - \langle\hat{u}_t, x\rangle)^2 \quad \text{(E.144)}$$

for some $\lambda$ which is a convex combination of $\langle\hat{u}, x\rangle$ and $\langle\bar{u}, x\rangle$. Rearranging gives

$$\sigma'(\langle\hat{u}_t, x\rangle)\langle\delta_t, x\rangle = \sigma'(\langle\hat{u}_t, x\rangle)\langle\hat{u}_t - \bar{u}_t, x\rangle = \sigma(\langle\hat{u}_t, x\rangle) - \sigma(\langle\bar{u}_t, x\rangle) + \frac{\sigma''(\lambda)}{2}\langle\delta_t, x\rangle^2 \quad \text{(E.145)}$$

Thus, taking the expectation of the second term of $B_t'$ over $(\hat{u}_t, \bar{u}_t) \sim \Gamma_t$ gives

$$-2\mathbb{E}_{(\hat{u},\bar{u})\sim\Gamma_t}\mathbb{E}_{x\sim\mathbb{S}^{d-1}}[(f_{\hat{\rho}}(x) - f_{\bar{\rho}}(x))\sigma'(\langle\hat{u}_t, x\rangle)\langle\delta_t, x\rangle] \quad \text{(E.146)}$$

$$= -2\mathbb{E}_{(\hat{u},\bar{u})\sim\Gamma_t}\mathbb{E}_{x\sim\mathbb{S}^{d-1}}[(f_{\hat{\rho}}(x) - f_{\bar{\rho}}(x))(\sigma(\langle\hat{u}_t, x\rangle) - \sigma(\langle\bar{u}_t, x\rangle))] \quad \text{(E.147)}$$

$$- \mathbb{E}_{(\hat{u},\bar{u})\sim\Gamma_t}\mathbb{E}_{x\sim\mathbb{S}^{d-1}}[(f_{\hat{\rho}}(x) - f_{\bar{\rho}}(x))\sigma''(\lambda)\langle\delta_t, x\rangle^2] \quad \text{(By Eq. (E.145))}$$

$$= -2\mathbb{E}_{x\sim\mathbb{S}^{d-1}}[(f_{\hat{\rho}}(x) - f_{\bar{\rho}}(x))^2] \quad \text{(E.148)}$$

$$- \mathbb{E}_{(\hat{u},\bar{u})\sim\Gamma_t}\mathbb{E}_{x\sim\mathbb{S}^{d-1}}[(f_{\hat{\rho}}(x) - f_{\bar{\rho}}(x))\sigma''(\lambda)\langle\delta_t, x\rangle^2] \quad \text{(By Eq. (3.1))}$$

$$\leq -\mathbb{E}_{(\hat{u},\bar{u})\sim\Gamma_t}\mathbb{E}_{x\sim\mathbb{S}^{d-1}}[(f_{\hat{\rho}}(x) - f_{\bar{\rho}}(x))\sigma''(\lambda)\langle\delta_t, x\rangle^2] \quad \text{(E.149)}$$

Next, let us bound the inner expectation in Eq. (E.149). By the Cauchy-Schwarz inequality, we have

$$\left|\mathbb{E}_{x\sim\mathbb{S}^{d-1}}[(f_{\hat{\rho}}(x) - f_{\bar{\rho}}(x))\sigma''(\lambda)\langle\delta_t, x\rangle^2]\right| \quad \text{(E.150)}$$

$$\leq \mathbb{E}_{x\sim\mathbb{S}^{d-1}}[(f_{\hat{\rho}}(x) - f_{\bar{\rho}}(x))^2]^{1/2}\mathbb{E}_{x\sim\mathbb{S}^{d-1}}[\sigma''(\lambda)^2\langle\delta_t, x\rangle^4]^{1/2} \quad \text{(E.151)}$$

$$\lesssim (\hat{\sigma}_{2,d}^2 + \hat{\sigma}_{4,d}^2)^{1/2}\overline{\Delta}_t \cdot \mathbb{E}_{x\sim\mathbb{S}^{d-1}}[\sigma''(\lambda)^2\langle\delta_t, x\rangle^4]^{1/2} \quad \text{(By Lemma E.16)}$$

$$\lesssim (\hat{\sigma}_{2,d}^2 + \hat{\sigma}_{4,d}^2)^{1/2}\overline{\Delta}_t \cdot \left(\mathbb{E}_{x\sim\mathbb{S}^{d-1}}[\sigma''(\langle\hat{u}_t, x\rangle)^2\langle\delta_t, x\rangle^4]^{1/2} + \mathbb{E}_{x\sim\mathbb{S}^{d-1}}[\sigma''(\langle\bar{u}_t, x\rangle)^2\langle\delta_t, x\rangle^4]^{1/2}\right) \quad \text{(E.152)}$$

Here to obtain the last inequality, we observe that $\sigma''$ is convex because $P_{2,d}{}''$ is a constant function and $P_{4,d}{}''$ is a quadratic function by Eq. (C.4) — additionally, we used the inequality $(a+b)^2 \lesssim a^2 + b^2$ and the fact that $\lambda$ is a convex combination of $\langle \hat{u}_t, x \rangle$ and $\langle \bar{u}_t, x \rangle$.

By an argument similar to the proof of Lemma E.15, we can show that $\mathbb{E}_{x \sim \mathbb{S}^{d-1}}[\sigma''(\langle \hat{u}_t, x \rangle)^2 \langle \delta_t, x \rangle^4]^{1/2} \lesssim (\hat{\sigma}_{2,d}^2 + \hat{\sigma}_{4,d}^2)^{1/2} \|\delta_t\|_2^2$ and $\mathbb{E}_{x \sim \mathbb{S}^{d-1}}[\sigma''(\langle \bar{u}_t, x \rangle)^2 \langle \delta_t, x \rangle^4]^{1/2} \lesssim (\hat{\sigma}_{2,d}^2 + \hat{\sigma}_{4,d}^2)^{1/2} \|\delta_t\|_2^2$. Combining this with Eq. (E.149), we have that

$$\left| \mathbb{E}_{x \sim \mathbb{S}^{d-1}} [(f_{\hat{\rho}}(x) - f_{\bar{\rho}}(x)) \sigma'(\langle \hat{u}_t, x \rangle) \langle \delta_t, x \rangle] \right| \lesssim (\hat{\sigma}_{2,d}^2 + \hat{\sigma}_{4,d}^2) \overline{\Delta}_t \|\delta_t\|_2^2 \tag{E.153}$$

and thus,

$$-2 \mathbb{E}_{(\hat{u}, \bar{u}) \sim \Gamma_t} \mathbb{E}_{x \sim \mathbb{S}^{d-1}} [(f_{\hat{\rho}}(x) - f_{\bar{\rho}}(x)) \sigma'(\langle \hat{u}_t, x \rangle) \langle \delta_t, x \rangle] \lesssim (\hat{\sigma}_{2,d}^2 + \hat{\sigma}_{4,d}^2) \overline{\Delta}_t^3 \tag{E.154}$$

by the definition of $\overline{\Delta}_t^3$. This gives a bound on the second term of $B_t'$. Combining this with Eq. (E.126) to bound the first term of $B_t'$, we have

$$\mathbb{E}_{(\hat{u}, \bar{u}) \sim \Gamma_t} [B_t'] \lesssim \frac{(\hat{\sigma}_{2,d}^2 + \hat{\sigma}_{4,d}^2) d (\log d)^{O(1)}}{\sqrt{m}} \mathbb{E}_{(\hat{u}, \bar{u}) \sim \Gamma_t} \|\delta_t\|_2 + (\hat{\sigma}_{2,d}^2 + \hat{\sigma}_{4,d}^2) \overline{\Delta}_t^3 \tag{E.155}$$

$$\lesssim \frac{(\hat{\sigma}_{2,d}^2 + \hat{\sigma}_{4,d}^2) d (\log d)^{O(1)}}{\sqrt{m}} \overline{\Delta}_t + (\hat{\sigma}_{2,d}^2 + \hat{\sigma}_{4,d}^2) \overline{\Delta}_t^3 \qquad \text{(By Cauchy-Schwarz)}$$

Combining this with Eq. (E.136) (for which we also take the expectation over $(\hat{u}_t, \bar{u}_t) \sim \Gamma_t$), we have that

$$\mathbb{E}_{(\hat{u}, \bar{u}) \sim \Gamma_t} |B_t''| \lesssim (\hat{\sigma}_{2,d}^2 + \hat{\sigma}_{4,d}^2) \cdot \left( \frac{d (\log d)^{O(1)}}{\sqrt{m}} + \overline{\Delta}_t \right) \cdot \overline{\Delta}_t^2 \tag{E.156}$$

which is at most the bound that we have for $B_t'$, and thus,

$$\mathbb{E}_{(\hat{u}, \bar{u}) \sim \Gamma_t} [B_t] \lesssim \frac{(\hat{\sigma}_{2,d}^2 + \hat{\sigma}_{4,d}^2) d (\log d)^{O(1)}}{\sqrt{m}} \overline{\Delta}_t + (\hat{\sigma}_{2,d}^2 + \hat{\sigma}_{4,d}^2) \overline{\Delta}_t^3 \tag{E.157}$$

as desired. $\qquad \square$

### E.3 Upper Bound on $C_t$

The goal of this section is to obtain an upper bound on

$$C_t(\chi) := -2 \langle \operatorname{grad}_{\hat{u}} \widehat{L}(\hat{\rho}_t) - \operatorname{grad}_{\hat{u}} L(\hat{\rho}_t), \hat{u}_t(\chi) - \bar{u}_t(\chi) \rangle. \tag{E.158}$$

This quantity is essentially the part of the growth of $\|\delta_t\|_2^2$ which is due to the difference between the gradient using finite samples and the gradient using infinitely many samples. When there is no risk of confusion, we simply use $C_t$, $\hat{u}_t$, and $\bar{u}_t$, omitting the $\chi$ dependency. We bound this term with the following lemma:

**Lemma E.11.** *Suppose we are in the setting of Theorem 3.4. Assume that the number of samples $x_i$ is $n \leq d^C$, for any universal constant $C > 0$. Assume that the width $m \leq d^C$ for any universal constant $C > 0$. Let $T > 0$. Assume for all $t \in [0, T]$ we have a bound $B_1$ on $\overline{\Delta}_t$ with $B_1 \leq \frac{1}{\sqrt{d}}$. Additionally, assume for all $t \in [0, T]$ that we have a bound $B_2$ on $\Delta_{\max,t}$ such that $B_2 \geq \frac{1}{\sqrt{d}}$. Then, with probability $1 - \frac{1}{d^{\Omega(\log d)}}$, for all $t \in [0, T]$ and $\chi \in \mathbb{S}^{d-1}$, we have*

$$|C_t(\chi)| \lesssim \hat{\sigma}_{\max}^2 (\log d)^{O(1)} \cdot \left( \sqrt{\frac{d}{n}} \|\delta_t\|_2 + \frac{d^2}{n} B_2^2 \overline{\Delta} \|\delta_t\|_2^2 + \frac{d^{2.5}}{n} \overline{\Delta}^2 B_2^2 \|\delta_t\|_2 + \frac{d^4}{n} B_2^4 \overline{\Delta}^2 \|\delta_t\|_2^2 \right). \tag{E.159}$$

*where $\delta_t = \hat{u}_t(\chi) - \bar{u}_t(\chi)$.*

*Proof.* We bound $C_t$ by expanding $\operatorname{grad}_{\hat{u}} L(\hat{\rho}_t)$ as the sum of several monomials, as well as expanding $\operatorname{grad}_{\hat{u}} \widehat{L}(\hat{\rho}_t)$, and then applying Lemma I.7 to each term from $\operatorname{grad}_{\hat{u}} L(\hat{\rho}_t)$ and the corresponding term

from $\text{grad}_{\hat{u}} \widehat{L}(\hat{\rho}_t)$. For convenience, we use $\ell^{(x)}(\hat{\rho})$ to denote the loss of $f_{\hat{\rho}}$ on a single data point $x \in \mathbb{S}^{d-1}$, and $\text{grad}_{\hat{u}} \ell^{(x)}(\hat{\rho})$ to denote the projected gradient of $\ell^{(x)}(\hat{\rho})$ with respect to $\hat{u}$. We have

$$\langle \text{grad}_{\hat{u}} \ell^{(x)}(\hat{\rho}_t), \hat{u}_t - \bar{u}_t \rangle = \langle (f_{\hat{\rho}}(x) - y(x)) \sigma'(\langle \hat{u}_t, x \rangle) x, (I - \hat{u}_t \hat{u}_t^\top)(\hat{u}_t - \bar{u}_t) \rangle \tag{E.160}$$

$$= \langle (f_{\hat{\rho}}(x) - y(x)) \sigma'(\langle \hat{u}_t, x \rangle) x, \hat{u}_t - \bar{u}_t \rangle \tag{E.161}$$

$$- \langle (f_{\hat{\rho}}(x) - y(x)) \sigma'(\langle \hat{u}_t, x \rangle) x, \hat{u}_t \rangle \cdot (1 - \langle \hat{u}_t, \bar{u}_t \rangle) \tag{E.162}$$

$$= \langle (f_{\hat{\rho}}(x) - y(x)) \sigma'(\langle \hat{u}_t, x \rangle) x, \hat{u}_t - \bar{u}_t \rangle \tag{E.163}$$

$$- \langle (f_{\hat{\rho}}(x) - y(x)) \sigma'(\langle \hat{u}_t, x \rangle) x, \hat{u}_t \rangle \cdot \frac{\|\hat{u}_t - \bar{u}_t\|_2^2}{2} \tag{E.164}$$

For convenience, define $\delta_t := \hat{u}_t - \bar{u}_t$. Then we can rewrite the above as

$$\langle \text{grad}_{\hat{u}} \ell^{(x)}(\hat{\rho}_t), \delta_t \rangle = \langle (f_{\hat{\rho}}(x) - y(x)) \sigma'(\langle \hat{u}_t, x \rangle) \langle x, \delta_t \rangle \tag{E.165}$$

$$- \langle (f_{\hat{\rho}}(x) - y(x)) \sigma'(\langle \hat{u}_t, x \rangle) x, \hat{u}_t \rangle \cdot \frac{\|\delta_t\|_2^2}{2} \tag{E.166}$$

For illustration, define $\mathcal{C}(g)$ as

$$\mathcal{C}(g) := \left| \frac{1}{n} \sum_{i=1}^n g(x_i) - \mathbb{E}_{x \sim \mathbb{S}^{d-1}}[g(x)] \right| \tag{E.167}$$

i.e. how close the empirical mean of the $g(x_i)$ is to its expected value. We will show that $\mathcal{C}(g)$ is small when $g$ corresponds to each of the terms on the right-hand side of Eq. (E.165), even when we take the supremum over the $\hat{u}_t$, which may depend on the $x_i$.

**Concentration for Mean of First Term in Eq. (E.165).** As a first step, we list all of the terms we obtain when expanding the first term on the right-hand side of Eq. (E.165). The $\sigma'(\langle \hat{u}_t, x \rangle)$ factor consists of the following terms: (1) a $\hat{\sigma}_{2,d} \sqrt{N_{2,d}} \langle \hat{u}_t, x \rangle$ term, (2) a $\hat{\sigma}_{4,d} \sqrt{N_{4,d}} \langle \hat{u}_t, x \rangle^3$ term, and (3) a $\frac{\hat{\sigma}_{4,d} \sqrt{N_{4,d}}}{d} \langle \hat{u}_t, x \rangle$ term — here, we have stated these terms up to absolute constant factors for convenience. In the $f_{\hat{\rho}}(x) - y(x)$ factor, $f_{\hat{\rho}}(x)$ contributes (1) a $\hat{\sigma}_{2,d} \sqrt{N_{2,d}} \langle \hat{u}_t', x \rangle^2$ term, (2) a $\hat{\sigma}_{4,d} \sqrt{N_{4,d}} \langle \hat{u}_t', x \rangle^4$ term, and (3) a $\frac{\hat{\sigma}_{4,d} \sqrt{N_{4,d}}}{d} \langle \hat{u}_t', x \rangle^2$ term. Here note that $\hat{u}_t'$ is not a single vector, but ranges over the entire support of $\hat{\rho}_t$. Thus we implicitly perform a union bound over the $m$ vectors in the support of $\hat{\rho}_t$ — this does not affect the proof since the failure probability of Lemma I.7 is $\frac{1}{d^{\Omega(\log d)}}$, while we assume the width $m$ is at most $d^C$ where $C$ can be any absolute constant. Additionally, $y(x)$ consists of the following terms: (1) a $\hat{\sigma}_{2,d} \gamma_2 \sqrt{N_{2,d}} \langle e_1, x \rangle^2$ term, (2) a $\hat{\sigma}_{4,d} \gamma_4 \sqrt{N_{4,d}} \langle e_1, x \rangle^4$ term, and (3) a $\frac{\hat{\sigma}_{4,d} \gamma_4 \sqrt{N_{4,d}}}{d} \langle e_1, x \rangle^2$ term. Finally, there is a factor of $\langle \delta_t, x \rangle$.

For convenience, throughout this proof, let $\bar{\delta}_t = \frac{\delta_t}{\|\delta_t\|_2}$. In each of the above factors, for every occurrence of $\langle \hat{u}_t, x \rangle$, we substitute $\hat{u}_t = \bar{u}_t + \delta_t$ and further expand. This is useful since originally $\hat{u}_t$ depends on the $x_i$ and thus, when we applied Lemma I.7 we would have to take the supremum over $\hat{u}_t$ — however, now $\bar{u}_t$ can be considered a fixed vector with respect to the $x_i$, and while we need to take the supremum over $\delta_t$, in each term where $\delta_t$ occurs, we obtain an additional factor of $\|\delta_t\|_2$, which reduces the error from uniform convergence. We now completely expand the first term on the right-hand side of Eq. (E.165). First we consider the contribution to $\mathcal{C}(g)$ of $f_{\hat{\rho}}(x)$ and later study $y(x)$. Up to constant factors, it suffices to obtain an upper bound on the concentration of

$$\sum_{\substack{p+r=1,3 \\ q+s=2,4}} \hat{\sigma}_{\max}^2 d^{\frac{p+1+q+r+s}{2}} \langle \delta_t, x \rangle^{p+1} \langle \delta_t', x \rangle^q \langle \bar{u}_t, x \rangle^r \langle \bar{u}_t', x \rangle^s \tag{E.168}$$

where we have let $\hat{\sigma}_{\max} = \max(|\hat{\sigma}_{2,d}|, |\hat{\sigma}_{4,d}|)$. Here, the sum ranges over $p+q = 1, 3$ and $q+s = 2, 4$ since the $\delta_t$ and $\bar{u}_t$ terms are due to the $\sigma'(\langle \hat{u}_t, x \rangle)$ factor, while the $\delta_t'$ and $\bar{u}_t'$ terms are due to the $f_{\hat{\rho}}(x)$ factors.

We now obtain concentration bounds for each of the terms in Eq. (E.168). First, we consider the case where $p + q \geq 1$, as in this case, Lemma I.7 can be applied (since we can choose $p_1, p_2, p_3, p_4$

in Lemma I.7 so that $u_1 = \bar{\delta}_t$ and $p_1 = p + 1$, and $u_2 = \bar{\delta}'_t$ and $p_2 = q$, and $p_3 = p_4 = 0$ — the hypotheses are then satisfied since $p_1 + p_2 + p_3 + p_4 \geq 2$). From Lemma I.7, we obtain the bound

$$\hat{\sigma}_{\max}^2 d^{\frac{p+q+r+s+1}{2}} \left| \frac{1}{n} \sum_{i=1}^n \langle \delta_t, x \rangle^{p+1} \langle \delta'_t, x \rangle^q \langle \bar{u}_t, x \rangle^r \langle \bar{u}'_t, x \rangle^s - \mathop{\mathbb{E}}_{x \sim \mathbb{S}^{d-1}} [\langle \delta_t, x \rangle^{p+1} \langle \delta'_t, x \rangle^q \langle \bar{u}_t, x \rangle^r \langle \bar{u}'_t, x \rangle^s] \right| \tag{E.169}$$

$$\lesssim \hat{\sigma}_{\max}^2 (\log d)^{O(1)} \left( \sqrt{\frac{d}{n}} + \frac{d^{\frac{p+1+q}{2}}}{n} \right) \|\delta_t\|_2^{p+1} \|\delta'_t\|_2^q \tag{E.170}$$

Here in applying Lemma I.7, we have assumed that the number of samples $n$ is at most $d^C$ for some absolute constant $C$, meaning that the $\frac{1}{d^{\Omega(\log d)}}$ term in the conclusion is a lower-order term. Lastly, we consider the case where $p + q = 0$. To apply Lemma I.7, we can let $u_1 = \bar{u}_t$, $p_1 = 1$, $u_2 = \frac{\delta_t}{\|\delta_t\|_2}$, $p_2 = 1$, and let $p_3 = p_4 = 0$, with $w_1 = \bar{u}_t$, $q_1 = r - 1$ and $w_2 = \bar{u}'_t$, $q_2 = s$ — since $p_1 + p_2 = 2$, the hypotheses of Lemma I.7 are still satisfied. This leads to us obtaining only one power of $\|\delta_t\|_2$ in the bound for this term: if $p = q = 0$, then we have the bound

$$\hat{\sigma}_{\max}^2 d^{\frac{p+q+r+s+1}{2}} \left| \frac{1}{n} \sum_{i=1}^n \langle \delta_t, x \rangle^{p+1} \langle \delta'_t, x \rangle^q \langle \bar{u}_t, x \rangle^r \langle \bar{u}'_t, x \rangle^s - \mathop{\mathbb{E}}_{x \sim \mathbb{S}^{d-1}} [\langle \delta_t, x \rangle^{p+1} \langle \delta'_t, x \rangle^q \langle \bar{u}_t, x \rangle^r \langle \bar{u}'_t, x \rangle^s] \right| \tag{E.171}$$

$$\lesssim \hat{\sigma}_{\max}^2 (\log d)^{O(1)} \left( \sqrt{\frac{d}{n}} + \frac{d}{n} \right) \|\delta_t\|_2 \tag{E.172}$$

Summing over the possible values of $p, q, r, s$, we find that the overall contribution to the concentration error of the term with $f_{\hat{\rho}}$ is, up to logarithmic factors and a $\hat{\sigma}_{\max}^2$ factor,

$$\mathop{\mathbb{E}}_{\delta' \sim \hat{\rho}} \sum_{p,q,r,s} \left( \sqrt{\frac{d}{n}} \|\delta_t\|_2^{p+1} \|\delta'_t\|_2^q + \frac{d^{\frac{p+1+q}{2}}}{n} \|\delta_t\|_2^{p+1} \|\delta'_t\|_2^q \right) \tag{E.173}$$

$$\lesssim \mathop{\mathbb{E}}_{\delta' \sim \hat{\rho}} \sum_{p \leq 3, q \leq 4} \left( \sqrt{\frac{d}{n}} \|\delta_t\|_2^{p+1} \|\delta'_t\|_2^q + \frac{d^{\frac{p+1+q}{2}}}{n} \|\delta_t\|_2^{p+1} \|\delta'_t\|_2^q \right) \tag{E.174}$$

$$\lesssim \sqrt{\frac{d}{n}} \|\delta_t\|_2 + \mathop{\mathbb{E}}_{\delta' \sim \hat{\rho}} \sum_{p \leq 3, q \leq 4} \frac{d^{\frac{p+q+1}{2}}}{n} \|\delta_t\|_2^{p+1} \|\delta'_t\|_2^q \tag{E.175}$$

$$\lesssim \sqrt{\frac{d}{n}} \|\delta_t\|_2 + \mathop{\mathbb{E}}_{\delta' \sim \hat{\rho}} \sum_{p \leq 3, q \leq 4} \frac{d^{\frac{p+q+1}{2}}}{n} \|\delta_t\|_2^{p+1} \|\delta'_t\|_2^q \tag{E.176}$$

$$\lesssim \sqrt{\frac{d}{n}} \|\delta_t\|_2 + \sum_{p \leq 3, q \leq 4} \frac{d^{\frac{p+q+1}{2}}}{n} \|\delta_t\|_2^{p+1} \mathop{\mathbb{E}}_{\delta' \sim \hat{\rho}} \|\delta'_t\|_2^q \tag{E.177}$$

Finally, in order to simplify this bound, we isolate the dominant terms among the terms of the form $\frac{d^{\frac{p+q+1}{2}}}{n} \|\delta_t\|_2^{p+1} \mathbb{E}_{\delta' \sim \hat{\rho}} \|\delta'_t\|_2^q$, with the following cases. First, consider all the terms where $p = 0$. For $q \leq 2$ we can bound these terms by $\frac{d^{\frac{p+q+1}{2}}}{n} \overline{\Delta}^q \|\delta_t\|_2$, and for $q > 2$ we can bound these terms by $\frac{d^{\frac{p+q+1}{2}}}{n} \cdot \overline{\Delta}^2 \cdot B_2^{q-2} \|\delta_t\|_2$. Thus, out of these terms, the term with $q = 4$ is dominant, and it contributes

$$\frac{d^{2.5}}{n} \cdot \overline{\Delta}^2 \cdot B_2^2 \|\delta_t\|_2 \tag{E.178}$$

Next, consider the case where $p \geq 1$. In this case, since the bound $B_2$ on $\Delta_{\max,t}$ is greater than $\frac{1}{d^{1/2}}$, the term with $p = 3$ and $q = 4$ dominates, and it contributes

$$\frac{d^4}{n} \cdot B_2^2 \overline{\Delta}^2 \cdot B_2^2 \|\delta_t\|_2^2 = \frac{d^4}{n} B_2^4 \overline{\Delta}^2 \|\delta_t\|_2^2 \tag{E.179}$$

However, we must lastly consider the case where $p \geq 1$ and $q < 2$ — the bound on the concentration error may be larger for $q = 1$ than $q = 2$ since $\overline{\Delta} < \frac{1}{\sqrt{d}}$. In this case, since $B_2 > \frac{1}{\sqrt{d}}$, we must

consider the $p = 3, q = 1$ and $p = 3, q = 0$ terms, which respectively contribute

$$\frac{d^{2.5}}{n}\overline{\Delta}B_2^2\|\delta_t\|_2^2 \tag{E.180}$$

and

$$\frac{d^2}{n}B_2^2\|\delta_t\|_2^2 \tag{E.181}$$

Note that the $p = 3, q = 0$ term dominates since our bound on $\overline{\Delta}$ is less than $\frac{1}{\sqrt{d}}$, meaning we can ignore the $p = 3, q = 1$ term. Thus our overall bound on the concentration error originating from the first term in Eq. (E.165) is

$$\hat{\sigma}_{\max}^2(\log d)^{O(1)} \cdot \left(\sqrt{\frac{d}{n}}\|\delta_t\|_2 + \frac{d^2}{n}B_2^2\overline{\Delta}\|\delta_t\|_2^2 + \frac{d^{2.5}}{n}\overline{\Delta}^2 B_2^2\|\delta_t\|_2 + \frac{d^4}{n}B_2^4\overline{\Delta}^2\|\delta_t\|_2^2\right) \tag{E.182}$$

To deal with the terms originating from $y(x)$, we observe that we can simply treat $\langle e_1, x\rangle$ in the same way as we deal with $\langle \bar{u}_t, x\rangle$, and thus the terms originating from $y(x)$ do not make any new contributions to the concentration error.

**Concentration for Mean of Second Term in Eq. (E.165).** Finally, to deal with the second term on the right-hand side of Eq. (E.165), we note that we can follow the same proof, but instead obtaining an upper bound on the concentration of

$$\|\delta_t\|_2^2 \sum_{\substack{p+r=2,4 \\ q+s=2,4}} \hat{\sigma}_{\max}^2 d^{\frac{p+q+r+s}{2}} \langle \delta_t, x\rangle^p \langle \delta_t', x\rangle^q \langle \bar{u}_t, x\rangle^r \langle \bar{u}_t', x\rangle^s \tag{E.183}$$

and as before, the contribution of the terms originating from $y(x)$ is at most the contribution of the terms in Eq. (E.183). Due to the additional $\|\delta_t\|_2^2$ factor, we can in fact ignore the contribution of the terms in Eq. (E.183), as the dominant terms will be less than the dominant terms from Eq. (E.168).

**Concluding to Bound $C_t$.** In summary, with probability at least $1 - \frac{1}{d^{\Omega(\log d)}}$, for all choices of $\delta_t, \delta_t'$, we have

$$|\mathcal{C}(g)| \lesssim \hat{\sigma}_{\max}^2(\log d)^{O(1)} \cdot \left(\sqrt{\frac{d}{n}}\|\delta_t\|_2 + \frac{d^2}{n}B_2^2\overline{\Delta}\|\delta_t\|_2^2 + \frac{d^{2.5}}{n}\overline{\Delta}^2 B_2^2\|\delta_t\|_2 + \frac{d^4}{n}B_2^4\overline{\Delta}^2\|\delta_t\|_2^2\right) \tag{E.184}$$

where $g(x) = \langle \mathrm{grad}_{\hat{u}}\ell^{(x)}(\hat{\rho}_t), \delta_t\rangle$. Since $C_t$ is exactly equal to

$$\frac{1}{n}\sum_{i=1}^n g(x_i) - \mathbb{E}_{x\sim\mathbb{S}^{d-1}}[g(x)] \tag{E.185}$$

up to signs and constant factors, we therefore have

$$|C_t| \lesssim \hat{\sigma}_{\max}^2(\log d)^{O(1)} \cdot \left(\sqrt{\frac{d}{n}}\|\delta_t\|_2 + \frac{d^2}{n}B_2^2\overline{\Delta}\|\delta_t\|_2^2 + \frac{d^{2.5}}{n}\overline{\Delta}^2 B_2^2\|\delta_t\|_2 + \frac{d^4}{n}B_2^4\overline{\Delta}^2\|\delta_t\|_2^2\right). \tag{E.186}$$

$\square$

## E.4 Additional Lemmas

**Lemma E.12** (Uniform Bound on $\sigma$ and $\sigma'$). *For any $t \in [-1, 1]$, we have $|\sigma(t)| \le |\hat{\sigma}_{2,d}|\sqrt{N_{2,d}} + |\hat{\sigma}_{4,d}|\sqrt{N_{4,d}}$, and additionally, $|\sigma'(t)| \lesssim |\hat{\sigma}_{2,d}|\sqrt{N_{2,d}} + |\hat{\sigma}_{4,d}|\sqrt{N_{4,d}}$. Finally, we have $|\sigma''(t)| \lesssim |\hat{\sigma}_{2,d}|\sqrt{N_{2,d}} + |\hat{\sigma}_{4,d}|\sqrt{N_{4,d}}$.*

*Proof of Lemma E.12.* The first statement is by the definition of $\sigma$, and because $|P_{2,d}(t)|, |P_{4,d}(t)| \le 1$ for any $t \in [-1, 1]$, by Atkinson and Han [12], Eq. (2.116). The second and third statements are by the definition of $\sigma$ and using Eq. (C.4). $\square$

**Lemma E.13** (Error from Finite Width). *Let $\rho$ be a distribution on $\mathbb{S}^{d-1}$, and let $\bar{\rho}$ be a distribution on $\mathbb{S}^{d-1}$ of support at most $m$, where the elements of the support of $\bar{\rho}$ are sampled i.i.d. from $\rho$. Additionally, let $\delta > 0$. Then, for some absolute constant $C > 0$,*

$$\mathbb{E}_{x\sim\mathbb{S}^{d-1}}[(f_\rho(x) - f_{\bar{\rho}}(x))^2] \lesssim \frac{(\hat{\sigma}_{2,d}^2 + \hat{\sigma}_{4,d}^2)(d^2 + \log\frac{1}{\delta})(\log d)^C}{m} + \frac{\hat{\sigma}_{2,d}^2 + \hat{\sigma}_{4,d}^2}{d^{\Omega(\log d)}} \tag{E.187}$$

*with probability at least $1 - \delta$, where $f_\rho$ and $f_{\bar{\rho}}$ are defined as in Eq. (3.1).*

*Proof of Lemma E.13.* We can write $f_\rho(x) = \mathbb{E}_{u\sim\rho}[\sigma(\langle u, x\rangle)]$, and $f_{\bar{\rho}}(x) = \frac{1}{m}\sum_{i=1}^m \sigma(\langle u_i, x\rangle)$. Now, let $B \in (0,1)$, which we will choose appropriately later, and define the function $\varphi_B : [-1, 1] \to [-1, 1]$ by

$$\varphi_B(t) = \begin{cases} t & t \in [-B, B] \\ B & t > B \\ -B & t < -B \end{cases} \tag{E.188}$$

Additionally, define $\tilde{f}_\rho(x) = \mathbb{E}_{u\sim\rho}[\sigma(\varphi_B(\langle u, x\rangle))]$, and define $\tilde{f}_{\bar{\rho}}(x) = \frac{1}{m}\sum_{i=1}^m \sigma(\varphi_B(\langle u_i, x\rangle))$. Let us analyze the error incurred from replacing $f_\rho$ be $\tilde{f}_\rho$ and $f_{\bar{\rho}}$ by $\tilde{f}_{\bar{\rho}}$. Observe that

$$\mathbb{E}_{x\sim\mathbb{S}^{d-1}}[(f_\rho(x) - \tilde{f}_\rho(x))^2] = \mathbb{E}_{x\sim\mathbb{S}^{d-1}}(f_\rho(x))^2 - 2\mathbb{E}_{x\sim\mathbb{S}^{d-1}}f_\rho(x)\tilde{f}_\rho(x) + \mathbb{E}_{x\sim\mathbb{S}^{d-1}}(\tilde{f}_\rho(x))^2 \tag{E.189}$$

We can simplify the second term on the right-hand side by expanding $f_\rho$ and $\tilde{f}_\rho$:

$$\mathbb{E}_{x\sim\mathbb{S}^{d-1}} f_\rho(x)\tilde{f}_\rho(x) = \mathbb{E}_{x\sim\mathbb{S}^{d-1}} \mathbb{E}_{u_1,u_2\sim\rho} \sigma(\langle u_1, x\rangle)\sigma(\varphi_B(\langle u_2, x\rangle)) \tag{E.190}$$

$$= \mathbb{E}_{u_1,u_2\sim\rho} \mathbb{E}_{x\sim\mathbb{S}^{d-1}} \sigma(\langle u_1, x\rangle)\sigma(\varphi_B(\langle u_2, x\rangle)) \tag{E.191}$$

To simplify the inner expectation, we use Proposition I.5 to find that $\varphi_B(\langle u_2, x\rangle) = \langle u_2, x\rangle$ with probability at least $1 - 2\exp(-\frac{B^2 d}{2})$. Thus,

$$\left| \mathbb{E}_{x\sim\mathbb{S}^{d-1}}(f_\rho(x))^2 - \mathbb{E}_{x\sim\mathbb{S}^{d-1}} f_\rho(x)\tilde{f}_\rho(x) \right| \tag{E.192}$$

$$\leq \mathbb{E}_{u_1,u_2\sim\rho} \mathbb{E}_{x\sim\mathbb{S}^{d-1}} |\sigma(\langle u_1, x\rangle)| \Big|\sigma(\varphi_B(\langle u_2, x\rangle)) - \sigma(\langle u_2, x\rangle)\Big| \tag{E.193}$$

$$\leq (|\hat{\sigma}_{2,d}|\sqrt{N_{2,d}} + |\hat{\sigma}_{4,d}|\sqrt{N_{4,d}}) \mathbb{E}_{u_2\sim\rho} \mathbb{E}_{x\sim\mathbb{S}^{d-1}} \Big|\sigma(\varphi_B(\langle u_2, x\rangle)) - \sigma(\langle u_2, x\rangle)\Big|$$
$$\text{(By Lemma E.12)}$$

$$\lesssim \exp\left(-\frac{B^2 d}{2}\right) \cdot (\hat{\sigma}_{2,d}^2 N_{2,d} + \hat{\sigma}_{4,d}^2 N_{4,d}) \quad \text{(By Proposition I.5 and Lemma E.12)}$$

Similarly, we can show that

$$\left| \mathbb{E}_{x\sim\mathbb{S}^{d-1}}(f_\rho(x))^2 - \mathbb{E}_{x\sim\mathbb{S}^{d-1}}(\tilde{f}_\rho(x))^2 \right| \lesssim \exp\left(-\frac{B^2 d}{2}\right) \cdot (\hat{\sigma}_{2,d}^2 N_{2,d} + \hat{\sigma}_{4,d}^2 N_{4,d}) \tag{E.194}$$

and therefore,

$$\mathbb{E}_{x\sim\mathbb{S}^{d-1}}[(f_\rho(x) - \tilde{f}_\rho(x))^2] \lesssim \exp\left(-\frac{B^2 d}{2}\right) \cdot (\hat{\sigma}_{2,d}^2 N_{2,d} + \hat{\sigma}_{4,d}^2 N_{4,d}) \tag{E.195}$$

By the same argument, we can show that

$$\mathbb{E}_{x\sim\mathbb{S}^{d-1}}[(f_{\bar{\rho}}(x) - \tilde{f}_{\bar{\rho}}(x))^2] \lesssim \exp\left(-\frac{B^2 d}{2}\right) \cdot (\hat{\sigma}_{2,d}^2 N_{2,d} + \hat{\sigma}_{4,d}^2 N_{4,d}) \tag{E.196}$$

Therefore, it suffices to show that $\mathbb{E}_{x\sim\mathbb{S}^{d-1}}[(\tilde{f}_\rho(x) - \tilde{f}_{\bar{\rho}}(x))^2]$ is small with high probability over $u_1, \ldots, u_m$. Observe that for $|t| \leq B$, we have $|P_{2,d}(t)| \leq B^2 + O(1/d)$, and $|P_{4,d}(t)| \leq B^4 + O(B^2/d) + O(1/d^2)$. Thus, if we choose $B \geq 1/\sqrt{d}$, then for $|t| \leq B$ we have $|P_{2,d}(t)| \lesssim B^2$ and $|P_{4,d}(t)| \lesssim B^4$. In particular, we have

$$|\sigma(t)| \lesssim |\hat{\sigma}_{2,d}|\sqrt{N_{2,d}}B^2 + |\hat{\sigma}_{4,d}|\sqrt{N_{4,d}}B^4. \tag{E.197}$$

Now, letting $x \in \mathbb{S}^{d-1}$, and using Hoeffding's inequality, we find that if $u_1, \ldots, u_m$ are i.i.d. samples from $\rho$, then

$$\left| \frac{1}{m} \sum_{i=1}^{m} \sigma(\varphi_B(\langle u_1, x \rangle)) - \mathbb{E}_{u \sim \rho}[\sigma(\varphi_B(\langle u, x \rangle))] \right| \leq t \tag{E.198}$$

with probability at least $1 - 2 \exp\left( - \frac{Cmt^2}{\hat{\sigma}_{2,d}^2 N_{2,d} B^4 + \hat{\sigma}_{4,d}^2 N_{4,d} B^8} \right)$ for some universal constant $C$. Now, by Corollary 4.2.13 of Vershynin [74], for any $\gamma > 0$, there exists $\mathcal{N} \subset \mathbb{S}^{d-1}$ of size at most $(\frac{C}{\gamma})^d$ such that for any $x \in \mathbb{S}^{d-1}$, there exists $y \in \mathbb{S}^{d-1}$ such that $\|x - y\|_2 \leq \gamma$. Thus, with probability at least $1 - 2\left(\frac{C}{\gamma}\right)^d \exp\left( - \frac{Cmt^2}{\hat{\sigma}_{2,d}^2 N_{2,d} B^4 + \hat{\sigma}_{4,d}^2 N_{4,d} B^8} \right)$, for all $y \in \mathcal{N}$, we have

$$\left| \frac{1}{m} \sum_{i=1}^{m} \sigma(\varphi_B(\langle u_1, y \rangle)) - \mathbb{E}_{u \sim \rho}[\sigma(\varphi_B(\langle u, y \rangle))] \right| \leq t. \tag{E.199}$$

If Eq. (E.199) holds for any $y \in \mathcal{N}$, then for any $x \in \mathbb{S}^{d-1}$, since $|\sigma'(t)| \lesssim |\hat{\sigma}_{2,d}| \sqrt{N_{2,d}} + |\hat{\sigma}_{4,d}| \sqrt{N_{4,d}}$ for any $t \in [-1, 1]$ (see Lemma E.12), if we let $y \in \mathcal{N}$ such that $\|x - y\|_2 \leq \gamma$, then we have

$$\left| \frac{1}{m} \sum_{i=1}^{m} \sigma(\varphi_B(\langle u_1, x \rangle)) - \mathbb{E}_{u \sim \rho}[\sigma(\varphi_B(\langle u, x \rangle))] \right| \leq t + (|\hat{\sigma}_{2,d}| \sqrt{N_{2,d}} + |\hat{\sigma}_{4,d}| \sqrt{N_{4,d}})\gamma. \tag{E.200}$$

In summary, with probability at least $1 - 2\left(\frac{C}{\gamma}\right)^d \exp\left( - \frac{Cmt^2}{\hat{\sigma}_{2,d}^2 N_{2,d} B^4 + \hat{\sigma}_{4,d}^2 N_{4,d} B^8} \right)$, we have

$$\mathbb{E}_{x \sim \mathbb{S}^{d-1}}[(\tilde{f}_\rho(x) - \tilde{f}_{\bar{\rho}}(x))^2] \leq t^2 + (\hat{\sigma}_{2,d}^2 N_{2,d} + \hat{\sigma}_{4,d}^2 N_{4,d})\gamma^2. \tag{E.201}$$

Combining Eq. (E.201), Eq. (E.195) and Eq. (E.196), we have that

$$\mathbb{E}_{x \sim \mathbb{S}^{d-1}}[(f_\rho(x) - f_{\bar{\rho}}(x))^2] \lesssim t^2 + (\hat{\sigma}_{2,d}^2 N_{2,d} + \hat{\sigma}_{4,d}^2 N_{4,d})\gamma^2 \tag{E.202}$$

$$+ \exp\left( - \frac{B^2 d}{2} \right) \cdot (\hat{\sigma}_{2,d}^2 N_{2,d} + \hat{\sigma}_{4,d}^2 N_{4,d}) \tag{E.203}$$

with probability at least $1 - 2\left(\frac{C}{\gamma}\right)^d \exp\left( - \frac{Cmt^2}{\hat{\sigma}_{2,d}^2 N_{2,d} B^4 + \hat{\sigma}_{4,d}^2 N_{4,d} B^8} \right)$. Choosing $B = \frac{C'(\log d)^2}{\sqrt{d}}$ for a sufficiently large universal constant $C'$, and using the fact that $N_{2,d} \lesssim d^2$ and $N_{4,d} \lesssim d^4$ (by Eq. (2.10) of Atkinson and Han [12]) we have that

$$\mathbb{E}_{x \sim \mathbb{S}^{d-1}}[(f_\rho(x) - f_{\bar{\rho}}(x))^2] \lesssim t^2 + (\hat{\sigma}_{2,d}^2 d^2 + \hat{\sigma}_{4,d}^2 d^4)\gamma^2 + \frac{\hat{\sigma}_{2,d}^2 d^2 + \hat{\sigma}_{4,d}^2 d^4}{d^{\Omega(\log d)}} \tag{E.204}$$

with probability at least $1 - 2\left(\frac{C}{\gamma}\right)^d \exp\left( - \Omega\left( \frac{mt^2}{(\hat{\sigma}_{2,d}^2 + \hat{\sigma}_{4,d}^2)(\log d)^{16}} \right) \right)$. We can rearrange the failure probability to find that, up to constant factors, it is at most $\exp\left( -\Omega\left( \frac{mt^2}{(\hat{\sigma}_{2,d}^2 + \hat{\sigma}_{4,d}^2)(\log d)^{16}} \right) + d \log \frac{C}{\gamma} \right)$. Finally, choosing $\gamma = \frac{1}{d^{\Omega(\log d)}}$, and choosing $t$ so that the failure probability is at most $\delta$, we obtain

$$\mathbb{E}_{x \sim \mathbb{S}^{d-1}}[(f_\rho(x) - f_{\bar{\rho}}(x))^2] \lesssim \frac{(\hat{\sigma}_{2,d}^2 + \hat{\sigma}_{4,d}^2)(\log d)^{16}(d \log \frac{C}{\gamma} + \log \frac{1}{\delta})}{m} + \frac{\hat{\sigma}_{2,d}^2 d^2 + \hat{\sigma}_{4,d}^2 d^4}{d^{\Omega(\log d)}} \tag{E.205}$$

$$= \frac{(\hat{\sigma}_{2,d}^2 + \hat{\sigma}_{4,d}^2)(\log d)^{16}(d^2 \log d + \log \frac{1}{\delta})}{m} + \frac{\hat{\sigma}_{2,d}^2 + \hat{\sigma}_{4,d}^2}{d^{\Omega(\log d)}}$$

$$\text{(B.c. } \gamma = \tfrac{1}{d^{\Omega(\log d)}})$$

with probability at least $1 - \delta$, as desired. $\qquad \square$

**Lemma E.14** (Bound on Gradient). *Let $u \in \mathbb{S}^{d-1}$ and $\rho$ be a distribution with support on $\mathbb{S}^{d-1}$. In addition, let $\nabla_u L(\rho)$ and $\mathrm{grad}_u L(\rho)$ be defined as in Eq. (3.4). Then, we have $\|\nabla_u L(\rho)\|_2 \lesssim (\hat{\sigma}_{2,d}^2 + \hat{\sigma}_{4,d}^2)d^4$ and $\|\mathrm{grad}_u L(\rho)\|_2 \lesssim (\hat{\sigma}_{2,d}^2 + \hat{\sigma}_{4,d}^2)d^4$.*

*Similarly, let $\nabla_u \widehat{L}(\rho)$ and $\mathrm{grad}_u \widehat{L}(\rho)$ be defined as in Eq. (3.5). Then, we have $\|\nabla_u \widehat{L}(\rho)\|_2 \lesssim (\hat{\sigma}_{2,d}^2 + \hat{\sigma}_{4,d}^2)d^4$ and $\|\mathrm{grad}_u \widehat{L}(\rho)\|_2 \lesssim (\hat{\sigma}_{2,d}^2 + \hat{\sigma}_{4,d}^2)d^4$.*

*Proof of Lemma E.14.* First, we will bound $\nabla_u L(\rho)$ for any $u \in \mathbb{S}^{d-1}$ and distribution $\rho$ with support in $\mathbb{S}^{d-1}$:

$$\|\nabla_u L(\rho)\|_2 = \left\| \underset{x\sim\mathbb{S}^{d-1}}{\mathbb{E}}[(f_\rho(x) - y(x))\sigma'(\langle u, x\rangle)x] \right\|_2 \tag{E.206}$$

$$\leq \underset{x\sim\mathbb{S}^{d-1}}{\mathbb{E}} |f_\rho(x) - y(x)||\sigma'(\langle u, x\rangle)| \tag{E.207}$$

By Lemma E.12, for any $u, x \in \mathbb{S}^{d-1}$, we have $|\sigma'(\langle u, x\rangle)| \lesssim |\hat{\sigma}_{2,d}|\sqrt{N_{2,d}} + |\hat{\sigma}_{4,d}|\sqrt{N_{4,d}}$. Additionally, by Eq. (3.1) and by the definition of $y(x)$, we have $|f_\rho(x) - y(x)| \leq |\hat{\sigma}_{2,d}|\sqrt{N_{2,d}} + |\hat{\sigma}_{4,d}|\sqrt{N_{4,d}}$ by Lemma E.12. Thus,

$$\|\nabla_u L(\rho)\|_2 \lesssim \hat{\sigma}_{2,d}^2 N_{2,d} + \hat{\sigma}_{4,d}^2 N_{4,d} \lesssim (\hat{\sigma}_{2,d}^2 + \hat{\sigma}_{4,d}^2)d^4 \tag{E.208}$$

because $N_{2,d} \lesssim d^2$ and $N_{4,d} \lesssim d^4$ by Eq. (2.10) of Atkinson and Han [12]. By the same argument, for any $u \in \mathbb{S}^{d-1}$,

$$\|(I - uu^\top)\nabla_u L(\rho)\|_2 \lesssim (\hat{\sigma}_{2,d}^2 + \hat{\sigma}_{4,d}^2)d^4 \tag{E.209}$$

because $I - uu^\top$ is an orthogonal projection matrix. This completes the proof — the proof of the analogous statements for $\nabla_u \widehat{L}(\rho)$ and $\mathrm{grad}_u \widehat{L}(\rho)$ follow from the same arguments. $\qquad\square$

**Lemma E.15.** *Let $u, u' \in \mathbb{S}^{d-1}$. Then, we have*

$$\underset{x\sim\mathbb{S}^{d-1}}{\mathbb{E}}[(\sigma'(u^\top x) - \sigma'(u'^\top x))^2(u^\top x - u'^\top x)^2] \lesssim (\hat{\sigma}_{2,d}^2 + \hat{\sigma}_{4,d}^2)\|u - u'\|_2^4 \tag{E.210}$$

*and*

$$\underset{x\sim\mathbb{S}^{d-1}}{\mathbb{E}}[\sigma'(u^\top x)^2(u'^\top x)^2] \lesssim (\hat{\sigma}_{2,d}^2 + \hat{\sigma}_{4,d}^2). \tag{E.211}$$

*Proof of Lemma E.15.* For convenience, let $\delta := u' - u$. Then, we have

$$\underset{x\sim\mathbb{S}^{d-1}}{\mathbb{E}}[(\sigma'(u^\top x) - \sigma'(u'^\top x))^2(u^\top x - u'^\top x)^2] \tag{E.212}$$

$$\lesssim \hat{\sigma}_{2,d}^2 N_{2,d} \underset{x\sim\mathbb{S}^{d-1}}{\mathbb{E}}[(P_{2,d}'(u^\top x) - P_{2,d}'(u'^\top x))^2(u^\top x - u'^\top x)^2] \tag{E.213}$$

$$+ \hat{\sigma}_{4,d}^2 N_{4,d} \underset{x\sim\mathbb{S}^{d-1}}{\mathbb{E}}[(P_{4,d}'(u^\top x) - P_{4,d}'(u'^\top x))^2(u^\top x - u'^\top x)^2]$$

$$\text{(By definition of } \sigma \text{ and } \overline{P}_{k,d} \text{ and b.c. } (a + b)^2 \lesssim a^2 + b^2\text{)}$$

$$\lesssim \hat{\sigma}_{2,d}^2 N_{2,d} \underset{x\sim\mathbb{S}^{d-1}}{\mathbb{E}}[(\delta^\top x)^4] + \hat{\sigma}_{4,d}^2 N_{4,d} \underset{x\sim\mathbb{S}^{d-1}}{\mathbb{E}}[((u^\top x)^3 - (u'^\top x)^3)^2(u^\top x - u'^\top x)^2] \tag{E.214}$$

$$+ \frac{\hat{\sigma}_{4,d}^2 N_{4,d}}{d^2} \underset{x\sim\mathbb{S}^{d-1}}{\mathbb{E}}[(u^\top x - u'^\top x)^4]$$

$$\text{(By Eq. (C.4) and expanding the term with } P_{4,d}')$$

$$\lesssim \hat{\sigma}_{2,d}^2 N_{2,d} \underset{x\sim\mathbb{S}^{d-1}}{\mathbb{E}}[(\delta^\top x)^4] + \hat{\sigma}_{4,d}^2 N_{4,d} \sum_{p+q=2} \underset{x\sim\mathbb{S}^{d-1}}{\mathbb{E}}[(u^\top x)^{2p}(u'^\top x)^{2q}(\delta^\top x)^4] \tag{E.215}$$

$$+ \frac{\hat{\sigma}_{4,d}^2 N_{4,d}}{d^2} \underset{x\sim\mathbb{S}^{d-1}}{\mathbb{E}}[(\delta^\top x)^4] \tag{E.216}$$

Here the last inequality is due to the identity $a^3 - b^3 = (a - b)(a^2 + ab + b^2)$, which gives $((u'^\top x)^3 - (u^\top x)^3)^2 = ((u'^\top x)^2 + (u'^\top x)(u^\top x) + (u^\top x)^2)^2(u'^\top x - u^\top x)^2$ — expanding the first factor fully gives terms of the form $(u'^\top x)^p(u^\top x)^q$ where $p$ and $q$ are both even and $p + q = 4$. Thus, applying Lemma I.17 to the final expression above gives

$$\underset{x\sim\mathbb{S}^{d-1}}{\mathbb{E}}[(\sigma'(u^\top x) - \sigma'(u'^\top x))^2(u^\top x - u'^\top x)^2] \tag{E.217}$$

$$\lesssim \hat{\sigma}_{2,d}^2 N_{2,d} \cdot \frac{1}{d^2}\|\delta\|_2^4 + \hat{\sigma}_{4,d}^2 N_{4,d} \sum_{p+q=2} \frac{1}{d^{p+q+2}}\|\delta\|_2^4 + \frac{\hat{\sigma}_{4,d}^2 N_{4,d}}{d^2} \cdot \frac{1}{d^2}\|\delta\|_2^4$$

$$\text{(By Lemma I.17)}$$

$$\lesssim \hat{\sigma}_{2,d}^2\|\delta\|_2^4 + \hat{\sigma}_{4,d}^2\|\delta\|_2^4$$

$$\text{(B.c. } N_{2,d} \lesssim d^2 \text{ and } N_{4,d} \lesssim d^4 \text{ (see Equation 2.10 of Atkinson and Han [12]))}$$

which completes the proof of the first statement. The proof of the second statement is similar — we expand using the definition of $\sigma$ and then apply Lemma I.17 to each of the terms. We omit the details of the calculation. $\qquad\square$

**Lemma E.16.** *Let $\rho$ and $\rho'$ be two distributions on $\mathbb{S}^{d-1}$ which both have support size $m < \infty$, and consider a coupling $(u, u') \sim (\rho, \rho')$ of the two distributions. Additionally, let $f_\rho$ and $f_{\rho'}$ be defined as in Eq. (3.1). Then,*

$$\mathbb{E}_{x \sim \mathbb{S}^{d-1}}[(f_\rho(x) - f_{\rho'}(x))^2] \lesssim (\hat{\sigma}_{2,d}^2 + \hat{\sigma}_{4,d}^2) \mathbb{E}_{(u,u') \sim (\rho,\rho')} \|u - u'\|_2^2 \qquad (\text{E.218})$$

*Proof of Lemma E.16.* We can write the left-hand side as

$$\mathbb{E}_{x \sim \mathbb{S}^{d-1}}[(f_\rho(x) - f_{\rho'}(x))^2] = \mathbb{E}_{x \sim \mathbb{S}^{d-1}} \left[ \left( \mathbb{E}_{(u,u') \sim (\rho,\rho')}[\sigma(\langle u, x \rangle) - \sigma(\langle u', x \rangle)] \right)^2 \right] \qquad (\text{E.219})$$

$$\leq \mathbb{E}_{x \sim \mathbb{S}^{d-1}} \mathbb{E}_{(u,u') \sim (\rho,\rho')} \left( \sigma(\langle u, x \rangle) - \sigma(\langle u', x \rangle) \right)^2 \qquad (\text{E.220})$$

$$= \mathbb{E}_{(u,u') \sim (\rho,\rho')} \mathbb{E}_{x \sim \mathbb{S}^{d-1}} \left( \sigma(\langle u, x \rangle) - \sigma(\langle u', x \rangle) \right)^2 \qquad (\text{E.221})$$

For any $u, u' \in \mathbb{S}^{d-1}$ and $x \in \mathbb{S}^{d-1}$, by the mean-value theorem, there exists $\lambda$ which is a convex combination of $\langle u, x \rangle$ and $\langle u', x \rangle$ such that

$$\sigma(\langle u, x \rangle) - \sigma(\langle u', x \rangle) = \sigma'(\lambda)(\langle u, x \rangle - \langle u', x \rangle). \qquad (\text{E.222})$$

Using a similar argument as in the proof of Lemma E.15, we can show that

$$\mathbb{E}_{x \sim \mathbb{S}^{d-1}} \left[ \sigma'(\lambda)^2 (\langle u, x \rangle - \langle u', x \rangle)^2 \right] \lesssim (\hat{\sigma}_{2,d}^2 + \hat{\sigma}_{4,d}^2) \|u - u'\|_2^2. \qquad (\text{E.223})$$

The proof proceeds by expanding $\sigma'(\lambda)$ into each of its terms and using the inequality $|a + b|^3 \lesssim |a|^3 + |b|^3$ (which holds for any two real numbers $a, b$) in order to consider only terms which involve one of $u$ or $u'$ — the resulting terms are of the same form as those which appear in Eq. (E.216). We omit the details of the calculation. Thus,

$$\mathbb{E}_{x \sim \mathbb{S}^{d-1}}[(f_\rho(x) - f_{\rho'}(x))^2] \leq \mathbb{E}_{(u,u') \sim (\rho,\rho')} \mathbb{E}_{x \sim \mathbb{S}^{d-1}} \left( \sigma(\langle u, x \rangle) - \sigma(\langle u', x \rangle) \right)^2 \qquad (\text{E.224})$$

$$\lesssim (\hat{\sigma}_{2,d}^2 + \hat{\sigma}_{4,d}^2) \mathbb{E}_{(u,u') \sim (\rho,\rho')} \|u - u'\|_2^2 \qquad (\text{E.225})$$

as desired. $\qquad\square$

# F    Proof of Theorem 3.5

In this section, we build on our analysis from Appendix E to show that projected gradient descent with $\frac{1}{\text{poly}(d)}$ learning rate can achieve low population loss. In the rest of the section, for convenience, we define $\tilde{\delta}_t = \tilde{u}_t - \hat{u}_{\eta t}$, and omit $\chi$ when it is clear from context.

**Lemma F.1.** *Let $\rho, \rho'$ be two distributions on $\mathbb{S}^{d-1}$, and let $\Gamma$ be a coupling between $\rho$ and $\rho'$. Additionally, let $u_1, u_2, v \in \mathbb{S}^{d-1}$. Then, we have*

$$|\nabla_{u_1} L(\rho) \cdot v - \nabla_{u_2} L(\rho') \cdot v| \lesssim (\hat{\sigma}_{2,d}^2 + \hat{\sigma}_{4,d}^2) \left( \mathbb{E}_{u,u' \sim \Gamma} \left[ \|u - u'\|_2^2 \right]^{1/2} + \|u_1 - u_2\|_2 \right). \quad (\text{F.1})$$

*Proof.* We can expand the left-hand side as

$$\left| \mathbb{E}_{x \sim \mathbb{S}^{d-1}} \left[ (f_\rho(x) - y(x))\sigma'(u_1 \cdot x)(v \cdot x) - (f_{\rho'}(x) - y(x))\sigma'(u_2 \cdot x)(v \cdot x) \right] \right|. \qquad (\text{F.2})$$

We first obtain a bound on

$$\left| \mathbb{E}_{x \sim \mathbb{S}^{d-1}} \left[ f_\rho(x)\sigma'(u_1 \cdot x)(v \cdot x) - f_{\rho'}(x)\sigma'(u_2 \cdot x)(v \cdot x) \right] \right|. \qquad (\text{F.3})$$

By the definition of $f_\rho$, we can rewrite Eq. (F.3) as

$$\left| \mathop{\mathbb{E}}_{u_1', u_2' \sim \Gamma} \mathop{\mathbb{E}}_{x \sim \mathbb{S}^{d-1}} \left[ \sigma(u_1' \cdot x)\sigma'(u_1 \cdot x)(v \cdot x) - \sigma(u_2' \cdot x)\sigma'(u_2 \cdot x)(v \cdot x) \right] \right| \tag{F.4}$$

By Eq. (C.4), $\sigma$ is an even fourth-degree polynomial and $\sigma'$ is an odd third-degree polynomial. Thus, the inner expectation can be written as a linear combination of a constant number of terms of the form

$$\left| \mathop{\mathbb{E}}_{x \sim \mathbb{S}^{d-1}} \left[ (u_1' \cdot x)^i (u_1 \cdot x)^j (v \cdot x) - (u_2' \cdot x)^i (u_2 \cdot x)^j (v \cdot x) \right] \right| \tag{F.5}$$

where $i = 0, 2, 4$ and $j = 1, 3$. We can also estimate the coefficient which accompanies each individual term Eq. (F.5). We first recall the facts that $\sigma(t) = \hat{\sigma}_{2,d}\bar{P}_{2,d}(t) + \hat{\sigma}_{4,d}\bar{P}_{4,d}(t)$, that $N_{2,d} = O(d^2)$ and $N_{4,d} = O(d^4)$ by Equation (2.10) of Atkinson and Han [12], and that the $i^{\text{th}}$ degree term of $P_{2,d}$ has a coefficient which is $O(d^{i/2-1})$ and the $i^{\text{th}}$ degree term of $P_{4,d}$ has a coefficient which is $O(d^{i/2-2})$. From these facts it follows that the $i^{th}$ degree term of $\bar{P}_{2,d} = \sqrt{N_{2,d}}P_{2,d}$ has a coefficient which is $O(d^{i/2})$ and the $i^{th}$ degree term of $\bar{P}_{4,d} = \sqrt{N_{4,d}}P_{4,d}$ has a coefficient which is $O(d^{i/2})$. Thus, each term of the form given in Eq. (F.5) has a coefficient which is at most $(\hat{\sigma}_{2,d}^2 + \hat{\sigma}_{4,d}^2)d^{(i+j+1)/2}$ in absolute value. We can now upper bound each of the terms of the form given in Eq. (F.5) as follows:

$$\left| \mathop{\mathbb{E}}_{x \sim \mathbb{S}^{d-1}} \left[ (u_1' \cdot x)^i (u_1 \cdot x)^j (v \cdot x) - (u_2' \cdot x)^i (u_2 \cdot x)^j (v \cdot x) \right] \right| \tag{F.6}$$

$$\leq \left| \mathop{\mathbb{E}}_{x \sim \mathbb{S}^{d-1}} \left[ \left( (u_1' \cdot x)^i - (u_2' \cdot x)^i \right)(u_1 \cdot x)^j (v \cdot x) \right] \right| \tag{F.7}$$

$$+ \left| \mathop{\mathbb{E}}_{x \sim \mathbb{S}^{d-1}} \left[ (u_2' \cdot x)^i \left( (u_1 \cdot x)^j - (u_2 \cdot x)^j \right)(v \cdot x) \right] \right| \tag{F.8}$$

$$\leq \sum_{k=0}^{i-1} \left| \mathop{\mathbb{E}}_{x \sim \mathbb{S}^{d-1}} \left[ \left( (u_1' - u_2') \cdot x \right)(u_1' \cdot x)^k (u_2' \cdot x)^{i-1-k}(u_1 \cdot x)^j (v \cdot x) \right] \right| \tag{F.9}$$

$$+ \sum_{k=0}^{j-1} \left| \mathop{\mathbb{E}}_{x \sim \mathbb{S}^{d-1}} \left[ (u_2' \cdot x)^i \left( (u_1 - u_2) \cdot x \right)(u_1 \cdot x)^k (u_2 \cdot x)^{j-1-k}(v \cdot x) \right] \right| \tag{F.10}$$

$$\lesssim \sum_{k=0}^{i-1} \|u_1' - u_2'\|_2 \cdot \frac{1}{d^{(1+k+i-1-k+j+1)/2}} + \sum_{k=0}^{j-1} \|u_1 - u_2\|_2 \cdot \frac{1}{d^{(i+1+k+j-1-k+1)/2}} \tag{By Lemma I.17}$$

$$\lesssim \|u_1' - u_2'\|_2 \frac{1}{d^{(i+j+1)/2}} + \|u_1 - u_2\|_2 \frac{1}{d^{(i+j+1)/2}} . \tag{F.11}$$

Since the coefficient of this term is at most $(\hat{\sigma}_{2,d}^2 + \hat{\sigma}_{4,d}^2)d^{(i+j+1)/2}$ in absolute value, this term contributes at most

$$(\hat{\sigma}_{2,d}^2 + \hat{\sigma}_{4,d}^2)\Big( \|u_1' - u_2'\|_2 + \|u_1 - u_2\|_2 \Big) . \tag{F.12}$$

Thus, taking the expectation over $u_1', u_2' \sim \Gamma$, we have

$$\left| \mathop{\mathbb{E}}_{x \sim \mathbb{S}^{d-1}} \left[ f_\rho(x)\sigma'(u_1 \cdot x)(v \cdot x) - f_{\rho'}(x)\sigma'(u_2 \cdot x)(v \cdot x) \right] \right| \tag{F.13}$$

$$\lesssim (\hat{\sigma}_{2,d}^2 + \hat{\sigma}_{4,d}^2)\Big( \mathop{\mathbb{E}}_{u,u' \sim \Gamma} \left[ \|u - u'\|_2 \right] + \|u_1 - u_2\|_2 \Big) \tag{F.14}$$

$$\lesssim (\hat{\sigma}_{2,d}^2 + \hat{\sigma}_{4,d}^2)\Big( \mathop{\mathbb{E}}_{u,u' \sim \Gamma} \left[ \|u - u'\|_2^2 \right]^{1/2} + \|u_1 - u_2\|_2 \Big) . \tag{F.15}$$

A similar but simpler argument can be used to bound

$$\left| \mathop{\mathbb{E}}_{x \sim \mathbb{S}^{d-1}} \left[ y(x)\sigma'(u_1 \cdot x)(v \cdot x) - y(x)\sigma'(u_2 \cdot x)(v \cdot x) \right] \right| \tag{F.16}$$

by expanding $y(x)$ and $\sigma'(x)$. Thus, we obtain

$$|\nabla_{u_1} L(\rho) \cdot v - \nabla_{u_2} L(\rho) \cdot v| \le (\hat{\sigma}_{2,d}^2 + \hat{\sigma}_{4,d}^2)\Big( \underset{u,u'\sim\Gamma}{\mathbb{E}}\Big[\|u - u'\|_2^2\Big]^{1/2} + \|u_1 - u_2\|_2 \Big) \quad \text{(F.17)}$$

as desired. $\qquad\square$

**Lemma F.2.** *Suppose we are in the setting of Theorem 3.5. Assume that $n \gtrsim d^3(\log d)^C$ for a sufficiently large universal constant $C > 0$. Additionally, assume that $\max_\chi \|\tilde{\delta}_t(\chi)\|_2 \le \frac{1}{\sqrt{d}}$. Then, we have*

$$|(\mathrm{grad}_{\tilde{u}_t}\widehat{L}(\tilde{\rho}_t) - \mathrm{grad}_{\hat{u}_{\eta t}}\widehat{L}(\hat{\rho}_{\eta t})) \cdot \tilde{\delta}_t| \lesssim (\hat{\sigma}_{2,d}^2 + \hat{\sigma}_{4,d}^2) \max_\chi \|\tilde{\delta}_t(\chi)\|_2^2 \quad \text{(F.18)}$$

*Proof.* We can write the left-hand side of Eq. (F.18) as

$$(\mathrm{grad}_{\tilde{u}_t}\widehat{L}(\tilde{\rho}_t) - \mathrm{grad}_{\hat{u}_{\eta t}}\widehat{L}(\hat{\rho}_t)) \cdot \tilde{\delta}_t \quad \text{(F.19)}$$

$$= \frac{1}{n}\sum_{i=1}^n \Big((f_{\tilde{\rho}_t}(x_i) - y(x_i))\sigma'(\tilde{u}_t \cdot x_i) - (f_{\hat{\rho}_{\eta t}}(x_i) - y(x_i))\sigma'(\hat{u}_{\eta t} \cdot x_i)\Big)x_i \cdot \tilde{\delta}_t. \quad \text{(F.20)}$$

We proceed by first showing using Lemma I.7 that we do not incur much error when we replace the empirical average over the $x_i$ by an expectation over $x \sim \mathbb{S}^{d-1}$. We can separate the right hand side of Eq. (F.19) into two parts:

$$S_1 = \frac{1}{n}\sum_{i=1}^n \Big(f_{\tilde{\rho}_t}(x_i)\sigma'(\tilde{u}_t \cdot x_i) - f_{\hat{\rho}_{\eta t}}(x_i)\sigma'(\hat{u}_{\eta t} \cdot x_i)\Big)(\tilde{\delta}_t \cdot x_i) \quad \text{(F.21)}$$

and

$$S_2 = \frac{1}{n}\sum_{i=1}^n \Big(y(x_i)\sigma'(\tilde{u}_t \cdot x_i) - y(x_i)\sigma'(\hat{u}_{\eta t} \cdot x_i)\Big)(\tilde{\delta}_t \cdot x_i). \quad \text{(F.22)}$$

We wish to show that $S_1$ and $S_2$ are very close to

$$M_1 = \underset{x\sim\mathbb{S}^{d-1}}{\mathbb{E}}\Big[\Big(f_{\tilde{\rho}_t}(x)\sigma'(\tilde{u}_t \cdot x) - f_{\hat{\rho}_{\eta t}}(x)\sigma'(\hat{u}_{\eta t} \cdot x)\Big)(\tilde{\delta}_t \cdot x)\Big] \quad \text{(F.23)}$$

and

$$M_2 = \underset{x\sim\mathbb{S}^{d-1}}{\mathbb{E}}\Big[\Big(y(x_i)\sigma'(\tilde{u}_t \cdot x_i) - y(x_i)\sigma'(\hat{u}_{\eta t} \cdot x_i)\Big)(\tilde{\delta}_t \cdot x_i)\Big] \quad \text{(F.24)}$$

respectively, with high probability — then, we will obtain bounds on $M_1$ and $M_2$. We will show that $S_1$ and $M_1$ are close — the proof that $S_2$ and $M_2$ are close is similar.

**Expanding into Monomials** Our proof strategy is to expand the following quantity, for $x \in \mathbb{S}^{d-1}$:

$$G(x) = \Big(f_{\tilde{\rho}_t}(x)\sigma'(\tilde{u}_t \cdot x) - f_{\hat{\rho}_{\eta t}}(x)\sigma'(\hat{u}_{\eta t} \cdot x)\Big)(\tilde{\delta}_t \cdot x) \quad \text{(F.25)}$$

and obtain concentration bounds for each of the terms. In this proof, we let $\Gamma$ denote the natural coupling between $\bar{\rho}, \hat{\rho}, \tilde{\rho}$ which is the distribution over triples $(\bar{u}_{\eta t}(\chi), \hat{u}_{\eta t}(\chi), \tilde{u}_{\eta t}(\chi))$, where $\chi$ is sampled from $\{\chi_1, \ldots, \chi_m\}$. Then, we can write $G(x)$ as

$$G(x) = \underset{\bar{u}'_{\eta t}, \hat{u}'_{\eta t}, \tilde{u}'_t\sim\Gamma}{\mathbb{E}}\Big[\Big(\sigma(\tilde{u}'_t \cdot x)\sigma'(\tilde{u}_t \cdot x) - \sigma(\hat{u}'_{\eta t} \cdot x)\sigma'(\hat{u}_{\eta t} \cdot x)\Big) \cdot (\tilde{\delta}_t \cdot x)\Big]. \quad \text{(F.26)}$$

For convenience, let us define

$$g(x) = \Big(\sigma(\tilde{u}'_t \cdot x)\sigma'(\tilde{u}_t \cdot x) - \sigma(\hat{u}'_{\eta t} \cdot x)\sigma'(\hat{u}_{\eta t} \cdot x)\Big) \cdot (\tilde{\delta}_t \cdot x). \quad \text{(F.27)}$$

Clearly we have $G(x) = \mathbb{E}_{\bar{u}'_{\eta t}, \hat{u}'_{\eta t}, \tilde{u}'_t\sim\Gamma}[g(x)]$. Now, we consider the effect of expanding $g(x)$ by expanding $\sigma$ and $\sigma'$ into monomials that involve $(\tilde{u}'_t \cdot x)$, $(\tilde{u}_t \cdot x)$, $(\hat{u}'_{\eta t} \cdot x)$ and $(\hat{u}_{\eta t} \cdot x)$. Since $\sigma$

is an even polynomial of degree 4 and $\sigma'$ is an odd polynomial of degree 3, we can write $g(x)$ as a linear combination of terms of the following form:

$$\left((\tilde{u}'_t \cdot x)^{i_1}(\tilde{u}_t \cdot x)^{i_2} - (\hat{u}'_{\eta t} \cdot x)^{i_1}(\hat{u}_{\eta t} \cdot x)^{i_2}\right)(\tilde{\delta}_t \cdot x) \tag{F.28}$$

where $i_1 = 0, 2, 4$ and $i_2 = 1, 3$ — each such term also has a constant factor of $O(\hat{\sigma}_{i_1,d}\hat{\sigma}_{i_2,d}\sqrt{N_{i_1,d}}\sqrt{N_{i_2,d}})$. We now consider the contribution of Eq. (F.28) for each value of $i_1, i_2$. In the case where $i_1 = 0$, we can rewrite Eq. (F.28) as $(\tilde{\delta}_t \cdot x)^2$ when $i_2 = 1$, and

$$\left((\tilde{u}_t \cdot x)^2 + (\tilde{u}_t \cdot x)(\hat{u}_{\eta t} \cdot x) + (\hat{u}_{\eta t} \cdot x)^2\right)(\tilde{\delta}_t \cdot x)^2 \tag{F.29}$$

when $i_2 = 3$. In the case where $i_1$ and $i_2$ are both nonzero, letting $\tilde{\delta}'_t = \tilde{u}'_t - \hat{u}'_{\eta t}$ we can write Eq. (F.28) as

$$\left((\tilde{u}'_t \cdot x)^{i_1}(\tilde{u}_t \cdot x)^{i_2} - (\hat{u}'_{\eta t} \cdot x)^{i_1}(\hat{u}_{\eta t} \cdot x)^{i_2}\right)(\tilde{\delta}_t \cdot x) \tag{F.30}$$

$$= \left((\tilde{u}'_t \cdot x)^{i_1} - (\hat{u}'_{\eta t} \cdot x)^{i_1}\right)(\tilde{u}_t \cdot x)^{i_2}(\tilde{\delta}_t \cdot x) \tag{F.31}$$

$$+ (\hat{u}'_{\eta t} \cdot x)^{i_1}\left((\tilde{u}_t \cdot x)^{i_2} - (\hat{u}_{\eta t} \cdot x)^{i_2}\right)(\tilde{\delta}_t \cdot x) \tag{F.32}$$

$$= \sum_{\substack{i_3+i_4=i_1-1 \\ i_3,i_4 \geq 0}} (\tilde{u}'_t \cdot x)^{i_3}(\hat{u}'_{\eta t} \cdot x)^{i_4}(\tilde{u}'_t \cdot x - \hat{u}'_{\eta t} \cdot x)(\tilde{u}_t \cdot x)^{i_2}(\tilde{\delta}_t \cdot x) \tag{F.33}$$

$$+ \sum_{\substack{i_3+i_4=i_2-1 \\ i_3,i_4 \geq 0}} (\hat{u}'_{\eta t} \cdot x)^{i_1}(\tilde{u}_t \cdot x)^{i_3}(\hat{u}_{\eta t} \cdot x)^{i_4}(\tilde{u}_t \cdot x - \hat{u}_{\eta t} \cdot x)(\tilde{\delta}_t \cdot x) \tag{F.34}$$

$$= \sum_{\substack{i_3+i_4=i_1-1 \\ i_3,i_4 \geq 0}} (\tilde{u}'_t \cdot x)^{i_3}(\hat{u}'_{\eta t} \cdot x)^{i_4}(\tilde{u}_t \cdot x)^{i_2}(\tilde{\delta}_t \cdot x)(\tilde{\delta}'_t \cdot x) \tag{F.35}$$

$$+ \sum_{\substack{i_3+i_4=i_2-1 \\ i_3,i_4 \geq 0}} (\hat{u}'_{\eta t} \cdot x)^{i_1}(\tilde{u}_t \cdot x)^{i_3}(\hat{u}_{\eta t} \cdot x)^{i_4}(\tilde{\delta}_t \cdot x)^2 . \tag{F.36}$$

In summary, Eq. (F.28) can be written as a linear combination of terms of the form

$$(\tilde{u}'_t \cdot x)^{i_3}(\hat{u}_{\eta t} \cdot x)^{i_4}(\tilde{u}_t \cdot x)^{i_2}(\tilde{\delta}_t \cdot x)(\tilde{\delta}'_t \cdot x) \tag{F.37}$$

where $i_3 + i_4 = 1, 3$ and $i_2 = 1, 3$, and terms of the form

$$(\hat{u}'_{\eta t} \cdot x)^{i_1}(\tilde{u}_t \cdot x)^{i_3}(\hat{u}_{\eta t} \cdot x)^{i_4}(\tilde{\delta}_t \cdot x)^2 \tag{F.38}$$

where $i_1 = 0, 2, 4$ and $i_3 + i_4 = 0, 2$.

**Contribution of Eq. (F.38) to Concentration Error** In Eq. (F.38), we can write $\tilde{u}_t = \hat{u}_{\eta t} + \tilde{\delta}_t$, and thus write Eq. (F.38) as a linear combination of terms of the form

$$(\hat{u}'_{\eta t} \cdot x)^{i_1}(\hat{u}_{\eta t} \cdot x)^{i_2}(\tilde{\delta}_t \cdot x)^{2+i_3} \tag{F.39}$$

where $i_1 = 0, 2, 4$ and $i_2 + i_3 = 0, 2$. Next, in order to use Lemma I.7, we write $\hat{u}_{\eta t} = \bar{u}_{\eta t} + \delta_t$ and expand (recalling the definition of $\delta_t$ from Lemma 5.4). Thus, Eq. (F.38) is a linear combination of terms of the form

$$(\bar{u}'_{\eta t} \cdot x)^{i_1}(\delta'_{\eta t} \cdot x)^{i_2}(\bar{u}_{\eta t} \cdot x)^{i_3}(\delta_{\eta t} \cdot x)^{i_4}(\tilde{\delta}_t \cdot x)^{2+i_5} \tag{F.40}$$

where $i_1 + i_2 = 0, 2, 4$, and $i_3 + i_4 + i_5 = 0, 2$. Finally, we obtain a concentration bound for Eq. (F.40) using Lemma I.7:

$$\left| \frac{1}{n} \sum_{i=1}^n (\bar{u}'_{\eta t} \cdot x_i)^{i_1}(\delta'_{\eta t} \cdot x_i)^{i_2}(\bar{u}_{\eta t} \cdot x_i)^{i_3}(\delta_{\eta t} \cdot x_i)^{i_4}(\tilde{\delta}_t \cdot x_i)^{2+i_5} \right. \tag{F.41}$$

$$\left. - \mathop{\mathbb{E}}_{x \sim \mathbb{S}^{d-1}}\left[(\bar{u}'_{\eta t} \cdot x)^{i_1}(\delta'_{\eta t} \cdot x)^{i_2}(\bar{u}_{\eta t} \cdot x)^{i_3}(\delta_{\eta t} \cdot x)^{i_4}(\tilde{\delta}_t \cdot x)^{2+i_5}\right] \right| \tag{F.42}$$

$$\lesssim (\log d)^{O(1)} \frac{\|\delta'_{\eta t}\|_2^{i_2}\|\delta_{\eta t}\|_2^{i_4}\|\tilde{\delta}_t\|_2^{2+i_5}}{d^{(i_1+i_2+i_3+i_4+i_5)/2+1}}\left(\sqrt{\frac{d}{n}} + \frac{d^{(i_2+i_4+i_5)/2+1}}{n} + \frac{1}{d^{\Omega(\log d)}}\right). \tag{F.43}$$

Here, when applying Lemma I.7, $\bar{u}_{\eta t}$ and $\bar{u}'_{\eta t}$ take the role of $w_1$ and $w_2$, while $\delta'_{\eta t}/\|\delta'_{\eta t}\|_2$, $\delta_{\eta t}/\|\delta_{\eta t}\|_2$ and $\tilde{\delta}_t/\|\tilde{\delta}_t\|_2$ take the role of $u_1$, $u_2$, $u_3$ and $u_4$ (we let $u_3 = u_4 = \tilde{\delta}_t/\|\tilde{\delta}_t\|_2$ and choose $p_3$ and $p_4$ so that $p_3 + p_4 = 2 + i_5$). Note that we implicitly perform a union bound over $\bar{u}'_{\eta t}$ and $\delta'_{\eta t}$ drawn from the support of $f_{\hat{\rho}_{\eta t}}$ — this does not affect the failure probability, since the failure probability for the result of Lemma I.7 is $\frac{1}{d^{\Omega(\log d)}}$, and it is assumed in Theorem 3.5 that $m \leq d^C$ for some universal constant $C$. When applying Lemma I.7 at later points in this proof, we implicitly perform this union bound.

In order to compute the final contribution of each such term to the concentration error between $S_1$ and $M_1$, we must first (1) incorporate the coefficients of $\sigma$ and $\sigma'$ with respect to the Legendre polynomials $P_{2,d}$ and $P_{4,d}$, (2) take the average of $\|\delta'_{\eta t}\|_2^b$ with respect to $\chi$ (which is implicit throughout this proof), and (3) use the fact that $n \gtrsim d^3 (\log d)^{O(1)}$.

Recall that $\sigma = \hat{\sigma}_{2,d}\sqrt{N_{2,d}}P_{2,d} + \hat{\sigma}_{4,d}\sqrt{N_{4,d}}P_{4,d}$. Using Eq. (C.4) and the fact that $N_{2,d} \asymp d^2$ and $N_{4,d} \asymp d^4$ (by Equation (2.10) of Atkinson and Han [12]), we find that, up to constant factors, the coefficient of the $0^{\text{th}}$-order term in $\sigma(t)$ can be bounded above by $\max(\hat{\sigma}_{2,d}, \hat{\sigma}_{4,d})$, the coefficient of the $2^{\text{nd}}$-order term in $\sigma(t)$ can be bounded above by $\max(\hat{\sigma}_{2,d}, \hat{\sigma}_{4,d})d$, and the coefficient of the $4^{\text{th}}$ order term can be bounded above by $\hat{\sigma}_{4,d}d^2$. Similarly, up to constant factors, the coefficient of the $1^{\text{st}}$-order term in $\sigma'(t)$ can be bounded above by $\max(\hat{\sigma}_{2,d}, \hat{\sigma}_{4,d})d$ and the coefficient of the $3^{\text{rd}}$-order term in $\sigma'(t)$ can be bounded above by $\max(\hat{\sigma}_{2,d}, \hat{\sigma}_{4,d})d^2$. Thus, each term of degree $k$ originating from $\sigma$ contributes a factor of $\max(\hat{\sigma}_{2,d}, \hat{\sigma}_{4,d})d^{k/2}$ to the coefficient of Eq. (F.38), and each term of degree $k-1$ originating from $\sigma'$ contributes a factor of $\max(\hat{\sigma}_{2,d}, \hat{\sigma}_{4,d})d^{k/2}$ to the coefficient of Eq. (F.38). Finally, for each term in Eq. (F.40) of total degree $i_1 + i_2 + i_3 + i_4 + i_5 + 2$, note that $\sigma$ and $\sigma'$ together contribute a factor of degree $i_1 + i_2 + i_3 + i_4 + i_5 + 1$, since in Eq. (F.27), aside from the factors contributed by $\sigma$ and $\sigma'$, only a single factor of $\tilde{\delta}_t \cdot x$ is present.

Thus, to obtain the final contribution of Eq. (F.41) to the overall error between $M_1$ and $S_1$, we multiply by an additional coefficient of $\max(\hat{\sigma}_{2,d}^2, \hat{\sigma}_{4,d}^2)d^{(i_1+i_2+i_3+i_4+i_5)/2+1}$. Taking this into account, the contribution of Eq. (F.41) is

$$(\hat{\sigma}_{2,d}^2 + \hat{\sigma}_{4,d}^2)(\log d)^{O(1)}\|\delta'_{\eta t}\|_2^{i_2}\|\delta_{\eta t}\|_2^{i_4}\|\tilde{\delta}_t\|_2^{2+i_5}\left(\sqrt{\frac{d}{n}} + \frac{d^{(i_2+i_4+i_5)/2+1}}{n} + \frac{1}{d^{\Omega(\log d)}}\right). \quad \text{(F.44)}$$

Next, we simplify this using casework on $i_2$. For the case $i_2 \leq 1$, since $i_4 + i_5 \leq 2$, the bound is at most

$$(\hat{\sigma}_{2,d}^2 + \hat{\sigma}_{4,d}^2)\|\delta'_{\eta t}\|_2^{i_2}\|\delta_{\eta t}\|_2^{i_4}\|\tilde{\delta}_t\|_2^{2+i_5} \lesssim (\hat{\sigma}_{2,d}^2 + \hat{\sigma}_{4,d}^2)\|\tilde{\delta}_t\|_2^2 \quad \text{(F.45)}$$

since $d^{(i_2+i_4+i_5)/2+1} \leq d^{2.5}$, and because $n \gtrsim d^3(\log d)^{O(1)}$, and we eliminate a factor of $\|\delta'_{\eta t}\|_2^{i_2}\|\delta_{\eta t}\|_2^{i_4}\|\tilde{\delta}_t\|_2^{i_5}$ because all of the particles have $\ell_2$ norm at most 1. Additionally, for the case $i_2 \geq 2$, after taking the average of $\|\delta'_{\eta t}\|_2^{i_2} \lesssim \|\delta'_{\eta t}\|_2^2$ across all $\chi$, the bound is at most

$$(\hat{\sigma}_{2,d}^2 + \hat{\sigma}_{4,d}^2)(\log d)^{O(1)}\overline{\Delta}_{\eta t}^2\|\tilde{\delta}_t\|_2^2\left(\sqrt{\frac{d}{n}} + \frac{d^{(i_2+i_4+i_5)/2+1}}{n} + \frac{1}{d^{\Omega(\log d)}}\right) \quad \text{(F.46)}$$

$$\lesssim (\hat{\sigma}_{2,d}^2 + \hat{\sigma}_{4,d}^2)\overline{\Delta}_{\eta t}^2\|\tilde{\delta}_t\|_2^2\left(d^{i_2/2-1} + O\left(\frac{1}{d}\right)\right)$$
$$\text{(B.c. } n \gtrsim d^3(\log d)^{O(1)} \text{ and } i_4 + i_5 \leq 2)$$

$$\lesssim (\hat{\sigma}_{2,d}^2 + \hat{\sigma}_{4,d}^2)\|\tilde{\delta}_t\|_2^2 \qquad \text{(B.c. } \overline{\Delta}_{\eta t} \leq \frac{1}{\sqrt{d}} \text{ and } i_2 \leq 4)$$

where we have $\overline{\Delta}_{\eta t} \leq \frac{1}{\sqrt{d}}$ because by Lemma E.3, we have that Assumption 5.3 holds as long as $\eta t \leq T_{*,\epsilon}$. In summary, we have shown that terms of the form Eq. (F.38) contribute at most $(\hat{\sigma}_{2,d}^2 + \hat{\sigma}_{4,d}^2)\|\tilde{\delta}_t\|_2^2$, up to constant factors, to the concentration error between $S_1$ and $M_2$.

**Contribution of Eq. (F.37) to Concentration Error** We expand Eq. (F.37) by using $\tilde{u}_t = \tilde{\delta}_t + \delta_{\eta t} + \bar{u}_{\eta t}$, $\hat{u}_{\eta t} = \delta_{\eta t} + \bar{u}_{\eta t}$ and $\tilde{u}'_t = \tilde{\delta}'_t + \delta'_{\eta t} + \bar{u}'_{\eta t}$. Thus, we can write Eq. (F.37) as a linear combination of terms of the form

$$(\tilde{\delta}_t \cdot x)^{1+i_1}(\tilde{\delta}'_t \cdot x)^{1+i_2}(\delta_{\eta t} \cdot x)^{i_3}(\delta'_{\eta t} \cdot x)^{i_4}(\bar{u}_{\eta t} \cdot x)^{i_5}(\bar{u}'_{\eta t} \cdot x)^{i_6} \quad \text{(F.47)}$$

where $i_1 + i_2 + i_3 + i_4 + i_5 + i_6 = 6$ (since the degrees of each of the monomials must be 8) and $i_3 \leq 3$ (since each of these terms originates from $(\tilde{u}'_t \cdot x)^{i_3} (\hat{u}'_{\eta t} \cdot x)^{i_4} (\tilde{u}_t \cdot x)^{i_2} (\tilde{\delta}_t \cdot x)(\tilde{\delta}'_t \cdot x)$ where $i_2 \leq 3$). Using Lemma I.7, we obtain a concentration bound for Eq. (F.47):

$$\left| \frac{1}{n} \sum_{i=1}^{n} (\tilde{\delta}_t \cdot x_i)^{1+i_1} (\tilde{\delta}'_t \cdot x_i)^{1+i_2} (\delta_{\eta t} \cdot x_i)^{i_3} (\delta'_{\eta t} \cdot x_i)^{i_4} (\bar{u}_{\eta t} \cdot x_i)^{i_5} (\bar{u}'_{\eta t} \cdot x_i)^{i_6} \right. \tag{F.48}$$

$$\left. - \mathop{\mathbb{E}}_{x \sim \mathbb{S}^{d-1}} \left[ (\tilde{\delta}_t \cdot x)^{1+i_1} (\tilde{\delta}'_t \cdot x)^{1+i_2} (\delta_{\eta t} \cdot x)^{i_3} (\delta'_{\eta t} \cdot x)^{i_4} (\bar{u}_{\eta t} \cdot x)^{i_5} (\bar{u}'_{\eta t} \cdot x)^{i_6} \right] \right| \tag{F.49}$$

$$\lesssim (\log d)^{O(1)} \frac{\|\tilde{\delta}_t\|_2^{1+i_1} \|\tilde{\delta}'_t\|_2^{1+i_2} \|\delta_{\eta t}\|_2^{i_3} \|\delta'_{\eta t}\|_2^{i_4}}{d^{1+(i_1+i_2+i_3+i_4+i_5+i_6)/2}} \left( \sqrt{\frac{d}{n}} + \frac{d^{1+(i_1+i_2+i_3+i_4)/2}}{n} + \frac{1}{d^{\Omega(\log d)}} \right). \tag{F.50}$$

As discussed in the previous case, after we consider the coefficients of $\sigma(t)$ and $\sigma'(t)$, we find that the term in Eq. (F.47) contributes

$$(\hat{\sigma}_{2,d}^2 + \hat{\sigma}_{4,d}^2)(\log d)^{O(1)} \|\tilde{\delta}_t\|_2^{1+i_1} \|\tilde{\delta}'_t\|_2^{1+i_2} \|\delta_{\eta t}\|_2^{i_3} \|\delta'_{\eta t}\|_2^{i_4} \left( \sqrt{\frac{d}{n}} + \frac{d^{1+(i_1+i_2+i_3+i_4)/2}}{n} + \frac{1}{d^{\Omega(\log d)}} \right). \tag{F.51}$$

Now, suppose $i_5 + i_6 \geq 2$. Then, $i_1 + i_2 + i_3 + i_4 \leq 4$, meaning the contribution to the concentration error is at most

$$(\hat{\sigma}_{2,d}^2 + \hat{\sigma}_{4,d}^2)\|\tilde{\delta}_t\|_2 \|\tilde{\delta}'_t\|_2 \leq (\hat{\sigma}_{2,d}^2 + \hat{\sigma}_{4,d}^2) \max_{\chi} \|\tilde{\delta}_t(\chi)\|_2^2 \tag{F.52}$$

using the fact that $n \gtrsim d^3 (\log d)^C$ for a sufficiently large constant $C$, and $\|\tilde{\delta}_t\|_2 \lesssim 1$ and $\|\delta_{\eta t}\|_2 \lesssim 1$. On the other hand, if $i_5 + i_6 \leq 1$, then $i_1 + i_2 + i_3 + i_4 \geq 5$, and since $i_3 \leq 3$, this implies that $i_1 + i_2 + i_4 \geq 2$. In this case, the bound is at most

$$(\hat{\sigma}_{2,d}^2 + \hat{\sigma}_{4,d}^2)(\log d)^{O(1)} \|\tilde{\delta}_t\|_2^{1+i_1} \|\tilde{\delta}'_t\|_2^{1+i_2} \|\delta_{\eta t}\|_2^{i_3} \|\delta'_{\eta t}\|_2^{i_4} \left( \sqrt{\frac{d}{n}} + \frac{d^{1+(i_1+i_2+i_3+i_4)/2}}{n} + \frac{1}{d^{\Omega(\log d)}} \right) \tag{F.53}$$

$$\lesssim (\hat{\sigma}_{2,d}^2 + \hat{\sigma}_{4,d}^2)\|\tilde{\delta}_t\|_2^{1+i_1} \|\tilde{\delta}'_t\|_2^{1+i_2} \|\delta_{\eta t}\|_2^{i_3} \|\delta'_{\eta t}\|_2^{i_4} d$$
$$\text{(B.c. } i_1 + i_2 + i_3 + i_4 \leq 6 \text{ and } n \gtrsim d^3 (\log d)^C)$$

$$\lesssim (\hat{\sigma}_{2,d}^2 + \hat{\sigma}_{4,d}^2) \max_{\chi} \|\tilde{\delta}_t(\chi)\|_2^2 \cdot \frac{1}{d^{(i_1+i_2)/2}} \|\delta'_{\eta t}\|_2^{i_4} d .$$
$$\text{(B.c. } \max_{\chi} \|\tilde{\delta}_t(\chi)\|_2 \leq \tfrac{1}{\sqrt{d}} \text{ and } \|\delta_{\eta t}\|_2 \lesssim 1)$$

Thus, if $i_1 + i_2 \geq 2$, then the contribution of this term to the overall concentration error is at most $(\hat{\sigma}_{2,d}^2 + \hat{\sigma}_{4,d}^2) \max_{\chi} \|\tilde{\delta}_t(\chi)\|_2^2$. On the other hand, if $i_4 = 1$, then we can take the average of $\|\delta'_{\eta t}\|_2$ across $\chi$, and by Lemma E.3, we have that Assumption 5.3 holds for $\eta t \leq T_{*,\epsilon}$, meaning that $\mathbb{E}_\chi \|\delta'_{\eta t}\|_2 \leq \overline{\Delta}_{\eta t} \leq \frac{1}{\sqrt{d}}$. In this case, the contribution of this term to the overall concentration error is at most $(\hat{\sigma}_{2,d}^2 + \hat{\sigma}_{4,d}^2) \max_{\chi} \|\tilde{\delta}_t(\chi)\|_2^2 \cdot \frac{1}{d^{(i_1+i_2+i_4)/2}} d = (\hat{\sigma}_{2,d}^2 + \hat{\sigma}_{4,d}^2) \max_{\chi} \|\tilde{\delta}_t(\chi)\|_2^2$. Finally, when $i_4 \geq 2$, we can take the average of $\|\delta'_{\eta t}\|_2^2$ over $\chi$ — we have $\mathbb{E}_\chi \|\delta'_{\eta t}\|_2^2 \leq \frac{1}{d}$, and therefore the contribution of this term to the overall concentration error is at most

$$(\hat{\sigma}_{2,d}^2 + \hat{\sigma}_{4,d}^2) \max_{\chi} \|\tilde{\delta}_t(\chi)\|_2^2 \cdot \frac{1}{d^{1+(i_1+i_2)/2}} d \leq (\hat{\sigma}_{2,d}^2 + \hat{\sigma}_{4,d}^2) \max_{\chi} \|\tilde{\delta}_t(\chi)\|_2^2 . \tag{F.54}$$

In all cases, the contribution of Eq. (F.37) to the overall concentration error between $S_1$ and $M_1$ is at most

$$(\hat{\sigma}_{2,d}^2 + \hat{\sigma}_{4,d}^2) \max_{\chi} \|\tilde{\delta}_t(\chi)\|_2^2 . \tag{F.55}$$

**Overall Concentration Error**

In summary, the overall concentration error between $S_1$ and $M_1$ is that of $G(x)$. We have expanded $G(x)$ into various monomials of dot products involving $x$, and shown that each monomial contributes at most $(\hat{\sigma}_{2,d}^2 + \hat{\sigma}_{4,d}^2) \max_{\chi} \|\tilde{\delta}_t(\chi)\|_2^2$ to the concentration error. Thus, with high probability,

$$|M_1 - S_1| \lesssim (\hat{\sigma}_{2,d}^2 + \hat{\sigma}_{4,d}^2) \max_{\chi} \|\tilde{\delta}_t(\chi)\|_2^2 . \tag{F.56}$$

In order to bound $|M_2 - S_2|$, an almost identical, but simpler, argument can be used, with $y(x)$ in the place of $f_{\tilde{\rho}_t}(x)$ or $f_{\hat{\rho}_{\eta t}}(x)$. In the modified argument, $\tilde{u}'_t$, $\hat{u}'_{\eta t}$ and $\bar{u}'_{\eta t}$ will all be replaced with $e_1$. Thus, in the analysis of Eq. (F.40), we can replace $\bar{u}'_{\eta t}$ with $e_1$ and assume that $i_2 = 0$ (since otherwise this term becomes $0$ due to the $\delta'_{\eta t} \cdot x$ factor) — we can then proceed similarly to the subcase where $i_2 \leq 1$ in the analysis of Eq. (F.40). Additionally, the term analogous to Eq. (F.47) does not arise in the case where $f_{\tilde{\rho}_t}(x)$ and $f_{\hat{\rho}_{\eta t}}(x)$ are replaced by $y(x)$, since $\tilde{u}'_t$ and $\hat{u}'_{\eta t}$ are both replaced by $e_1$, so the corresponding term is $0$. We also note that the Legendre coefficients of $y(x)$ have absolute values which are less than those of $\sigma$. Thus, we have

$$|M_1 - S_1| + |M_2 - S_2| \leq (\hat{\sigma}_{2,d}^2 + \hat{\sigma}_{4,d}^2) \max_{\chi} \|\tilde{\delta}_t(\chi)\|_2^2. \tag{F.57}$$

**Bounding the Expectation** To complete the proof, it suffices to bound $|M_1 - M_2|$. We have that

$$M_1 - M_2 = \nabla_{\tilde{u}_t} L(\tilde{\rho}_t) \cdot \tilde{\delta}_t - \nabla_{\hat{u}_{\eta t}} L(\hat{\rho}_{\eta t}) \cdot \tilde{\delta}_t \tag{F.58}$$

and by Lemma F.1, we have

$$|M_1 - M_2| \lesssim (\hat{\sigma}_{2,d}^2 + \hat{\sigma}_{4,d}^2)\Big( \underset{(\tilde{u}'_t, \hat{u}'_{\eta t}) \sim (\tilde{\rho}_t, \hat{\rho}_{\eta t})}{\mathbb{E}} \Big[ \|\tilde{u}'_t - \hat{u}'_{\eta t}\|_2^2 \Big]^{1/2} + \|\tilde{u}_t - \hat{u}_{\eta t}\|_2 \Big) \|\tilde{\delta}_t\|_2 \tag{F.59}$$

$$\lesssim (\hat{\sigma}_{2,d}^2 + \hat{\sigma}_{4,d}^2) \max_{\chi} \|\tilde{\delta}_t(\chi)\|_2^2 \tag{F.60}$$

Combining this with our bound on $|M_1 - S_1| + |M_2 - S_2|$, we find that

$$|(\mathrm{grad}_{\tilde{u}_t} \widehat{L}(\tilde{\rho}_t) - \mathrm{grad}_{\hat{u}_{\eta t}} \widehat{L}(\hat{\rho}_{\eta t})) \cdot \tilde{\delta}_t| \lesssim (\hat{\sigma}_{2,d}^2 + \hat{\sigma}_{4,d}^2) \max_{\chi} \|\tilde{\delta}_t(\chi)\|_2^2 \tag{F.61}$$

as desired. $\qquad \square$

**Lemma F.3.** *Suppose we are in the setting of Theorem 3.5. Suppose $s, t \geq 0$. Then,*

$$\|\mathrm{grad}_{\hat{u}_t} \widehat{L}(\hat{\rho}_t) - \mathrm{grad}_{\hat{u}_s} \widehat{L}(\hat{\rho}_s)\|_2 \leq (\hat{\sigma}_{2,d}^2 + \hat{\sigma}_{4,d}^2) d^4 \cdot \max_{i \in [m]} \|\hat{u}_t(\chi_i) - \hat{u}_s(\chi_i)\|_2 \tag{F.62}$$

*Proof.* We have

$$\|\mathrm{grad}_{\hat{u}_t} \widehat{L}(\hat{\rho}_t) - \mathrm{grad}_{\hat{u}_s} \widehat{L}(\hat{\rho}_s)\|_2 \tag{F.63}$$

$$= \|(I - \hat{u}_t \hat{u}_t^\top) \nabla_{\hat{u}_t} \widehat{L}(\hat{\rho}_t) - (I - \hat{u}_s \hat{u}_s^\top) \nabla_{\hat{u}_s} \widehat{L}(\hat{\rho}_s)\|_2 \tag{F.64}$$

$$= \Big\| (I - \hat{u}_t \hat{u}_t^\top) \cdot \frac{1}{n} \sum_{i=1}^{n} (f_{\hat{\rho}_t}(x_i) - y(x_i)) \sigma'(\hat{u}_t^\top x_i) x_i \tag{F.65}$$

$$- (I - \hat{u}_s \hat{u}_s^\top) \cdot \frac{1}{n} \sum_{i=1}^{n} (f_{\hat{\rho}_s}(x_i) - y(x_i)) \sigma'(\hat{u}_s^\top x_i) x_i \Big\|_2 \tag{F.66}$$

$$\leq \frac{1}{n} \sum_{i=1}^{n} \|(f_{\hat{\rho}_t}(x_i) - y(x_i)) \sigma'(\hat{u}_t^\top x_i)(I - \hat{u}_t \hat{u}_t^\top) x_i \tag{F.67}$$

$$- (f_{\hat{\rho}_s}(x_i) - y(x_i)) \sigma'(\hat{u}_s^\top x_i)(I - \hat{u}_s \hat{u}_s^\top) x_i \|_2. \tag{F.68}$$

We bound the right-hand side by several applications of the triangle inequality. First, observe that for any $x \in \mathbb{S}^{d-1}$, we have

$$|f_{\hat{\rho}_t}(x) - f_{\hat{\rho}_s}(x)| \leq \frac{1}{m} \sum_{j=1}^{m} |\sigma(\hat{u}_t(\chi_i) \cdot x) - \sigma(\hat{u}_s(\chi_i) \cdot x)| \tag{F.69}$$

$$\lesssim \frac{1}{m} \sum_{j=1}^{m} (|\hat{\sigma}_{2,d}| \sqrt{N_{2,d}} + |\hat{\sigma}_{4,d}| \sqrt{N_{4,d}}) |\hat{u}_t(\chi_i) \cdot x - \hat{u}_s(\chi_i) \cdot x|$$

$$\text{(By Lemma E.12)}$$

$$\lesssim \frac{(|\hat{\sigma}_{2,d}| \sqrt{N_{2,d}} + |\hat{\sigma}_{4,d}| \sqrt{N_{4,d}})}{m} \sum_{j=1}^{m} \|\hat{u}_t(\chi_i) - \hat{u}_s(\chi_i)\|_2 \tag{F.70}$$

$$\lesssim (|\hat{\sigma}_{2,d}| \sqrt{N_{2,d}} + |\hat{\sigma}_{4,d}| \sqrt{N_{4,d}}) \max_{i \in [m]} \|\hat{u}_t(\chi_i) - \hat{u}_s(\chi_i)\|_2. \tag{F.71}$$

Additionally, for $\hat{u}_t$ and $\hat{u}_s$ with the same initialization $\chi$, we have

$$|\sigma'(\hat{u}_t^\top x) - \sigma'(\hat{u}_s^\top x)| \lesssim (|\hat{\sigma}_{2,d}|\sqrt{N_{2,d}} + |\hat{\sigma}_{4,d}|\sqrt{N_{4,d}}) \cdot |\hat{u}_t^\top x - \hat{u}_s^\top x| \qquad \text{(By Lemma E.12)}$$

$$\lesssim (|\hat{\sigma}_{2,d}|\sqrt{N_{2,d}} + |\hat{\sigma}_{4,d}|\sqrt{N_{4,d}})\|\hat{u}_t - \hat{u}_s\|_2. \qquad \text{(F.72)}$$

$$\lesssim (|\hat{\sigma}_{2,d}|\sqrt{N_{2,d}} + |\hat{\sigma}_{4,d}|\sqrt{N_{4,d}}) \cdot \max_{i \in [m]}\|\hat{u}_t(\chi_i) - \hat{u}_s(\chi_i)\|_2. \qquad \text{(F.73)}$$

Also, for $\hat{u}_t$ and $\hat{u}_s$ with the same initialization $\chi$, we have

$$\|(I - \hat{u}_t\hat{u}_t^\top)x_i - (I - \hat{u}_s\hat{u}_s^\top)x_i\|_2 = \|(\hat{u}_t \cdot x_i)\hat{u}_t - (\hat{u}_s \cdot x_i)\hat{u}_s\|_2 \qquad \text{(F.74)}$$

$$\leq \|(\hat{u}_t \cdot x_i - \hat{u}_s \cdot x_i)\hat{u}_t\|_2 + |\hat{u}_s \cdot x_i|\|\hat{u}_t - \hat{u}_s\|_2 \qquad \text{(F.75)}$$

$$\lesssim \max_{i \in [m]}\|\hat{u}_t(\chi_i) - \hat{u}_s(\chi_i)\|_2. \qquad \text{(F.76)}$$

Finally, for $x \in \mathbb{S}^{d-1}$ and any distribution $\rho$ on $\mathbb{S}^{d-1}$, we have

$$|f_\rho(x) - y(x)| \lesssim |\hat{\sigma}_{2,d}|\sqrt{N_{2,d}} + |\hat{\sigma}_{4,d}|\sqrt{N_{4,d}} \qquad \text{(F.77)}$$

since we can bound $|f_\rho(x)|$ using Lemma E.12, and because (ignoring $\hat{h}_{0,d}$ since it is equal to $\hat{\sigma}_{0,d}$) $y(x) = \gamma_2\hat{\sigma}_{2,d}P_{2,d} + \gamma_4\hat{\sigma}_{4,d}P_{4,d}$, and using Assumption 3.2.

Thus, we can simplify the right-hand side of Eq. (F.68) using the triangle inequality, and combining Eq. (F.71), Eq. (F.73), Eq. (F.76) and Eq. (F.77):

$$\|(f_{\hat{\rho}_t}(x_i) - y(x_i))\sigma'(\hat{u}_t^\top x_i)(I - \hat{u}_t\hat{u}_t^\top)x_i - (f_{\hat{\rho}_s}(x_i) - y(x_i))\sigma'(\hat{u}_s^\top x_i)(I - \hat{u}_s\hat{u}_s^\top)x_i\|_2 \quad \text{(F.78)}$$

$$\leq |f_{\hat{\rho}_t}(x_i) - f_{\hat{\rho}_s}(x_i)|\|\sigma'(\hat{u}_t^\top x_i)(I - \hat{u}_t\hat{u}_t^\top)x_i\|_2 \qquad \text{(F.79)}$$

$$+ |f_{\hat{\rho}_s}(x_i) - y(x_i)||\sigma'(\hat{u}_t^\top x_i) - \sigma'(\hat{u}_s^\top x_i)|\|(I - \hat{u}_t\hat{u}_t^\top)x_i\|_2 \qquad \text{(F.80)}$$

$$+ |f_{\hat{\rho}_s}(x_i) - y(x_i)||\sigma'(\hat{u}_s^\top x_i)|\|(I - \hat{u}_t\hat{u}_t^\top)x_i - (I - \hat{u}_s\hat{u}_s^\top)x_i\|_2 \qquad \text{(F.81)}$$

$$\lesssim (\hat{\sigma}_{2,d}^2 N_{2,d} + \hat{\sigma}_{4,d}^2 N_{4,d}) \cdot \max_{i \in [m]}\|\hat{u}_t(\chi_i) - \hat{u}_s(\chi_i)\|_2 \qquad \text{(F.82)}$$

$$+ |f_{\hat{\rho}_s}(x_i) - y(x_i)||\sigma'(\hat{u}_t^\top x_i) - \sigma'(\hat{u}_s^\top x_i)|\|(I - \hat{u}_t\hat{u}_t^\top)x_i\|_2 \qquad \text{(F.83)}$$

$$+ |f_{\hat{\rho}_s}(x_i) - y(x_i)||\sigma'(\hat{u}_s^\top x_i)|\|(I - \hat{u}_t\hat{u}_t^\top)x_i - (I - \hat{u}_s\hat{u}_s^\top)x_i\|_2$$
$$\text{(By Eq. (F.71), bounding } |\sigma'(\hat{u}_t^\top x_i)| \text{ by Lemma E.12)}$$

$$\lesssim (\hat{\sigma}_{2,d}^2 N_{2,d} + \hat{\sigma}_{4,d}^2 N_{4,d}) \cdot \max_{i \in [m]}\|\hat{u}_t(\chi_i) - \hat{u}_s(\chi_i)\|_2 \qquad \text{(F.84)}$$

$$+ (\hat{\sigma}_{2,d}^2 N_{2,d} + \hat{\sigma}_{4,d}^2 N_{4,d}) \cdot \max_{i \in [m]}\|\hat{u}_t(\chi_i) - \hat{u}_s(\chi_i)\|_2 \qquad \text{(F.85)}$$

$$+ |f_{\hat{\rho}_s}(x_i) - y(x_i)||\sigma'(\hat{u}_s^\top x_i)|\|(I - \hat{u}_t\hat{u}_t^\top)x_i - (I - \hat{u}_s\hat{u}_s^\top)x_i\|_2$$
$$\text{(By Eq. (F.77) and Eq. (F.73))}$$

$$\lesssim (\hat{\sigma}_{2,d}^2 N_{2,d} + \hat{\sigma}_{4,d}^2 N_{4,d}) \cdot \max_{i \in [m]}\|\hat{u}_t(\chi_i) - \hat{u}_s(\chi_i)\|_2 \qquad \text{(F.86)}$$

$$+ (\hat{\sigma}_{2,d}^2 N_{2,d} + \hat{\sigma}_{4,d}^2 N_{4,d}) \cdot \max_{i \in [m]}\|\hat{u}_t(\chi_i) - \hat{u}_s(\chi_i)\|_2 \qquad \text{(F.87)}$$

$$+ (\hat{\sigma}_{2,d}^2 N_{2,d} + \hat{\sigma}_{4,d}^2 N_{4,d}) \cdot \max_{i \in [m]}\|\hat{u}_t(\chi_i) - \hat{u}_s(\chi_i)\|_2$$
$$\text{(By Eq. (F.76), Eq. (F.77) and Lemma E.12)}$$

$$\lesssim (\hat{\sigma}_{2,d}^2 N_{2,d} + \hat{\sigma}_{4,d}^2 N_{4,d}) \cdot \max_{i \in [m]}\|\hat{u}_t(\chi_i) - \hat{u}_s(\chi_i)\|_2. \qquad \text{(F.88)}$$

Therefore, Eq. (F.68) gives

$$\|\text{grad}_{\hat{u}_t}\widehat{L}(\hat{\rho}_t) - \text{grad}_{\hat{u}_s}\widehat{L}(\hat{\rho}_s)\|_2 \lesssim (\hat{\sigma}_{2,d}^2 N_{2,d} + \hat{\sigma}_{4,d}^2 N_{4,d}) \cdot \max_{i \in [m]}\|\hat{u}_t(\chi_i) - \hat{u}_s(\chi_i)\|_2 \qquad \text{(F.89)}$$

as desired. $\qquad\square$

*Proof of Theorem 3.5.* To show that projected gradient descent achieves low population loss, we show that it does not diverge far from projected gradient descent. Suppose $\hat{\rho}_0$ and $\tilde{\rho}_0$ are both equal to $\mathrm{unif}(\chi_1, \ldots, \chi_m\})$. Then, by induction over $t$, we bound the difference between $\tilde{u}_t(\chi)$ and $\hat{u}_{\eta t}(\chi)$ for all $\chi \in \{\chi_1, \ldots, \chi_m\}$. At $t = 0$, we have $\tilde{u}_0(\chi) = \hat{u}_0(\chi)$. Let $t$ be a nonnegative integer, and assume that for all $s \le t$ we have $\max_\chi \|\tilde{\delta}_t(\chi)\|_2 \le \frac{1}{\sqrt{d}}$.

By the definition of projected gradient descent, we have

$$\tilde{u}_{t+1}(\chi) = \frac{\tilde{u}_t(\chi) - \eta \cdot \mathrm{grad}_{\tilde{u}_t(\chi)} \widehat{L}(\tilde{\rho}_t)}{\|\tilde{u}_t(\chi) - \eta \cdot \mathrm{grad}_{\tilde{u}_t(\chi)} \widehat{L}(\tilde{\rho}_t)\|_2} \,. \tag{F.90}$$

Note that $\|\tilde{u}_t - \eta \cdot \mathrm{grad}_{\tilde{u}_t} \widehat{L}(\tilde{\rho}_t)\|_2 \ge \|\tilde{u}_t\|_2$, since $\mathrm{grad}_{\tilde{u}_t} \widehat{L}(\tilde{\rho}_t)$ is orthogonal to $\tilde{u}_t$. Thus, letting $\Pi$ denote the projection onto the $\ell_2$ unit ball, we have

$$\|\tilde{u}_{t+1} - \hat{u}_{\eta(t+1)}\|_2 = \|\Pi(\tilde{u}_t - \eta \cdot \mathrm{grad}_{\tilde{u}_t} \widehat{L}(\tilde{\rho}_t)) - \Pi(\hat{u}_{\eta(t+1)})\|_2 \tag{F.91}$$

$$\le \|\tilde{u}_t - \eta \cdot \mathrm{grad}_{\tilde{u}_t} \widehat{L}(\tilde{\rho}_t) - \hat{u}_{\eta(t+1)}\|_2 \,. \tag{F.92}$$

Also, we have

$$\hat{u}_{\eta(t+1)}(\chi) = \hat{u}_{\eta t}(\chi) - \int_{\eta t}^{\eta(t+1)} \mathrm{grad}_{\hat{u}_s(\chi)} \widehat{L}(\hat{\rho}_s) ds \,. \tag{F.93}$$

Thus,

$$\|\tilde{u}_t - \eta \cdot \mathrm{grad}_{\tilde{u}_t} \widehat{L}(\tilde{\rho}_t) - \hat{u}_{\eta(t+1)}\|_2^2 = \left\| \tilde{u}_t - \eta \cdot \mathrm{grad}_{\tilde{u}_t} \widehat{L}(\tilde{\rho}_t) - \hat{u}_{\eta t} + \int_{\eta t}^{\eta(t+1)} \mathrm{grad}_{\hat{u}_s} \widehat{L}(\hat{\rho}_s) ds \right\|_2^2$$
$$\tag{F.94}$$

$$= \|\tilde{u}_t - \hat{u}_{\eta t}\|_2^2 + \left\| \eta \cdot \mathrm{grad}_{\tilde{u}_t} \widehat{L}(\tilde{\rho}_t) - \int_{\eta t}^{\eta(t+1)} \mathrm{grad}_{\hat{u}_s} \widehat{L}(\hat{\rho}_s) ds \right\|_2^2$$
$$\tag{F.95}$$

$$+ 2(\tilde{u}_t - \hat{u}_{\eta t}) \cdot \int_{\eta t}^{\eta(t+1)} (\mathrm{grad}_{\tilde{u}_t} \widehat{L}(\tilde{\rho}_t) - \mathrm{grad}_{\hat{u}_s} \widehat{L}(\hat{\rho}_s)) ds$$
$$\tag{F.96}$$

$$= \|\tilde{\delta}_t\|_2^2 + O\left( \eta^2 (\hat{\sigma}_{2,d}^2 + \hat{\sigma}_{4,d}^2)^2 d^8 \right) \tag{F.97}$$

$$+ 2\eta \tilde{\delta}_t \cdot (\mathrm{grad}_{\tilde{u}_t} \widehat{L}(\tilde{\rho}_t) - \mathrm{grad}_{\hat{u}_{\eta t}} \widehat{L}(\hat{\rho}_{\eta t})) \tag{F.98}$$

$$+ 2\tilde{\delta}_t \cdot \int_{\eta t}^{\eta(t+1)} (\mathrm{grad}_{\hat{u}_{\eta t}} \widehat{L}(\hat{\rho}_{\eta t}) - \mathrm{grad}_{\hat{u}_s} \widehat{L}(\hat{\rho}_s)) ds$$
$$\text{(By Lemma E.14)}$$

$$\le \|\tilde{\delta}_t\|_2^2 + O\left( \eta^2 (\hat{\sigma}_{2,d}^2 + \hat{\sigma}_{4,d}^2)^2 d^8 \right) \tag{F.99}$$

$$+ O\left( \eta (\hat{\sigma}_{2,d}^2 + \hat{\sigma}_{4,d}^2) \max_\chi \|\tilde{\delta}_t(\chi)\|_2^2 \right) \tag{F.100}$$

$$+ 2\tilde{\delta}_t \cdot \int_{\eta t}^{\eta(t+1)} (\mathrm{grad}_{\hat{u}_{\eta t}} \widehat{L}(\hat{\rho}_{\eta t}) - \mathrm{grad}_{\hat{u}_s} \widehat{L}(\hat{\rho}_s)) ds \,.$$
$$\text{(By Lemma F.2)}$$

Further simplifying using the Cauchy-Schwarz inequality, we have

$$\|\tilde{u}_t - \eta \cdot \mathrm{grad}_{\tilde{u}_t} \widehat{L}(\tilde{\rho}_t) - \hat{u}_{\eta(t+1)}\|_2^2 \leq \|\tilde{\delta}_t\|_2^2 + O\left(\eta^2(\hat{\sigma}_{2,d}^2 + \hat{\sigma}_{4,d}^2)^2 d^8\right) \tag{F.101}$$

$$+ O\left(\eta(\hat{\sigma}_{2,d}^2 + \hat{\sigma}_{4,d}^2) \max_{\chi} \|\tilde{\delta}_t(\chi)\|_2^2\right) \tag{F.102}$$

$$+ 2\eta\|\tilde{\delta}_t\|_2 \max_{s\in[\eta t, \eta(t+1)]} \|\mathrm{grad}_{\hat{u}_{\eta t}} \widehat{L}(\hat{\rho}_{\eta t}) - \mathrm{grad}_{\hat{u}_s} \widehat{L}(\hat{\rho}_s)\|_2 \tag{F.103}$$

$$\leq \|\tilde{\delta}_t\|_2^2 + O\left(\eta^2(\hat{\sigma}_{2,d}^2 + \hat{\sigma}_{4,d}^2)^2 d^8\right) \tag{F.104}$$

$$+ O\left(\eta(\hat{\sigma}_{2,d}^2 + \hat{\sigma}_{4,d}^2) \max_{\chi} \|\tilde{\delta}_t(\chi)\|_2^2\right) \tag{F.105}$$

$$+ 2\eta\|\tilde{\delta}_t\|_2(\hat{\sigma}_{2,d}^2 + \hat{\sigma}_{4,d}^2)d^4 \tag{F.106}$$

$$\cdot \max_{s\in[\eta t, \eta(t+1)]} \max_{i\in[m]} \|\hat{u}_s(\chi_i) - \hat{u}_{\eta t}(\chi_i)\|_2$$

$$\text{(By Lemma F.3)}$$

$$\leq \|\tilde{\delta}_t\|_2^2 + O\left(\eta^2(\hat{\sigma}_{2,d}^2 + \hat{\sigma}_{4,d}^2)^2 d^8\right) \tag{F.107}$$

$$+ O\left(\eta(\hat{\sigma}_{2,d}^2 + \hat{\sigma}_{4,d}^2) \max_{\chi} \|\tilde{\delta}_t(\chi)\|_2^2\right) \tag{F.108}$$

$$+ 2\eta\|\tilde{\delta}_t\|_2(\hat{\sigma}_{2,d}^2 + \hat{\sigma}_{4,d}^2)d^4 \cdot \eta(\hat{\sigma}_{2,d}^2 + \hat{\sigma}_{4,d}^2)d^4 \tag{F.109}$$

$$\text{(By Lemma E.14)}$$

$$\leq \|\tilde{\delta}_t\|_2^2 + O\left(\eta^2(\hat{\sigma}_{2,d}^2 + \hat{\sigma}_{4,d}^2)^2 d^8\right) \tag{F.109}$$

$$+ O\left(\eta(\hat{\sigma}_{2,d}^2 + \hat{\sigma}_{4,d}^2) \max_{\chi} \|\tilde{\delta}_t(\chi)\|_2^2\right). \tag{F.110}$$

Thus, we have

$$\max_{\chi} \|\tilde{\delta}_{t+1}(\chi)\|_2^2 \leq \left(1 + O\left(\eta(\hat{\sigma}_{2,d}^2 + \hat{\sigma}_{4,d}^2)\right)\right) \max_{\chi} \|\tilde{\delta}_t(\chi)\|_2^2 + O\left(\eta^2(\hat{\sigma}_{2,d}^2 + \hat{\sigma}_{4,d}^2)^2 d^8\right) \tag{F.111}$$

which means that for some universal constant $C$,

$$\max_{\chi} \|\tilde{\delta}_{t+1}(\chi)\|_2^2 \lesssim \sum_{i=0}^{t} \left(1 + C\eta(\hat{\sigma}_{2,d}^2 + \hat{\sigma}_{4,d}^2)\right)^i \cdot \eta^2(\hat{\sigma}_{2,d}^2 + \hat{\sigma}_{4,d}^2)^2 d^8 \tag{F.112}$$

$$\lesssim \eta^2(\hat{\sigma}_{2,d}^2 + \hat{\sigma}_{4,d}^2)^2 d^8 \cdot \frac{\left(1 + C\eta(\hat{\sigma}_{2,d}^2 + \hat{\sigma}_{4,d}^2)\right)^{t+1} - 1}{\left(1 + C\eta(\hat{\sigma}_{2,d}^2 + \hat{\sigma}_{4,d}^2)\right) - 1} \tag{F.113}$$

$$\lesssim \eta^2(\hat{\sigma}_{2,d}^2 + \hat{\sigma}_{4,d}^2)^2 d^8 \cdot \frac{\left(1 + C\eta(\hat{\sigma}_{2,d}^2 + \hat{\sigma}_{4,d}^2)\right)^{t+1} - 1}{C\eta(\hat{\sigma}_{2,d}^2 + \hat{\sigma}_{4,d}^2)} \tag{F.114}$$

$$\lesssim \eta(\hat{\sigma}_{2,d}^2 + \hat{\sigma}_{4,d}^2)d^8 \cdot \left(1 + C\eta(\hat{\sigma}_{2,d}^2 + \hat{\sigma}_{4,d}^2)\right)^{t+1} \tag{F.115}$$

$$\lesssim \eta(\hat{\sigma}_{2,d}^2 + \hat{\sigma}_{4,d}^2)d^8 \cdot e^{C\eta(\hat{\sigma}_{2,d}^2 + \hat{\sigma}_{4,d}^2)(t+1)}. \tag{F.116}$$

Thus, as long as $\eta = \frac{1}{(\hat{\sigma}_{2,d}^2 + \hat{\sigma}_{4,d}^2)d^B}$ for a sufficiently large universal constant $B$, and choosing $\eta$ so that $T_{*,\epsilon}$ is an integer multiple of $\eta$, we have $\max_{\chi} \|\tilde{\delta}_{t+1}(\chi)\|_2 \leq \frac{1}{\sqrt{d}}$, because if $t \leq \frac{T_{*,\epsilon}}{\eta}$, then

$$\max_{\chi} \|\tilde{\delta}_{t+1}(\chi)\|_2^2 \leq \eta(\hat{\sigma}_{2,d}^2 + \hat{\sigma}_{4,d}^2)d^8 \cdot e^{C(\hat{\sigma}_{2,d}^2 + \hat{\sigma}_{4,d}^2)T_{*,\epsilon}} \tag{F.117}$$

$$\leq \eta(\hat{\sigma}_{2,d}^2 + \hat{\sigma}_{4,d}^2)d^8 \cdot e^{O(\log d)} \tag{F.118}$$

$$\leq \eta(\hat{\sigma}_{2,d}^2 + \hat{\sigma}_{4,d}^2)d^{O(1)}. \tag{F.119}$$

This completes the induction, and therefore for all $t \leq \frac{T_{*,\epsilon}}{\eta}$, we have $\max_\chi \|\tilde{\delta}_t(\chi)\|_2 \leq \frac{1}{\sqrt{d}}$. Additionally, by Lemma E.16, and using the coupling between $\tilde{\rho}_t$ and $\hat{\rho}_{\eta t}$ which is the joint distribution of $(\tilde{u}_t(\chi), \hat{u}_{\eta t}(\chi))$, we have

$$\mathbb{E}_{x \sim \mathbb{S}^{d-1}}[(f_{\tilde{\rho}_t}(x) - f_{\hat{\rho}_{\eta t}}(x))^2] \lesssim (\hat{\sigma}_{2,d}^2 + \hat{\sigma}_{4,d}^2) \max_\chi \|\tilde{\delta}_t(\chi)\|_2^2 \lesssim \frac{\hat{\sigma}_{2,d}^2 + \hat{\sigma}_{4,d}^2}{d} \lesssim (\hat{\sigma}_{2,d}^2 + \hat{\sigma}_{4,d}^2)\epsilon \tag{F.120}$$

which completes the proof. $\qquad \square$

## G  Missing Proofs in Section 3

### G.1  Proof of Lemma 3.1

In this section, we prove sufficient and necessary conditions for $\gamma_2$ and $\gamma_4$ when the population loss is 0.

**Proposition G.1** (Necessary condition). *Let $f_\rho(x) = \mathbb{E}_{u \sim \rho}[\sigma(u^\top x)]$ be a (not necessarily symmetric nor rotational invariant) two-layer neural network and $y(x) = h(e_1^\top x)$ the target function. Define $\gamma_k = \frac{\hat{h}_{k,d}}{\hat{\sigma}_{k,d}}, \forall k \geq 0$. Suppose $\mathbb{E}_{x \sim \mathbb{S}^{d-1}}(f_\rho(x) - y(x))^2 = 0$ and $\hat{\sigma}_{2,d} \neq 0, \hat{\sigma}_{4,d} \neq 0$, we have*

$$\gamma_2^2 \leq \gamma_4 + O\left(\frac{1}{\sqrt{d}}\right), \tag{G.1}$$

$$\gamma_4 \leq \gamma_2 + O\left(\frac{1}{\sqrt{d}}\right), \tag{G.2}$$

$$\gamma_2 \leq 1. \tag{G.3}$$

*Proof.* Similar to the proof of Lemma 4.2, we have

$$\mathbb{E}_{x \sim \mathbb{S}^{d-1}}[(f_\rho(x) - h(x^\top e_1))^2] = \mathbb{E}_{x \sim \mathbb{S}^{d-1}}\left(\mathbb{E}_{u \sim \rho}[\sigma(u^\top x)] - h(x^\top e_1)\right)^2 \tag{G.4}$$

$$= \mathbb{E}_{x \sim \mathbb{S}^{d-1}}\left(\mathbb{E}_{u \sim \rho}\left[\sum_{k=0}^\infty \hat{\sigma}_{k,d}\overline{P_{k,d}}(u^\top x)\right] - \sum_{k=0}^\infty \hat{h}_{k,d}\overline{P}_{k,d}(x^\top e_1)\right)^2 \tag{G.5}$$

$$= \mathbb{E}_{x \sim \mathbb{S}^{d-1}}\left(\sum_{k=0}^\infty \left(\hat{\sigma}_{k,d}\mathbb{E}_{u \sim \rho}[\overline{P}_{k,d}(u^\top x)] - \hat{h}_{k,d}\overline{P}_{k,d}(x^\top e_1)\right)\right)^2. \tag{G.6}$$

Continuing the equation by invoking Lemma C.5, we get

$$\mathbb{E}_{x \sim \mathbb{S}^{d-1}}\left(\sum_{k=0}^\infty \left(\hat{\sigma}_{k,d}\mathbb{E}_{u \sim \rho}[\overline{P}_{k,d}(u^\top x)] - \hat{h}_{k,d}\overline{P}_{k,d}(x^\top e_1)\right)\right)^2 \tag{G.7}$$

$$= \sum_{k=0}^\infty \mathbb{E}_{x \sim \mathbb{S}^{d-1}}\left(\hat{\sigma}_{k,d}\mathbb{E}_{u \sim \rho}[\overline{P}_{k,d}(u^\top x)] - \hat{h}_{k,d}\overline{P}_{k,d}(x^\top e_1)\right)^2 \tag{G.8}$$

$$= \sum_{k=0}^\infty \hat{\sigma}_{k,d}^2\left(\mathbb{E}_{u,u' \sim \rho}[P_{k,d}(u^\top u') - 2\gamma_k P_{k,d}(u^\top e_1) + \gamma_k^2 P_{k,d}(e_1^\top e_1)]\right)^2. \tag{G.9}$$

Therefore, $\mathbb{E}_{x \sim \mathbb{S}^{d-1}}[(f_\rho(x) - h(x^\top e_1))^2] = 0$ implies that

$$\mathbb{E}_{u,u' \sim \rho}[P_{2,d}(u^\top u') - 2\gamma_2 P_{2,d}(u^\top e_1) + \gamma_2^2 P_{2,d}(e_1^\top e_1)] = 0, \tag{G.10}$$

$$\mathbb{E}_{u,u' \sim \rho}[P_{4,d}(u^\top u') - 2\gamma_4 P_{4,d}(u^\top e_1) + \gamma_4^2 P_{4,d}(e_1^\top e_1)] = 0. \tag{G.11}$$

First we prove Eq. (G.3). By Lemma C.6, there is a feature mapping $\phi : \mathbb{S}^{d-1} \to \mathbb{R}^{N_{2,d}}$ such that for every $u, v \in \mathbb{S}^{d-1}$,

$$\langle \phi(u), \phi(v) \rangle = N_{2,d}P_{2,d}(u^\top v), \tag{G.12}$$

$$\|\phi_u\|_2^2 = N_{2,d}. \tag{G.13}$$

Rewrite Eq. (G.10) using $\phi_2$ we get

$$\| \mathop{\mathbb{E}}_{u\sim\rho}[\phi_2(u)] - \gamma_2\phi_2(e_1)\|_2^2 = 0. \tag{G.14}$$

Or equivalently,

$$\gamma_2\phi_2(e_1) = \mathop{\mathbb{E}}_{u\sim\rho}[\phi_2(u)] \tag{G.15}$$

As a result,

$$\gamma_2 = \left\|\frac{\phi_2(e_1)}{\sqrt{N_{k,d}}}\right\|_2 = \left\|\frac{\mathbb{E}_{u\sim\rho}[\phi_2(u)]}{\sqrt{N_{k,d}}}\right\|_2 \leq \frac{\mathbb{E}_{u\sim\rho}[\|\phi_2(u)\|_2]}{\sqrt{N_{2,d}}} = 1. \tag{G.16}$$

Similarly, we also have $\gamma_4 \leq 1$.

Now we prove Eq. (G.1). Recall that

$$P_{2,d}(t) = \frac{d}{d-1}t^2 - \frac{1}{d-1}, \tag{G.17}$$

$$P_{4,d}(t) = \frac{(d+2)(d+4)}{d^2-1}t^4 - \frac{6d+12}{d^2-1}t^2 + \frac{3}{d^2-1}. \tag{G.18}$$

Combining Eq. (G.10) and Eq. (G.17) we have

$$0 = \frac{d-1}{d} \mathop{\mathbb{E}}_{u,u'\sim\rho}[P_{2,d}(u^\top u') - 2\gamma_2 P_{2,d}(u^\top e_1) + \gamma_2^2 P_{2,d}(e_1^\top e_1)] \tag{G.19}$$

$$= \mathop{\mathbb{E}}_{u,u'\sim\rho}[(u^\top u')^2 - 2\gamma_2(u^\top e_1)^2 + \gamma_2^2(e_1^\top e_1)^2] - \frac{1}{d}(1 - 2\gamma_2 + \gamma_2^2) \tag{G.20}$$

$$= \| \mathop{\mathbb{E}}_{u\sim\rho}[u^{\otimes 2} - \gamma_2 e_1^{\otimes 2}]\|_F^2 - \frac{1}{d}(1 - \gamma_2)^2. \tag{G.21}$$

Consequently,

$$\| \mathop{\mathbb{E}}_{u\sim\rho}[u^{\otimes 2} - \gamma_2 e_1^{\otimes 2}]\|_F^2 = \frac{1}{d}(1 - \gamma_2)^2. \tag{G.22}$$

Similarly, combining Eq. (G.11) and Eq. (G.18) we get

$$0 = \frac{d^2-1}{(d+2)(d+4)} \mathop{\mathbb{E}}_{u,u'\sim\rho}[P_{4,d}(u^\top u') - 4\gamma_4 P_{4,d}(u^\top e_1) + \gamma_4^2 P_{4,d}(e_1^\top e_1)] \tag{G.23}$$

$$= \mathop{\mathbb{E}}_{u,u'\sim\rho}[(u^\top u')^4 - 2\gamma_4(u^\top e_1)^4 + \gamma_4^2(e_1^\top e_1)^4] + \frac{3}{(d+2)(d+4)}(1 - 2\gamma_4 + \gamma_4^2) \tag{G.24}$$

$$- \frac{6d+12}{(d+2)(d+4)} \mathop{\mathbb{E}}_{u,u'\sim\rho}[(u^\top u')^2 - 2\gamma_4(u^\top e_1)^2 + \gamma_4^2(e_1^\top e_1)^2]. \tag{G.25}$$

Consequently,

$$\| \mathop{\mathbb{E}}_{u\sim\rho}[u^{\otimes 4} - \gamma_4 e_1^{\otimes 4}]\|_F^2 \leq \frac{3}{(d+2)(d+4)}(1 - \gamma_4)^2 + \frac{6d+12}{(d+2)(d+4)}(\gamma_4 + 1)^2. \tag{G.26}$$

Now for any fixed $\xi \in \mathbb{S}^{d-1}$, our key observation is that by Jensen's inequality,

$$\mathop{\mathbb{E}}_{u\sim\rho}[(u^\top\xi)^2]^2 \leq \mathop{\mathbb{E}}_{u\sim\rho}[(u^\top\xi)^4]. \tag{G.27}$$

On the one hand, by Eq. (G.22) we have

$$\left|\mathop{\mathbb{E}}_{u\sim\rho}[(u^\top\xi)^2] - \gamma_2(e_1^\top\xi)^2\right| = \left|\mathop{\mathbb{E}}_{u\sim\rho}[\langle u^{\otimes 2} - \gamma_2 e_1^{\otimes 2}, \xi^{\otimes 2}\rangle]\right| \tag{G.28}$$

$$\leq \| \mathop{\mathbb{E}}_{u\sim\rho}[u^{\otimes 2} - \gamma_2 e_1^{\otimes 2}]\|_F \|\xi^{\otimes 2}\|_F = \sqrt{\frac{(1-\gamma_2)^2}{d}}. \tag{G.29}$$

On the other hand, by Eq. (G.26) we have

$$\left| \mathbb{E}_{u\sim\rho}[(u^\top\xi)^4] - \gamma_4(e_1^\top\xi)^4 \right| = \left| \mathbb{E}_{u\sim\rho}[\langle u^{\otimes 4} - \gamma_4 e_1^{\otimes 4}, \xi^{\otimes 4}\rangle] \right| \tag{G.30}$$

$$\leq \| \mathbb{E}_{u\sim\rho}[u^{\otimes 4} - \gamma_4 e_1^{\otimes 4}]\|_F \|\xi^{\otimes 4}\|_F \leq 2\sqrt{\frac{6d+12}{(d+2)(d+4)}}(|\gamma_4+1| + |\gamma_4-1|). \tag{G.31}$$

Finally, combining Eqs (G.27)-(G.31) and set $\xi = e_1$ we get

$$\left( \gamma_2 - \sqrt{\frac{(1-\gamma_2)^2}{d}} \right)^2 \leq \gamma_4 + 2\sqrt{\frac{6d+12}{(d+2)(d+4)}}(|\gamma_4+1| + |\gamma_4-1|). \tag{G.32}$$

It follows directly that

$$\gamma_2^2 \leq \gamma_4 + O\left(\frac{1}{\sqrt{d}}\right). \tag{G.33}$$

Finally we prove Eq. (G.2). For any $\xi, u \in \mathbb{S}^{d-1}$ we have $|u^\top\xi| \leq 1$. As a result,

$$\mathbb{E}_{u\sim\rho}[(u^\top\xi)^4] \leq \mathbb{E}_{u\sim\rho}[(u^\top\xi)^2]. \tag{G.34}$$

Combining with Eq. (G.29) and Eq. (G.31) and setting $\xi = e_1$ we get

$$\gamma_4 \leq O(1/\sqrt{d}) + \mathbb{E}_{u\sim\rho}[(u^\top\xi)^4] \leq O(1/\sqrt{d}) + \mathbb{E}_{u\sim\rho}[(u^\top\xi)^2] \leq O(1/\sqrt{d}) + \gamma_2. \tag{G.35}$$

$\square$

**Proposition G.2** (Sufficient condition). *Let $f_\rho(x) = \mathbb{E}_{u\sim\rho}[\sigma(u^\top x)]$ be a two-layer neural network and $y(x) = h(e_1^\top x)$ the target function. Let $\gamma_k = \frac{\hat{h}_{k,d}}{\hat{\sigma}_{k,d}}, \forall k \geq 0$. Suppose $\hat{h}_{k,d} = 0, \forall k \notin \{0,2,4\}$, $\hat{h}_{0,d} = \hat{\sigma}_{0,d}$, $\sigma$ is a degree-4 polynomial, and*

$$\gamma_2^2 \leq \gamma_4 - O\left(1/d\right), \tag{G.36}$$
$$\gamma_4 \leq \gamma_2 - O\left(1/d\right), \tag{G.37}$$
$$\gamma_2 \leq 1 - O\left(1/d\right). \tag{G.38}$$

*Then there exists a symmetric and rotational invariant neural network $\rho$ such that $\mathbb{E}_{x\sim\mathbb{S}^{d-1}}(f_\rho(x) - y(x))^2 = 0$.*

*Proof.* Recall that by Lemma 4.2, for every symmetric and rotational invariant neural network $\rho$ we have

$$L(\rho) = \frac{\hat{\sigma}_{2,d}^2}{2}\left( \mathbb{E}_{u\sim\rho}[P_{2,d}(w)] - \gamma_2 \right)^2 + \frac{\hat{\sigma}_{4,d}^2}{2}\left( \mathbb{E}_{u\sim\rho}[P_{4,d}(w)] - \gamma_4 \right)^2 \tag{G.39}$$

where $u = (w, z)$, $w \in [-1,1]$, and $z \in \sqrt{1-w^2}\mathbb{S}^{d-2}$. Therefore, we only need to show that there exists a one-dimensional distribution $\mu$ such that

$$\mathbb{E}_{w\sim\mu}[P_{2,d}(w)] = \gamma_2, \tag{G.40}$$

$$\mathbb{E}_{w\sim\mu}[P_{4,d}(w)] = \gamma_4. \tag{G.41}$$

Recall that

$$P_{2,d}(t) = \frac{d}{d-1}t^2 - \frac{1}{d-1}, \tag{G.42}$$

$$P_{4,d}(t) = \frac{(d+2)(d+4)}{d^2-1}t^4 - \frac{6d+12}{d^2-1}t^2 + \frac{3}{d^2-1}. \tag{G.43}$$

Consequently, Eq. (G.40) and Eq. (G.41) is equivalent to

$$\mathbb{E}_{w\sim\mu}[w^2] = \frac{d-1}{d}\gamma_2 + \frac{1}{d}, \tag{G.44}$$

$$\mathbb{E}_{w\sim\mu}[w^4] = \frac{d^2-1}{(d+2)(d+4)}\gamma_4 + \frac{6d+12}{(d+2)(d+4)}\left(\frac{d-1}{d}\gamma_2 + \frac{1}{d}\right) - \frac{3}{(d+2)(d+4)}. \tag{G.45}$$

Let $\beta_2 = \frac{d-1}{d}\gamma_2 + \frac{1}{d}$ and $\beta_4 = \frac{d^2-1}{(d+2)(d+4)}\gamma_4 + \frac{6d+12}{(d+2)(d+4)}\left(\frac{d-1}{d}\gamma_2 + \frac{1}{d}\right) - \frac{3}{(d+2)(d+4)}$. Then we have $|\beta_2 - \gamma_2| \le O(1/d)$ and $|\beta_4 - \gamma_4| \le O(1/d)$. Rewriting Eqs. (G.36)-(G.38) using $\beta_2$ and $\beta_4$, we have

$$\beta_2^2 \le \beta_4, \tag{G.46}$$

$$\beta_4 \le \beta_2, \tag{G.47}$$

$$\beta_2 \le 1. \tag{G.48}$$

By Proposition I.14, there exists a distribution $\mu^\star$ such that

$$\mathbb{E}_{w\sim\mu^\star}[w^2] = \beta_2, \tag{G.49}$$

$$\mathbb{E}_{w\sim\mu^\star}[w^4] = \beta_4. \tag{G.50}$$

Hence, by setting the marginal distribution of $w$ equal to $\mu^\star$, we have

$$\mathbb{E}_{w\sim\mu^\star}[P_{2,d}(w)] = \gamma_2, \qquad \mathbb{E}_{w\sim\mu^\star}[P_{4,d}(w)] = \gamma_4, \tag{G.51}$$

which proves the desired result. $\qquad\qquad\qquad\qquad\qquad\qquad\qquad\qquad\qquad\qquad\qquad\qquad$ $\square$

# H  Proofs for Sample Complexity Lower Bounds for Kernel Method

In this section, we present a sample complexity lower bound for kernel methods with any inner product kernel. That is, the kernel $K : \mathbb{S}^{d-1} \times \mathbb{S}^{d-1} \to \mathbb{R}$ can be written as $K(x,z) = \kappa(x^\top z)$ for some one-dimensional function $\kappa : [-1,1] \to \mathbb{R}$. Inner product kernels are rotational invariant and do not specialize to any coordinate. In particular, NTK of two-layer neural networks is an inner product kernel (e.g., Du et al. [30], Wei et al. [76]).

To prove Theorem 3.6, we only need to work on the degree-$k$ spherical harmonics component of the target function and its estimation because, by the orthogonality of spherical harmonics, any lower bound for the degree-$k$ components directly translates to the original setting. The following lemma states the lower bound for general $k \ge 0$, while Theorem 3.6 only requires the case $k = 4$.

**Lemma H.1.** *Fixed any $k \ge 1$, let $g^\star(x) = \overline{P}_{k,d}(u^\top x)$ for some fixed $u \in \mathbb{S}^{d-1}$. When $d$ is larger than a sufficiently large universal constant and $n < d^k(8k)^{-(3k+1)}(\ln d)^{-(k+2)}$, with probability at least $1/2$ over the randomness of $n$ i.i.d. data $\{x_1, \cdots, x_n\}$ drawn uniformly from $\mathbb{S}^{d-1}$, any estimation $g$ of the form $g(x) = \sum_{i=1}^n \beta_i \overline{P}_{k,d}(x^\top x_i)$ must have a constant error:*

$$\mathbb{E}_{x\sim\mathbb{S}^{d-1}}(g(x) - g(x)^\star)^2 \ge 3/4. \tag{H.1}$$

Proof of Lemma H.1 is deferred to Appendix H.2, where our key observation is that $\overline{P}_{k,d}(\langle x_i, \cdot \rangle)$ cannot correlate too much with $g^\star$ when $x_i$ is uniformly randomly drawn from the unit sphere. Indeed, $\langle \overline{P}_{k,d}(\langle x_i, \cdot \rangle), g^\star \rangle = P_{k,d}(x_i^\top u)$ concentrates within $\pm 1/N_{k,d}$ (Lemma C.7). Hence, we can upperbound $\langle g, g^\star \rangle = \sum_{i=1}^n \beta_i P_{k,d}(x_i^\top u) \lesssim \|\beta\|_2 \sqrt{n/N_{k,d}}$. However, we also have $\|g\|_2^2 = \sum_{i,j\in[n]} \beta_i\beta_j P_{k,d}(x_i^\top x_j) \gtrsim \|\beta\|_2^2$ because the matrix $(P_{k,d}(x_i^\top x_j))_{i,j\in[n]}$ concentrates around $I$ (Lemma H.3). Since $\|g - g^\star\|_2^2 < 3/4$ requires $\langle g, g^\star \rangle > \|g\|_2/2$, we must have $n \gtrsim N_{k,d} \approx d^{-k}$.

## H.1  Proof of Theorem 3.6

The following theorem states that for every $k \ge 0$, to achieve a constant population error, we need at least $C_k d^{-k}(\ln d)^{-(k+2)}$ samples where $C_k = (8k)^{-(3k+1)}$ is a constant that only depends on $k$. Hence, Theorem 3.6 is a direct corollary of Theorem H.2 by taking $k = 4$.

**Theorem H.2.** *Let $y : \mathbb{S}^{d-1} \to \mathbb{R}$ be a function of the form $y(x) = h(q_\star^\top x)$ where $q_\star \in \mathbb{S}^{d-1}$ is a fixed vector and $h : [-1, 1] \to \mathbb{R}$ is a one-dimensional function. Let $K : \mathbb{S}^{d-1} \times \mathbb{S}^{d-1} \to \mathbb{R}$ be any inner product kernel. When $d$ is larger than a universal constant and $n < d^k(8k)^{-(3k+1)}(\ln d)^{-(k+2)}$, with probability at least $1/2$ over the randomness of $n$ i.i.d. data points $\{x_i\}_{i=1}^n$, any estimation $f$ of the form $f(x) = \sum_{i=1}^n \beta_i K(x_i, x)$ must have a significant error:*

$$\|y - f\|_2^2 \geq \frac{3}{4}(\hat{h}_{k,d})^2.$$

*Proof of Theorem H.2.* In the following we first fix a degree $k \geq 1$. For every $l > 0$, let $\mathbb{Y}_{l,d}$ be the space of degree-$l$ spherical harmonics and $\Pi_l$ the projection to $\mathbb{Y}_{l,d}$. By the orthogonality of $\mathbb{Y}_{l,d}$, we have

$$\|y - f\|_2^2 = \sum_{l=0}^\infty \|\Pi_l(y - f)\|_2^2 \geq \|\Pi_k(y - f)\|_2^2. \tag{H.2}$$

By Atkinson and Han [12, Section 2.3], the projection operator $\Pi_k$ is given by

$$(\Pi_k f)(x) = \sqrt{N_{k,d}} \mathop{\mathbb{E}}_{\xi \sim \mathbb{S}^{d-1}} [\overline{P}_{k,d}(x^\top \xi) f(\xi)]. \tag{H.3}$$

Consequently, we have $(\Pi_k y)(x) = \sqrt{N_{k,d}} \mathbb{E}_{\xi \sim \mathbb{S}^{d-1}} [\overline{P}_{k,d}(x^\top \xi) h(q_\star^\top \xi)]$. Recall that

$$\mu_d(t) \triangleq (1 - t^2)^{\frac{d-3}{2}} \frac{\Gamma(d/2)}{\Gamma((d-1)/2)} \frac{1}{\sqrt{\pi}} \tag{H.4}$$

is the density of $u_1$ when $u = (u_1, \cdots, u_d)$ is drawn uniformly from $\mathbb{S}^{d-1}$, and $\hat{h}_{k,d} = \mathbb{E}_{t \sim \mu_d}[\overline{P}_{k,d}(t)h(t)]$. By Funk-Hecke formula (Theorem C.1) we get

$$(\Pi_k y)(x) = \hat{h}_{k,d} \overline{P}_{k,d}(x^\top q_\star), \quad \forall x \in \mathbb{S}^{d-1}. \tag{H.5}$$

Similarly, we can write the inner product kernel $K$ as $K(x, z) = \kappa(x^\top z)$. Define $\hat{\kappa}_{k,d} = \mathbb{E}_{t \sim \mu_d}[\overline{P}_{k,d}(t)\kappa(t)]$. Then we have

$$(\Pi_k f)(x) = \hat{\kappa}_{k,d} \sum_{i=1}^n \beta_i \overline{P}_{k,d}(x^\top x_i), \quad \forall x \in \mathbb{S}^{d-1}. \tag{H.6}$$

Now we apply Lemma H.1 to the function $(\hat{h}_{k,d})^{-1}(\Pi_k y) = \overline{P}_{k,d}(\langle q_\star, \cdot \rangle)$. As a result, with probability at least $1/2$ over the randomness of $\{x_i\}_{i=1}^n$, for any function $f$ of the form $f(\cdot) = \sum_{i=1}^n \beta_i K(x_i, \cdot)$, we have $\|(\hat{h}_{k,d})^{-1} \Pi_4(y - f)\|_2^2 \geq 3/4$, which implies $\|y - f\|_2^2 \geq \frac{3}{4}(\hat{h}_{k,d})^2$. $\square$

## H.2 Proof of Lemma H.1

*Proof of Lemma H.1.* First we prove that when $\|g^\star\|_2 = 1$, $\|g - g^\star\|_2^2 < 3/4$ implies $\langle g, g^\star \rangle \geq \|g\|_2/2$. To this end, by basic algebra we get

$$3/4 > \|g - g^\star\|_2^2 = \|g\|_2^2 + \|g^\star\|_2^2 - 2\langle g, g^\star \rangle. \tag{H.7}$$

Consequently,

$$\langle g, g^\star \rangle \geq \frac{1}{2}\left(\|g\|_2^2 + \frac{1}{4}\right) \geq \frac{1}{2}\|g\|_2, \tag{H.8}$$

where the last inequality follows from AM-GM. In the following, we prove that with probability at least $1/2$, Eq. (H.8) is impossible when $n < d^k(8k)^{-(3k+1)}(\ln d)^{-(k+2)}$.

Recall that Lemma C.5 states that for every $x, z \in \mathbb{S}^{d-1}$ and $k \geq 0$,

$$\langle \overline{P}_{k,d}(\langle x, \cdot \rangle), \overline{P}_{k,d}(\langle z, \cdot \rangle) \rangle = P_{k,d}(\langle x, z \rangle). \tag{H.9}$$

As a result,

$$\|g\|_2^2 = \sum_{i,j=1}^n \beta_i \beta_j \langle \overline{P}_{k,d}(\langle x_i, \cdot \rangle), \overline{P}_{k,d}(\langle x_j, \cdot \rangle) \rangle = \sum_{i,j=1}^n \beta_i \beta_j P_{k,d}(\langle x_i, x_j \rangle). \tag{H.10}$$

Invoking Lemma H.3, with probability at least $3/4$ over the randomness of $\{x_i\}_{i \in [n]}$, we have

$$\|g\|_2^2 = \sum_{i,j=1}^n \beta_i \beta_j P_{k,d}(\langle x_i, x_j \rangle) \geq \frac{1}{2} \|\beta\|_2^2. \tag{H.11}$$

For the LHS of Eq. (H.8),

$$\langle g, g^\star \rangle = \sum_{i=1}^n \beta_i \langle \overline{P}_{k,d}(\langle x_i, \cdot \rangle), \overline{P}_{k,d}(\langle u, \cdot \rangle) \rangle = \sum_{i=1}^n \beta_i P_{k,d}(\langle x_i, u \rangle) \tag{H.12}$$

$$\leq \left( \sum_{i=1}^n \beta_i^2 \right)^{1/2} \left( \sum_{i=1}^n P_{k,d}(\langle x_i, u \rangle)^2 \right)^{1/2} = \|\beta\|_2 \left( \sum_{i=1}^n P_{k,d}(\langle x_i, u \rangle)^2 \right)^{1/2}. \tag{H.13}$$

By Lemma C.7, when $n < d^k(8k)^{-(3k+1)}(\ln d)^{-(k+2)}$, with probability at least $3/4$ we have

$$\langle g, g^\star \rangle \leq \|\beta\|_2 \left( \sum_{i=1}^n P_{k,d}(\langle x_i, u \rangle)^2 \right)^{1/2} \leq \|\beta\|_2 \left( n \max_{i \in [n]} P_{k,d}(\langle x_i, u \rangle)^2 \right)^{1/2} \tag{H.14}$$

$$\leq \|\beta\|_2 \sqrt{\frac{n(32k \ln(dn))^k}{d^k}} \leq \|\beta\|_2 \sqrt{\frac{1}{\ln d}}. \tag{H.15}$$

Combining Eq. (H.11) and Eq. (H.15), there exists a universal constant $d_0 > 0$ such that when $d > d_0$, with probability at least $1/2$ over the randomness of $\{x_1, \cdots, x_n\}$,

$$\langle g, g^\star \rangle < \|\beta\|_2/4 \leq \|g\|_2/2. \tag{H.16}$$

Recall that $\|g - g^\star\|_2^2 < 3/4$ implies $\langle g, g^\star \rangle \geq \|g\|_2/2$. Consequently, with probability at least $1/2$, when $n < d^k(8k)^{-(3k+1)}(\ln d)^{-(k+2)}$ and $d > d_0$ we have $\|g - g^\star\|_2^2 \geq 3/4$. $\qquad \square$

**Lemma H.3.** *Let $x_1, \cdots, x_n$ be i.i.d. random variables drawn uniformly from $\mathbb{S}^{d-1}$, and $P_{k,d}$ the degree-$k$ Legendre polynomial. For any fixed $k \geq 1$, define the matrix $M_k \in \mathbb{R}^{n \times n}$ where $[M_k]_{i,j} = P_{k,d}(\langle x_i, x_j \rangle)$. There exists universal constants $d_0$ such that, when $n < d^k(8k)^{-(3k+1)}(\ln d)^{-(k+2)}$ and $d > d_0$, with probability at least $3/4$,*

$$\lambda_{\min}(M_k) \geq 1/2. \tag{H.17}$$

*Proof.* In the following, we fix $k \geq 1$. First, we prove that

$$\mathbb{E}[\lambda_{\min}(M_k)] \geq 7/8. \tag{H.18}$$

To this end, we invoke Theorem I.4 by constructing a matrix $A$ such that $\frac{1}{N_{k,d}} A^\top A = M_k$.

By Lemma C.6, there exists a feature mapping $\phi : \mathbb{S}^{d-1} \to \mathbb{R}^{N_{k,d}}$ such that for every $u, v \in \mathbb{S}^{d-1}$

$$\langle \phi(u), \phi(v) \rangle = N_{k,d} P_{k,d}(u^\top v), \tag{H.19}$$

$$\|\phi(u)\|_2^2 = N_{k,d}, \tag{H.20}$$

$$\mathbb{E}_{u \sim \mathbb{S}^{d-1}}[\phi(u)\phi(u)^\top] = I. \tag{H.21}$$

We set $A_i = \phi(x_i), \forall 1 \leq i \leq n$. Consequently,

$$\frac{1}{N_{k,d}} A^\top A = M_k, \tag{H.22}$$

$$\|A_i\|_2^2 = N_{k,d}, \tag{H.23}$$

$$\mathbb{E}[A_i A_i^\top] = I. \tag{H.24}$$

In the following, we upper bound the incoherence parameter $p$ defined in Theorem I.4:

$$p = \frac{1}{N_{k,d}} \mathbb{E}\left[ \max_{i \leq n} \sum_{j \in [n], j \neq i} \langle A_i, A_j \rangle^2 \right]. \tag{H.25}$$

By Eq. (H.19) we get

$$\frac{1}{N_{k,d}} \mathbb{E}\left[\max_{i\leq n} \sum_{j\in[n],j\neq i} \langle A_i, A_j\rangle^2\right] = N_{k,d} \mathbb{E}\left[\max_{i\leq n} \sum_{j\in[n],j\neq i} P_{k,d}(\langle x_i, x_j\rangle)^2\right] \qquad \text{(H.26)}$$

$$\leq n N_{k,d} \mathbb{E}\left[\max_{i\leq n} \max_{j\neq i} P_{k,d}(\langle x_i, x_j\rangle)^2\right]. \qquad \text{(H.27)}$$

Invoking Lemma C.7 we get

$$N_{k,d} \mathbb{E}\left[\max_{i\leq n} \max_{j\neq i} P_{k,d}(\langle x_i, x_j\rangle)^2\right] \lesssim (32k\ln(dn))^k. \qquad \text{(H.28)}$$

It follows that $p \lesssim n(32k\ln(dn))^k$.

Now that we have established the conditions, by Theorem I.4 we have

$$\mathbb{E}[\|M - I\|] \lesssim \sqrt{\frac{n(\ln n)(32k\ln(dn))^k}{N_{k,d}}}. \qquad \text{(H.29)}$$

Note that $N_{k,d} \geq \binom{k+d-2}{k} \geq (d/k)^k$ when $k \geq 2$, and clearly $N_{1,d} \geq d$. Hence, there exists a universal constant $d_0$ such that, when $n < d^k(8k)^{-(3k+1)}(\ln d)^{-(k+2)}$.

$$\frac{n(\ln n)(32k\ln(dn))^k}{N_{k,d}} \leq \frac{n(8k)^{2k}\ln(dn)^{k+1}}{d^k} \leq \frac{n(8k)^{2k}\ln(d^{k+1})^{k+1}}{d^k} \leq \frac{1}{\ln d}. \qquad \text{(H.30)}$$

Hence, there exists a universal constant $d_0$ such that when $d \geq d_0$,

$$\mathbb{E}[\lambda_{\min}(M)] \geq 1 - O(1)\sqrt{\frac{n(\ln n)(32k\ln(dn))^k}{N_{k,d}}} \geq 1 - O(1)\sqrt{\frac{1}{\ln d}} \geq 7/8. \qquad \text{(H.31)}$$

Now we prove Eq. (H.17). Let $t = 1 - \lambda_{\min}(M)$ be a random variable. Note that by Eq. (H.18) we have $\mathbb{E}[t] \leq 1/8$. By basic algebra, we also have

$$\lambda_{\min}(M) \leq M_{1,1} = P_{k,d}(\langle x_1, x_1\rangle) = P_{k,d}(1) = 1, \qquad \text{(H.32)}$$

which implies $t \geq 0$. Therefore, by Markov inequality we get

$$\Pr(\lambda_{\min}(M) \leq 1/2) = \Pr(t \geq 1/2) \leq 2\mathbb{E}[t] \leq 1/4. \qquad \text{(H.33)}$$

$\square$

# I  Toolbox

## I.1  Rademacher Complexity and Generalization Bounds

**Lemma I.1** (Special Case of Contraction Principle (Lemma 4.6 of Adamczak et al. [4])). *Let $\mathcal{F}$ be a family of functions from $D$ to $\mathbb{R}$, and let $\varphi$ be an L-Lipschitz function with $\varphi(0) = 0$. Then, for any $S = \{x_1, \dots, x_N\} \subset D$,*

$$\mathbb{E}_{\epsilon_i} \sup_{f\in\mathcal{F}} \frac{1}{N} \sum_{i=1}^{N} \epsilon_i \varphi(f(x_i)) \leq L \cdot \mathbb{E}_{\epsilon_i} \sup_{f\in\mathcal{F}} \frac{1}{N} \sum_{i=1}^{N} \epsilon_i f(x_i) \qquad \text{(I.1)}$$

*where the expectations are taken over i.i.d. Rademacher random variables $\epsilon_1, \dots, \epsilon_N$.*

In the following, for a function family $\mathcal{H}$, we let $R_N(\mathcal{H})$ denote its Rademacher complexity and $R_S(\mathcal{H})$ denote its empirical Rademacher complexity (given a fixed dataset $S$) — see Section 4.4.2 of Ma [53] for a definition of these concepts.

**Lemma I.2** (Empirical Rademacher Complexity of Products). *Suppose $\mathcal{F}$ and $\mathcal{G}$ are two families of functions from $D$ to $\mathbb{R}$, with $D \subset \mathbb{R}^d$, which are uniformly bounded by $B > 0$. Then, for any set $S = \{x_1, \dots, x_N\} \subset D$,*

$$R_S(\mathcal{F}\cdot\mathcal{G}) \leq B\cdot(R_S(\mathcal{F}) + R_S(\mathcal{G})), \qquad \text{(I.2)}$$

*where $\mathcal{F}\cdot\mathcal{G} = \{fg \mid f\in\mathcal{F}, g\in\mathcal{G}\}$. As a corollary,*

$$R_N(\mathcal{F}\cdot\mathcal{G}) \leq B\cdot(R_N(\mathcal{F}) + R_N(\mathcal{G})). \qquad \text{(I.3)}$$

*Proof of Lemma I.2.* We can write

$$R_S(\mathcal{F} \cdot \mathcal{G}) = \mathbb{E}_{\epsilon_i} \sup_{f \in \mathcal{F}, g \in \mathcal{G}} \frac{1}{N} \sum_{i=1}^{N} \epsilon_i f(x_i) g(x_i) \tag{I.4}$$

$$= \mathbb{E}_{\epsilon_i} \sup_{f \in \mathcal{F}, g \in \mathcal{G}} \frac{1}{N} \sum_{i=1}^{N} \epsilon_i \cdot \frac{(f(x_i) + g(x_i))^2 - (f(x_i) - g(x_i))^2}{4} \tag{I.5}$$

$$\leq \mathbb{E}_{\epsilon_i} \sup_{f \in \mathcal{F}, g \in \mathcal{G}} \frac{1}{N} \sum_{i=1}^{N} \epsilon_i \cdot \frac{(f(x_i) + g(x_i))^2}{4} + \mathbb{E}_{\epsilon_i} \sup_{f \in \mathcal{F}, g \in \mathcal{G}} \frac{1}{N} \sum_{i=1}^{N} \epsilon_i \frac{(f(x_i) - g(x_i))^2}{4} \cdot \tag{I.6}$$

Using Lemma I.1 and the fact that the function $x \mapsto x^2$ is $2B$ Lipschitz on the interval $[-B, B]$, we have

$$R_S(\mathcal{F} \cdot \mathcal{G}) \leq 2B \cdot \mathbb{E}_{\epsilon_i} \sup_{f \in \mathcal{F}, g \in \mathcal{G}} \frac{1}{N} \sum_{i=1}^{N} \epsilon_i \cdot \frac{f(x) + g(x)}{4} + 2B \cdot \mathbb{E}_{\epsilon_i} \sup_{f \in \mathcal{F}, g \in \mathcal{G}} \frac{1}{N} \sum_{i=1}^{N} \epsilon_i \frac{f(x_i) - g(x_i)}{4} \tag{I.7}$$

$$\leq \frac{B}{2} \cdot (R_S(\mathcal{F}) + R_S(\mathcal{G})) + \frac{B}{2} \cdot (R_S(\mathcal{F}) + R_S(\mathcal{G})) \tag{I.8}$$

$$= B \cdot (R_S(\mathcal{F}) + R_S(\mathcal{G})), \tag{I.9}$$

as desired. $\qquad\square$

**Lemma I.3** (Symmetrization — Analogous to Corollary 4.7 of Adamczak et al. [4])**.** *Let $X_1, \ldots, X_N$ be independent random variables. Let $\mathcal{F}$ be a family of functions from $D \subset \mathbb{R}^d$ to $\mathbb{R}$ that is uniformly bounded by $B_1 \geq 1$, and let $\mathcal{G}$ be a family of functions from $D \subset \mathbb{R}^d$ to $\mathbb{R}$. Then, for nonnegative integers $p_1, p_2, p_3, p_4$ and a fixed $g \in \mathcal{G}$ that is bounded by $B_2 \geq 1$, we have*

$$\mathbb{E}_{X_i} \sup_{f_1, f_2, f_3, f_4 \in \mathcal{F}, g \in \mathcal{G}} \left| \frac{1}{N} \sum_{i=1}^{N} \left( |f_1(X_i)|^{p_1} |f_2(X_i)|^{p_2} |f_3(X_i)|^{p_3} |f_4(X_i)|^{p_4} |g(X_i)| \right. \tag{I.10}$$

$$\left. - \mathbb{E}_{X_i} \left[ |f_1(X_i)|^{p_1} |f_2(X_i)|^{p_2} |f_3(X_i)|^{p_3} |f_4(X_i)|^{p_4} |g(X_i)| \right] \right) \right| \tag{I.11}$$

$$\leq \mathrm{poly}(p_1, p_2, p_3, p_4, B_1^{p_1+p_2+p_3+p_4}, B_2) \cdot R_N(\mathcal{F}) \tag{I.12}$$

*Proof of Lemma I.3.* By Theorem 4.13 of Ma [53],

$$\mathbb{E}_{X_i} \sup_{f_1, f_2, f_3, f_4 \in \mathcal{F}, g \in \mathcal{G}} \left| \frac{1}{N} \sum_{i=1}^{N} \left( |f_1(X_i)|^{p_1} |f_2(X_i)|^{p_2} |f_3(X_i)|^{p_3} |f_4(X_i)|^{p_4} |g(X_i)| \right. \tag{I.13}$$

$$\left. - \mathbb{E}_{X_i} \left[ |f_1(X_i)|^{p_1} |f_2(X_i)|^{p_2} |f_3(X_i)|^{p_3} |f_4(X_i)|^{p_4} |g(X_i)| \right] \right) \right| \tag{I.14}$$

$$\lesssim R_N(\mathcal{F}^{p_1} \cdot \mathcal{F}^{p_2} \cdot \mathcal{F}^{p_3} \cdot \mathcal{F}^{p_4} \cdot g) \qquad \text{(By Theorem 4.13 of Ma [53])}$$

$$\lesssim \max(B_1^{p_1+p_2+p_3+p_4}, B_2) \cdot \left( R_N(\mathcal{F}^{p_1} \cdot \mathcal{F}^{p_2} \cdot \mathcal{F}^{p_3} \cdot \mathcal{F}^{p_4}) + R_N(\{g\}) \right)$$
$$\text{(By Lemma I.2)}$$

$$\lesssim \max(B_1^{p_1+p_2+p_3+p_4}, B_2) \cdot R_N(\mathcal{F}^{p_1} \cdot \mathcal{F}^{p_2} \cdot \mathcal{F}^{p_3} \cdot \mathcal{F}^{p_4}) \tag{I.15}$$

$$\lesssim \max(B_1^{p_1+p_2+p_3+p_4}, B_2) \cdot \max(B_1^{p_1}, B_1^{p_2+p_3+p_4}) \cdot \max(B_1^{p_2}, B_1^{p_3+p_4}) \tag{I.16}$$

$$\cdot \max(B_1^{p_3}, B_1^{p_4}) \cdot (R_N(\mathcal{F}^{p_1}) + R_N(\mathcal{F}^{p_2}) + R_N(\mathcal{F}^{p_3}) + R_N(\mathcal{F}^{p_4}))$$
$$\text{(By repeated applications of Lemma I.2)}$$

$$\lesssim \max(B_1^{O(p_1+p_2+p_3+p_4)}, B_2) \cdot (R_N(\mathcal{F}^{p_1}) + R_N(\mathcal{F}^{p_2}) + R_N(\mathcal{F}^{p_3}) + R_N(\mathcal{F}^{p_4}))$$
$$\tag{I.17}$$

Finally, we simplify $R_N(\mathcal{F}^{p_i})$ for each $i$. For nonnegative integers $p$, since $\mathcal{F}$ is uniformly bounded by $B_1$, the function $x \mapsto x^p$ is $pB_1^{p-1}$-Lipschitz on $[-B_1, B_1]$ for $p \geq 1$ and is 1-Lipschitz for $p = 0$. Thus, by Lemma I.1, we have $R_N(\mathcal{F}^p) \leq \min(1, pB_1^{p-1})R_N(\mathcal{F})$, and

$$\mathbb{E}_{X_i} \sup_{f_1, f_2, f_3, f_4 \in \mathcal{F}, g \in \mathcal{G}} \left| \frac{1}{N} \sum_{i=1}^{N} \left( |f_1(X_i)|^{p_1} |f_2(X_i)|^{p_2} |f_3(X_i)|^{p_3} |f_4(X_i)|^{p_4} |g(X_i)| \right. \right. \tag{I.18}$$

$$\left. \left. - \mathbb{E}_{X_i} \left[ |f_1(X_i)|^{p_1} |f_2(X_i)|^{p_2} |f_3(X_i)|^{p_3} |f_4(X_i)|^{p_4} |g(X_i)| \right] \right) \right| \tag{I.19}$$

$$\lesssim \max(B_1^{O(p_1+p_2+p_3+p_4)}, B_2) \cdot \max(p_1 B_1^{p_1}, p_2 B_1^{p_2}, p_3 B_1^{p_3}, p_4 B_1^{p_4}) \cdot R_N(\mathcal{F}) \tag{I.20}$$

$$\lesssim \text{poly}(p_1, p_2, p_3, p_4, B_1^{p_1+p_2+p_3+p_4}, B_2) \cdot R_N(\mathcal{F}) \tag{I.21}$$

as desired. $\qquad\square$

## I.2   Concentration Lemmas

**Theorem I.4** (Theorem 5.62 of Vershynin [73]). *Let $A$ be an $m \times n$ matrix with independent columns $\{A_j\}_{j=1}^{n}$ and $m \geq n$. Suppose $A_j$ is isotropic, i.e., $\mathbb{E}[A_j A_j^\top] = I$ and $\|A_j\|_2 = \sqrt{m}$ for every $j \in [n]$. Let $p$ be the incoherence parameter defined by*

$$p \triangleq \frac{1}{m} \mathbb{E} \left[ \max_{i \leq n} \sum_{j \in [n], j \neq i} \langle A_i, A_j \rangle^2 \right]. \tag{I.22}$$

*Then there is a universal constant $C_0$ such that*

$$\mathbb{E} \left[ \left\| \frac{1}{m} A^\top A - I \right\| \right] \leq C_0 \sqrt{\frac{p \ln n}{m}}. \tag{I.23}$$

The following proposition gives the tail bound for inner products of random vectors in $\mathbb{S}^{d-1}$ (c.f. Theorem 3.4.6 of Vershynin [74]).

**Proposition I.5.** *Let $u$ be a vector drawn uniformly at random from the unit sphere $\mathbb{S}^{d-1}$. For any fixed vector $v \in \mathbb{S}^{d-1}$, we have*

$$\forall t > 0, \quad \Pr(|u^\top v| \geq t) \leq 2 \exp \left( -\frac{t^2 d}{2} \right). \tag{I.24}$$

*Proof of Proposition I.5.* By the symmetricity of $u$ and $v$, we can assume that $v = (1, 0, \cdots, 0)$ without loss of generality. Let $x = \frac{1}{2}(u^\top v + 1)$. Then we have $x \sim \text{Beta}(\frac{d-1}{2}, \frac{d-1}{2})$. Hence, the desired result follows directly from Skorski [63, Theorem 1]. $\qquad\square$

**Lemma I.6** (Modification of Proposition 4.4 of Adamczak et al. [4]). *Let $X_1, \ldots, X_N$ be i.i.d. random vectors drawn uniformly from $\sqrt{d}\mathbb{S}^{d-1}$, and let $p_1, p_2, p_3, p_4, q_1, q_2$ be nonnegative integers. Suppose $p_1 + p_2 + p_3 + p_4 \geq 2$. Then, for fixed $w_1, w_2 \in \mathbb{S}^{d-1}$, as long as $N \leq d^C$ for any universal constant $C > 0$, we have*

$$\sup_{u_1, u_2, u_3, u_4 \in \mathbb{S}^{d-1}} \left| \frac{1}{N} \sum_{i=1}^{N} |\langle X_i, u_1 \rangle|^{p_1} |\langle X_i, u_2 \rangle|^{p_2} |\langle X_i, u_3 \rangle|^{p_3} |\langle X_i, u_4 \rangle|^{p_4} |\langle X_i, w_1 \rangle|^{q_1} |\langle X_i, w_2 \rangle|^{q_2} \right.$$
$$\tag{I.25}$$

$$\left. - \mathbb{E}_{X \sim \sqrt{d}\mathbb{S}^{d-1}} \left[ |\langle X, u_1 \rangle|^{p_1} |\langle X, u_2 \rangle|^{p_2} |\langle X, u_3 \rangle|^{p_3} |\langle X, u_4 \rangle|^{p_4} |\langle X, w_1 \rangle|^{q_1} |\langle X, w_2 \rangle|^{q_2} \right] \right|$$
$$\tag{I.26}$$

$$\leq C_{p_1, p_2, p_3, p_4, q_1, q_2} (\log d)^{O(p_1+p_2+p_3+p_4+q_1+q_2)} \cdot \sqrt{\frac{d}{N}} \tag{I.27}$$

$$+ C_{p_1, p_2, p_3, p_4} (\log d)^{O(q_1+q_2)} \cdot \frac{d^{\frac{p_1+p_2+p_3+p_4}{2}}}{N} + \frac{C_{p_1, p_2, p_3, p_4, q_1, q_2}}{d^{\Omega(\log d)}} \tag{I.28}$$

*with probability at least* $1 - \frac{1}{d^{\Omega(\log d)}}$. *Here,* $C_{p_1,p_2,p_3,p_4,q_1,q_2}$ *is a sufficiently large constant which depends on* $p_1, p_2, p_3, p_4, q_1, q_2$ *and* $C_{p_1,p_2,p_3,p_4}$ *is a constant which depends on* $p_1, p_2, p_3, p_4$.

*Proof of Lemma I.6.* We largely follow the proof of Proposition 4.4 of Adamczak et al. [4], with some minor modifications. First, let $E_1$ be the event that $|\langle X_i, w_1 \rangle| \leq (\log d)^2$ for all $i = 1, \ldots, N$, and define $E_2$ analogously for $w_2$. By Lemma I.10, the sub-exponential norm of $\langle X_i, w_1 \rangle$ is at most an absolute constant $K > 0$. Thus, by Proposition 2.7.1 of Vershynin [74], we have

$$\mathbb{P}\Big( |\langle X_i, w_1 \rangle| \geq (\log d)^2 \Big) \leq 2 \exp\Big( -\Omega\Big( \frac{(\log d)^2}{\|\langle X_i, w_1 \rangle\|_{\psi_1}} \Big) \Big) \tag{I.29}$$

$$\lesssim \exp\Big( -\Omega(\log d)^2 \Big) \tag{I.30}$$

$$\lesssim \frac{1}{d^{\Omega(\log d)}}. \tag{I.31}$$

Thus, by a union bound, with probability at least $1 - \frac{N}{d^{\Omega(\log d)}}$, for all $i = 1, \ldots, N$, we have $|\langle X_i, w_1 \rangle| \leq (\log d)^2$, and $|\langle X_i, w_2 \rangle| \leq (\log d)^2$ (by performing a similar union bound using the sub-exponential norm of $\langle X_i, w_2 \rangle$). In summary, $E_1$ and $E_2$ simultaneously hold with probability at least $1 - \frac{1}{d^{\Omega(\log d)}}$ since $N \leq d^C$ for some universal constant $C > 0$.

Now, as in Adamczak et al. [4], define $B > 1$, which we will specify later — this is the amount by which we will truncate $\langle X_i, u_1 \rangle, \langle X_i, u_2 \rangle, \langle X_i, u_3 \rangle$ and $\langle X_i, u_4 \rangle$. For convenience, define the function $\varphi_B : \mathbb{R} \to \mathbb{R}$ given by

$$\varphi_B(t) = \begin{cases} t & t \in [-B, B] \\ B & t > B \\ -B & t < B \end{cases} \tag{I.32}$$

Define $B' = (\log d)^2$, and define $\varphi_{B'}$ similarly to $\varphi_B$, but with $B$ replaced by $B'$. If $E_1$ and $E_2$ hold, then for all $u_1, u_2, u_3, u_4 \in \mathbb{S}^{d-1}$, we have

$$\Big| \frac{1}{N} \sum_{i=1}^{N} |\langle X_i, u_1 \rangle|^{p_1} |\langle X_i, u_2 \rangle|^{p_2} |\langle X_i, u_3 \rangle|^{p_3} |\langle X_i, u_4 \rangle|^{p_4} |\varphi_{B'}(\langle X_i, w_1 \rangle)|^{q_1} |\varphi_{B'}(\langle X_i, w_2 \rangle)|^{q_2}$$

$$\tag{I.33}$$

$$- \mathbb{E}_{X \sim \sqrt{d}\mathbb{S}^{d-1}} \Big[ |\langle X, u_1 \rangle|^{p_1} |\langle X, u_2 \rangle|^{p_2} |\langle X, u_3 \rangle|^{p_3} |\langle X, u_4 \rangle|^{p_4} |\varphi_{B'}(\langle X, w_1 \rangle)|^{q_1} |\varphi_{B'}(\langle X, w_2 \rangle)|^{q_2} \Big] \Big|$$

$$\tag{I.34}$$

$$= \Big| \frac{1}{N} \sum_{i=1}^{N} |\langle X_i, u_1 \rangle|^{p_1} |\langle X_i, u_2 \rangle|^{p_2} |\langle X_i, u_3 \rangle|^{p_3} |\langle X_i, u_4 \rangle|^{p_4} |\langle X_i, w_1 \rangle|^{q_1} |\langle X_i, w_2 \rangle|^{q_2} \tag{I.35}$$

$$- \mathbb{E}_{X \sim \sqrt{d}\mathbb{S}^{d-1}} \Big[ |\langle X, u_1 \rangle|^{p_1} |\langle X, u_2 \rangle|^{p_2} |\langle X, u_3 \rangle|^{p_3} |\langle X, u_4 \rangle|^{p_4} |\varphi_{B'}(\langle X, w_1 \rangle)|^{q_1} |\varphi_{B'}(\langle X, w_2 \rangle)|^{q_2} \Big] \Big|$$

$$\text{(B.c. } E_1 \text{ and } E_2 \text{ hold)}$$

$$= \Big| \frac{1}{N} \sum_{i=1}^{N} |\langle X_i, u_1 \rangle|^{p_1} |\langle X_i, u_2 \rangle|^{p_2} |\langle X_i, u_3 \rangle|^{p_3} |\langle X_i, u_4 \rangle|^{p_4} |\langle X_i, w_1 \rangle|^{q_1} |\langle X_i, w_2 \rangle|^{q_2} \tag{I.36}$$

$$- \mathbb{E}_{X \sim \sqrt{d}\mathbb{S}^{d-1}} \Big[ |\langle X, u_1 \rangle|^{p_1} |\langle X, u_2 \rangle|^{p_2} |\langle X, u_3 \rangle|^{p_3} |\langle X, u_4 \rangle|^{p_4} |\langle X, w_1 \rangle|^{q_1} |\langle X, w_2 \rangle|^{q_2} \Big] \Big|$$

$$\tag{I.37}$$

$$\pm C_{p_1, \ldots, p_4, q_1, q_2} e^{-\Omega(B')}. \tag{By Lemma I.8}$$

Thus, the rest of the proof will focus on obtaining an upper bound on

$$U := \sup_{u_1, u_2, u_3, u_4} \Big| \frac{1}{N} \sum_{i=1}^{N} |\langle X_i, u_1 \rangle|^{p_1} |\langle X_i, u_2 \rangle|^{p_2} |\langle X_i, u_3 \rangle|^{p_3} |\langle X_i, u_4 \rangle|^{p_4} |\varphi_{B'}(\langle X_i, w_1 \rangle)|^{q_1} |\varphi_{B'}(\langle X_i, w_2 \rangle)|^{q_2}$$

$$\tag{I.38}$$

$$- \mathbb{E}_{X \sim \sqrt{d}\mathbb{S}^{d-1}} \Big[ |\langle X, u_1 \rangle|^{p_1} |\langle X, u_2 \rangle|^{p_2} |\langle X, u_3 \rangle|^{p_3} |\langle X, u_4 \rangle|^{p_4} |\varphi_{B'}(\langle X, w_1 \rangle)|^{q_1} |\varphi_{B'}(\langle X, w_2 \rangle)|^{q_2} \Big] \Big|$$

$$\tag{I.39}$$

First define

$$U' := \sup_{u_1, u_2, u_3, u_4} \left| \frac{1}{N} \sum_{i=1}^{N} |\varphi_B(\langle X_i, u_1 \rangle)|^{p_1} |\varphi_B(\langle X_i, u_2 \rangle)|^{p_2} |\varphi_B(\langle X_i, u_3 \rangle)|^{p_3} |\varphi_B(\langle X_i, u_4 \rangle)|^{p_4} \right.$$

$$\tag{I.40}$$

$$|\varphi_{B'}(\langle X_i, w_1 \rangle)|^{q_1} |\varphi_{B'}(\langle X_i, w_2 \rangle)|^{q_2} \tag{I.41}$$

$$- \mathop{\mathbb{E}}_{X \sim \sqrt{d}\mathbb{S}^{d-1}} \left[ |\varphi_B(\langle X, u_1 \rangle)|^{p_1} |\varphi_B(\langle X, u_2 \rangle)|^{p_2} |\varphi_B(\langle X, u_3 \rangle)|^{p_3} |\varphi_B(\langle X, u_4 \rangle)|^{p_4} \right. \tag{I.42}$$

$$\left. \left. |\varphi_{B'}(\langle X, w_1 \rangle)|^{q_1} |\varphi_{B'}(\langle X, w_2 \rangle)|^{q_2} \right] \right| \tag{I.43}$$

We will first obtain an upper bound on $U'$, then show that $U'$ and $U$ are close. We apply Lemma I.3 with $\mathcal{F} = \{x \mapsto \varphi_B(\langle x, u \rangle) \mid u \in \mathbb{S}^{d-1}\}$, and with $f_i$ corresponding to $\varphi_B(\langle x, u_i \rangle)$, and finally with $g(x) = |\varphi_{B'}(\langle x, w_1 \rangle)|^{q_1} |\varphi_{B'}(\langle x, w_2 \rangle)|^{q_2}$, to find that

$$\mathop{\mathbb{E}}_{X_i} U' \leq \mathrm{poly}(p_1, p_2, p_3, p_4, B^{p_1+p_2+p_3+p_4}, (B')^{q_1+q_2}) \cdot R_N(\mathcal{F}) \qquad \text{(By Lemma I.3)}$$

$$\leq \mathrm{poly}(p_1, p_2, p_3, p_4, B^{p_1+p_2+p_3+p_4}, (B')^{q_1+q_2}) \cdot \mathop{\mathbb{E}}_{X_i, \epsilon_i} \sup_{u \in \mathbb{S}^{d-1}} \left| \frac{1}{N} \sum_{i=1}^{N} \epsilon_i \varphi_B(\langle X_i, u \rangle) \right|$$

$$\tag{I.44}$$

$$\leq \mathrm{poly}(p_1, p_2, p_3, p_4, B^{p_1+p_2+p_3+p_4}, (B')^{q_1+q_2}) \cdot \mathop{\mathbb{E}}_{X_i, \epsilon_i} \sup_{u \in \mathbb{S}^{d-1}} \left| \frac{1}{N} \sum_{i=1}^{N} \epsilon \langle X_i, u \rangle \right|$$

$$\text{(By Lemma I.1 since } \varphi_B \text{ is 1-Lipschitz)}$$

$$\leq \mathrm{poly}(p_1, p_2, p_3, p_4, B^{p_1+p_2+p_3+p_4}, (B')^{q_1+q_2}) \cdot \mathop{\mathbb{E}}_{X_i, \epsilon_i} \left\| \frac{1}{N} \sum_{i=1}^{N} \epsilon_i X_i \right\|_2 \tag{I.45}$$

$$\leq \mathrm{poly}(p_1, p_2, p_3, p_4, B^{p_1+p_2+p_3+p_4}, (B')^{q_1+q_2}) \cdot \sqrt{\frac{d}{N}}. \qquad \text{(By Lemma I.9)}$$

Now, to show a high-probability upper bound on $U'$, we apply Lemma 4.8 of Adamczak et al. [4] (Talagrand's concentration inequality), and we ensure that all of the conditions are satisfied:

- We set $X_1, \ldots, X_N$ in Lemma 4.8 of Adamczak et al. [4] to be as defined in the statement of Lemma I.6.

- For convenience, define

$$g_{B,B',u_1,u_2,u_3,u_4,w_1,w_2}(x) = |\varphi_B(\langle x, u_1 \rangle)|^{p_1} |\varphi_B(\langle x, u_2 \rangle)|^{p_2} |\varphi_B(\langle x, u_3 \rangle)|^{p_3} |\varphi_B(\langle x, u_4 \rangle)|^{p_4}$$

$$\tag{I.46}$$

$$|\varphi_{B'}(\langle x, w_1 \rangle)|^{q_1} |\varphi_{B'}(\langle x, w_2 \rangle)|^{q_2}. \tag{I.47}$$

Then, we let

$$\mathcal{F} = \{x \mapsto g_{B,B',u_1,u_2,u_3,u_4,w_1,w_2}(x) - \mathop{\mathbb{E}}_{X \sim \sqrt{d}\mathbb{S}^{d-1}} [g_{B,B',u_1,u_2,u_3,u_4,w_1,w_2}(X)] \mid u_1, u_2, u_3, u_4 \in \mathbb{S}^{d-1}\}$$

$$\tag{I.48}$$

where $w_1$ and $w_2$ are fixed (i.e. as defined in the statement of Lemma I.6).

- We let $a = 2B^{p_1+p_2+p_3+p_4}(B')^{q_1+q_2}$, which is a uniform bound on all the functions in $\mathcal{F}$.

- Observe that for all $f \in \mathcal{F}$, $\mathbb{E}_{X \sim \sqrt{d}\mathbb{S}^{d-1}} f(X) = 0$.

- We let $Z = \sup_{f \in \mathcal{F}} \sum_{i=1}^{N} f(X_i)$ and $\sigma^2 = \sup_{f \in \mathcal{F}} \sum_{i=1}^{N} \mathbb{E} f(X_i)^2$. Observe that $\sigma^2 \leq a^2 N$.

Thus, we apply Lemma 4.8 of Adamczak et al. [4] with

$$t = a\,\mathrm{poly}(p_1, p_2, p_3, p_4, B^{p_1+p_2+p_3+p_4}, (B')^{q_1+q_2}) \cdot \sqrt{\frac{d}{N}} \cdot (\log d)^2 \geq a(\log d)^2 \mathop{\mathbb{E}}_{X_i} U' \tag{I.49}$$

to find that

$$U' \leq \mathop{\mathbb{E}}_{X_i} U' + a\,\mathrm{poly}(p_1, p_2, p_3, p_4, B^{p_1+p_2+p_3+p_4}, (B')^{q_1+q_2}) \cdot \sqrt{\frac{d}{N}} \qquad \text{(I.50)}$$

with failure probability at most

$$\exp\Big(-\frac{t^2 N^2}{2(\sigma^2 + 2aN\,\mathbb{E}\,U') + 3atN}\Big) \leq \exp\Big(-\frac{t^2 N^2}{12\max(\sigma^2 + 2a \cdot N\,\mathbb{E}\,U', atN)}\Big) \qquad \text{(I.51)}$$

$$= \exp\Big(-\frac{1}{12}\min\Big(\frac{t^2 N^2}{\sigma^2 + 2aN\,\mathbb{E}\,U'}, \frac{tN}{a}\Big)\Big) \qquad \text{(I.52)}$$

$$\leq \exp\Big(-\frac{1}{12}\min\Big(\frac{t^2 N^2}{a^2 N + 2aN\,\mathbb{E}\,U'}, \frac{tN}{a}\Big)\Big) \qquad \text{(I.53)}$$

$$\leq \exp\Big(-\frac{1}{12}\min\Big(\frac{t^2 N^2}{a^2 N + 2aN\,\mathbb{E}\,U'}, \sqrt{dN}\Big)\Big) \qquad \text{(I.54)}$$

$$\leq \exp\Big(-\frac{1}{12}\min\Big(\frac{t^2 N}{a^2 + 2a\,\mathbb{E}\,U'}, \sqrt{dN}\Big)\Big) \qquad \text{(I.55)}$$

$$\leq \exp\Big(-\frac{1}{24}\min\Big(\frac{t^2 N}{a^2}, \frac{t^2 N}{2a\,\mathbb{E}\,U'}, \sqrt{dN}\Big)\Big) \qquad \text{(I.56)}$$

$$\leq \exp\Big(-\frac{1}{24}\min\Big(d, \frac{t^2 N}{2a\,\mathbb{E}\,U'}, \sqrt{dN}\Big)\Big) \qquad \text{(I.57)}$$

$$\leq \exp\Big(-\frac{1}{24}\min\Big(d, tN, \sqrt{dN}\Big)\Big) \qquad \text{(I.58)}$$

$$\leq \exp\Big(-\frac{1}{24}\min\Big(d, \sqrt{dN}\Big)\Big) \qquad \text{(I.59)}$$

$$\leq \exp(-\Omega(\sqrt{d})). \qquad \text{(I.60)}$$

Observe that Talagrand's concentration inequality only gives us a bound on $\sup_{f\in\mathcal{F}}\sum_{i=1}^N f(X_i)$, while we wish to obtain a high-probability bound on $\sup_{f\in\mathcal{F}}|\sum_{i=1}^N f(X_i)|$. However, using the same argument, we can obtain a similar high-probability bound on $\sup_{f\in\mathcal{F}}\sum_{i=1}^N -f(X_i) = -\inf_{f\in\mathcal{F}}\sum_{i=1}^N f(X_i)$. Thus, we obtain a lower bound on $\inf_{f\in\mathcal{F}}\sum_{i=1}^N f(X_i)$ which is the negative of the upper bound that we obtained for $\sup_{f\in\mathcal{F}}\sum_{i=1}^N f(X_i)$, meaning that the upper bound we obtained for $\sup_{f\in\mathcal{F}}\sum_{i=1}^N f(X_i)$ also holds for $\sup_{f\in\mathcal{F}}|\sum_{i=1}^N f(X_i)|$. In summary,

$$U' \leq \mathrm{poly}(p_1, p_2, p_3, p_4, B^{p_1+p_2+p_3+p_4}, (B')^{q_1+q_2}) \cdot (\log d)^2 \cdot \sqrt{\frac{d}{N}} \qquad \text{(I.61)}$$

with probability at least $1 - \exp(-\Omega(\sqrt{d}))$.

Now that we have obtained a high-probability bound on $U'$, let us analyze the difference between $U'$ and $U$. Recall the definition of $g_{B,B',u_1,u_2,u_3,u_4,w_1,w_2}$ from above. Additionally, for convenience, let $g_{B',u_1,u_2,u_3,u_4,w_1,w_2}(x) = |\langle x, u_1\rangle|^{p_1}|\langle x, u_2\rangle|^{p_2}|\langle x, u_3\rangle|^{p_3}|\langle x, u_4\rangle|^{p_4}|\varphi_{B'}(\langle x, w_1\rangle)|^{q_1}|\varphi_{B'}(\langle x, w_2\rangle)|^{q_2}$, i.e. we are simply removing the truncation by $B$. Then, following the proof of Proposition 4.4 in Adamczak et al. [4], we have

$$U \leq U' \qquad \text{(I.62)}$$

$$+ \sup_{u_1,u_2,u_3,u_4\in\mathbb{S}^{d-1}} \frac{1}{N}\sum_{i=1}^N |g_{B',u_1,u_2,u_3,u_4,w_1,w_2}(X_i) - g_{B,B',u_1,u_2,u_3,u_4,w_1,w_2}(X_i)|\mathbb{1}\Big(\bigcup_{j=1}^4\{|\langle X_i, u_j\rangle| \geq B\}\Big) \qquad \text{(I.63)}$$

$$+ \sup_{u_1,u_2,u_3,u_4\in\mathbb{S}^{d-1}} \frac{1}{N}\sum_{i=1}^N \mathop{\mathbb{E}}_{X_i} |g_{B',u_1,u_2,u_3,u_4,w_1,w_2}(X_i) - g_{B,B',u_1,u_2,u_3,u_4,w_1,w_2}(X_i)|\mathbb{1}\Big(\bigcup_{j=1}^4\{|\langle X_i, u_j\rangle| \geq B\}\Big), \qquad \text{(I.64)}$$

where $\{C\}$ denotes the event that the condition $C$ holds. Here the inequality holds because the difference $g_{B',u_1,u_2,u_3,u_4,w_1,w_2}(X_i) - g_{B,B',u_1,u_2,u_3,u_4,w_1,w_2}(X_i)$ is nonzero if and only if $1\left(\bigcup_{j=1}^{4}\{|\langle X_i, u_j\rangle| \geq B\}\right)$ is nonzero. We have obtained a high probability bound on $U'$. Let us now obtain a bound on the third term on the right-hand side of Eq. (I.62) (here, $C_{p_1,p_2,p_3,p_4,q_1,q_2}$ is a constant that depends on $p_1, p_2, p_3, p_4, q_1, q_2$):

$$\mathbb{E}_{X_i} |g_{B',u_1,u_2,u_3,u_4,w_1,w_2}(X_i) - g_{B,B',u_1,u_2,u_3,u_4,w_1,w_2}(X_i)|1\left(\bigcup_{j=1}^{4}\{|\langle X_i, u_j\rangle| \geq B\}\right) \quad \text{(I.65)}$$

$$\leq \mathbb{E}_{X_i} |\langle X_i, u_1\rangle|^{p_1}|\langle X_i, u_2\rangle|^{p_2}|\langle X_i, u_3\rangle|^{p_3}|\langle X_i, u_4\rangle|^{p_4}|\langle X_i, w_1\rangle|^{q_1}|\langle X_i, w_2\rangle|^{q_2}1\left(\bigcup_{j=1}^{4}\{|\langle X_i, u_j\rangle| \geq B\}\right)$$
$$\text{(I.66)}$$

$$\leq \mathbb{E}_{X_i}[|\langle X_i, u_1\rangle|^{2p_1}|\langle X_i, u_2\rangle|^{2p_2}|\langle X_i, u_3\rangle|^{2p_3}|\langle X_i, u_4\rangle|^{2p_4}|\langle X_i, w_1\rangle|^{2q_1}|\langle X_i, w_2\rangle|^{2q_2}]^{1/2} \quad \text{(I.67)}$$

$$\mathbb{P}\left(\bigcup_{j=1}^{4}\{|\langle X_i, u_j\rangle| \geq B\}\right)^{1/2}$$

(By Cauchy-Schwarz inequality)

$$\leq C_{p_1,p_2,p_3,p_4,q_1,q_2}\mathbb{P}\left(\bigcup_{j=1}^{4}\{|\langle X_i, u_j\rangle| \geq B\}\right)^{1/2}. \quad \text{(I.68)}$$

To obtain the last inequality, we apply the Cauchy-Schwarz inequality multiple times to write

$$\mathbb{E}_{X_i}[|\langle X_i, u_1\rangle|^{2p_1}|\langle X_i, u_2\rangle|^{2p_2}|\langle X_i, u_3\rangle|^{2p_3}|\langle X_i, u_4\rangle|^{2p_4}|\langle X_i, w_1\rangle|^{2q_1}|\langle X_i, w_2\rangle|^{2q_2}]^{1/2} \quad \text{(I.69)}$$

as a product of moments of $|\langle X_i, u_j\rangle|$ and $|\langle X_i, w_j\rangle|$, and by Lemma I.10, each of these moments can be bounded above by a constant that depends on $p_1, p_2, p_3, p_4, q_1, q_2$. Additionally, by Lemma I.10 and Proposition 2.7.1, part (a) of Vershynin [74], we have $\mathbb{P}(|\langle X_i, u_j\rangle| \geq B) \lesssim e^{-\Omega(B)}$, since $|\langle X_i, u_j\rangle|$ has constant sub-exponential norm. In summary,

$$\mathbb{E}_{X_i} |g_{B',u_1,u_2,u_3,u_4,w_1,w_2}(X_i) - g_{B,B',u_1,u_2,u_3,u_4,w_1,w_2}(X_i)|1\left(\bigcup_{j=1}^{4}\{|\langle X_i, u_j\rangle| \geq B\}\right) \quad \text{(I.70)}$$

$$\leq C_{p_1,p_2,p_3,p_4,q_1,q_2}e^{-\Omega(B)}. \quad \text{(I.71)}$$

To complete the proof, we bound the second term on the right-hand side of Eq. (I.62). Following the proof of Proposition 4.4 of Adamczak et al. [4], we apply Lemma I.11 (a variant of Theorem 3.6 of Adamczak et al. [4]) to find that, with failure probability at most $e^{-\Omega(\sqrt{d})}$, for all $m$-sparse vectors $z \in \mathbb{S}^{N-1}$, we have

$$\left\|\sum_{i=1}^{N} z_i X_i\right\|_2 \lesssim \sqrt{d} + \sqrt{m}\log\left(\frac{2N}{m}\right). \quad \text{(I.72)}$$

Let $E_{\text{sparse}}$ be the event that Eq. (I.72) holds. For convenience, let $A \in \mathbb{R}^{d\times N}$ be the matrix whose $i^{th}$ column is $X_i$. Additionally, for any subset $S \subset [N]$, let $A_S$ be the matrix whose columns have indices in $S$. If $S \subset [N]$ with $|S| = m$, then assuming $E_{\text{sparse}}$ holds, the operator norm of $A_S$ is at

most $\sqrt{d} + \sqrt{m}\log\left(\frac{2N}{m}\right)$ up to a constant factor. Thus, for any set $S \subset [N]$, if $E_{\text{sparse}}$ holds,

$$\sup_{u_1,u_2,u_3,u_4\in\mathbb{S}^{d-1}}\left(\sum_{i\in S}|\langle X_i,u_1\rangle|^{p_1}|\langle X_i,u_2\rangle|^{p_2}|\langle X_i,u_3\rangle|^{p_3}|\langle X_i,u_4\rangle|^{p_4}|\varphi_{B'}(\langle X_i,w_1\rangle)|^{q_1}|\varphi_{B'}(\langle X_i,w_2\rangle)|^{q_2}\right)^{\frac{1}{p_1+p_2+p_3+p_4}}$$

$$\tag{I.73}$$

$$\leq (B')^{\frac{q_1+q_2}{p_1+p_2+p_3+p_4}}\sup_{u_1,u_2,u_3,u_4\in\mathbb{S}^{d-1}}\left(\sum_{i\in S}|\langle X_i,u_1\rangle|^{p_1}|\langle X_i,u_2\rangle|^{p_2}|\langle X_i,u_3\rangle|^{p_3}|\langle X_i,u_4\rangle|^{p_4}\right)^{\frac{1}{p_1+p_2+p_3+p_4}}$$

$$\tag{I.74}$$

$$\leq (B')^{\frac{q_1+q_2}{p_1+p_2+p_3+p_4}}\sup_{u_1,u_2,u_3,u_4\in\mathbb{S}^{d-1}}\left(\sum_{i\in S}|\langle X_i,u_1\rangle|^{\frac{2p_1}{p_1+p_2+p_3+p_4}}|\langle X_i,u_2\rangle|^{\frac{2p_2}{p_1+p_2+p_3+p_4}}\right.$$

$$\tag{I.75}$$

$$\left.|\langle X_i,u_3\rangle|^{\frac{2p_3}{p_1+p_2+p_3+p_4}}|\langle X_i,u_4\rangle|^{\frac{2p_4}{p_1+p_2+p_3+p_4}}\right)^{\frac{1}{2}}$$

(B.c. $\|x\|_2 \geq \|x\|_p$ for $p \geq 2$ and $x \in \mathbb{R}^d$, and $p_1 + p_2 + p_3 + p_4 \geq 2$)

$$\leq C_{p_1,p_2,p_3,p_4}(B')^{\frac{q_1+q_2}{p_1+p_2+p_3+p_4}}\sup_{u_1,u_2,u_3,u_4\in\mathbb{S}^{d-1}}\left(\sum_{i\in S}|\langle X_i,u_1\rangle|^2+|\langle X_i,u_2\rangle|^2\right.$$

$$\tag{I.76}$$

$$\left.+|\langle X_i,u_3\rangle|^2+|\langle X_i,u_4\rangle|^2\right)^{\frac{1}{2}}$$

(By Weighted AM-GM Inequality, with $C_{p_1,p_2,p_3,p_4}$ a constant depending on $p_1,p_2,p_3,p_4$)

$$\lesssim C_{p_1,p_2,p_3,p_4}(B')^{\frac{q_1+q_2}{p_1+p_2+p_3+p_4}}\cdot\left(\sqrt{d}+\sqrt{|S|}\log\left(\frac{2N}{|S|}\right)\right).\tag{I.77}$$

where the last inequality is because $E_{\text{sparse}}$ holds.

Following the proof of Proposition 4.4 of Adamczak et al. [4], we now define $S_B = \{i \in [N] \mid |\langle X_i,u_j\rangle| \geq B$ for some $j \in [4]\}$. Note that $S_B$ depends on $u_1,u_2,u_3,u_4$, and we will now show that if we select $B$ to be a certain value which is not too large, this will still ensure that $S_B$ is small. Observe that

$$\left(\sum_{i\in S_B}|\langle X_i,u_1\rangle|^2+|\langle X_i,u_2\rangle|^2+|\langle X_i,u_3\rangle|^2+|\langle X_i,u_4\rangle|^2\right)^{1/2}\geq\left(\sum_{i\in S_B}B^2\right)^{1/2}\geq B\sqrt{|S_B|}$$

$$\tag{I.78}$$

and

$$\left(\sum_{i\in S_B}|\langle X_i,u_1\rangle|^2+|\langle X_i,u_2\rangle|^2+|\langle X_i,u_3\rangle|^2+|\langle X_i,u_4\rangle|^2\right)^{1/2}\lesssim\sqrt{d}+\sqrt{|S_B|}\log\left(\frac{N}{|S_B|}\right).$$

(B.c. $E_{\text{sparse}}$ holds)

Combining these we obtain

$$B\sqrt{|S_B|}\lesssim\sqrt{d}+\sqrt{|S_B|}\log\left(\frac{N}{|S_B|}\right).\tag{I.79}$$

Therefore, if we select $B \gtrsim \log\left(\frac{N}{|S_B|}\right)$, we can eliminate the second term on the right-hand side of Eq. (I.79), finding that $|S_B| \lesssim \frac{d}{B^2}$. Combining this with Eq. (I.73), if $B \gtrsim \log N$ and $E_{\text{sparse}}$ holds we obtain

$$\sup_{u_1,u_2,u_3,u_4\in\mathbb{S}^{d-1}}\left(\sum_{i\in S_B}|\langle X_i,u_1\rangle|^{p_1}|\langle X_i,u_2\rangle|^{p_2}|\langle X_i,u_3\rangle|^{p_3}|\langle X_i,u_4\rangle|^{p_4}|\varphi_{B'}(\langle X_i,w_1\rangle)|^{q_1}|\varphi_{B'}(\langle X_i,w_2\rangle)|^{q_2}\right)^{\frac{1}{p_1+p_2+p_3+p_4}}$$

$$\tag{I.80}$$

$$\lesssim C_{p_1,p_2,p_3,p_4}(B')^{\frac{q_1+q_2}{p_1+p_2+p_3+p_4}}\cdot\left(\sqrt{d}+\sqrt{|S_B|}\log N\right)\tag{I.81}$$

$$\lesssim C_{p_1,p_2,p_3,p_4}(B')^{\frac{q_1+q_2}{p_1+p_2+p_3+p_4}}\cdot\left(\sqrt{d}+\frac{\sqrt{d}}{B}\log N\right)\qquad\text{(B.c. }|S_B|\lesssim\frac{d}{B^2}\text{)}$$

$$\lesssim C_{p_1,p_2,p_3,p_4}(B')^{\frac{q_1+q_2}{p_1+p_2+p_3+p_4}}\sqrt{d}.\qquad\text{(B.c. }B\gtrsim\log N\text{)}$$

Raising both sides to the $(p_1 + p_2 + p_3 + p_4)^{\text{th}}$ power, we finally obtain

$$\sup_{u_1,u_2,u_3,u_4 \in \mathbb{S}^{d-1}} \sum_{i \in S_B} |\langle X_i, u_1 \rangle|^{p_1} |\langle X_i, u_2 \rangle|^{p_2} |\langle X_i, u_3 \rangle|^{p_3} |\langle X_i, u_4 \rangle|^{p_4} |\varphi_{B'}(\langle X_i, w_1 \rangle)|^{q_1} |\varphi_{B'}(\langle X_i, w_2 \rangle)|^{q_2}$$

$$\tag{I.82}$$

$$\lesssim C'_{p_1,p_2,p_3,p_4} (B')^{q_1+q_2} \cdot d^{\frac{p_1+p_2+p_3+p_4}{2}} \tag{I.83}$$

with probability $1 - e^{-\Omega(\sqrt{d})}$, where $C'_{p_1,p_2,p_3,p_4}$ is a constant which depends on $p_1, p_2, p_3, p_4$.

In summary, assuming $E_{\text{sparse}}$ holds and the high-probability bound on $U'$ holds (and these both hold simultaneously with probability at least $1 - \exp(-\Omega(\sqrt{d}))$), we obtain

$$U \lesssim \text{poly}(p_1,p_2,p_3,p_4, B^{p_1+p_2+p_3+p_4}, (B')^{q_1+q_2}) \cdot (\log d)^2 \cdot \sqrt{\frac{d}{N}} + C'_{p_1,p_2,p_3,p_4} (B')^{q_1+q_2} \cdot \frac{d^{\frac{p_1+p_2+p_3+p_4}{2}}}{N}$$

$$\tag{I.84}$$

$$+ C_{p_1,p_2,p_3,p_4,q_1,q_2} e^{-\Omega(B)} \tag{I.85}$$

by Eq. (I.62). Finally, by Eq. (I.33), we have

$$\sup_{u_1,u_2,u_3,u_4 \in \mathbb{S}^{d-1}} \left| \frac{1}{N} \sum_{i=1}^{N} |\langle X_i, u_1 \rangle|^{p_1} |\langle X_i, u_2 \rangle|^{p_2} |\langle X_i, u_3 \rangle|^{p_3} |\langle X_i, u_4 \rangle|^{p_4} |\langle X_i, w_1 \rangle|^{q_1} |\langle X_i, w_2 \rangle|^{q_2} \right.$$

$$\tag{I.86}$$

$$\left. - \mathop{\mathbb{E}}_{X \sim \sqrt{d}\mathbb{S}^{d-1}} \left[ |\langle X, u_1 \rangle|^{p_1} |\langle X, u_2 \rangle|^{p_2} |\langle X, u_3 \rangle|^{p_3} |\langle X, u_4 \rangle|^{p_4} |\langle X, w_1 \rangle|^{q_1} |\langle X, w_2 \rangle|^{q_2} \right] \right|$$

$$\tag{I.87}$$

$$= U \pm C_{p_1,p_2,p_3,p_4,q_1,q_2} e^{-\Omega(B')} \tag{I.88}$$

as long as the events $E_1$ and $E_2$ hold, and these events simultaneously hold with probability at least $1 - \frac{1}{d^{\Omega(\log d)}}$. Thus, by our above bound on $U$ we have

$$\sup_{u_1,u_2,u_3,u_4 \in \mathbb{S}^{d-1}} \left| \frac{1}{N} \sum_{i=1}^{N} |\langle X_i, u_1 \rangle|^{p_1} |\langle X_i, u_2 \rangle|^{p_2} |\langle X_i, u_3 \rangle|^{p_3} |\langle X_i, u_4 \rangle|^{p_4} |\langle X_i, w_1 \rangle|^{q_1} |\langle X_i, w_2 \rangle|^{q_2} \right.$$

$$\tag{I.89}$$

$$\left. - \mathop{\mathbb{E}}_{X \sim \sqrt{d}\mathbb{S}^{d-1}} \left[ |\langle X, u_1 \rangle|^{p_1} |\langle X, u_2 \rangle|^{p_2} |\langle X, u_3 \rangle|^{p_3} |\langle X, u_4 \rangle|^{p_4} |\langle X, w_1 \rangle|^{q_1} |\langle X, w_2 \rangle|^{q_2} \right] \right|$$

$$\tag{I.90}$$

$$\lesssim \text{poly}(p_1,p_2,p_3,p_4, B^{p_1+p_2+p_3+p_4}, (B')^{q_1+q_2}) \cdot (\log d)^2 \cdot \sqrt{\frac{d}{N}} \tag{I.91}$$

$$+ C'_{p_1,p_2,p_3,p_4} (B')^{q_1+q_2} \cdot \frac{d^{\frac{p_1+p_2+p_3+p_4}{2}}}{N} + C_{p_1,p_2,p_3,p_4,q_1,q_2} e^{-\Omega(B)}$$

$$\tag{I.92}$$

$$+ C_{p_1,p_2,p_3,p_4,q_1,q_2} e^{-\Omega(B')} . \tag{I.93}$$

Setting $B' = (\log d)^2$ and $B \gtrsim \max(\log N, (\log d)^2) \asymp (\log d)^2$ (recall that $N \leq d^C$ for a universal constant $C$), we have

$$\sup_{u_1, u_2, u_3, u_4 \in \mathbb{S}^{d-1}} \left| \frac{1}{N} \sum_{i=1}^{N} |\langle X_i, u_1 \rangle|^{p_1} |\langle X_i, u_2 \rangle|^{p_2} |\langle X_i, u_3 \rangle|^{p_3} |\langle X_i, u_4 \rangle|^{p_4} |\langle X_i, w_1 \rangle|^{q_1} |\langle X_i, w_2 \rangle|^{q_2} \right.$$

(I.94)

$$\left. - \mathop{\mathbb{E}}_{X \sim \sqrt{d} \mathbb{S}^{d-1}} \left[ |\langle X, u_1 \rangle|^{p_1} |\langle X, u_2 \rangle|^{p_2} |\langle X, u_3 \rangle|^{p_3} |\langle X, u_4 \rangle|^{p_4} |\langle X, w_1 \rangle|^{q_1} |\langle X, w_2 \rangle|^{q_2} \right] \right|$$

(I.95)

$$\leq C_{p_1, p_2, p_3, p_4, q_1, q_2} (\log d)^{O(p_1 + p_2 + p_3 + p_4 + q_1 + q_2)} \cdot \sqrt{\frac{d}{N}}$$

(I.96)

$$+ C_{p_1, p_2, p_3, p_4} (\log d)^{O(q_1 + q_2)} \cdot \frac{d^{\frac{p_1 + p_2 + p_3 + p_4}{2}}}{N} + \frac{C_{p_1, p_2, p_3, p_4, q_1, q_2}}{d^{\Omega(\log d)}}$$

(I.97)

with probability at least $1 - \frac{1}{d^{\Omega(\log d)}}$, as desired. $\qquad\square$

**Lemma I.7** (Corollary for Uniform Samples from $\mathbb{S}^{d-1}$). *Let $x_1, \ldots, x_N$ be i.i.d. random vectors sampled uniformly from $\mathbb{S}^{d-1}$, and let $p_1, p_2, p_3, p_4, q_1, q_2$ be nonnegative integers. Suppose $p_1 + p_2 + p_3 + p_4 \geq 2$. Then, for fixed $w_1, w_2 \in \mathbb{S}^{d-1}$, as long as $N \leq d^C$ for any universal constant $C > 0$, we have*

$$\sup_{u_1, u_2, u_3, u_4 \in \mathbb{S}^{d-1}} \left| \frac{1}{N} \sum_{i=1}^{N} |\langle x_i, u_1 \rangle|^{p_1} |\langle x_i, u_2 \rangle|^{p_2} |\langle x_i, u_3 \rangle|^{p_3} |\langle x_i, u_4 \rangle|^{p_4} |\langle x_i, w_1 \rangle|^{q_1} |\langle x_i, w_2 \rangle|^{q_2} \right.$$

(I.98)

$$\left. - \mathop{\mathbb{E}}_{x \sim \mathbb{S}^{d-1}} \left[ |\langle x, u_1 \rangle|^{p_1} |\langle x, u_2 \rangle|^{p_2} |\langle x, u_3 \rangle|^{p_3} |\langle x, u_4 \rangle|^{p_4} |\langle x, w_1 \rangle|^{q_1} |\langle x, w_2 \rangle|^{q_2} \right] \right|$$

(I.99)

$$\leq \frac{1}{d^{\frac{p_1 + p_2 + p_3 + p_4 + q_1 + q_2}{2}}} \left( C_{p_1, p_2, p_3, p_4, q_1, q_2} (\log d)^{O(p_1 + p_2 + p_3 + p_4 + q_1 + q_2)} \cdot \sqrt{\frac{d}{N}} \right.$$

(I.100)

$$\left. + C_{p_1, p_2, p_3, p_4} (\log d)^{O(q_1 + q_2)} \cdot \frac{d^{\frac{p_1 + p_2 + p_3 + p_4}{2}}}{N} + \frac{C_{p_1, p_2, p_3, p_4, q_1, q_2}}{d^{\Omega(\log d)}} \right)$$

(I.101)

*with probability at least $1 - \frac{1}{d^{\Omega(\log d)}}$. Here, $C_{p_1, p_2, p_3, p_4, q_1, q_2}$ is a constant which depends on $p_1, p_2, p_3, p_4, q_1, q_2$, and $C_{p_1, p_2, p_3, p_4}$ is a constant which depends on $p_1, p_2, p_3, p_4$.*

*Proof of Lemma I.7.* This is a direct corollary of Lemma I.6, obtained by dividing both sides of Lemma I.6 by $\frac{1}{d^{(p_1 + p_2 + p_3 + p_4 + q_1 + q_2)/2}}$. $\qquad\square$

**Lemma I.8** (Moments for Truncated Dot Products with Random Spherical Vectors). *Let $P$ be a positive integer, and for $i = 1, \ldots, P$, let $u_i \in \mathbb{S}^{d-1}$ and $p_i$ be a nonnegative integer. Additionally, let $B \geq 1$. Finally, let $X$ be a random vector drawn uniformly from $\sqrt{d} \mathbb{S}^{d-1}$. Then, for some absolute constant $c > 0$ we have*

$$\left| \mathop{\mathbb{E}}_X \left[ \prod_{i=1}^{P} |\langle X, u_i \rangle|^{p_i} \right] - \mathop{\mathbb{E}}_X \left[ \prod_{i=1}^{P} |\varphi_B(\langle X, u_i \rangle)|^{p_i} \right] \right| \leq C_{p_1, \ldots, p_P} e^{-cB}$$

(I.102)

*where $\varphi_B$ is defined as*

$$\varphi_B(t) = \begin{cases} t & t \in [-B, B] \\ B & t > B \\ -B & t < B \end{cases}$$

(I.103)

*and $C_{p_1, \ldots, p_P}$ is a constant which depends only on $p_1, \ldots, p_P$.*

*Proof of Lemma I.8.* We use an argument similar to one used in part of the proof of Proposition 4.4 in Adamczak et al. [4]. First, for $i \in [P]$, we can write

$$|\langle X, u_i\rangle|^{p_i} = |\varphi_B(\langle X, u_i\rangle)|^{p_i} + (|\langle X, u_i\rangle|^{p_i} - B^{p_i})1_{|\langle X, u_i\rangle \geq B}. \tag{I.104}$$

Thus, in the expression

$$\prod_{i=1}^{P}|\langle X, u_i\rangle|^{p_i} - \prod_{i=1}^{P}|\varphi_B(\langle X, u_i\rangle)|^{p_i}, \tag{I.105}$$

if we expand the first product using Eq. (I.104), then we obtain terms of the form $\prod_{i=1}^{P} F_i$, where $F_i$ is either $|\varphi_B(\langle X, u_i\rangle)|^{p_i}$ or $(|\langle X, u_i\rangle|^{p_i} - B^{p_i})1(|\langle X, u_i\rangle| \geq B)$. Observe that the term where $F_i = |\varphi_B(\langle X, u_i\rangle)|^{p_i}$ for all $i \in [P]$ cancels with the second term in Eq. (I.105). Thus, in order to obtain an upper bound on

$$\mathbb{E}_{X}\left[\prod_{i=1}^{P}|\langle X, u_i\rangle|^{p_i} - \prod_{i=1}^{P}|\varphi_B(\langle X, u_i\rangle)|^{p_i}\right], \tag{I.106}$$

it suffices to upper bound the terms $\prod_{i=1}^{P} F_i$ where at least one of the $F_i$ is $(|\langle X, u_i\rangle|^{p_i} - B^{p_i})1(|\langle X, u_i\rangle| \geq B)$. Observe however that

$$(|\langle X, u_i\rangle|^{p_i} - B^{p_i})1(|\langle X, u_i\rangle| \geq B) \leq |\langle X, u_i\rangle|^{p_i}1(|\langle X, u_i\rangle| \geq B). \tag{I.107}$$

Thus, because $|\varphi_B(\langle X, u_i\rangle)| \leq |\langle X, u_i\rangle|$, in order to upper bound the terms $\prod_{i=1}^{P} F_i$ where at least one of the $F_i$ is $(|\langle X, u_i\rangle|^{p_i} - B^{p_i})1(|\langle X, u_i\rangle \geq B)$, it suffices to obtain an upper bound on the expected value of

$$1(|\langle X, u_i\rangle| \geq B \text{ for some } i \in [P])\prod_{i=1}^{P}|\langle X, u_i\rangle|^{p_i} \leq \sum_{i=1}^{P}1(|\langle X, u_i\rangle| \geq B)\prod_{j=1}^{P}|\langle X, u_j\rangle|^{p_i}. \tag{I.108}$$

Without loss of generality, let us bound the expected value of the first term in the sum above:

$$\mathbb{E}_{X}\left[1(|\langle X, u_1\rangle| \geq B)\prod_{i=1}^{P}|\langle X, u_j\rangle|^{p_i}\right] \leq \mathbb{E}_{X}[1(|\langle X, u_1\rangle| \geq B)]^{1/2}\,\mathbb{E}_{X}\left[\prod_{i=1}^{P}|\langle X, u_j\rangle|^{2p_i}\right]^{1/2}$$

(By the Cauchy-Schwarz Inequality)

$$\leq \mathbb{E}_{X}[1(|\langle X, u_1\rangle| \geq B)]^{1/2} \cdot \prod_{i=1}^{P}\mathbb{E}_{X}[|\langle X, u_j\rangle|^{2^{i+1}p_i}]^{\frac{1}{2^{i+1}}}$$

(By applying the Cauchy-Schwarz inequality $P - 1$ times)

$$\leq C_{p_1,\ldots,p_P}\,\mathbb{E}_{X}[1(|\langle X, u_1\rangle| \geq B)]^{1/2}, \tag{I.109}$$

where the last inequality is by Lemma I.10 and Proposition 2.7.1 of Vershynin [74]. Finally, since the sub-exponential norm of $\langle X, u_1\rangle$ is at most an absolute constant by Lemma I.10, we have that

$$\mathbb{E}_{X}[1(|\langle X, u_1\rangle| \geq B] = \mathbb{P}(|\langle X, u_1\rangle| \geq B) \lesssim e^{-cB} \tag{I.110}$$

for some absolute constant $c > 0$. Thus, by Eq. (I.109), we obtain

$$\mathbb{E}_{X}\left[1(|\langle X, u_1\rangle| \geq B)\prod_{i=1}^{P}|\langle X, u_j\rangle|^{p_j}\right] \lesssim C_{p_1,\ldots,p_P}e^{-cB} \tag{I.111}$$

for some absolute constant $c > 0$. This completes the proof. $\square$

**Lemma I.9.** *Let $N, d \in \mathbb{N}$. Let $X_1, \ldots, X_N$ i.i.d. random vectors drawn uniformly from $\sqrt{d}\mathbb{S}^{d-1}$. Additionally, let $\epsilon_1, \ldots, \epsilon_N$ be i.i.d. Rademacher random variables. Then,*

$$\mathbb{E}_{X,\epsilon_i}\left\|\frac{1}{N}\sum_{i=1}^{N}\epsilon_i X_i\right\|_2 \leq \sqrt{\frac{d}{N}}. \tag{I.112}$$

*Proof of Lemma I.9.* We first bound the second moment of the norm:

$$\mathop{\mathbb{E}}_{X,\epsilon_i} \Big\| \frac{1}{N} \sum_{i=1}^{N} \epsilon_i X_i \Big\|_2^2 = \frac{1}{N^2} \sum_{i=1}^{N} \sum_{j=1}^{N} \mathop{\mathbb{E}}_{\epsilon_i, X_i} \epsilon_i \epsilon_j \langle X_i, X_j \rangle \tag{I.113}$$

$$= \frac{1}{N^2} \sum_{i=1}^{N} \mathop{\mathbb{E}}_{X_i} \|X_i\|_2^2 \qquad (\text{B.c. } \mathbb{E}[\epsilon_i \epsilon_j] = 0 \text{ for } i \neq j)$$

$$= \frac{1}{N^2} \sum_{i=1}^{N} d \qquad (\text{B.c. } \|X_i\|_2 = \sqrt{d} \text{ with probability } 1)$$

$$= \frac{Nd}{N^2} \tag{I.114}$$

$$= \frac{d}{N}. \tag{I.115}$$

Thus, $\mathbb{E}_{X,\epsilon_i} \Big\| \frac{1}{N} \sum_{i=1}^{N} \epsilon_i X_i \Big\|_2 \leq \Big( \mathbb{E}_{X,\epsilon_i} \Big\| \frac{1}{N} \sum_{i=1}^{N} \epsilon_i X_i \Big\|_2^2 \Big)^{1/2} \leq \sqrt{\frac{d}{N}}$, as desired. $\qquad\square$

**Lemma I.10** (Sub-Exponential Norm of Uniform Distribution on $\sqrt{d}\mathbb{S}^{d-1}$)**.** *Assume $d \geq C$ for some sufficiently large universal constant $C$. Then there exists an absolute constant $K > 0$ such that, if $X$ is a random vector drawn uniformly from $\sqrt{d}\mathbb{S}^{d-1}$ and $v \in \mathbb{S}^{d-1}$ is a fixed vector, then $\|\langle X, v \rangle\|_{\psi_1} \leq K$, where $\|\cdot\|_{\psi_1}$ denotes the sub-exponential norm of a random variable. (See Definition 2.7.5 of Vershynin [74].) As a corollary, for any $p \geq 1$, there exists a constant $C_p > 0$ depending on $p$ such that for any $v \in \mathbb{S}^{d-1}$, $\mathbb{E}_{X \sim \sqrt{d}\mathbb{S}^{d-1}}[|\langle X, v \rangle|^p] \leq C_p$.*

*Proof of Lemma I.10.* Let $X$ be drawn uniformly at random from the uniform distribution on $\sqrt{d}\mathbb{S}^{d-1}$, and let $v \in \mathbb{S}^{d-1}$. Observe that the distribution of $\frac{1}{\sqrt{d}}\langle X, v \rangle$ is $\mu_d$. Thus, the density of $\frac{1}{\sqrt{d}}\langle X, v \rangle$ is $p(t) = \frac{\Gamma(d/2)}{\sqrt{\pi}\Gamma((d-1)/2)}(1 - t^2)^{\frac{d-3}{2}}$ by Eqs. (1.16) and (1.18) of Atkinson and Han [12]. By a change of variables, the density of $\langle X, v \rangle$ is

$$p(t) = \frac{\Gamma(d/2)}{\sqrt{\pi}\Gamma((d-1)/2)} \Big(1 - \frac{t^2}{d}\Big)^{\frac{d-3}{2}} \cdot \frac{1}{\sqrt{d}}. \tag{I.116}$$

Thus, for any $t \geq 0$,

$$\mathbb{P}(|\langle X, v \rangle| \geq t) \lesssim 2 \int_t^\infty p(s)ds \tag{I.117}$$

$$\lesssim 2 \int_t^\infty \frac{\Gamma(d/2)}{\sqrt{\pi}\Gamma((d-1)/2)} \Big(1 - \frac{s^2}{d}\Big)^{\frac{d-3}{2}} \cdot \frac{1}{\sqrt{d}} ds \tag{I.118}$$

$$\lesssim \int_t^\infty \Big(1 - \frac{s^2}{d}\Big)^{\frac{d-3}{2}} ds \qquad (\text{By Stirling's formula})$$

$$\lesssim \int_t^\infty e^{-\frac{d-3}{2} \cdot \frac{s^2}{d}} ds \qquad (\text{B.c. } 1 + x \leq e^x)$$

$$\lesssim \int_t^\infty e^{-\frac{s^2}{3}} ds \qquad (\text{B.c. } \frac{d-3}{2d} \geq \frac{1}{3} \text{ for } d \text{ sufficiently large})$$

Thus, for some absolute constant $C > 0$ and all $t \geq 0$,

$$\mathbb{P}(|\langle X, v \rangle| \geq t) \leq C \int_t^\infty e^{-\frac{s^2}{3}} ds \leq C \int_t^\infty e^{-\frac{st}{3}} ds = C\Big(-\frac{3}{t}e^{-\frac{st}{3}}\Big)\Big|_t^\infty = \frac{3C}{t}e^{-\frac{t^2}{3}} \leq e^{-\frac{t^2}{3} + \log\frac{3C}{t}} \tag{I.119}$$

Suppose $t \geq (18C)^{1/3}$. Then, $\frac{t^2}{6} \geq \log\frac{3C}{t}$, meaning that $-\frac{t^2}{3} + \log\frac{3C}{t} \leq -\frac{t^2}{6}$, and therefore

$$\mathbb{P}(|\langle X, v \rangle| \geq t) \leq e^{-\frac{t^2}{6}} \tag{I.120}$$

Thus, by the definition of sub-Gaussian norm (Definition 2.5.6 and Proposition 2.5.2 of Vershynin [74]), the sub-Gaussian norm of $\langle X, v \rangle$ is bounded by a constant, i.e. for any $v \in \mathbb{S}^{d-1}$, we have $\|\langle X, v \rangle\|_{\psi_2} \leq K$ for some absolute constant $K > 0$ (here $\|\cdot\|_{\psi_2}$ denotes the sub-Gaussian norm of a random variable). By Propositions 2.5.2 and 2.7.1, and Definition 2.7.5, of Vershynin [74], this implies that $\|\langle X, v \rangle\|_{\psi_1} \lesssim 1$, since sub-Gaussian random variables are always sub-exponential (specifically, using item (ii) of Proposition 2.5.2 and item (b) of Proposition 2.7.1 of Vershynin [74]). The last statement of the lemma follows directly from Proposition 2.7.1, item (b) of Vershynin [74]. $\square$

**Lemma I.11** (Modification of Theorem 3.6 of Adamczak et al. [4]). *Let $d \geq C$ for some absolute constant $C$, and $1 \leq N \leq e^{\sqrt{d}}$. Let $X_1, \ldots, X_N$ be i.i.d. random vectors sampled from the uniform distribution on $\sqrt{d}\mathbb{S}^{d-1}$. Finally, let $t \geq 1$. Then, for some absolute constant $c > 0$, with probability at least $1 - e^{-ct\sqrt{d}}$, for all $m \in [N]$,*

$$\sup_{\substack{z \in \mathbb{S}^{d-1} \\ |\operatorname{supp} z| \leq m}} \Big\| \sum_{i=1}^{N} z_i X_i \Big\|_2 \lesssim t \Big( \sqrt{d} + \sqrt{m} \log \Big( \frac{2N}{m} \Big) \Big) \tag{I.121}$$

*Here, we use $\operatorname{supp} z$ to denote the set of nonzero coordinates of $z$.*

*Proof of Lemma I.11.* The proof is the same as that of Theorem 3.6 of Adamczak et al. [4] — while Theorem 3.6 of Adamczak et al. [4] is shown for the case where $X_1, \ldots, X_N$ are sampled from an isotropic, log-concave probability distribution, the proof can be generalized to the case where the $X_i$ are sampled from $\sqrt{d}\mathbb{S}^{d-1}$. To see why, observe that the proof only uses the fact that the distribution of the $X_i$ is log-concave in the following places: (i) when applying Lemma 3.1 (also of Adamczak et al. [4]), of which the conclusion is that $\max_{i \leq N} \|X_i\|_2 \leq C_0 K \sqrt{d}$ with probability at least $1 - e^{-K\sqrt{d}}$, for some absolute constant $C_0$, (ii) when applying Lemmas 3.3 and 3.4 (also of Adamczak et al. [4]) the proof requires that $\|\langle X_i, y \rangle\|_{\psi_1} \leq \psi$ for some absolute constant $\psi > 0$ and any $y \in \mathbb{S}^{d-1}$. However, property (i) trivially holds when the $X_i$ are on $\sqrt{d}\mathbb{S}^{d-1}$ since $\|X_i\|_2 = \sqrt{d}$, and property (ii) holds by Lemma I.10. $\square$

## I.3 Helper Lemmas

**Lemma I.12.** *Let $f : [0, \infty) \to \mathbb{R}_{\geq 0}$ such that*

$$\frac{d}{dt} f(t)^2 \leq p_t f(t) + q_t f(t)^2 \tag{I.122}$$

*where $p_t, q_t > 0$ for $t \in [0, \infty)$. Then, for $T > 0$ we have*

$$f(T)^2 \leq \Big( f(0) + \frac{\max_t p_t}{\min_t q_t} \Big)^2 \exp \Big( \int_0^T q_t dt \Big). \tag{I.123}$$

*Proof.*

$$\frac{df(t)}{dt} = \frac{1}{2f(t)} \frac{df(t)^2}{dt} \leq \frac{p_t}{2} + \frac{q_t}{2} f(t) \leq \frac{\max_t p_t}{2} + \frac{q_t}{2} f(t). \tag{I.124}$$

Thus, we have

$$\frac{d}{dt} \Big( f(t) + \frac{\max_t p_t}{\min_t q_t} \Big) \leq \frac{q_t}{2} \Big( f(t) + \frac{\max_t p_t}{\min_t q_t} \Big). \tag{I.125}$$

As a result, by Fact I.13,

$$f(T) \leq \Big( f(0) + \frac{\max_t p_t}{\min_t q_t} \Big) \exp \Big( \int_0^T \frac{q_t}{2} dt \Big). \tag{I.126}$$

Taking the square of both sides finishes the proof. $\square$

**Fact I.13** (Gronwall's Inequality [13]). *Let $a < b$, and let $f : [a, b] \to \mathbb{R}_{\geq 0}$ and $g : [a, b] \to \mathbb{R}$ be differentiable functions, such that*

$$f'(t) \leq g(t)f(t) \tag{I.127}$$

*for all $t \in [a, b]$. Then, for all $t \in [a, b]$, $f(t) \leq f(a) \exp(\int_a^s g(s)ds)$. Additionally, suppose $f, g$ satisfy*

$$f'(t) \geq g(t)f(t) \tag{I.128}$$

*for all $t \in [a, b]$. Then, for all $t \in [a, b]$, $f(t) \geq f(a) \exp(\int_a^s g(s)ds)$.*

**Proposition I.14.** *For any $\beta_2, \beta_4$ such that $0 \leq \beta_2^2 \leq \beta_4 \leq \beta_2 \leq 1$, there exists a one-dimensional distribution $\mu$ that satisfies*

$$\underset{w \sim \mu}{\mathbb{E}}[w^2] = \beta_2, \tag{I.129}$$

$$\underset{w \sim \mu}{\mathbb{E}}[w^4] = \beta_4, \tag{I.130}$$

$$\operatorname{supp}(\mu) \subseteq [-1, 1]. \tag{I.131}$$

*Proof.* Let $p = \beta_2^2/\beta_4$. By the assumptions we have $0 \leq p \leq 1$. Consider the distribution

$$\mu(x) = \frac{\beta_2^2}{\beta_4} \delta_{x = \beta_4^{1/2}\beta_2^{-1/2}} + \left(1 - \frac{\beta_2^2}{\beta_4}\right)\delta_{x=0}, \tag{I.132}$$

where $\delta_{x=t}$ denotes the Dirac measure at $x = t$. First of all, since $\beta_2^2 \leq \beta_4$, the distribution $\mu(x)$ is well-defined. And we can verify that

$$\underset{w \sim \mu}{\mathbb{E}}[w^2] = \frac{\beta_2^2}{\beta_4}(\beta_4^{1/2}\beta_2^{-1/2})^2 = \beta_2, \tag{I.133}$$

$$\underset{w \sim \mu}{\mathbb{E}}[w^4] = \frac{\beta_2^2}{\beta_4}(\beta_4^{1/2}\beta_2^{-1/2})^4 = \beta_4. \tag{I.134}$$

Note that $0 \leq \beta_4 \leq \beta_2$ implies $\beta_4^{1/2}\beta_2^{-1/2} \in [0, 1]$. Consequently,

$$\operatorname{supp}(\mu) \subseteq [-1, 1]. \tag{I.135}$$

$\square$

**Lemma I.15.** *Let $\rho$ be a distribution which is rotationally invariant as in Definition 4.1. Let $\nabla_u L(\rho)$ be defined as in Eq. (3.4), and let $\nabla_z L(\rho)$ denote the last $(d - 1)$ coordinates of $\nabla_u L(\rho)$. Then, for any particle $u = (w, z)$, we have that $\nabla_z L(\rho)$ is a scalar multiple of $z$. In particular, if $\operatorname{grad}_u L(\rho)$ is defined as in Eq. (3.4) and $\operatorname{grad}_z L(\rho)$ denotes the last $(d - 1)$ coordinates of $\operatorname{grad}_u L(\rho)$, then $\operatorname{grad}_z L(\rho)$ is a scalar multiple of $z$.*

*Proof.* From Eq. (3.4), we have

$$\nabla_u L(\rho) = \underset{x \sim \mathbb{S}^{d-1}}{\mathbb{E}}[(f_\rho(x) - y(x))\sigma'(u^\top x)x]. \tag{I.136}$$

Since $\rho$ is rotationally invariant, if $x, x' \in \mathbb{S}^{d-1}$ such that $\langle e_1, x \rangle = \langle e_1, x' \rangle$, then $f_\rho(x) - y(x) = f_\rho(x') - y(x')$. Thus, $(f_\rho(x) - y(x))\sigma'(u^\top x)$ only depends on $\langle e_1, x \rangle$ and $\langle u, x \rangle$.

Let $P_{e_1, u} \in \mathbb{R}^{d \times d}$ denote the projection matrix into the subspace spanned by $e_1$ and $u$. Then,

$$\underset{x \sim \mathbb{S}^{d-1}}{\mathbb{E}}[(f_\rho(x) - y(x))\sigma'(u^\top x)(x - P_{e_1, u}x)] = 0 \tag{I.137}$$

because if $v \in \mathbb{S}^{d-1}$ is orthogonal to $e_1$ and $u$, then the distribution of $\langle v, x \rangle$ conditioned on $\langle e_1, x \rangle$ and $\langle u, x \rangle$ is symmetric around 0. Thus,

$$\nabla_u L(\rho) = \underset{x \sim \mathbb{S}^{d-1}}{\mathbb{E}}[(f_\rho(x) - y(x))\sigma'(u^\top x)P_{e_1, u}x] = P_{e_1, u}(\nabla_u L(\rho)) \tag{I.138}$$

and we have that $\nabla_u L(\rho)$ is in the subspace spanned by $e_1$ and $u$. In other words, there are scalars $c_1, c_2 \in \mathbb{R}$ such that

$$\nabla_u L(\rho) = c_1 e_1 + c_2 u \tag{I.139}$$

and taking the last $(d - 1)$ coordinates of both sides of this equation gives $\nabla_u L(\rho) = c_2 z$, as desired. The last statement of the lemma holds because $\operatorname{grad}_u L(\rho) = (I - uu^\top)\nabla_u L(\rho)$, which is also a linear combination of $e_1$ and $u$. $\square$

**Lemma I.16.** *Let $P$ be a polynomial of degree at most $k > 0$, where $k$ is bounded by a universal constant, and let $w_1, w_2 \in [-1, 1]$. Suppose the coefficients of $P$ are bounded by $B$. Then, $|P(w_1) - P(w_2)| \lesssim B|w_1 - w_2|$.*

*Proof.* Let $c_i$ be the coefficient of the $i^{th}$ degree term in $P$. Then, the lemma follows from expanding $P(w_1) - P(w_2)$ as

$$|P(w_1) - P(w_2)| \leq \sum_{i=0}^{k} |c_i||w_1^i - w_2^i| \tag{I.140}$$

$$\leq \sum_{i=0}^{k} |c_i||w_1 - w_2| \cdot \sum_{j=0}^{i} |w_1|^j |w_2|^{i-j} \tag{I.141}$$

$$\leq \sum_{i=0}^{k} (i+1) \cdot |c_i||w_1 - w_2| \qquad \text{(B.c. } |w_1|, |w_2| \leq 1\text{)} \tag{}$$

$$\leq B(k+1) \sum_{i=0}^{k} |w_1 - w_2| \tag{I.142}$$

$$= B(k+1)^2 |w_1 - w_2| \tag{I.143}$$

$$\lesssim B|w_1 - w_2| \qquad \text{(B.c. } k \text{ is at most an absolute constant)}$$

as desired. $\qquad\qquad\square$

**Lemma I.17** (Powers of Dot Products on $\mathbb{S}^{d-1}$)**.** *Let $u_1, u_2, u_3, u_4, u_5 \in \mathbb{S}^{d-1}$, and $p, q, r, s, t \geq 1$. Then,*

$$\underset{x \sim \mathbb{S}^{d-1}}{\mathbb{E}}[|\langle u_1, x\rangle|^p |\langle u_2, x\rangle|^q |\langle u_3, x\rangle|^r |\langle u_4, x\rangle|^s |\langle u_5, x\rangle|^t] \leq C_{p,q,r,s,t} \frac{1}{d^{(p+q+r+s+t)/2}} \cdot \tag{I.144}$$

*Here, $C_{p,q,r,s,t}$ is a constant depending on $p, q, r, s, t$.*

*Proof of Lemma I.17.* By repeated applications of the Cauchy-Schwarz inequality, we have

$$\underset{X \sim \sqrt{d}\mathbb{S}^{d-1}}{\mathbb{E}}[|\langle u_1, X\rangle|^p |\langle u_2, X\rangle|^q |\langle u_3, X\rangle|^r |\langle u_4, X\rangle|^s |\langle u_5, X\rangle|^t] \tag{I.145}$$

$$\leq \underset{X \sim \sqrt{d}\mathbb{S}^{d-1}}{\mathbb{E}}[\langle u_1, X\rangle|^{2p}]^{1/2} \underset{X \sim \sqrt{d}\mathbb{S}^{d-1}}{\mathbb{E}}[|\langle u_2, X\rangle|^{2q} |\langle u_3, X\rangle|^{2r} |\langle u_4, X\rangle|^{2s} |\langle u_5, X\rangle|^{2t}]^{1/2} \tag{I.146}$$

$$\leq \underset{X \sim \sqrt{d}\mathbb{S}^{d-1}}{\mathbb{E}}[\langle u_1, X\rangle|^{2p}]^{1/2} \underset{X \sim \sqrt{d}\mathbb{S}^{d-1}}{\mathbb{E}}[|\langle u_2, X\rangle|^{4q}]^{1/4} \underset{X \sim \sqrt{d}\mathbb{S}^{d-1}}{\mathbb{E}}[|\langle u_3, X\rangle|^{4r} |\langle u_4, X\rangle|^{4s} |\langle u_5, X\rangle|^{4t}]^{1/4} \tag{I.147}$$

$$\leq \underset{X \sim \sqrt{d}\mathbb{S}^{d-1}}{\mathbb{E}}[\langle u_1, X\rangle|^{2p}]^{1/2} \underset{X \sim \sqrt{d}\mathbb{S}^{d-1}}{\mathbb{E}}[|\langle u_2, X\rangle|^{4q}]^{1/4} \underset{X \sim \sqrt{d}\mathbb{S}^{d-1}}{\mathbb{E}}[|\langle u_3, X\rangle|^{8r}]^{1/8} \tag{I.148}$$

$$\underset{X \sim \sqrt{d}\mathbb{S}^{d-1}}{\mathbb{E}}[|\langle u_4, X\rangle|^{8s} |\langle u_5, X\rangle|^{8t}]^{1/8} \tag{I.149}$$

$$\leq \underset{X \sim \sqrt{d}\mathbb{S}^{d-1}}{\mathbb{E}}[\langle u_1, X\rangle|^{2p}]^{1/2} \underset{X \sim \sqrt{d}\mathbb{S}^{d-1}}{\mathbb{E}}[|\langle u_2, X\rangle|^{4q}]^{1/4} \underset{X \sim \sqrt{d}\mathbb{S}^{d-1}}{\mathbb{E}}[|\langle u_3, X\rangle|^{8r}]^{1/8} \tag{I.150}$$

$$\underset{X \sim \sqrt{d}\mathbb{S}^{d-1}}{\mathbb{E}}[|\langle u_4, X\rangle|^{16s}]^{1/16} \underset{X \sim \sqrt{d}\mathbb{S}^{d-1}}{\mathbb{E}}[|\langle u_5, X\rangle|^{16t}]^{1/16} \tag{I.151}$$

where each line results from a single application of the Cauchy-Schwarz inequality. Thus, by Lemma I.10, we have

$$\underset{X \sim \sqrt{d}\mathbb{S}^{d-1}}{\mathbb{E}}[|\langle u_1, X\rangle|^p |\langle u_2, X\rangle|^q |\langle u_3, X\rangle|^r |\langle u_4, X\rangle|^s |\langle u_5, X\rangle|^t] \leq C_{p,q,r,s,t} \cdot \tag{I.152}$$

Dividing both sides by $d^{(p+q+r+s+t)/2}$, we obtain

$$\underset{x \sim \mathbb{S}^{d-1}}{\mathbb{E}}[|\langle u_1, x\rangle|^p |\langle u_2, x\rangle|^q |\langle u_3, x\rangle|^r |\langle u_4, x\rangle|^s |\langle u_5, x\rangle|^t] \leq C_{p,q,r,s,t} \frac{1}{d^{(p+q+r+s+t)/2}} \tag{I.153}$$

as desired. $\qquad\qquad\square$

