# OpenReview forum: "Beyond NTK with Vanilla Gradient Descent: A Mean-Field Analysis of Neural Networks with Polynomial Width, Samples, and Time"
_NeurIPS.cc/2023/Conference — NeurIPS 2023 poster_

### Official Review · Reviewer_HQjX · 2023-07-05

**Soundness:** 4 excellent
**Presentation:** 3 good
**Contribution:** 4 excellent
**Rating:** 8
**Confidence:** 5

**Summary:**

This paper studies the global convergence of gradient descent for training two-layer networks in learning a high dimensional quartic function. The authors show that GD converges when the sample size $n = O(d^{3.1})$ and the width of the neural network grows at most polynomially in d. The authors also showed that any kernel method with $n \ll d^4$ cannot achieve the same order of accuracy.

**Strengths:**

First of all, this is an excellent paper on the non-asymptotic mean field analysis of GD training for two-layer neural networks and its separation from kernel method. Albeit the simplicity of the target function, the convergence analysis requests new ideas and tools. One crucial ingredient of the proof lies in showing that the projected gradient flow (population) dynamics can escape saddle points. The other key ingredient is bounding the coupling error between the empirical particle dynamics and the population dynamics. For the later, the authors tame the usual exponential growth rate in the coupling error with a delicate investigation on the relationship between the growth of the error and that of the signal.

**Weaknesses:**

I do not have many comments on the weakness, but have few questions, which are mostly relevant to future considerations. See below.


**Questions:**

The authors established the results for learning the quartic function. Does the proof generalize to a more general class of functions? If so, can the authors highlight the further work to be done? If not, it would be good if the authors can comment on the major challenges.

---

> ### Author Rebuttal · Authors · 2023-08-10
>
> We thank the reviewer for their positive review, and for mentioning that “this is an excellent paper on the non-asymptotic mean field analysis of GD for training two-layer neural networks.”
>
> # Extending to more general activations
>
> The framework that we use for analyzing the population gradient flow can potentially be generalized to higher-degree activations. We divide the population gradient flow into three phases, as described in Section 4. Phases 1 and 2 would likely have very similar analyses even if the activations/target function had higher degree. Analyzing Phase 3 with higher-degree activations/target functions would be more challenging, since the velocity function would have more roots.
>
> Separately from our techniques for analyzing the population dynamics, we also believe our techniques for analyzing the finite-width and finite-sample setting, by coupling the infinite-width/population loss and finite-width/empirical loss trajectories, can generalize to higher-degree polynomial activations with relatively modest effort, to obtain sample complexity better than NTK.

---

> > ### Comment · Reviewer_HQjX · 2023-08-19
> > **Keep my rating unchanged**
> >
> > I have read the rebuttal. Thank the authors for addressing my comments. I keep my rating unchanged.

---

> > > ### Author Response · Authors · 2023-08-19
> > > **Thank you!**
> > >
> > > Thank you very much for your reply!

---

### Official Review · Reviewer_ZhLf · 2023-07-07

**Soundness:** 3 good
**Presentation:** 3 good
**Contribution:** 3 good
**Rating:** 5
**Confidence:** 4

**Summary:**

This paper studied the projected gradient flow on two-layer neural networks in the mean-field regime with polynomial width and quartic activation function. With data sampled uniformly on the sphere, the authors proved that to learn a single-index model with an even quartic link function, this neural network needs $n=O(d^{3.1})$ training samples. However, we know for any inner product kernel ridge regression requires $n\gg d^4$ to learn a quartic target function.  This provides a concrete example of practical feature learning of neural networks outperforming the kernel methods and NTK regime when the target function has a low-rank structure.

**Strengths:**

Overall, I found the paper well-written and easy to follow. The paper presents interesting insights into more accurate models of neural network training. The results obtained by the authors are as sharp as can be reasonably expected given the problem in certain cases. Unlike previous analyses of feature learning with two-stage training procedures, this work shows that neural networks can learn both feature direction and nonlinear link function when training in a more practical way with the projected gradient flow. The methods of the proof may provide further insights to obtain more general results on feature learning.

**Weaknesses:**

1. The main limitation is that the activation function and the target link function are special, only quartic functions. It would be nice to check if the proof techniques can handle more general activation and target functions.  Another thing that can be improved is the precise order of the width: how wide the neural network is sufficient for this feature learning.
2. There should be a comparison between the results of the current submission and [10]. [10] uses information exponent to determine the sample complexity for online SGD to learn a single-index target. When $\gamma_2=0$ in Assumption 3.2, [10] implies that sample size should be $n\gg d^{3}\log n$ which is similar to the results in the current submission. Further explanations are needed.
3.  And it should be more convincing to visualize Phases 1-3 of the training dynamic in Section 5 if there is some simulation of synthetic data or real-world data for this finite-width training dynamic.

**Questions:**

1. [Ba et al. 2022] also study the feature learning and single-index model when $n\asymp d \asymp m$ with two-stage training processes.  And [Arnaboldi et al. 2023] and [Berthier et al. 2023] also studied the gradient flow dynamics of two-layer neural networks in high-dimension to a single-index model. Some comparisons with the current paper could be made.
2. In line 88, you mentioned that a similar coupling error was shown in [49] but [49] is for random feature regression with proportional limit. Can you explain more about it?
3. Line 147 typo.
4. In Theorem 3.4, the sample complexity is $n\ge d^\mu$. Is there an explicit formula or upper bound for the power $\mu$?
5. In Theorem 3.5, you considered the limitation of the inner-product kernel with data points drawn from the unit sphere. Does the data distribution have to be uniform unit sphere distribution? Can Theorem 3.5 be extended to a general rotational invariant kernel or Euclidean distance kernel?
6. Line 233 double "Appendix"
7. Line 274 $w$ should be $w_t$?
8.  In the analysis, Phases 1 and 2 are controlled by the quadratic components in the activation and target functions, and the dynamics are analogous to a power method update. Is there any relation between this and the PCA warmup proposed by [Chen and Meka 2020] before SGD training?
9. In Assumption 3.2, we need $\gamma_4\ge 1.1\gamma_2^2$. Why do we need this assumption and coefficient $1.1$? What if $\gamma_2=\gamma_4=1$?

===============================================================================================
- Chen, S. and Meka, R., 2020, July. Learning polynomials in few relevant dimensions. In Conference on Learning Theory (pp. 1161-1227). PMLR.
- Ba, J., Erdogdu, M.A., Suzuki, T., Wang, Z., Wu, D. and Yang, G., 2022. High-dimensional asymptotics of feature learning: How one gradient step improves the representation. Advances in Neural Information Processing Systems, 35, pp.37932-37946.
- Arnaboldi, L., Stephan, L., Krzakala, F. and Loureiro, B., 2023. From high-dimensional & mean-field dynamics to dimensionless ODEs: A unifying approach to SGD in two-layers networks. arXiv preprint arXiv:2302.05882.
- Berthier, R., Montanari, A. and Zhou, K., 2023. Learning time-scales in two-layers neural networks. arXiv preprint arXiv:2303.00055.

**Limitations:**

I think the authors have adequately addressed the potential negative social impact of their work. The conclusion lays out some suggested next steps and in doing so highlights certain current limitations.

---

> ### Author Rebuttal · Authors · 2023-08-10
>
> We thank the reviewer for their detailed feedback and for noting that our work “presents interesting insights into more accurate models of neural network training.” We now address the reviewer’s questions. We will incorporate presentation-related comments in the revision and include a simulation to illustrate Phases 1-3.
>
> # Generality of Activation Function
>
> We agree that our activation/target function deviates from more realistic settings. We keep the activations and target functions simple to not distract from our main objective of studying unmodified gradient descent in a non-trivial setting which requires avoiding bad stationary points. We note that while many works such as Abbe et al. (2022) study more general activation functions, they use modified algorithms which bypass the question of whether gradient descent (GD) can avoid bad stationary points. To our knowledge, our work is the first to study unmodified GD in the mean-field regime that goes beyond linear/quadratic activations.
>
> Also, the framework that we use for analyzing the population gradient flow can potentially be generalized to higher-degree activations. We divide the population gradient flow into three phases, as described in Section 4. Phases 1 and 2 would likely have very similar analyses even if the activations/target function had higher degree. Analyzing Phase 3 with higher-degree activations would be more challenging, since the velocity function would have more roots.
>
> Separately from our techniques for analyzing the population dynamics, we also believe our techniques for analyzing the finite-width and finite-sample setting, by coupling the infinite-width/population loss and finite-width/empirical loss trajectories, can generalize to higher-degree polynomial activations with relatively modest effort, to obtain sample complexity better than NTK.
>
> # Precise Width
>
> Up to logarithmic factors and other factors depending on the Legendre coefficients of the activation, the width m can be $d^{3 + \gamma}$, where $\gamma$ can be any arbitrarily small, but positive, universal constant.
>
> # Comparison with Ben Arous et al. (2021)
>
> Ben Arous et al. (2021) consider a single-neuron student network - this is significantly simpler than our setting where we consider a neural network with $\text{poly}(d)$ width. Still, it is plausible that the sample complexity of GD in our setting is also $d (\log d)^{O(1)}$, since the lowest order term in the Legendre decomposition (which is comparable to the information exponent for the case of spherical data) of our activation and target functions has degree 2. We leave the tight sample complexity for GD as an open question for future work.
>
> Our Assumption 3.2 excludes the case $\gamma_2 = 0$, due to the condition that $\gamma_4 \leq c_1 \gamma_2^2$. Our analysis of Phases 1 and 2 makes use of the fact that $\gamma_2$ is nonzero because the particles grow uniformly in magnitude while the second-order term is dominant.
>
> # Other Related Works
>
> Ba et al. (2022) perform one step of gradient descent on the hidden layer, followed by linear regression to fit the second layer. Meanwhile, we train the hidden layer for a long period of time.
>
> Arnaboldi et al. (2023) bound the error between dimension-free dynamics and the true dynamics. We also obtain one-dimensional dynamics, but we do not claim that it is novel. Rather, our main contributions are our convergence analysis of population dynamics (Section 4.2) and of coupling error between population and empirical dynamics (Section 5).
>
> Berthier et al. (2023) obtain a super-exponential coupling error, which gives a super-polynomial coupling error in our setting.
>
> # Exponent of sample complexity
>
> In the sample complexity $d^\mu$, we can set $\mu = 3 + \gamma$ where $\gamma$ is a positive, but arbitrarily small, universal constant.
>
> # Data Distribution for Kernel Lower Bound
>
> While our current proof requires a uniform distribution over the unit sphere, we believe that our proof technique is quite general and can be used to prove similar results for other data distributions. Essentially we only require that the minimum eigenvalue of the matrix $(K(x_i, x_j))_{i,j\in [n]}$ concentrates to its mean with constant probability, and $K(x_i,z)^2$ concentrates to its mean for a fixed $z$. Thus, we leave the technical details as future work.
>
> # PCA Warmup
>
> Our phases 1 and 2 are not directly related to the PCA warmup of Chen and Meka (2020) - they explicitly threshold points $x$ based on $y(x)$ to identify the low-rank subspace, while we show the neurons automatically achieve high correlation with $e_1$ in Phase 1 and 2. Additionally, in the rank-1 case, Chen and Meka (2020) perform Riemannian GD on a teacher-student setting where both the teacher and student have only one neuron and the same activation, while our model has poly(d) neurons.
>
> # Loosening assumption that $\gamma_4 \geq 1.1 \gamma_2^2$
>
> We make use of this assumption in the analysis of Phase 3, Case 2 (omitted from main body due to space constraints). In Phase 3, Case 2, we have $D_{4, t} > 0$ and $D_{2, t} < 0$, which by the assumption $\gamma_4 \geq 1.1 \gamma_2^2$, implies that the distribution $\rho_t$ cannot have all of its mass at a root of the velocity function. Thus the population dynamics will continue to make progress. More generally, we could instead have assumed that $\gamma_4 \geq c \gamma_2^2$ for any constant $c > 1$.
>
> Our proof could likely extend to the case $\gamma_2 = \gamma_4 = 1$ (and more generally $\gamma_4 = \gamma_2^2$), using a similar proof by contradiction to show that if all the particles are close to the root of the velocity, then this root must be $\sqrt{\gamma_2}$, or else a contradiction is obtained. We exclude this case to simplify the analysis while still having a nontrivial setting where unmodified GD achieves good sample complexity.
>
> # Citation to [49]
>
> This was a typo - we intended to cite Mei et al. (2019).
>
> # References
>
> (Mei et al. 2019) arxiv ID: 1902.06015

---

> > ### Author Response · Authors · 2023-08-18
> > **Minor Clarification on Precise Width**
> >
> > We wish to add a minor clarification to the "Precise Width" section of our rebuttal - this is not related to any of the other points that we make in the rebuttal. We just wanted to clarify that the width $m$ does not depend on the Legendre coefficients of the activations - we incorrectly said in the rebuttal that it has some factors depending on the Legendre coefficients. So the width $m$, up to $(\log d)^{O(1)}$ factors, is $d^{3 + \gamma}$ where $\gamma > 0$ is an arbitrarily small constant.
> >
> > We note that this change does not affect our submitted manuscript - in our actual submitted manuscript, we correctly calculated that the width only depends on $d$ and does not depend on the Legendre coefficients of the activation.

---

> > ### Comment · Reviewer_ZhLf · 2023-08-21
> > **Thanks for the rebuttal**
> >
> > Thanks for the authors' detailed response. I thank the authors for their clarifications in the rebuttal and appreciate the theoretical contributions of this work. But considering the model assumptions and the lack of a tight sample complexity for GD, I will keep my score.

---

> ### Author Response · Authors · 2023-08-19
> **Please let us know if you have any questions!**
>
> Dear reviewer, since the end of the discussion period is approaching, we just wanted to check if we have addressed your questions, or if you have any additional questions. Thank you very much!

---

### Official Review · Reviewer_c2Ri · 2023-07-08

**Soundness:** 4 excellent
**Presentation:** 3 good
**Contribution:** 3 good
**Rating:** 7
**Confidence:** 4

**Summary:**

This paper studies the statistical efficiency of the projected gradient dynamics on the sphere for (polynomial-width) two-layer neural networks under the mean-field regime. In particular, this work proves the sample complexity of $O(d^{3.1})$ for learning the single-index model with an unknown quartic link function.

**Strengths:**

The results can be of interest to the readers and are technically sound. The remarkable contributions compared to most related works are that this work

- studies not unnatural optimization dynamics but the standard projected gradient flow on the sphere,

- shows the separation between two-layer neural networks in the mean field regime and kernel methods by providing the sample complexities.

Technically, the improvement in particle complexity is also worth noting. In a naive way, the approximation accuracy to the distribution by using finite particles exponentially deteriorates in time, but this study prevents such deterioration by making use of the problem and model structures. Hence, I think this work certainly contributes to the context.

**Weaknesses:**

- This theory deals with not standard activation such as ReLU, sigmoid, and tanh but a special type of activation function (i.e., fourth-order polynomial).

- In my understanding, Abbe et al. (2023) [2] also studies the standard optimization dynamics and shows the superiority of the mean-field regime against linear models. A detailed discussion of the relationship to [2] would help clarify the position of the work.

**Questions:**

Could you specify the dependence of $m$ stated in the next sentence of Lemma 5.1: ``network width $m$ is sufficiently large polynomial of $d$?'' What does the highest degree of this polynomial depend on?

**Limitations:**

The limitation of the paper is well addressed.

---

> ### Author Rebuttal · Authors · 2023-08-10
>
> We thank the reviewer for their feedback and for noting that our “results can be of interest to the readers and are technically sound.” Below we address the reviewer’s questions and concerns.
>
> # Generality of Activation Function
>
> We agree with the reviewer that our activation/target function deviates from realistic settings. We chose to keep the activations and target functions simple to not distract from our main objective of studying unmodified gradient descent in a non-trivial setting which requires avoiding bad stationary points. We note that while many works such as Abbe et al. (2022) study more general activation functions, they use modified algorithms (such as the layerwise/two-stage algorithm used by Abbe et al. (2022)) which avoid the question of whether gradient descent can avoid bad stationary points. To our knowledge, our work is the first to study unmodified gradient descent that goes beyond linear/quadratic activations.
>
> Also, the framework that we use for analyzing the population gradient flow can potentially be generalized to higher-degree activations. We divide the population gradient flow into three phases, as described in Section 4. Phases 1 and 2 would likely have very similar analyses even if the activations/target function had higher degree. Analyzing Phase 3 with higher-degree activations/target functions would be more challenging, since the velocity function would have more roots.
>
> Separately from our techniques for analyzing the population dynamics, we also believe our techniques for analyzing the finite-width and finite-sample setting, by coupling the infinite-width/population loss and finite-width/empirical loss trajectories, can generalize to higher-degree polynomial activations with relatively modest effort, to obtain sample complexity better than NTK.
>
> # Comparison with Abbe et al. (2022)
>
> ### We study how gradient descent can escape bad stationary points - Abbe et al. (2022) use a two-stage algorithm, only training the hidden layer for O(1) time, which bypasses this challenge.
> Abbe et al. (2022) use a two-stage algorithm for the dimension-free population dynamics. They train the first layer for O(1) time, and show no guarantees on the error obtained by training the first layer. They then train the second layer weights, which is a kernel regression problem.
>
> ### We have a much tighter bound on the coupling error which allows us to analyze the training for O(log d) steps, while Abbe et al. (2022) have a much looser error bound which is super-exponential in the number of steps and could only work with O(1) training time.
> Technically speaking, in Abbe et al. (2022) the upper bound on coupling error is based on a loose Lipschitzness bound (also see Theorem 1B from Mei et al. (2019)), while we tighten the bound by comparing the growth of the error with the growth of the signal.
>
> ### We do not use fresh samples in every iteration, unlike Abbe et al. (2022) - ours is a more realistic setting.
> This setting causes the weights to be correlated with the samples, thus requiring a more intricate induction on the coupling error together with a uniform concentration bound from Adamczak et al. (2010).
>
> ### Our target function does not satisfy the merged-staircase property (MSP).
> The lowest degree term in our target function has degree 2. The more recent work of Abbe et al. (2023) generalizes the MSP to functions with more “leaps,” but they use a layerwise training algorithm, and apply non-standard projection steps while training the first layer (separating the coordinates into large and small coordinates and applying different projections to each subset). Thus, Abbe et al. (2023) does not study unmodified gradient descent.
>
> # Precise Order of Width
>
> Up to logarithmic factors and other factors depending on the Legendre coefficients of the activation, the width m can be $d^{3 + \gamma}$, where $\gamma$ can be any arbitrarily small, but positive, universal constant. We will explicitly mention it in the revised version.
>
> # References
>
> (Adamczak et al. 2010) Quantitative estimates of the convergence of the empirical covariance matrix in Log-concave Ensembles
>
> (Mei et al. 2019) Mean-field theory of two-layers neural networks: dimension-free bounds and kernel limit
>
> (Abbe et al. 2023) SGD learning on neural networks: leap complexity and saddle-to-saddle dynamics

---

> > ### Comment · Reviewer_c2Ri · 2023-08-18
> > **reply to authors**
> >
> > I appreciate the author's response. The authors have adequately addressed my concerns. I will keep the evaluation.

---

> > > ### Author Response · Authors · 2023-08-18
> > > **Thank you!**
> > >
> > > Thank you very much for your reply and for your review!

---

> > ### Author Response · Authors · 2023-08-18
> > **Minor Clarification**
> >
> > We wish to add a minor clarification to the "Precise Width" section of our rebuttal - this is not related to any of the other points that we make in the rebuttal. We just wanted to clarify that the width $m$ does not depend on the Legendre coefficients of the activations - we incorrectly said in the rebuttal that it has some factors depending on the Legendre coefficients. So the width $m$, up to $(\log d)^{O(1)}$ factors, is $d^{3 + \gamma}$ where $\gamma > 0$ is an arbitrarily small constant.
> >
> > We note that this change does not affect our submitted manuscript - in our actual submitted manuscript, we correctly calculated that the width only depends on $d$ and does not depend on the Legendre coefficients of the activation.

---

### Official Review · Reviewer_aN4G · 2023-07-11

**Soundness:** 3 good
**Presentation:** 3 good
**Contribution:** 2 fair
**Rating:** 5
**Confidence:** 3

**Summary:**

Analyzed the (projected) gradient flow dynamics of a two-layer neural network in the mean-field regime in learning a specific degree-4 single-index target function. The main contribution is a polynomial-time convergence guarantee and a sample complexity that outperforms kernel methods. This differs from the naive mean-field analysis where the network width can grow exponentially.

**Strengths:**

Most existing mean-field analyses of two-layer neural network focus on the optimization aspect, with the goal of showing global convergence of the training loss, but the generalization properties and sample complexity in learning certain class of target functions are largely unknown. The only exception that I know of is (Abbe et al. 2022). So this submission definitely tackles a challenging and interesting problem. On the technical side, the analysis differs from many previous works on learning single-index model with two-layer neural network, where the link function mismatch is typically handled by training the second-layer parameters.

**Weaknesses:**


I have the following concerns.

1. The problem setting is restrictive and convoluted, which is rather underwhelming given the promise of a *clean mean-field analysis* with no unnatural modifications. (i) The training algorithm is the unmodified projected gradient flow, but time discretization is not studied. (ii) The target function is assumed to be an even degree-4 function which is quite restrictive. Moreover, the trained neural network also uses a somewhat unnatural degree-4 activation function, and the Legendre coefficients are restricted by Assumption 3.2. The fact that this submission is lengthy and technically demanding despite these strong assumptions raises the question of whether the same analysis can be extended to more general problem settings.

2. The comparison against prior works is not sufficient, and consequently, the significance of the results cannot be easily evaluated.
* In the context of learning single-index model, the sample complexity of gradient descent is decided by the information exponent from (Ben Arous et al. 2021). This definition is originally defined in the analysis of online SGD, but the authors should discuss whether the same mechanism is present in the ERM setting. Specifically, my current reading is that this submission assumes an information exponent of 2, due to the factor of $\sigma_2^2\gamma_2$ in the denominator of the stopping time $T_*$ in Theorem 3.3.
Please clarify if this is the case.
(Ben Arous et al. 2021) *Online stochastic gradient descent on non-convex losses from high-dimensional inference*

* Related to the previous point, it is also well-known that trained two-layer neural network can outperform kernel models when the link function is a high-degree polynomial with low information exponent. For example, the staircase functions in (Abbe et al. 2022) can be learned in linear sample complexity. While these prior results typically employs a layer-wise training procedure, the similarity in the main message may undermine the significance of the theoretical results in this submission. Therefore, I feel that the authors should use more space to highlight the technical differences and challenges (no bias units and retraining of second layer, etc.).
(Abbe et al. 2022) *The merged-staircase property: a necessary and nearly sufficient condition for sgd learning of sparse functions on two-layer neural networks*

* The simplification of population dynamics into some low-dimensional object via symmetry has appeared in many prior works, such as the dimension-free PDE in (Abbe et al. 2022) and (Hajjar and Chizat 2022).
The authors need to explain the difference and new ingredients in Section 4.1.
(Hajjar and Chizat 2022) *Symmetries in the dynamics of wide two-layer neural networks*

* The claim that prior mean-field analysis only proved "generic exponential growth rate in the coupling error" is not entirely accurate. For the noisy gradient descent setting, recent papers have provided uniform-in-time propagation of chaos estimates, for example see (Chen et al. 2022) (Suzuki et al. 2023). These results hold for finite-width network and discrete-time algorithm, but the current submission only handles the finite-width error. It would be a good idea to comment on the possibility/difficulty of  handling the discrete projected gradient descent algorithm.
(Chen et al. 2022) *Uniform-in-time propagation of chaos for mean field Langevin dynamics*
(Suzuki et al. 2023) *Convergence of mean-field Langevin dynamics: Time and space discretization, stochastic gradient, and variance reduction*

**Questions:**

I would be happy to update my evaluation if the authors could address the concerns and questions in the Weaknesses section.

#####################**Post-rebuttal Update**#####################

The authors addressed some of my concerns; I have therefore increased my score to 5.
In the revised manuscript, please include a detailed comparison against relevant prior works, as done in the rebuttal.

**Limitations:**

See weaknesses.

---

> ### Author Rebuttal · Authors · 2023-08-10
>
> We thank the reviewer for their very helpful comments, and for noting that our submission “definitely tackles a challenging and interesting problem.” We now respond to each of the reviewer’s concerns.
>
> # Discrete Time
>
> We were able to extend our results to time discretization, and we will include this result upon revision. Specifically, we can prove that projected gradient descent with 1/poly(d) step size and a poly(d) width neural network can achieve low population loss in poly(d) iterations as a relatively direct extension of our existing results (please see the next paragraph).
>
> To extend to discrete time, we emulate standard bounds on the discretization error of Euler’s method. To bound the error per time step, we bound the smoothness of the empirical gradient, using concentration bounds similar to those we used in the analysis of the empirical dynamics.
>
> # Generality of Activations
>
> We agree with the reviewer that our activation/target function deviates from realistic settings. We chose to keep the activations and target functions simple to not distract from our main objective of studying unmodified gradient descent in a non-trivial setting which requires avoiding bad stationary points. We note that while many works such as Abbe et al. (2022) study more general activation functions, they use modified algorithms (such as the layerwise/two-stage algorithm used by Abbe et al. (2022)) which avoid the question of whether gradient descent (GD) can avoid bad stationary points. To our knowledge, our work is the first to study unmodified GD in the mean-field regime beyond linear/quadratic activations.
>
> Also, the framework that we use for analyzing the population gradient flow can potentially be generalized to higher-degree activations. We divide the population gradient flow into three phases, as described in Section 4. Phases 1 and 2 would likely have very similar analyses even if the activations/target function had higher degree. Analyzing Phase 3 would be more challenging, since the velocity function would have more roots.
>
> Separately, we believe our techniques for coupling error of the finite-width/finite-sample trajectory can generalize to higher-degree polynomial activations with relatively modest effort, to obtain sample complexity better than NTK.
>
> # Comparison to Ben Arous et al. (2021)
>
> In the setting of Ben Arous et al. (2021), the goal is to learn a student neural network which has a single neuron - this is significantly simpler than our setting where we consider a neural network with $\text{poly}(d)$ width. It is plausible that the sample complexity of GD in our setting is also $d (\log d)^{O(1)}$, since the lowest order term in the Legendre decomposition (which is comparable to the information exponent for the case of spherical data) of our activation and target functions has degree 2. We leave the tight sample complexity for GD as an open question for future work.
>
> # Comparison to Abbe et al. (2022)
>
> ### We study how gradient descent can escape bad stationary points - Abbe et al. (2022) use a two-stage algorithm, only training the hidden layer for O(1) time, which bypasses this challenge.
> Abbe et al. (2022) use a two-stage algorithm for the dimension-free population dynamics. They train the first layer for O(1) time, and show no guarantees on the error obtained by training the first layer. They then train the second layer weights, which is a kernel regression problem.
>
> ### We have a much tighter bound on the coupling error which allows us to analyze the training for O(log d) steps, while Abbe et al. (2022) have a much looser error bound which is super-exponential in the number of steps and could only work with O(1) training time.
> Technically speaking, in Abbe et al. (2022) the upper bound on coupling error is based on a loose Lipschitzness bound (also see Theorem 1B from Mei et al. (2019)), while we tighten the bound by comparing the growth of the error with the growth of the signal.
>
> ### We do not use fresh samples in every iteration, unlike Abbe et al. (2022) - ours is a more realistic setting.
> This setting causes the weights to be correlated with the samples, thus requiring a more intricate induction on the coupling error together with a uniform concentration bound from Adamczak et al. (2010).
>
> ### Our target function does not satisfy the merged-staircase property (MSP).
> The lowest degree term in our target function has degree 2. The more recent work of Abbe et al. (2023) generalizes the MSP to functions with more “leaps,” but they apply non-standard projection steps while training the first layer (separating the coordinates into large and small coordinates and applying different projections to each subset), rather than studying unmodified GD.
>
> # Uniform-in-time propagation of chaos
>
> We will cite Suzuki et al. (2023) and Chen et al. (2022) (though the work of Suzuki et al. (2023) was posted after the NeurIPS deadline). **However, it is likely highly non-trivial to apply current analyses of mean-field Langevin dynamics to obtain good test error and sample complexity.** Assuming Theorem 4 of Mei et al. (2018) is tight, the inverse temperature $\lambda$ has to be at least proportional to the dimension $D$ for Langevin dynamics to achieve good test error. However, this causes the log-Sobolev constant in Suzuki et al. (2023) to be $e^{-D}$, and hence their runtime is $e^D$. In comparison, **we are able to extend our techniques to discrete-time projected gradient descent with $\text{poly}(d)$ iterations.** Also, Chen et al. (2022) do not study finite data or discrete time.
>
> # Dimension-free dynamics
>
> We do not claim that our reduction to one-dimensional dynamics is novel. The main novelty of our work is in our convergence analysis of the 1-dimensional dynamics and the coupling error between the population and empirical dynamics.
>
> # References
>
> (Adamczak et al. 2010) arxiv ID: 0903.2323
> (Mei et al. 2018) 1804.06561
> (Mei et al. 2019) 1902.06015
> (Abbe et al. 2023) 2302.11055

---

### Decision · Program_Chairs · 2023-09-21

**Decision:**

Accept (poster)

**Comment:**

The paper studies the dynamics of gradient descent for learning a single index function on the sphere. It performs a mean field analysis, which yields a generalization guarantee for O(d^{3.1}) samples and a polynomial-width network. In contrast, spherical kernel methods (including NTK) require d^4 samples to guarantee generalization. At a technical level, the analysis is able to show that gradient descent avoids suboptimal critical points, without requiring layerwise training.

Reviewers expressed a uniformly positive evaluation of the paper’s contributions: it shows a separation between neural network training and kernel methods, with polynomial network width, without requiring unnatural modifications to the training procedure. The main limitation seems to quartic activation function, and its assumptions on the target function. Major points of discussion on review included the relationship to previous works on learning single index functions and the aforementioned limitations. After considering author responses, reviewers uniformly recommended acceptance, based on the quality and novelty of the paper’s theoretical results.